# Approximation and Generalization Abilities of Score-based Neural Network Generative Models for Sub-Gaussian Distributions

**Guoji Fu**
School of Computing
National University of Singapore
guojifu@comp.nus.edu.sg

**Wee Sun Lee**
School of Computing
National University of Singapore
leews@comp.nus.edu.sg

## Abstract

This paper studies the approximation and generalization abilities of score-based neural network generative models (SGMs) in estimating an unknown distribution $P_0$ from $n$ i.i.d. observations in $d$ dimensions. Assuming merely that $P_0$ is $\alpha$-sub-Gaussian, we prove that for any time step $t \in [t_0, n^{\mathcal{O}(1)}]$, where $t_0 > \mathcal{O}(\alpha^2 n^{-2/d} \log n)$, there exists a deep ReLU neural network with width $\leq \mathcal{O}(n^{\frac{3}{d}} \log_2 n)$ and depth $\leq \mathcal{O}(\log^2 n)$ that can approximate the scores with $\tilde{\mathcal{O}}(n^{-1})$ mean square error and achieve a nearly optimal rate of $\tilde{\mathcal{O}}(n^{-1} t_0^{-d/2})$ for score estimation, as measured by the score matching loss. Our framework is universal and can be used to establish convergence rates for SGMs under milder assumptions than previous work. For example, assuming further that the target density function $p_0$ lies in Sobolev or Besov classes, with an appropriately early stopping strategy, we demonstrate that neural network-based SGMs can attain nearly minimax convergence rates up to logarithmic factors. Our analysis removes several crucial assumptions, such as Lipschitz continuity of the score function or a strictly positive lower bound on the target density.

## 1 Introduction

Score-based generative modeling (SGM) [1–5], also called diffusion modeling, has emerged as a powerful tool of generative models, demonstrating exceptional performance in a wide range of applications, such as image and text generation [4, 6], video generation [3]. SGM typically encompasses two Markov processes: a forward process that gradually adds noise to convert samples drawn from a data distribution, denoted as $P_0$, into noise (e.g., Gaussian noise), and a reverse process that effectively reverses the forward process to recover the samples from noise. Specifically, SGM uses score functions (i.e., gradients of the log probability density functions) to transform the Gaussian noise into the target data distribution via solving a stochastic differential equation (SDE). Implementing the reverse process requires accurately estimating the score functions, which is typically accomplished through training neural networks on a finite number of samples using a score matching objective [7, 8].

Despite their remarkable empirical success across a wide range of applications, the theoretical understanding of SGMs remains in its infancy. In particular, the following fundamental questions remain inadequately addressed in the literature:

*How effectively do diffusion models approximate the true data distribution? What is the optimal number of diffusion steps required for high-quality generation? How many training samples are necessary for diffusion models to estimate the true distribution accurately? In which scenarios do diffusion models excel, and where do they encounter limitations?*

39th Conference on Neural Information Processing Systems (NeurIPS 2025).

To address these questions, existing theoretical analyses of SGMs primarily focus on two aspects: (i) *the convergence rates of SGMs*, which aims to quantify how quickly SGMs converge to the target distribution, assuming access to accurate score estimators; (ii) *the generalization bounds of SGMs*, which, on the other hand, investigates the score estimation error bounds throughout the diffusion process given a finite number of observations.

Early works on convergence analysis literature either often relied on strong structural assumptions about the data distribution, such as requiring it to satisfy the log-Sobolev inequality (LSI) [9, 10], be log-Concave [11], or they exploit exponential convergence rates [12, 13]. Subsequent research [14–16] achieved polynomial convergence rates under milder assumptions, requiring that the data distributions have finite second moments and Lipschitz continuous score functions along the diffusion process. More recently, [17, 18] have established nearly linear convergence rates in data dimension, requiring only that the data has finite second moments or finite Fisher information with respect to (w.r.t) the Gaussian distribution. Notably, the best-known convergence rates for Langevin Monte Carlo (LMC) under various functional inequalities [19–21] also scale linearly with the data dimension, up to logarithmic factors, thus matching the rates achieved by [17, 18]. This observation shows that when arbitrarily accurate score estimators are available, SGMs can approximate the data distribution effectively without imposing stringent regularity conditions such as isoperimetry, log-concavity, LSI, or even smoothness on the target distribution.

However, assuming perfect score estimation in the convergence analysis of SGMs is highly restrictive and generally unattainable in practice, especially when only a finite number of observations are available. Recent studies have delved into analyzing score estimation errors and investigated how these errors influence the final distribution estimation. [22–28] have studied the statistical guarantees of neural network-based score estimators, showing that neural network-based SGMs are effective distribution learners for distributions on bounded support [22] or smooth low-dimensional manifolds [26, 27] with lower-bounded densities; distributions on low-dimensional linear subspace [23] or manifolds which are the images of Hölder smooth maps [28]; distributions on bounded support with Lipschiz continuous score functions [25]; and sub-Gaussian distributions with Barron class of density [24]. Additionally, [29–32] have focused on kernel-based score estimators and demonstrated that kernel-based SGMs achieve minimax optimal convergence rates for distributions that are sub-Gaussian with Sobolev class of density [29] or Lipschitz continuous score functions [30] as well as for distributions on bounded support with Hölder smooth density [31]. [32] investigated the sample complexity results for scenarios where the target distribution is either a standard Gaussian or has bounded support, and discussed challenges related to the potential memorization of training samples when using KDE-based score estimators. Table 1 summarizes some recent studies on generalization analysis for SGMs.

Despite these advances, some assumptions on the data adopted in existing work remain quite restrictive. For example, the assumption of Lipschitz continuous score functions used in [25, 30] excludes many distributions of interest, such as those supported on a submanifold. Additionally, the Lipschitz constant can conceal additional dependence on the data dimension in some cases, especially when the data are approximately supported on a submanifold. Moreover, as highlighted in [29], the density lower-bound assumption, as employed in [22, 26, 27], prevents their results from being applicable to many natural distribution classes, such as multimodal distributions and mixtures with well-separated components, significantly restricting their ability to explain the practical success achieved by SGMs. Under such stringent requirements, the Holley–Stroock perturbation principle [33] allows us to conclude that the true density satisfies the LSI. When the LSI holds, it is well-established that Langevin dynamics are sufficient to achieve statistical efficiency [34]. However, diffusion models are designed to be effective for a broader range of distributions by incorporating smoothed versions of the data distributions. On the other hand, several studies, such as [29–31], have been focusing on kernel-based estimators, whereas neural network-based estimators are more widely used in practice. These works do not address the theoretical challenges associated with neural network-based score estimation, leaving a gap in understanding the practical effectiveness of SGMs.

## 1.1 Our contributions

In this paper, we develop a new theoretical framework for analyzing the approximation and generalization capabilities of neural network-based SGMs. Assuming that the data distribution is $\alpha$-sub-Gaussian on $\mathbb{R}^d$, we first establish a bound on the score matching loss between the true score functions, $\frac{\nabla p_t(\cdot)}{p_t(\cdot)}$, and a regularized empirical counterpart $\frac{\nabla \hat{p}_t(\cdot)}{\hat{p}_t(\cdot) \vee \rho_{n,t}}$ with a regularization $\rho_{n,t} > 0$,

Table 1: A summary of recently developed generalization bounds for SDE-based SGMs. Bounds are expressed in terms of the distribution estimation error in total variation (TV) distance. ($P_0, \widehat{P}_0$: true and learned data distributions; $p_0$: true density function; KDE: kernel-based density estimator; DNNs: deep neural networks; $\hat{\psi}_t(\cdot)$ kernel function; $s$: smoothness parameter; $t_0$: early stopping time)

| Paper | Assumption | Estimator | | Metric | Bound |
|---|---|---|---|---|---|
| [29] | Sub-Gaussian $P_0$ | KDE: | $\frac{\nabla\hat{p}_t(\cdot)}{\hat{p}_t(\cdot)}\mathbb{1}_{\hat{p}_t(\cdot)>\rho_n}$ | $\mathrm{TV}(P_{t_0}, \widehat{P}_{t_0})$ | $\tilde{\mathcal{O}}(n^{-1/2}t_0^{-d/4})$ |
| | Sub-Gaussian $P_0$ Sobolev class $p_0$ | | | $\mathrm{TV}(P_0, \widehat{P}_0)$ | $\tilde{\mathcal{O}}\big(n^{\frac{-s}{d+2s}}\big)$ |
| [30] | Sub-Gaussian $P_0$ $L$-Lipschitz score | KDE: | $\frac{\nabla\hat{p}_t(\cdot)}{\hat{p}_t(\cdot)\vee\rho_n}$ | $\mathrm{TV}(P_0, \widehat{P}_0)$ | $\tilde{\mathcal{O}}\big(L^{\frac{d+2}{d+4}}n^{\frac{-1}{d+4}}\big)$ |
| [31] | $\mathrm{supp}(P_0)=[0,1]$ Hölder class $p_0$ | KDE: | $\frac{\nabla\hat{\psi}_t(\cdot)}{\hat{p}_t(\cdot)\vee\rho(\cdot,t)}$ | $\mathrm{TV}(P_0, \widehat{P}_0)$ | $\mathcal{O}\big(n^{\frac{-s}{1+2s}}\big)$ |
| [22] | $\mathrm{supp}(P_0)=[0,1]^d$ lower bounded $p_0$ Besov class $p_0$ | ReLU DNNs | | $\mathrm{TV}(P_0, \widehat{P}_0)$ | $\tilde{\mathcal{O}}(n^{\frac{-s}{d+2s}})$ |
| Theorem 3 | Sub-Gaussian $P_0$ | | | $\mathrm{TV}(P_{t_0}, \widehat{P}_{t_0})$ | $\tilde{\mathcal{O}}(n^{-1/2}t_0^{-d/4})$ |
| Corollary 1 | Sub-Gaussian $P_0$ Sobolev class $p_0$ | ReLU DNNs | | $\mathrm{TV}(P_0, \widehat{P}_0)$ | $\tilde{\mathcal{O}}(n^{\frac{-s}{d+2s}})$ |
| Corollary 2 | $\mathrm{supp}(P_0)=[0,1]^d$ Besov class $p_0$ | | | | |

which is the KDE-based estimator introduced in [30]. We then derive the approximation and estimation rates of ReLU DNNs for learning the true score function $\frac{\nabla p_t(\cdot)}{p_t(\cdot)}$ (see Theorems 1 and 3, respectively) by approximating and estimating the surrogate $\frac{\nabla\hat{p}_t(\cdot)}{\hat{p}_t(\cdot)\vee\rho_{n,t}}$.

**Score estimation via empirical Bayes smoothing.** Inspired by [30], we employ empirical Bayes smoothing techniques to establish a score estimation rate of $\tilde{\mathcal{O}}\big(n^{-1}\sigma_t^{-d-2}(\sigma_t^d \vee 1)\big)$ for the estimator $\frac{\nabla\hat{p}_t(\cdot)}{\hat{p}_t(\cdot)\vee\rho_{n,t}}$ under only a sub-Gaussian assumption (see Lemma 1). Notably, we improve upon [30] by removing the Lipschitz score requirement, demonstrating that regularity assumptions are unnecessary for achieving minimax optimal rates for $\frac{\nabla\hat{p}_t(\cdot)}{\hat{p}_t(\cdot)\vee\rho_{n,t}}$, which matches the result obtained by [29] for the truncated score estimator $\frac{\nabla\hat{p}_t(\cdot)}{\hat{p}_t(\cdot)}\mathbb{1}_{\{\hat{p}_t(\cdot)>\rho_n\}}$.

**Neural network score approximation.** We demonstrate in Lemma 12 that there exists a ReLU DNN of width $\mathcal{O}(\log^3 n)$ and depth $\mathcal{O}(n^{3/d}\log_2 n)$ that approximates $\frac{\nabla\hat{p}_t(\cdot)}{\hat{p}_t(\cdot)\vee\rho_{n,t}}$ with an $\tilde{\mathcal{O}}(n^{-1})$ rate for time steps $t \in [n^{-2/d}, \infty)$. While DNN approximation rates for smooth functions have been well studied (e.g., [35–39]), a naive application of existing results, e.g., assigning one sub-network per exponential component in KDE, would cause the network size to grow linearly with the sample size $n$, making practical estimation infeasible. In contrast, our proof constructs a more compact DNN architecture, which not only prevents the size from blowing up with $n$ but also yields nearly optimal estimation error bounds. Moreover, unlike [22], our approximation results do not require the density lower bound assumption. The approximation rate for the true score function $\frac{\nabla p_t(\cdot)}{p_t(\cdot)}$ (i.e., Theorem 1) follows immediately by combining Lemmas 1 and 12.

**Neural network score and distribution estimations.** [28] recently identified a flaw in Theorem C.4 of [22], which invalidates the proofs for the convergence rates claimed in that work and in subsequent papers [26, 27] that rely on it. In contrast, [28] establish a corrected score estimation rate of $\tilde{\mathcal{O}}(t_0^{-1}(t_0^{-1}n)^{-\frac{2s}{2s+d'}})$ for data supported on a $d'$-dimensional manifold ($d' \ll d$), where the manifold is the image of a $s$-Hölder smooth map. In this paper, we aim to derive optimal convergence rates for SGMs under broader conditions. Specifically, we provide a new proof strategy that removes the need for a density lower bound condition. A key observation is that the surrogate $\frac{\nabla\hat{p}_t(\cdot)}{\hat{p}_t(\cdot)\vee\rho_{n,t}}$ can be uniformly bounded (see Lemma 5) which allows us to verify Bernstein's condition for the excess risk associated with learning $\frac{\nabla\hat{p}_t(\cdot)}{\hat{p}_t(\cdot)\vee\rho_{n,t}}$. Instead of directly deriving a high probability bound for

the true score function, we first establish a uniform bound for the constructed DNN class trained to learn $\frac{\nabla \hat{p}_t(\cdot)}{\hat{p}_t(\cdot) \vee \rho_{n,t}}$, using Bernstein's inequality and a $\varepsilon$-net argument. Combining this with the above empirical Bayes score estimation bound (Lemma 1) and the score approximation rate (Theorem 1), we obtain a neural network-based score estimation rate of $\tilde{\mathcal{O}}(n^{-1}t_0^{-d/2})$ (see Theorem 2). This approach enables us to avoid the need for the density lower bound assumption (see Appendix E.1 for more details). Finally, applying Girsanov's theorem [15, 17] yields a $\tilde{\mathcal{O}}(n^{-1/2}t_0^{-d/4})$ bound in total variation (TV) distance for the distribution estimation error at the early-stopping time $t_0$ (see Theorem 3). Moreover, if the target density belongs to a Sobolev or Besov class, controlling the truncation error at $t_0$ allows us to achieve nearly minimax optimal rates up to a logarithmic factor.

To summarize, we remove the Lipschitz score assumption used in [30] and establish minimax optimal rates for the empirical score function under a sub-Gaussian assumption. We derive score approximation and estimation error bounds without the density lower bound condition as used in [22, 26, 27] and show that neural network-based SGMs can achieve nearly minimax optimality in TV distance, even under mild regularity assumptions.

The remainder of the paper is organized as follows. Section 2 introduces the notation and definitions used throughout the paper, as well as the background of SGMs. Section 3 presents our main results concerning the error bounds for score estimation and approximation as well as distribution estimation in total variation distance for SGMs. Section 4 offers proof sketches for the score estimation and approximation errors. Finally, we conclude in Section 5. We defer all proofs to the appendix.

## 2 Preliminaries and Background

### 2.1 Notations and definitions

We use $\mathbb{R}_+ := \{x \in \mathbb{R} | x \geq 0\}$ to denote the space of non-negative real values. Denote by $\mathbb{N} := \{0, 1, 2, \dots\}$ the set of natural numbers and $\mathbb{N}_+ := \mathbb{N} \setminus 0$. We denote by $\mathcal{N}(0, \sigma^2 \boldsymbol{I}_d)$ the Gaussian distribution with mean vector 0 and covariance matrix $\sigma^2 \boldsymbol{I}_d$ and write $\varphi_\sigma$ for its density. The standard Gaussian distribution on $\mathbb{R}^d$ is represented by $\gamma_d := \mathcal{N}(0, \boldsymbol{I}_d)$. For any function or distribution on $\Omega$, $\text{supp}(\Omega)$ denotes its support. Let $(\boldsymbol{X}_t)_{[0,T]}$ be a process with $\text{law}(\boldsymbol{X}_t) = P_t$ and corresponding density $p_t$. We refer to $(P_0, p_0)$ as the target data distribution and density. We write $a \vee b := \max\{a, b\}$ and $a \wedge b := \min\{a, b\}$. The notation $a = O(b)$ means $a \leq Cb$ for a universal constant $C > 0$ and we use $\tilde{O}(\cdot)$ to hide logarithmic factors. Throughout, $\lesssim$ suppress constants that depend on the dimension $d$.

**Definition 1** (Sub-Gaussian Distribution [40]). *We say a probability distribution $P$ on $\mathbb{R}^d$ is $\alpha$-sub-Gaussian for some $0 < \alpha < \infty$ if for all $\boldsymbol{\theta} \in \mathbb{R}^d$:*

$$\mathbb{E}_{\boldsymbol{X} \sim P}\left[\exp\left(\theta^\top(\boldsymbol{X} - \mathbb{E}_{\boldsymbol{X} \sim P}[\boldsymbol{X}])\right)\right] \leq \exp\left(\alpha^2 \|\boldsymbol{\theta}\|_2^2 / 2\right). \tag{1}$$

**Deep ReLU neural networks.** We follow the notation used in [38] for ReLU neural networks, please refer to Appendix D.1 for a more detailed introduction. We say that a neural network (architecture) with width $N$ and depth $L$ if the maximum width of any hidden layer in the network is at most $N$, and the total number of hidden layers does not exceed $L$.

### 2.2 Score-based generative models

In this section, we introduce the background of SGMs. A SGM typically encompasses two Markov processes: a forward process $(\boldsymbol{X}_t)_{[0,T]}$ that starts from the target distribution $\boldsymbol{X}_0 \sim P_0$, the model gradually adds noise to transform the signal into noise $\boldsymbol{X}_0 \to \boldsymbol{X}_1 \to \cdots \to \boldsymbol{X}_T \sim P_T$ and a reverse process $\boldsymbol{Y}_t := \boldsymbol{X}_{T-t}, 0 \leq t \leq T$ starts with the noise $\boldsymbol{Y}_0 \sim P_T$, and reverse the forward process to recover the signal from noise $\boldsymbol{Y}_0 \to \boldsymbol{Y}_1 \to \cdots \to \boldsymbol{Y}_T \sim P_0$.

**OU process.** We consider the following Ornstein–Ulhenbeck (OU) process as the forward process:

$$\mathrm{d}\boldsymbol{X}_t = -\boldsymbol{X}_t \mathrm{d}t + \sqrt{2}\mathrm{d}\boldsymbol{B}_t \quad (0 \leq t \leq T), \qquad \boldsymbol{X}_0 \sim P_0, \tag{2}$$

where $\boldsymbol{B}_t$ denotes a $d$-dimensional standard Brownian motion and we have $\boldsymbol{X}_t = e^{-t}\boldsymbol{X}_0 + \sqrt{1 - e^{-2t}}\boldsymbol{Z}$, with $\boldsymbol{Z} \sim \gamma_d$. The OU process is well-defined and has a reverse process $(\boldsymbol{Y}_t)_{[0,T]}$:

$$\mathrm{d}\boldsymbol{Y}_t = (\boldsymbol{Y}_t + 2\nabla \log p_{T-t}(\boldsymbol{Y}_t)) \, \mathrm{d}t + \sqrt{2}\mathrm{d}\boldsymbol{B}_t' \quad (0 \leq t \leq T), \qquad \boldsymbol{Y}_0 \sim P_T, \tag{3}$$

where $\boldsymbol{B}_t'$ denotes another $d$-dimensional standard Brownian motion and $\nabla \log p_t(\cdot)$ is called the score function. The noise distribution and score function are unknown. Using the fact that the OU

process Eq. (2) converges to standard Gaussian distribution $\gamma_d$ exponentially [16, 17], for sufficiently large $T$, we can replace $P_T$ by $\gamma_d$.

**Score matching.** Given a finite set of samples, we can train a neural network $\phi_{\text{score}}(\cdot, t)$ approximate $\nabla \log p_t(\cdot)$ for $t \in [0, T]$ by minimizing the score matching loss [7]:

$$\mathcal{L}_{\text{SM}}(\phi_{\text{score}}) := \int_{t=0}^{T} \mathbb{E}_{\boldsymbol{X}_t} \left[ \| \phi_{\text{score}}(\boldsymbol{X}_t, t) - \nabla \log p_t(\boldsymbol{X}_t) \|_2^2 \right] \mathrm{d}t, \tag{4}$$

which is equivalent to the *denoising score matching* $\mathbb{E}_{\boldsymbol{X}_0}[\ell(\phi_{\text{score}}, \boldsymbol{X}_0)]$ up to a constant [7, 8], where

$$\ell(\phi_{\text{score}}, \boldsymbol{X}_0) := \int_{t=0}^{T} \mathbb{E}_{\boldsymbol{X}_t | \boldsymbol{X}_0} \left[ \| \phi_{\text{score}}(\boldsymbol{X}_t, t) - \nabla \log p_t(\boldsymbol{X}_t | \boldsymbol{X}_0) \|_2^2 \right] \mathrm{d}t. \tag{5}$$

We replace $\nabla \log p_{T-t}(x)$ by $\phi_\theta(x, T-t)$ and obtain the *score-based process:*

$$\mathrm{d}\widehat{\boldsymbol{Y}}_t = \left( \widehat{\boldsymbol{Y}}_t + 2\phi_\theta(\widehat{\boldsymbol{Y}}_t, T-t) \right) \mathrm{d}t + \sqrt{2} \mathrm{d}\boldsymbol{B}_t' \quad (0 \le t \le T), \qquad \widehat{\boldsymbol{Y}}_0^{\gamma_d} \sim P_T. \tag{6}$$

We replace $P_T$ by $\gamma_d$ in Eq. (6) and obtain a reverse process $(\widehat{\boldsymbol{Y}}_t^{\gamma_d})_{[0,T]}$ that starting from $\widehat{\boldsymbol{Y}}_0^{\gamma_d} \sim \gamma_d$.

**Early stopping.** Instead of running Eq. (6) back to the start time $t = 0$, we stop early at a small time $t_0 > 0$. Hence, the diffusion model will approximate $P_{t_0}$ rather than $P_0$, i.e., we want $\widehat{P}_{t_0}^{\gamma_d} \approx P_{t_0}$.

**Problem statement.** Given the ground-truth data distribution $P_0$ and a set of $n$ i.i.d observations $\{\boldsymbol{x}^{(i)}\}_{i=1}^n \sim P_0^{\otimes n}$, we learn the score function $\nabla \log p_t, \forall t \in [t_0, T]$ via the empirical risk minimizer:

$$\widehat{\phi} \in \arg\min_{\phi \in \mathcal{NN}} \frac{1}{n} \sum_{i=1}^{n} \ell(\phi, \boldsymbol{x}^{(i)}), \tag{7}$$

and plugin $\widehat{\phi}$ to the process Eq. (6) to generate new samples $\boldsymbol{X}' \sim \widehat{P}_{t_0}^{\gamma_d}$, where $\widehat{P}_{t_0}^{\gamma_d}$ is the modeled distribution of SGMs. Our goal is to study the estimation error in total variation distance $\mathsf{TV}(P_0, \widehat{P}_{t_0}^{\gamma_d})$.

## 3 Main Results

In this section, we present our main results regarding the error bounds for score estimation and approximation and establish the nearly optimal convergence rates for diffusion models.

### 3.1 Assumptions

Here, we outline the assumptions imposed on the target data distribution $P_0$ and density function $p_0$ in our analysis. In particular, our score approximation and estimation results are not specified to Assumptions 2 and 3, where they are used only in Corollaries 1 and 2 to control errors induced by early stopping, respectively. We follow [29, 22] for the definitions of the Sobolev ball and the Besov space, respectively, which are deferred to Appendix G due to space limitations.

**Assumption 1** (Sub-Gaussian Distribution). *The target data distribution $P_0$ is $\alpha$-sub-Gaussian.*

**Assumption 2** (Sobolev Class of Density). *The density $p_0$ belongs to the Sobolev ball with $0 < s \le 2$.*

**Assumption 3** (Besov Class of Density). *The density $p_0 \in L^q([0,1]^d) \cap U(B_{q,q'}^s([0,1]^d); C)$ for some $C > 0$, where $1 \le q \le \infty, 0 < q' \le \infty$, and $0 < s \le 2$.*

### 3.2 Score estimation by regularized empirical score functions

Given a set of $n$ i.i.d. observations $\{\boldsymbol{x}^{(i)}\}_{i=1}^n$ drawn from an unknown target distribution $P_0$, let $\hat{P}_0^{(n)} := \frac{1}{n} \sum_{i=1}^n \delta_{\boldsymbol{x}^{(i)}}$ be the empirical measure. For OU process Eq. (2), we have $\hat{P}_t^{(n)} = \frac{1}{n} \sum_{i=1}^n \mathcal{N}(e^{-t}\boldsymbol{x}^{(i)}, (1 - e^{-2t})\boldsymbol{I}_d))$ and $\hat{p}_t(\cdot) = \frac{1}{n} \sum_{i=}^n \varphi_{\sigma_t}(\cdot - m_t \boldsymbol{x}^{(i)})$, where $m_t := \exp(-t), \sigma_t := \sqrt{1 - \exp(-2t)}$. [30] showed that for an $\alpha$-sub-Gaussian distribution $P_0$ with $L$-Lipschitz continuous scores, the following regularized empirical score function, with bandwidth $h = \left( \frac{d^3(\alpha^2 \log n)^{d/2}}{L^2 n} \right)^{2/(d+4)}$ and regularizer $\rho_n = (2\pi h)^{-d/2} e^{-1} n^{-2}$:

$$\frac{\nabla \hat{p}_h(\cdot)}{\hat{p}_h(\cdot) \vee \rho_n} = \frac{\nabla \left( \frac{1}{n} \sum_{i=1}^n \varphi_{\sqrt{h}}(\cdot - \boldsymbol{x}^{(i)}) \right)}{\frac{1}{n} \sum_{i=1}^n \varphi_{\sqrt{h}}(\cdot - \boldsymbol{x}^{(i)}) \vee \rho_n} \tag{8}$$

achieves a nearly minimax optimal rate of $\mathcal{O}\big(d\alpha^2 L^2 n^{\frac{-2}{d+4}} \log^{\frac{d}{d+4}} n\big)$ score estimation error in term of score matching loss. However, the Lipschitz continuous score assumption excludes many distributions of interest, such as those supported on a submanifold and the Lipschitz constant can conceal additional dependence on the data dimension in some cases, especially when the data are approximately supported on a submanifold. On the other hand, [29] demonstrates that a similar truncated score estimator $\frac{\nabla \tilde{p}_t(\cdot)}{\hat{p}_t(\cdot)} \mathbb{1}_{\{\hat{p}_t(\cdot) > \rho_n\}}$ attains a minimax optimal rate of $\tilde{\mathcal{O}}(n^{-1} \sigma_t^{-d-2}(\sigma_t^d \vee 1))$ under merely sub-Gaussian distribution, indicating that regularity conditions on the score are not necessary to derive optimal rates. We resolve this question with the following lemma, demonstrating that the estimator in Eq. (8) can achieve minimax optimal rates as long as the data distribution is sub-Gaussian:

**Lemma 1.** *For any $d \geq 1, n \geq 3$, Let $P$ be a $\alpha$-sub-Gaussian distribution on $\mathbb{R}^d$ and $\hat{P}$ be its empirical distribution associated to a sample $\{\boldsymbol{x}^{(i)}\}_{i=1}^n$. For any $\sigma \gtrsim \alpha n^{-1/d} \log^{1/2} n$, let $P_\sigma = P * \mathcal{N}(0, \sigma^2 \boldsymbol{I}_d), \hat{P}_\sigma^{(n)} = \hat{P}^{(n)} * \mathcal{N}(0, \sigma^2 \boldsymbol{I}_d)$ with density functions $p_\sigma, \hat{p}_\sigma : \mathbb{R}^d \to \mathbb{R}_+$. Fix $0 < \rho_n \leq (2\pi\sigma^2)^{-d/2} e^{-1} n^{-1}$, then we have*

$$\mathbb{E}_{\{\boldsymbol{x}^{(i)}\}_{i=1}^n}\Big[\int_{\mathbb{R}^d} \Big\|\frac{\nabla p_\sigma(\boldsymbol{x})}{p_\sigma(\boldsymbol{x})} - \frac{\nabla \hat{p}_\sigma(\boldsymbol{x})}{\hat{p}_\sigma(\boldsymbol{x}) \vee \rho_n}\Big\|_2^2 p_\sigma(\boldsymbol{x})\mathrm{d}\boldsymbol{x}\Big] \lesssim \sigma^{-d-2}\big(\sigma^d + \alpha^d\big) \log^3\Big(\frac{(2\pi\sigma^2)^{-\frac{d}{2}}}{\rho_n}\Big)\frac{\log^{d/2} n}{n}.$$

We provide a proof sketch for Lemma 1 in Section 4.1 and defer the detailed proof to Appendix C.2.

**Remark 1.** *To establish our neural network approximation and estimation bounds, we adopt the empirical regularized score estimator rather than the truncated estimator as our surrogate. The regularized estimator is globally smooth and stable, ensuring compatibility with established approximation theories for smooth functions, while the truncated estimator's discontinuities at low-density regions violate these assumptions and complicate both approximation and generalization analyses.*

### 3.3 Score approximation and estimation by deep neural networks

#### 3.3.1 Neural network score approximation

Lemma 12 shows that the regularized empirical score function Eq. (8) can be well approximated by a ReLU DNN in $L_2$-distance. Combining Lemmas 1 and 12, we obtain the neural network score approximation error by the following theorem. For the proof, please refer to Appendix D.4.

**Theorem 1** (Neural Network Score Approximation for Sub-Gaussian Distributions). *Suppose that $P_0$ satisfies Assumption 1. For any $1 \leq d \lesssim \sqrt{\log n}, n \geq 3$ and any $\frac{1}{2}\alpha^2 n^{-2/d} \log n < t_0 \leq 1$ and $T = n^{\mathcal{O}(1)}$, let $\{\boldsymbol{x}_t\}_{t \in [t_0, T]}$ be the solutions of the process Eq. (2) with density function $p_t : \mathbb{R}^d \to \mathbb{R}_+$. Fix $k \in \mathbb{N}_+$ with $d/2 \leq k \lesssim \frac{\log n}{\log \log n}$. Then, there exists a ReLU DNN $\phi_{score}$ with width $\leq \mathcal{O}\big(n^{\frac{3}{2k}} \log_2 n\big)$ and depth $\leq \mathcal{O}\big(\log^2 n\big)$ constructed from i.i.d. samples $\{\boldsymbol{x}^{(i)}\}_{i=1}^n$ such that*

$$\mathbb{E}_{\{\boldsymbol{x}^{(i)}\}_{i=1}^n}\Big[\mathbb{E}_{\boldsymbol{x}_t \sim P_t}\big[\|\nabla \log p_t(\boldsymbol{x}_t) - \phi_{score}(\boldsymbol{x}_t, t)\|_2^2\big]\Big] \lesssim \sigma_t^{-d-2}\big(\sigma_t^d + \alpha^d\big)\frac{\log^{d/2+3} n}{n},$$

*and we have $\|\phi_{score}(\cdot, t)\|_\infty \lesssim \sigma_t^{-1}\sqrt{\log n}$. Moreover, let $T = n^{\mathcal{O}(1)}$, we have*

$$\mathbb{E}_{\{\boldsymbol{x}^{(i)}\}_{i=1}^n}\Big[\int_{t=t_0}^T \mathbb{E}_{\boldsymbol{x}_t \sim P_t}\big[\|\nabla \log p_t(\boldsymbol{x}_t) - \phi_{score}(\boldsymbol{x}_t, t)\|_2^2\big]\mathrm{d}t\Big] \lesssim \alpha^d t_0^{-d/2} n^{-1} \log^{\frac{d}{2}+4} n.$$

#### 3.3.2 Neural network score estimation

According to Theorem 1, $\widehat{\phi}(\boldsymbol{x}, t)$ can be taken so that $\|\widehat{s}(\cdot, t)\|_\infty \lesssim \sigma_t^{-1}\sqrt{\log n}$. Hence, we limit the neural network class of Theorem 1 into

$$\mathcal{NN} := \big\{\phi \in \mathcal{NN}(\text{width} \leq \mathcal{O}(n^{\frac{3}{2k}} \log_2 n); \text{depth} \leq \mathcal{O}(\log^2 n)) \mid \|\phi(\cdot, t)\|_\infty \lesssim \frac{\sqrt{\log n}}{\sigma_t}\big\}. \quad (9)$$

Together with the score approximation error bound from Theorem 1, applying Bernstein's inequality and an $\epsilon$-net argument, we obtain the following neural network score estimation error bound in terms of score matching loss. A proof sketch is provided in Section 4.3 and see Appendix E.1 for details.

**Theorem 2** (Neural Network Score Estimation for Sub-Gaussian Distributions). *Assume that the conditions of Theorem 1 hold. For $3 \leq d \lesssim \sqrt{\log n}$, fix $k \in \mathbb{N}_+$ with $\frac{6 \log n}{(d-2)\log(t_0^{-1})} \vee d/2 \leq k \lesssim \frac{\log n}{\log \log n}$ in Eq. (9). Then, for any $\delta \in (0,1)$, with probability at least $1 - \delta$, the excess risk of an empirical risk minimizer Eq. (7) over the neural network class $\mathcal{NN}$ satisfies that*

$$\int_{t_0}^T \mathbb{E}_{\boldsymbol{X}_t}\big[\|\widehat{\phi}(\boldsymbol{X}_t, t) - \nabla \log p_t(\boldsymbol{X}_t)\|_2^2\big]\mathrm{d}t \lesssim t_0^{-d/2} n^{-1} \log^{\frac{d}{2}+4} n + t_0^{-1} n^{-1} \log n \cdot \log \frac{2}{\delta}.$$

Our rate in Theorem 2 matches the rate for regularized kernel-based estimators in Lemma 1 and aligns with the nearly minimax optimal rate derived in [29] for a similar truncated estimator.

### 3.4 Distribution estimation errors of SGMs for sub-Gaussian distributions

Here, we evaluate the distribution estimation error of neural network-based SGMs in TV distance. The following theorem aims to bound the TV distance between the true marginal distribution $P_{t_0}$ at time $t_0 \geq \alpha^2 n^{-2/d} \log n$ and the learned distribution $\widehat{P}_{t_0}^{\gamma_d}$ by SGMs. Proof refer to Appendix E.2.

**Theorem 3** (Distribution Estimation Error of $P_{t_0}$). *Assume that the conditions of Theorem 2 hold. Then, for any $\delta \in (0,1)$, with probability at least $1 - \delta$,*

$$\mathbb{E}_{\{\boldsymbol{x}^{(i)}\}_{i=1}^n}\left[\mathsf{TV}(P_{t_0}, \widehat{P}_{t_0}^{\gamma_d})\right] \lesssim \alpha^{d/2} t_0^{-d/4} n^{-1/2} \log^{\frac{d}{4}+2} n + t_0^{-1/2} n^{-1/2} \log^{1/2} n \cdot \sqrt{\log(2/\delta)}.$$

Somewhat unexpectedly, by choosing $\delta = 1/n$, Theorem 3 shows that with only a sub-Gaussian assumption, neural network-based SGMs achieve a TV distance bound of $\tilde{\mathcal{O}}(\alpha^{d/2} t_0^{-d/4} n^{-1/2})$ for the distribution estimation error at the early stopping time. By triangle inequality, $\mathsf{TV}(P_0, \widehat{P}_{t_0}) \leq \mathsf{TV}(P_0, P_{t_0}) + \mathsf{TV}(P_{t_0}, \widehat{P}_{t_0})$, indicating that to bound $\mathsf{TV}(P_0, \widehat{P}_{t_0})$, it is necessary to control the error introduced by stopping early. To this end, we introduce nonparametric class assumptions.

**Bounding the early stopping induced error.** Assume that the target density function $p_0$ belongs to Sobolev space $\mathcal{W}_2^s(\mathbb{R}^d)$ (Assumption 2), or Besov space $B_{p,q}^s([0,1]^d)$ (Assumption 3), with $t_0 = n^{-\frac{2}{d+2s}}$, it can be upper-bounded by Lemma 26 and Theorem 9, respectively:

$$\mathsf{TV}(P_0, P_{t_0}) \lesssim n^{-\frac{s}{d+2s}}. \tag{10}$$

Theorem 3 and Lemma 26 immediately imply Corollary 1:

**Corollary 1** (Distribution Estimation Error for Sobolev Class of Density). *Assume Assumptions 1 and 2 hold. Let $t_0 = n^{-\frac{2}{d+2s}}, T = n^{\mathcal{O}(1)}$. Then, with probability at least $1 - 1/n$, it holds that*

$$\mathbb{E}_{\{\boldsymbol{x}^{(i)}\}_{i=1}^n}\left[\mathsf{TV}(P_0, \widehat{P}_0^{\gamma_d})\right] \lesssim \mathrm{polylog}(n) n^{-\frac{s}{d+2s}}.$$

Notice that $P_0$ on $[0,1]^d$ is $\sqrt{d}$-sub-Gaussian, Corollary 2 immediately follows by Theorems 3 and 9:

**Corollary 2** (Distribution Estimation Error for Besov Class of Density). *Assume that Assumption 3 holds. Let $t_0 = n^{-\frac{2}{d+2s}}, T = n^{\mathcal{O}(1)}$. Then, with probability at least $1 - 1/n$, it holds that*

$$\mathbb{E}_{\{\boldsymbol{x}^{(i)}\}_{i=1}^n}\left[\mathsf{TV}(P_0, \widehat{P}_0^{\gamma_d})\right] \lesssim n^{-\frac{s}{d+2s}}.$$

Therefore, we have proved that neural network-based SGMs achieve minimax estimation rates for Sobolev and Besov class densities in TV distance up to logarithmic factors, even without the Lipschitz score assumption (as opposed to [30]) and the lower bounded density assumption (as opposed to [22, 26, 27]).

**Remark 2** (Dependence of Network Width on Dimension $d$ and the Bias-Variance Trade-Off). *The $\tilde{\mathcal{O}}(n^{3/d})$ network width appearing in Theorems 1 and 2 suggests that the required width decreases with dimension $d$, which may seem counter-intuitive. This is a direct consequence of the interplay between the smoothness of $P_{t_0}$ and the network architecture in our analysis, which has a clear theoretical motivation: (1) **Higher dimensions force more smoothing**. Our condition on the early stopping time ($t_0 \geq \tilde{\mathcal{O}}(n^{-2/d})$ in Theorems 1 and 2 or $t_0 = n^{-\frac{2}{d+2s}}$ in Corollaries 1 and 2) forces larger $t_0$ for higher $d$ to ensure theoretical guarantees. (2) **More smoothing simplifies the learning of $P_{t_0}$**. A larger $t_0$ means that $P_{t_0}$ is convolved with a Gaussian of higher variance. This makes $P_{t_0}$ inherently smoother and less complex. Its score can thus be approximated by a network whose size scales less severely with the sample size $n$. Hence, while higher dimension usually implies increased statistical difficulty (as seen in the convergence rate $n^{-\frac{s}{d+2s}}$ suffering from the curse of dimensionality), the necessity of stronger smoothing to achieve uniform control in high dimensions makes the intermediate learning problem architecturally less demanding in terms of network size. This is a reflection of the **bias–variance trade-off**: we reduce variance (by smoothing), but incur bias, which is ultimately what limits the rate.*

# 4 Proof Sketch

## 4.1 Proof sketch of Lemma 1

We use the following two ingredients to prove Lemma 1:

**(1) From score matching to Hellinger distance via empirical Bayes smoothing.** The following lemma shows that the score matching loss can be upper-bounded by the Hellinger distance $\mathsf{H}^2(P_t, \hat{P}_t^{(n)})$ plus the $L_2$-norm of the score over low-density regions:

**Lemma 2.** *Given any distributions $P, Q$ on $\mathbb{R}^d$ with density functions $p, q : \mathbb{R}^d \to \mathbb{R}_+$, respectively. Fix $\sigma > 0$, let $P_\sigma = P * \mathcal{N}(0, \sigma^2 \mathbf{I}_d), Q_\sigma = Q * \mathcal{N}(0, \sigma^2 \mathbf{I}_d)$ with density functions $p_\sigma, q_\sigma : \mathbb{R}^d \to \mathbb{R}_+$. For all $d \geq 1, n \geq 1$, let $0 < \rho_n \leq (2\pi\sigma^2)^{-d/2} e^{-1/2}$ and let $\mathcal{G} := \{x \in \mathbb{R}^d : p_\sigma(\boldsymbol{x}) \leq \rho_n\}$, then there exists a universal constant $C > 0$ such that*

$$\int_{\mathbb{R}^d} \Big\| \frac{\nabla p_\sigma(\boldsymbol{x})}{p_\sigma(\boldsymbol{x})} - \frac{\nabla q_\sigma(\boldsymbol{x})}{q_\sigma(\boldsymbol{x}) \vee \rho_n} \Big\|_2^2 p_\sigma(\boldsymbol{x}) \mathrm{d}x$$
$$\leq C \Big( \frac{d}{\sigma^2} \max \Big\{ \log^3 \Big( \frac{(2\pi\sigma^2)^{-d/2}}{\rho_n} \Big), \log\big(\mathsf{H}^{-2}(P_\sigma, Q_\sigma)\big) \Big\} \mathsf{H}^2(P_\sigma, Q_\sigma) + \int_{\mathcal{G}} \Big\| \frac{\nabla p_\sigma(\boldsymbol{x})}{p_\sigma(\boldsymbol{x})} \Big\|_2^2 p_\sigma(\boldsymbol{x}) \mathrm{d}x \Big).$$

For the proof, please refer to Appendix A.1. The second term in Lemma 2 is the $L_2$-norm of the score function over the low-density region $\{x \in \mathbb{R}^d : p_\sigma(\boldsymbol{x}) \leq \rho_n\}$. By Assumption 1, $P$ is $\alpha$-sub-Gaussian and once convolved with Gaussian noise, the resulting marginal $P_\sigma$ will become $\sqrt{\alpha^2 + \sigma^2}$-sub-Gaussian. Using Lemma 5 to bound the $L_2$-norm of the score function and leveraging the sub-Gaussian property of $P_\sigma$, we find that this term is bounded above by $\mathcal{O}(\sigma^{-2} \rho_n \log(\sigma^{-d} \rho_n^{-1}))$ (see Lemma 7). For the first term, recall the fact that $\mathsf{H}^2(\hat{P}_t^{(n)}, P_t) \leq \mathrm{KL}(\hat{P}_t^{(n)} \| P_t)$ (see Lemma 32).

**Remark 3.** *Lemma 2 does not require the density functions to exist for $P$ and $Q$.*

**Remark 4.** *Notably, [30, Lemma 1] employs a rescaling argument to adjust the random variables and apply [41, Theorem E.1], yielding a bound similar to Lemma 2. However, a careful examination reveals that a term $h^{-d/2}$ has been missing in their proof. Simply following their strategy results in a bound that scales as $\sigma^{-d-2}$ instead of $\sigma^{-2}$. On the contrary, we employ the rescaling argument to get a generalization result (see Lemma 4) of [41, Lemma F.2], and adopt a similar proof strategy of [41, Lemma E.1] to derive a bound that scales as $\sigma^{-2}$.*

**(2) KL-divergence rate of smoothed empirical distribution.** The following lemma is a restatement of [42, Theorem 3], in which we explicitly demonstrate the dependence of the bound on the Gaussian parameter $\sigma$. In particular, following the original proof of [42], one obtains a bound that scales *exponentially* with $\frac{1}{\sigma}$, i.e, $\mathbb{E}\big[\mathrm{KL}(\hat{P}_\sigma \| P_\sigma^*)\big] \leq \mathcal{O}(\exp(\frac{1}{\sigma}) \frac{\log^d n}{n})$, which blows up when $\sigma$ is sufficiently small. We refine their proof and establish a rate that scales *polynomially* in $\frac{1}{\sigma}$, as given below:

**Lemma 3** (Convergence Rate of Smoothed Empirical Sub-Gaussian Distributions)**.** *Given $d \geq 1, n \geq 1, \sigma > 0$. Suppose that $P$ is a $d$-dimensional $\alpha$-sub-Gaussian distribution. Let $P_\sigma = P * \mathcal{N}(0, \sigma^2 \mathbf{I}_d)$ and $\hat{P}$ be the empirical measure of an i.i.d. sample of size $n$ drawn from $P$ and $\hat{P}_\sigma = \hat{P} * \mathcal{N}(0, \sigma^2 \mathbf{I}_d)$. Then we have*

$$\mathbb{E}_{P^{\otimes n}} \Big[ \mathrm{KL}(\hat{P}_\sigma \| P_\sigma) \Big] \leq C_d \Big( \frac{\alpha}{\sigma} \Big)^d \frac{\log^{d/2} n}{n}. \tag{11}$$

For the proof, please refer to Appendix B.2. To upper bound the first term in Lemma 2, notice that $x \mapsto x \log x^{-1}$ is concave and is increasing in $(0, e^{-1})$, by Jensen's inequality we have

$$\mathbb{E}_{P^{\otimes n}} \Big[ \log\big(\mathsf{H}^{-2}(P_\sigma, \hat{P}_\sigma^{(n)})\big) \mathsf{H}^2(P_\sigma, \hat{P}_\sigma^{(n)}) \Big] \leq \log \Big( \frac{1}{\mathbb{E}_{P^{\otimes n}}[\mathsf{H}^2(P_\sigma, \hat{P}_\sigma^{(n)})]} \Big) \mathbb{E}_{P^{\otimes n}}[\mathsf{H}^2(P_\sigma, \hat{P}_\sigma^{(n)})]$$
$$\lesssim \log \Big( \Big(\frac{\alpha}{\sigma}\Big)^d \frac{\log^{d/2} n}{n} \Big) \Big(\frac{\alpha}{\sigma}\Big)^d \frac{\log^{d/2} n}{n}. \tag{12}$$

where the last inequality holds by letting $C_d \alpha^d \sigma^{-d} n^{-1} \log^{d/2} n \leq e^{-1}$, which can be satisfied when $(\alpha/\sigma)^d \lesssim n \log^{-d/2} n$, i.e., $\sigma \gtrsim \alpha n^{-1/d} \log^{1/2} n$. Combining the sub-Gaussian tail Lemma 7 an Eq. (12) we complete the proof for Lemma 2. Please refer to Appendix A.1 for a more detailed proof.

## 4.2 Proof sketch of Theorem 1

The approximation rates of DNNs for smooth functions have been extensively investigated in the literature [35–39, 43]. At first glance, one might try to construct a sub-network for each exponential component of the KDE, leveraging these existing results. However, such an approach would require $n$ sub-networks, causing the overall network size to scale at least linearly with $n$. In contrast, our proof constructs a more compact DNN architecture, which not only prevents the DNN size from blowing up with $n$ but also yields an optimal estimation error bound as shown in Theorem 2. Moreover, unlike [22], our approximation results do not require the density lower bound assumption. Recall that the regularized empirical score function with regularizer $\rho_n = (2\pi\sigma^2)^{-d/2}e^{-1}n^{-1}$ for the OU process Eq. (2) can be expressed as

$$\frac{\nabla \hat{p}_t(\boldsymbol{y})}{\hat{p}_t(\boldsymbol{y}) \vee \rho_{n,t}} = \frac{1}{\sigma_t^2} \frac{m_t \times f_{\text{kde}}^{(3)}(\boldsymbol{y},t) - \boldsymbol{y} \times f_{\text{kde}}^{(2)}(\boldsymbol{y},t)}{f_{\text{kde}}^{(1)}(\boldsymbol{y},t) \vee e^{-1}n^{-1}},$$

where we denote by

$$f_{\text{kde}}^{(1)}(\boldsymbol{y},t), f_{\text{kde}}^{(2)}(\boldsymbol{y},t) := \frac{1}{n}\sum_{i=1}^{n} \exp\left(-\frac{\|\boldsymbol{y}-m_t\boldsymbol{x}^{(i)}\|_2^2}{2\sigma_t^2}\right), f_{\text{kde}}^{(3)}(\boldsymbol{y},t) := \frac{1}{n}\sum_{i=1}^{n} \exp\left(-\frac{\|\boldsymbol{y}-m_t\boldsymbol{x}^{(i)}\|_2^2}{2\sigma_t^2}\right)\boldsymbol{x}^{(i)}.$$

Thus, we conduct two steps to construct a ReLU DNN to approximate $\frac{\nabla \hat{p}_t(\boldsymbol{y})}{\hat{p}_t(\boldsymbol{y}) \vee \rho_{n,t}}$:

**Approximating Gaussian kernel density estimators (Lemma 8).** Let $h^{(i)} := \frac{\|\boldsymbol{y}-m_t\boldsymbol{x}^{(i)}\|_2^2}{2\sigma_t^2}$ and $\tilde{h}(\boldsymbol{y},t) := \left(\frac{1}{n}\sum_{i=1}^{n}\left(h^{(i)}(\boldsymbol{y},t)\right)^s\right)^{1/s}$ for some $s \in \mathbb{N}_+$. Whenever $h^{(i)} \gtrsim \tilde{C}$, where $\tilde{C} = \mathcal{O}(\sqrt{s\log(\epsilon^{-1})})$ for some $0 < \epsilon < 1$, the term $\exp(-h^{(i)})$ becomes small enough to be trivially approximated by a DNN. Consequently, we only need to approximate $f_{\text{kde}}$ for $h \in [0, \tilde{C}]$. We further divide $[0, \tilde{C}]$ into $K$ subintervals. For each $\beta \in \{0, 1, \ldots, K-1\}$, define

$$Q_\beta := \left\{h \in \mathbb{R} : h \in \left[\frac{\beta\tilde{C}}{K}, \frac{(\beta+1)\tilde{C}}{K} - \delta \cdot \mathbb{1}_{\{\beta \le K-2\}}\right]\right\}.$$

For each $\beta$, we define $\tilde{h}_\beta := \frac{\beta\tilde{C}}{K}, \beta \in \{0, 1, \ldots, K-1\}$ as the vertex of $Q_\beta$. The Taylor expansion of the Gaussian kernel density estimator at $\tilde{h}_\beta$ up to order $s-1$ is given by

$$f_{\text{kde}}(\boldsymbol{y},t) = \sum_{k=0}^{s-1} \frac{(-1)^k \exp(-\tilde{h}_\beta)}{k!} \frac{1}{n}\sum_{i=1}^{n}\left(h^{(i)} - \tilde{h}_\beta\right)^k + \frac{1}{n}\sum_{i=1}^{n} \frac{(-1)^s \exp(-\theta^{(i)})}{s!}\left(h^{(i)} - \tilde{h}_\beta\right)^s := \text{(a)} + \text{(b)},$$

The second term (b) can be upper-bounded by $\frac{1}{s!}|\tilde{h} - \tilde{h}_\beta|^s$ (c.f. Eq. (62)). We can approximate this term by constructing a sub-network to learn $\tilde{h}_\beta$ such that $|\tilde{h} - \tilde{h}_\beta\| \lesssim \epsilon^s$ for some $0 < \epsilon < 1$ (c.f. Proposition 4). For the first term (a), some elementary calculations lead to

$$\text{(a)} = \exp(-\tilde{h}_\beta)\sum_{k=0}^{s-1}\sum_{\|\tilde{\boldsymbol{\nu}}\|_1+a_4=k} \frac{(-2)^{-(k-\|\boldsymbol{\nu}_2\|_1-a_4)}m_t^{\|\boldsymbol{\nu}_2\|_1+2\|\boldsymbol{\nu}_3\|_1}}{\sigma_t^{2(k-a_4)}\boldsymbol{\nu}_1!\boldsymbol{\nu}_2!\boldsymbol{\nu}_3!a_4!}\tilde{h}_\beta^{a_4}\boldsymbol{y}^{2\boldsymbol{\nu}_1+\boldsymbol{\nu}_2}\left(\frac{1}{n}\sum_{i=1}^{n}(\boldsymbol{x}^{(i)})^{\boldsymbol{\nu}_2+2\boldsymbol{\nu}_3}\right).$$

we construct each sub-network to approximate $\sigma_t^{-2k}$ by $\phi_{1/\sigma^2}^k$ (c.f. Proposition 2), $\boldsymbol{y}^{\boldsymbol{\nu}}$ by $\phi_{\text{poly}}^{\boldsymbol{\nu}}$ (c.f. Proposition 3), $\tilde{h}(\boldsymbol{y},t)$ by $\phi_{\tilde{h}}$ (c.f. Proposition 4), $\tilde{h}_\beta(\boldsymbol{y},t)$ by $\phi_{\tilde{h}_\beta}$ (c.f. Proposition 5), $\tilde{h}_\beta^k(\boldsymbol{y},t)$ by $\phi_{\tilde{h}_\beta}^k$ (c.f. Proposition 6), $\exp(-\tilde{h}_\beta(\boldsymbol{y},t))$ by $\phi_{\tilde{h}_\beta}^{\exp}$ (c.f. Proposition 7). By combining the sizes and approximation errors of all the sub-networks described above, we obtain sub-networks that approximate $f_{\text{kde}}^{(1)}, f_{\text{kde}^{(2)}}$, respectively. A similar argument applies to the approximation of $f_{\text{kde}}^{(3)}$.

**Approximating regularized empirical score functions (Lemma 10).** After constructing sub-networks that approximate $f_{\text{kde}^{(1)}}, f_{\text{kde}^{(2)}}, f_{\text{kde}^{(3)}}, m_t, \sigma_t^{-2}$, we can compose them into a single ReLU DNN to approximate the regularized empirical score function. For more details, see Appendix D.3.

The approximation rate for the true score functions follows immediately by combining Lemma 1 and the above approximation rate for the regularized empirical score functions.

## 4.3 Proof sketch of Theorem 2

Denote by $\phi^*(\cdot, t) := \nabla \log p_t(\cdot)$ and $\hat{\bar{\phi}}(\cdot, t) := \frac{\nabla \hat{p}_t(\cdot)}{\hat{p}_t(\cdot) \vee \rho_{n,t}}$. For any $\phi \in \mathcal{NN}$ from Eq. (9), we have $\int_{t_0}^T \mathbb{E}_{\boldsymbol{X}_t}[\|\phi(\boldsymbol{X}_t, t) - \phi^*(\boldsymbol{X}_t, t)\|_2^2]\mathrm{d}t = \mathbb{E}_{\boldsymbol{X}_0}[\ell(\phi, \boldsymbol{X}_0) - \ell(\phi^*, \boldsymbol{X}_0)] = \mathbb{E}_{\boldsymbol{X}_0}[\mathbb{E}_{\{\boldsymbol{x}^{(i)}\}}[\ell(\phi, \boldsymbol{X}_0) - \ell(\hat{\bar{\phi}}, \boldsymbol{X}_0)]] + \mathbb{E}_{\{\boldsymbol{x}^{(i)}\}}[\mathbb{E}_{\boldsymbol{X}_0}[\ell(\hat{\bar{\phi}}, \boldsymbol{X}_0) - \ell(\phi^*, \boldsymbol{X}_0)]]$, where the second term can be controlled using the empirical Bayes score estimation result from Lemma 1. For the first term, a key observation is that the surrogate $\hat{\bar{\phi}}(\cdot, t)$ can be uniformly bounded (see Lemma 5), allowing us to verify Bernstein's condition for the random variable $\mathbb{E}_{\{\boldsymbol{x}^{(i)}\}}[\ell(\phi, \boldsymbol{X}_0) - \ell(\hat{\bar{\phi}}, \boldsymbol{X}_0)]$. This enables us to apply Bernstein's inequality together with an $\varepsilon$-net argument to obtain a uniform high-probability bound for the first term over $\mathcal{NN}$. Combining these two terms and applying them to the empirical risk minimizer $\hat{\phi}$ trained over $\mathcal{NN}$, together with the score approximation results from Theorem 1, we obtain an upper bound on $\int_{t_0}^T \mathbb{E}_{\boldsymbol{X}_t}[\|\hat{\phi}(\boldsymbol{X}_t, t) - \phi^*(\boldsymbol{X}_t, t)\|_2^2]\mathrm{d}t$. This approach allows us to establish the desired generalization bound without requiring a lower bound on the density (see Appendix E.1 for details).

# 5 Conclusion and Discussion

## 5.1 Conclusion

In this paper, we introduce a theoretical framework for studying the approximation and generalization abilities of neural network-based SGMs for estimating sub-Gaussian distributions on $\mathbb{R}^d$. Using empirical Bayes smoothing techniques and neural network approximation theory, we established nearly minimax optimal convergence rates for SGMs without requiring strong regularities, such as Lipschitz score and density lower bound assumptions.

## 5.2 Limitations and future work.

**Curse of dimensionality (CoD).** One of our limitations is that our current bounds still suffer from the curse of dimensionality (CoD). Incorporating structural assumptions, such as manifold assumption explored in [44, 45], or low-rank structures [46]) is a crucial next step to mitigate the CoD. This would require non-trivial extensions to our analysis, particularly in adapting our empirical Bayes smoothing techniques and neural network approximation theory to efficiently exploit low-dimensional geometry. In particular, our analysis relies on an isotropic Gaussian kernel for the empirical score estimator. To effectively leverage a manifold structure, this would need to be replaced with an estimator that respects the underlying geometry, such as one using anisotropic or manifold-intrinsic kernels (e.g., heat kernels), to prevent smoothing data off the manifold. A manifold-aware analysis, following, e.g., the path of [44, 45], would require establishing new bounds for score estimation and approximation that depend on the intrinsic dimension of the data, rather than the ambient dimension.

**Going beyond sub-Gaussian distributions.** Our current analysis leverages the sub-Gaussian assumption in two essential components: controlling the score function in low- density regions (Lemma 7) and bounding the KL divergence of the Gaussian-smoothed empirical distribution (Lemma 3). Relaxing this assumption to encompass broader tail behaviors, such as sub-exponential or sub-Weibull distributions [47, 48], poses both technical and conceptual challenges, requiring finer control of tail integrals and stability properties of the score. We believe that developing such extensions would substantially deepen the theoretical foundations of score-based models, bridging the gap between idealized sub-Gaussian settings and the heavy-tailed distributions often encountered in practice. In particular, leveraging recent progress on the convergence of Gaussian-smoothed empirical measures under heavy-tailed assumptions offers a promising pathway toward this goal.

**Higher-order smoothness.** Our nearly minimax optimal rates in Corollaries 1 and 2 currently apply to smoothness parameters $0 < s \le 2$. This restriction arises because our analysis employs an isotropic Gaussian kernel-based score estimator, which is a linear estimator and thus cannot fully exploit higher-order smoothness [49, 50]. Exploring nonlinear or adaptive estimators that can leverage higher-order smoothness represents another compelling direction for extending our framework.

# Acknowledgments

G. Fu is grateful to Jonathan Scarlett for insightful discussions on Lemma 3 and to Taiji Suzuki for valuable feedback on this work. We thank the anonymous reviewers for their helpful comments and suggestions, which improved the quality of this paper. This research is supported by the Ministry of Education, Singapore, under its Academic Research Fund Tier 1 (A-8001814-00-00).

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

## Broader Impacts

This work advances the theoretical understanding of score-based generative models (SGMs) by providing approximation and generalization guarantees under minimal assumptions. By removing restrictive conditions such as Lipschitz continuity or density lower bounds, our results broaden the applicability of SGMs to more complex and realistic data distributions. These insights may inform the design of more robust and sample-efficient generative models, particularly in high-dimensional or structured data settings. While primarily theoretical, our findings contribute to a deeper understanding of SGMs, which is essential for their safe and responsible use in real-world applications.


# Appendix

## Contents

## A    From Score Matching to Hellinger Distance via Empirical Bayes

The following lemma extends [41, Lemma F.2] by generalizing it from densities convolved with a standard Gaussian distribution $\mathcal{N}(\mathbf{0}, \boldsymbol{I}_d)$ to densities convolved with a general isotropic Gaussian distribution $\mathcal{N}(\mathbf{0}, \sigma^2 \boldsymbol{I}_d)$ for any $\sigma > 0$.

**Lemma 4.** *For every pair of probability distributions $P$ and $Q$ on $\mathbb{R}^d$ with density functions $p, q :$ $\mathbb{R}^d \to \mathbb{R}_+$ respectively. Let $p_\sigma(\boldsymbol{y}) := \int_{\mathbb{R}^d} \varphi_\sigma(\boldsymbol{y} - \boldsymbol{x}) p(\boldsymbol{x}) \mathrm{d}\boldsymbol{x}$ and $q_\sigma(\boldsymbol{y}) := \int_{\mathbb{R}^d} \varphi_\sigma(\boldsymbol{y} - \boldsymbol{x}) q(\boldsymbol{x}) \mathrm{d}\boldsymbol{x}$. For $1 \le j \le d$ and $k \ge 1$, we have*

$$\int_{\mathbb{R}^d} \left( \partial_j^k p_\sigma(\boldsymbol{x}) - \partial_j^k q_\sigma(\boldsymbol{x}) \right)^2 \mathrm{d}\boldsymbol{x} \le 4\sigma^{-2k+1} (2\pi\sigma^2)^{-d/2} \inf_{a \ge \sqrt{2k-1}} \left\{ a^{2k} \mathsf{H}^2(P_\sigma, Q_\sigma) + \sqrt{\frac{2}{\pi}} a^{2k-1} e^{-a^2} \right\}.$$

*Proof.* For $\boldsymbol{X} \sim P, \boldsymbol{Y} \sim Q$, we let

$$P' = \mathsf{law}\left(\frac{\boldsymbol{X}}{\sigma}\right) * \mathcal{N}(0, \boldsymbol{I}_d),$$

$$Q' = \mathsf{law}\left(\frac{\boldsymbol{Y}}{\sigma}\right) * \mathcal{N}(0, \boldsymbol{I}_d),$$

and their density functions $p', q' : \mathbb{R}^d \to \mathbb{R}$, respectively. Let

$$\boldsymbol{X}_\sigma = \boldsymbol{X} + \mathcal{N}(0, \sigma^2 \boldsymbol{I}_d) \sim P * \mathcal{N}(0, \sigma^2 \boldsymbol{I}_d),$$

$$\boldsymbol{Y}_\sigma = \boldsymbol{Y} + \mathcal{N}(0, \sigma^2 \boldsymbol{I}_d) \sim Q * \mathcal{N}(0, \sigma^2 \boldsymbol{I}_d),$$

$$\boldsymbol{X}' = \frac{\boldsymbol{X}}{\sigma} + \mathcal{N}(0, \boldsymbol{I}_d) \sim P'$$

$$\boldsymbol{Y}' = \frac{\boldsymbol{Y}}{\sigma} + \mathcal{N}(0, \boldsymbol{I}_d) \sim Q'.$$

Then we have

$$\boldsymbol{X}_\sigma = \sigma \boldsymbol{X}',$$
$$\boldsymbol{Y}_\sigma = \sigma \boldsymbol{Y}',$$

and $p_\sigma(\boldsymbol{x}) = \sigma^{-d} p'(\frac{\boldsymbol{x}}{\sigma}), q_\sigma(\boldsymbol{x}) = \sigma^{-d} q'(\frac{\boldsymbol{x}}{\sigma})$. Therefore,

$$\int_{\mathbb{R}^d} \left( \partial_j^k p_\sigma(\boldsymbol{x}) - \partial_j^k q_\sigma(\boldsymbol{x}) \right)^2 \mathrm{d}\boldsymbol{x} = \int_{\mathbb{R}^d} \sigma \left( \sigma^{-d-k} \partial_j^k p'\left(\frac{\boldsymbol{x}}{\sigma}\right) - \sigma^{-d-k} \partial_j^k q'\left(\frac{\boldsymbol{x}}{\sigma}\right) \right)^2 \mathrm{d}\left(\frac{\boldsymbol{x}}{\sigma}\right)$$

$$= \sigma^{-2(d+k)+1} \int_{\mathbb{R}^d} \left( \partial_j^k p'\left(\frac{\boldsymbol{x}}{\sigma}\right) - \partial_j^k q'\left(\frac{\boldsymbol{x}}{\sigma}\right) \right)^2 \mathrm{d}\left(\frac{\boldsymbol{x}}{\sigma}\right).$$

By [41, Lemma F.2], we have

$$\int_{\mathbb{R}^d} \left( \partial_j^k p'\left(\frac{\boldsymbol{x}}{\sigma}\right) - \partial_j^k q'\left(\frac{\boldsymbol{x}}{\sigma}\right) \right)^2 \mathrm{d}\left(\frac{\boldsymbol{x}}{\sigma}\right) \le 4(2\pi)^{-d/2} \inf_{a \ge \sqrt{2k-1}} \left\{ a^{2k} \mathsf{H}^2(P', Q') + \sqrt{\frac{2}{\pi}} a^{2k-1} e^{-a^2} \right\}.$$

By the scale-invariance of the Hellinger distance, i.e.,

$$\mathsf{H}^2(P_\sigma, Q_\sigma) = \mathsf{H}^2(P', Q'),$$

we obtain

$$\int_{\mathbb{R}^d} \left( \partial_j^k p_\sigma(\boldsymbol{x}) - \partial_j^k q_\sigma(\boldsymbol{x}) \right)^2 \mathrm{d}\boldsymbol{x} \le 4\sigma^{-2k+1} (2\pi\sigma^2)^{-d/2} \inf_{a \ge \sqrt{2k-1}} \left\{ a^{2k} \mathsf{H}^2(P_\sigma, Q_\sigma) + \sqrt{\frac{2}{\pi}} a^{2k-1} e^{-a^2} \right\}.$$

$\square$

The following lemma extends [41, Lemma F.1] by generalizing it from densities convolved with a standard Gaussian distribution $\mathcal{N}(\mathbf{0}, \boldsymbol{I}_d)$ to densities convolved with a general isotropic Gaussian distribution $\mathcal{N}(\mathbf{0}, \sigma^2 \boldsymbol{I}_d)$ for any $\sigma > 0$.

**Lemma 5.** *Fix a probability distribution $P$ on $\mathbb{R}^d$ with density function $p : \mathbb{R}^d \to \mathbb{R}$. For all $\mu, \sigma > 0$, let $\boldsymbol{Y} = \mu \boldsymbol{X} + \sigma \boldsymbol{Z}$, where $\boldsymbol{Z} \sim \mathcal{N}(\mathbf{0}, \sigma^2 \boldsymbol{I}_d)$. Denote by $P_\sigma$ the marginal distribution of $\boldsymbol{Y}$ with density function $p_\sigma(\boldsymbol{y}) := \int_{\mathbb{R}^d} \varphi_\sigma(\boldsymbol{y} - \mu \boldsymbol{x}) p(\boldsymbol{x}) \mathrm{d}\boldsymbol{x}$ the density function of $\boldsymbol{Y}$. For all $\boldsymbol{y} \in \mathbb{R}^d$, we have*

$$\Big( \frac{\|\nabla p_\sigma(\boldsymbol{y})\|_2}{p_\sigma(\boldsymbol{y})} \Big)^2 \leq \mathrm{Tr}\Big( \frac{1}{\sigma^2} \boldsymbol{I}_d + \frac{\nabla^2(p_\sigma(\boldsymbol{y}))}{p_\sigma(\boldsymbol{y})} \Big) \leq \frac{2}{\sigma^2} \log\Big( \frac{(2\pi\sigma^2)^{-d/2}}{p_\sigma(\boldsymbol{y})} \Big). \tag{13}$$

*Moreover, we have*

$$\Big( \frac{\|\nabla p_\sigma(\boldsymbol{y})\|_2}{p_\sigma(\boldsymbol{y}) \vee \rho} \Big)^2 \leq \begin{cases} \frac{2}{\sigma^2} \log\Big( \frac{(2\pi\sigma^2)^{-d/2}}{\rho} \Big) & \text{if } 0 < \rho \leq (2\pi\sigma^2)^{-d/2} e^{-1/2}, \\ \frac{1}{\sigma^2} & \text{if } \rho > (2\pi\sigma^2)^{-d/2} e^{-1/2}. \end{cases} \tag{14}$$

*Proof.* If $\boldsymbol{X} \sim P$ and $\boldsymbol{Y} | \boldsymbol{X} \sim \mathcal{N}(\mu \boldsymbol{X}, \sigma^2 \boldsymbol{I}_d)$, then by Tweedie's formula, for every $\boldsymbol{y} \in \mathbb{R}^d$,

$$\frac{\nabla p_\sigma(\cdot)}{p_\sigma(\cdot)} \Big|_{\cdot = \boldsymbol{y}} = \frac{1}{\sigma^2} \mathbb{E}[\mu \boldsymbol{X} - \boldsymbol{Y} | \boldsymbol{Y} = \boldsymbol{y}]. \tag{15}$$

Then we have

$$\frac{\nabla^2(p_\sigma(\boldsymbol{y}))}{p_\sigma(\boldsymbol{y})} = \int \frac{\nabla^2(p_\sigma(\cdot | \boldsymbol{x}))}{p_\sigma(\cdot | \boldsymbol{x})} p_\sigma(\boldsymbol{x} | \cdot) \mathrm{d}\boldsymbol{x} \Big|_{\cdot = \boldsymbol{y}}$$

$$= \mathbb{E}_{\boldsymbol{X} | \boldsymbol{Y} = \boldsymbol{y}} \Big[ \frac{-\frac{1}{\sigma^2} \boldsymbol{I}_d \exp\Big( -\frac{\|\boldsymbol{Y} - \mu \boldsymbol{X}\|_2^2}{2\sigma^2} \Big) + \frac{1}{\sigma^4}(\boldsymbol{Y} - \mu \boldsymbol{X})(\boldsymbol{Y} - \mu \boldsymbol{X})^\top \exp\Big( -\frac{\|\boldsymbol{Y} - \mu \boldsymbol{X}\|_2^2}{2\sigma^2} \Big)}{\exp\Big( -\frac{\|\boldsymbol{Y} - \mu \boldsymbol{X}\|_2^2}{2\sigma^2} \Big)} \Big]$$

$$= \mathbb{E}_{\boldsymbol{X} | \boldsymbol{Y} = \boldsymbol{y}} \Big[ -\frac{1}{\sigma^2} \boldsymbol{I}_d + \frac{1}{\sigma^4}(\boldsymbol{Y} - \mu \boldsymbol{X})(\boldsymbol{Y} - \mu \boldsymbol{X})^\top \Big]$$

$$= -\frac{1}{\sigma^2} \boldsymbol{I}_d + \frac{1}{\sigma^4} \mathbb{E}[(\mu \boldsymbol{X} - \boldsymbol{Y})(\mu \boldsymbol{X} - \boldsymbol{Y})^\top | \boldsymbol{Y} = \boldsymbol{y}].$$

Then we have

$$\frac{1}{\sigma^2} \boldsymbol{I}_d + \frac{\nabla^2(p_\sigma(\boldsymbol{y}))}{p_\sigma(\boldsymbol{y})} = \frac{1}{\sigma^4} \mathbb{E}[(\mu \boldsymbol{X} - \boldsymbol{Y})(\mu \boldsymbol{X} - \boldsymbol{Y})^\top | \boldsymbol{Y} = \boldsymbol{y}]$$

$$= \frac{1}{\sigma^4} \Big( \big( \mathbb{E}[\mu \boldsymbol{X} - \boldsymbol{Y} | \boldsymbol{Y} = \boldsymbol{y}] \big) \big( \mathbb{E}[\mu \boldsymbol{X} - \boldsymbol{Y} | \boldsymbol{Y} = \boldsymbol{y}] \big)^\top + \mathrm{Cov}(\mu \boldsymbol{X} - \boldsymbol{Y} | \boldsymbol{Y} = \boldsymbol{y}) \Big)$$

$$= \frac{1}{\sigma^4} \Big( \big( \mathbb{E}[\mu \boldsymbol{X} - \boldsymbol{Y} | \boldsymbol{Y} = \boldsymbol{y}] \big) \big( \mathbb{E}[\mu \boldsymbol{X} - \boldsymbol{Y} | \boldsymbol{Y} = \boldsymbol{y}] \big)^\top + \mathrm{Cov}(\mu \boldsymbol{X} | \boldsymbol{Y} = \boldsymbol{y}) \Big)$$

$$= \frac{1}{\sigma^4} \Big( \sigma^2 \frac{\nabla p_\sigma(\boldsymbol{y})}{p_\sigma(\boldsymbol{y})} \cdot \sigma^2 \frac{(\nabla p_\sigma(\boldsymbol{y}))^\top}{p_\sigma(\boldsymbol{y})} \Big) + \frac{1}{\sigma^4} \mathrm{Cov}(\mu \boldsymbol{X} | \boldsymbol{Y} = \boldsymbol{y}) \quad \text{(by Eq. (15))}$$

$$= \frac{\nabla p_\sigma(\boldsymbol{y})}{p_\sigma(\boldsymbol{y})} \frac{(\nabla p_\sigma(\boldsymbol{y}))^\top}{p_\sigma(\boldsymbol{y})} + \frac{\mu^2}{\sigma^4} \mathrm{Cov}(\boldsymbol{X} | \boldsymbol{Y} = \boldsymbol{y}).$$

Hence,

$$\frac{\nabla p_\sigma(\boldsymbol{y})}{p_\sigma(\boldsymbol{y})} \frac{(\nabla p_\sigma(\boldsymbol{y}))^\top}{p_\sigma(\boldsymbol{y})} = \frac{1}{\sigma^2} \boldsymbol{I}_d + \frac{\nabla^2(p_\sigma(\boldsymbol{y}))}{p_\sigma(\boldsymbol{y})} - \frac{\mu^2}{\sigma^4} \mathrm{Cov}(\boldsymbol{X} | \boldsymbol{Y} = \boldsymbol{y}), \tag{16}$$

which gives that

$$\frac{\nabla p_\sigma(\boldsymbol{y})}{p_\sigma(\boldsymbol{y})} \frac{(\nabla p_\sigma(\boldsymbol{y}))^\top}{p_\sigma(\boldsymbol{y})} \preceq \frac{1}{\sigma^2} \boldsymbol{I}_d + \frac{\nabla^2(p_\sigma(\boldsymbol{y}))}{p_\sigma(\boldsymbol{y})}. \tag{17}$$

By the convexity of $x \mapsto \exp(\frac{1}{2}\mathrm{Tr}(\frac{1}{\sigma^2}x))$, we have

$$\exp\Big(\frac{1}{2}\mathrm{Tr}\Big(\boldsymbol{I}_d + \frac{\sigma^2 \nabla^2(p_\sigma(\boldsymbol{y}))}{p_\sigma(\boldsymbol{y})}\Big)\Big)$$

$$= \exp\Big(\frac{1}{2}\mathrm{Tr}\Big(\frac{1}{\sigma^2}\mathbb{E}[(\mu\boldsymbol{X} - \boldsymbol{Y})(\mu\boldsymbol{X} - \boldsymbol{Y})^\top|\boldsymbol{Y} = \boldsymbol{y}]\Big)\Big)$$

$$\leq \mathbb{E}\Big[\exp\Big(\frac{1}{2}\mathrm{Tr}\Big(\frac{1}{\sigma^2}(\mu\boldsymbol{X} - \boldsymbol{Y})(\mu\boldsymbol{X} - \boldsymbol{Y})^\top\Big)\Big)|\boldsymbol{Y} = \boldsymbol{y}\Big] \qquad \text{(by Jensen's inequality)}$$

$$= \mathbb{E}\Big[\exp\Big(\frac{\|\mu\boldsymbol{X} - \boldsymbol{Y}\|_2^2}{2\sigma^2}\Big)|\boldsymbol{Y} = \boldsymbol{y}\Big]$$

$$= \mathbb{E}_{\boldsymbol{X}|\boldsymbol{Y}=\boldsymbol{y}}\Big[\exp\Big(\frac{\|\mu\boldsymbol{X} - \boldsymbol{Y}\|_2^2}{2\sigma^2}\Big)\Big]$$

$$= \int \exp\Big(\frac{\|\mu\boldsymbol{x} - \boldsymbol{y}\|_2^2}{2\sigma^2}\Big)p_\sigma(\boldsymbol{x}|\boldsymbol{y})\mathrm{d}\boldsymbol{x}$$

$$= \int \frac{\exp\Big(\frac{\|\mu\boldsymbol{x} - \boldsymbol{y}\|_2^2}{2\sigma^2}\Big)p_\sigma(\boldsymbol{y}|\boldsymbol{x})p_\sigma(\boldsymbol{x})}{p_\sigma(\boldsymbol{y})}\mathrm{d}\boldsymbol{x}$$

$$= \frac{\int \exp\Big(\frac{\|\mu\boldsymbol{x} - \boldsymbol{y}\|_2^2}{2\sigma^2}\Big)p_\sigma(\boldsymbol{y}|\boldsymbol{x})p_\sigma(\boldsymbol{x})\mathrm{d}\boldsymbol{x}}{p_\sigma(\boldsymbol{y})}$$

$$= \frac{\int \exp\Big(\frac{\|\mu\boldsymbol{x} - \boldsymbol{y}\|_2^2}{2\sigma^2}\Big)(2\pi\sigma^2)^{-d/2}\exp\Big(-\frac{\|\mu\boldsymbol{x} - \boldsymbol{y}\|_2^2}{2\sigma^2}\Big)p_\sigma(\boldsymbol{x})\mathrm{d}\boldsymbol{x}}{p_\sigma(\boldsymbol{y})} \qquad (\boldsymbol{Y}|\boldsymbol{X} \sim \mathcal{N}(\mu\boldsymbol{X}, \sigma^2\boldsymbol{I}_d))$$

$$= \frac{(2\pi\sigma^2)^{-d/2}}{p_\sigma(\boldsymbol{y})}.$$

Therefore,

$$\mathrm{Tr}\Big(\frac{1}{\sigma^2}\boldsymbol{I}_d + \frac{\nabla^2(p_\sigma(\boldsymbol{y}))}{p_\sigma(\boldsymbol{y})}\Big) \leq \frac{2}{\sigma^2}\log\Big(\frac{(2\pi\sigma^2)^{-d/2}}{p_\sigma(\boldsymbol{y})}\Big),$$

together with Eq. (17) which gives

$$\frac{\|\nabla p_\sigma(\boldsymbol{y})\|_2^2}{p_\sigma^2(\boldsymbol{y})} \leq \mathrm{Tr}\Big(\frac{1}{\sigma^2}\boldsymbol{I}_d + \frac{\nabla^2(p_\sigma(\boldsymbol{y}))}{p_\sigma(\boldsymbol{y})}\Big) \leq \frac{2}{\sigma^2}\log\Big(\frac{(2\pi\sigma^2)^{-d/2}}{p_\sigma(\boldsymbol{y})}\Big).$$

Then, we have

$$\frac{\|\nabla p_\sigma(\boldsymbol{y})\|_2}{p_\sigma(\boldsymbol{y}) \vee \rho} \leq \frac{p_\sigma(\boldsymbol{y})}{p_\sigma(\boldsymbol{y}) \vee \rho}\sqrt{\frac{2}{\sigma^2}\log\Big(\frac{(2\pi\sigma^2)^{-d/2}}{p_\sigma(\boldsymbol{y})}\Big)}$$

$$= \begin{cases} \sqrt{\frac{2}{\sigma^2}\log\Big(\frac{(2\pi\sigma^2)^{-d/2}}{p_\sigma(\boldsymbol{y})}\Big)} \leq \sqrt{\frac{2}{\sigma^2}\log\Big(\frac{(2\pi\sigma^2)^{-d/2}}{\rho}\Big)} & \text{if } p_\sigma(\boldsymbol{y}) > \rho, \\[2ex] \frac{p_\sigma(\boldsymbol{y})}{\rho}\sqrt{\frac{2}{\sigma^2}\log\Big(\frac{(2\pi\sigma^2)^{-d/2}}{p_\sigma(\boldsymbol{y})}\Big)} & \text{if } p_\sigma(\boldsymbol{y}) \leq \rho. \end{cases}$$

When $p_\sigma(\boldsymbol{y}) \leq \rho \leq (2\pi\sigma^2)^{-d/2}e^{-1/2}$, since $x \mapsto x\sqrt{\log\big(\frac{c}{x}\big)}$ is non-decreasing on $(0, \frac{c}{\sqrt{e}}]$, we have

$$\frac{\|\nabla p_\sigma(\boldsymbol{y})\|_2}{p_\sigma(\boldsymbol{y}) \vee \rho} \leq \sqrt{\frac{2}{\sigma^2}\log\Big(\frac{(2\pi\sigma^2)^{-d/2}}{\rho}\Big)}.$$

Therefore, if $\rho \leq (2\pi\sigma^2)^{-d/2}e^{-1/2}$, we conclude that

$$\frac{\|\nabla p_\sigma(\boldsymbol{y})\|_2}{p_\sigma(\boldsymbol{y}) \vee \rho} \leq \sqrt{\frac{2}{\sigma^2}\log\Big(\frac{(2\pi\sigma^2)^{-d/2}}{\rho}\Big)}. \tag{18}$$

If $\rho > (2\pi\sigma^2)^{-d/2}e^{-1/2}$, since $x\sqrt{\log\left(\frac{c}{x}\right)} \le (2e)^{-1/2}$, where the equality is obtain when $x = \frac{c}{\sqrt{e}}$. Then,

$$
\frac{\|\nabla p_\sigma(\boldsymbol{y})\|_2}{p_\sigma(\boldsymbol{y}) \vee \rho} \le \frac{p_\sigma(\boldsymbol{y})}{p_\sigma(\boldsymbol{y}) \vee \rho}\sqrt{\frac{2}{\sigma^2}\log\left(\frac{(2\pi\sigma^2)^{-d/2}}{p_\sigma(\boldsymbol{y})}\right)}
$$
$$
\le \frac{(2\pi\sigma^2)^{-d/2}e^{-1/2}}{p_\sigma(\boldsymbol{y}) \vee \rho}\sqrt{\frac{1}{\sigma^2}} \le \frac{1}{\sigma}.
$$

$\square$

## A.1 Proof of Lemma 2

**Lemma 2** *Given any distributions $P, Q$ on $\mathbb{R}^d$ with density functions $p, q : \mathbb{R}^d \to \mathbb{R}_+$, respectively. Fix $\sigma > 0$, let $P_\sigma = P * \mathcal{N}(0, \sigma^2\boldsymbol{I}_d), Q_\sigma = Q * \mathcal{N}(0, \sigma^2\boldsymbol{I}_d)$ with density functions $p_\sigma, q_\sigma : \mathbb{R}^d \to \mathbb{R}_+$. For all $d \ge 1, n \ge 1$, let $0 < \rho_n \le (2\pi\sigma^2)^{-d/2}e^{-1/2}$ and let $\mathcal{G} := \{\boldsymbol{x} \in \mathbb{R}^d : p_\sigma(\boldsymbol{x}) \le \rho_n\}$, then there exists a universal constant $C > 0$ such that*

$$
\int_{\mathbb{R}^d}\left\|\frac{\nabla p_\sigma(\boldsymbol{x})}{p_\sigma(\boldsymbol{x})} - \frac{\nabla q_\sigma(\boldsymbol{x})}{q_\sigma(\boldsymbol{x}) \vee \rho_n}\right\|_2^2 p_\sigma(\boldsymbol{x})\mathrm{d}\boldsymbol{x}
$$
$$
\le C\left(\frac{d}{\sigma^2}\max\left\{\log^3\left(\frac{(2\pi\sigma^2)^{-d/2}}{\rho_n}\right), \log(\mathsf{H}^{-2}(P_\sigma, Q_\sigma))\right\}\mathsf{H}^2(P_\sigma, Q_\sigma) + \int_{\mathcal{G}}\left\|\frac{\nabla p_\sigma(\boldsymbol{x})}{p_\sigma(\boldsymbol{x})}\right\|_2^2 p_\sigma(\boldsymbol{x})\mathrm{d}\boldsymbol{x}\right).
$$

*Proof.* With $0 < \rho_n \le (2\pi\sigma^2)^{-d/2}e^{-1/2}$, let $\mathcal{G}_1 := \{x \in \mathbb{R}^d : p_\sigma(\boldsymbol{x}) \ge \rho_n\}, \mathcal{G}_2 := \{x \in \mathbb{R}^d : \rho_n > p_\sigma(\boldsymbol{x})\}$ such that $\mathbb{R}^d = \mathcal{G}_1 \cup \mathcal{G}_2$.

$$
\int_{\mathbb{R}^d}\left\|\frac{\nabla p_\sigma}{p_\sigma} - \frac{\nabla q_\sigma}{q_\sigma \vee \rho_n}\right\|_2^2 p_\sigma \le \underbrace{\int_{\mathcal{G}_1}\left\|\frac{\nabla p_\sigma}{p_\sigma} - \frac{\nabla q_\sigma}{q_\sigma \vee \rho_n}\right\|_2^2 p_\sigma}_{:= \text{(I)}} + \underbrace{\int_{\mathcal{G}_2}\left\|\frac{\nabla p_\sigma}{p_\sigma} - \frac{\nabla q_\sigma}{q_\sigma \vee \rho_n}\right\|_2^2 p_\sigma}_{:= \text{(II)}}.
$$

**Case 1:** $\mathcal{G}_1 := \{x \in \mathbb{R}^d : p_\sigma(\boldsymbol{x}) \ge \rho_n\}$,

$$
\int_{\mathcal{G}_1}\left\|\frac{\nabla p_\sigma}{p_\sigma} - \frac{\nabla q_\sigma}{q_\sigma \vee \rho_n}\right\|_2^2 \cdot p_\sigma
$$
$$
= \int_{\mathcal{G}_1}\left\|\frac{\nabla p_\sigma}{p_\sigma} - \frac{2\nabla p_\sigma}{p_\sigma + q_\sigma \vee \rho_n} + \frac{2(\nabla p_\sigma - \nabla q_\sigma)}{p_\sigma + q_\sigma \vee \rho_n} + \frac{2\nabla q_\sigma}{p_\sigma + q_\sigma \vee \rho_n} - \frac{\nabla q_\sigma}{q_\sigma \vee \rho_n}\right\|_2^2 \cdot p_\sigma
$$
$$
= 2\int_{\mathcal{G}_1}\left\|\frac{q_\sigma \vee \rho_n - p_\sigma}{p_\sigma + q_\sigma \vee \rho_n}\left(\frac{\nabla p_\sigma}{p_\sigma} + \frac{\nabla q_\sigma}{q_\sigma \vee \rho_n}\right)\right\|_2^2 \cdot p_\sigma + 2\int_{\mathcal{G}_1}\left\|\frac{2(\nabla p_\sigma - \nabla q_\sigma)}{p_\sigma + q_\sigma \vee \rho_n}\right\|_2^2 \cdot p_\sigma
$$
$$
\le \underbrace{2\int_{\mathcal{G}_1}\left\|\frac{\nabla p_\sigma}{p_\sigma} + \frac{\nabla q_\sigma}{q_\sigma \vee \rho_n}\right\|_2^2 \cdot \frac{(q_\sigma \vee \rho_n - p_\sigma)^2}{(p_\sigma + q_\sigma \vee \rho_n)^2} \cdot p_\sigma}_{:= \text{(I-a)}} + \underbrace{4\int_{\mathcal{G}_1}\left\|\frac{\nabla p_\sigma - \nabla q_\sigma}{p_\sigma + q_\sigma \vee \rho_n}\right\|_2^2 \cdot p_\sigma}_{:= \text{(I-b)}}
$$

**Bounding (I-a)**

$$
\begin{aligned}
\text{(I-a)} \ &\le 4 \int_{\mathcal{G}_1} \left( \left\| \frac{\nabla p_\sigma}{p_\sigma} \right\|_2^2 + \left\| \frac{\nabla q_\sigma}{q_\sigma \vee \rho_n} \right\|_2^2 \right) \frac{(q_\sigma \vee \rho_n - p_\sigma)^2 p_\sigma}{(p_\sigma + q_\sigma \vee \rho_n)^2} \\
&= 4 \int_{\mathcal{G}_1} \left( \left\| \frac{\nabla p_\sigma}{p_\sigma} \right\|_2^2 + \left\| \frac{\nabla q_\sigma}{q_\sigma \vee \rho_n} \right\|_2^2 \right) \frac{(\sqrt{q_\sigma} \vee \sqrt{\rho_n} - \sqrt{p_\sigma})^2 (\sqrt{q_\sigma} \vee \sqrt{\rho_n} + \sqrt{p_\sigma})^2 p_\sigma}{(p_\sigma + q_\sigma \vee \rho_n)^2} \\
&\le 8 \int_{\mathcal{G}_1} \left( \left\| \frac{\nabla p_\sigma}{p_\sigma} \right\|_2^2 + \left\| \frac{\nabla q_\sigma}{q_\sigma \vee \rho_n} \right\|_2^2 \right) \frac{(\sqrt{q_\sigma} \vee \sqrt{\rho_n} - \sqrt{p_\sigma})^2 p_\sigma}{p_\sigma + q_\sigma \vee \rho_n} \\
&\le 8 \int_{\mathcal{G}_1} \left( \left\| \frac{\nabla p_\sigma}{p_\sigma} \right\|_2^2 + \left\| \frac{\nabla q_\sigma}{q_\sigma \vee \rho_n} \right\|_2^2 \right) \frac{(\sqrt{q_\sigma} - \sqrt{p_\sigma})^2 p_\sigma}{p_\sigma + q_\sigma \vee \rho_n} && \text{(by } p_\sigma \ge \rho_n) \\
&\le 8 \int_{\mathcal{G}_1} \left( \left\| \frac{\nabla p_\sigma}{p_\sigma} \right\|_2^2 + \left\| \frac{\nabla q_\sigma}{q_\sigma \vee \rho_n} \right\|_2^2 \right) (\sqrt{q_\sigma} - \sqrt{p_\sigma})^2 \\
&\le \frac{32}{\sigma^2} \log\!\left( \frac{(2\pi\sigma^2)^{-d/2}}{\rho_n} \right) \int_{\mathcal{G}_1} (\sqrt{p_\sigma} - \sqrt{q_\sigma})^2. && \text{(by Lemma 5)}
\end{aligned}
$$

**Bounding (I-b)**

$$
\text{(I-b)} = 4 \int_{\mathcal{G}_1} \frac{\|\nabla p_\sigma - \nabla q_\sigma\|_2^2}{(p_\sigma + q_\sigma \vee \rho_n)^2} \cdot p_\sigma \le 4 \int_{\mathcal{G}_1} \frac{\|\nabla p_\sigma - \nabla q_\sigma\|_2^2}{p_\sigma + q_\sigma \vee \rho_n}. \tag{19}
$$

**Case 2:** $\mathcal{G}_2 := \{x \in \mathbb{R}^d : \rho_n > p_\sigma(\boldsymbol{x})\}$

**Bouding (II)**

$$
\begin{aligned}
\int_{\mathcal{G}_2} \left\| \frac{\nabla p_\sigma}{p_\sigma} - \frac{\nabla q_\sigma}{q_\sigma \vee \rho_n} \right\|_2^2 \cdot p_\sigma &= \int_{\mathcal{G}_2} \left\| \frac{(q_\sigma \vee \rho_n)\nabla p_\sigma - p_\sigma \nabla q_\sigma}{p_\sigma(q_\sigma \vee \rho_n)} \right\|_2^2 \cdot p_\sigma \\
&= \int_{\mathcal{G}_2} \left\| \frac{(q_\sigma \vee \rho_n)\nabla p_\sigma - p_\sigma \nabla p_\sigma + p_\sigma \nabla p_\sigma - p_\sigma \nabla q_\sigma}{p_\sigma(q_\sigma \vee \rho_n)} \right\|_2^2 \cdot p_\sigma \\
&= \int_{\mathcal{G}_2} \left\| \frac{(q_\sigma \vee \rho_n - p_\sigma)\nabla p_\sigma + p_\sigma(\nabla p_\sigma - \nabla q_\sigma)}{p_\sigma(q_\sigma \vee \rho_n)} \right\|_2^2 \cdot p_\sigma \\
&\le \underbrace{2 \int_{\mathcal{G}_2} \left\| \frac{(q_\sigma \vee \rho_n - p_\sigma)}{q_\sigma \vee \rho_n} \cdot \frac{\nabla p_\sigma}{p_\sigma} \right\|_2^2 \cdot p_\sigma}_{:= \text{(II-a)}} + \underbrace{2 \int_{\mathcal{G}_2} \left\| \frac{\nabla p_\sigma - \nabla q_\sigma}{q_\sigma \vee \rho_n} \right\|_2^2 \cdot p_\sigma}_{:= \text{(II-b)}}
\end{aligned}
$$

**Bounding (II-a)**

$$
\begin{aligned}
\text{(II-a)} &= 2\int_{\mathcal{G}_2}\Big\|\frac{\nabla p_\sigma}{p_\sigma}\Big\|_2^2\frac{(q_\sigma\vee\rho_n-p_\sigma)^2}{q_\sigma^2\vee\rho_n^2}\cdot p_\sigma\\
&= 2\int_{\mathcal{G}_2}\Big\|\frac{\nabla p_\sigma}{p_\sigma}\Big\|_2^2\frac{(\sqrt{q_\sigma}\vee\sqrt{\rho_n}-\sqrt{p_\sigma})^2(\sqrt{q_\sigma}\vee\sqrt{\rho_n}+\sqrt{p_\sigma})^2}{q_\sigma^2\vee\rho_n^2}\cdot p_\sigma\\
&\le 4\int_{\mathcal{G}_2}\Big\|\frac{\nabla p_\sigma}{p_\sigma}\Big\|_2^2\frac{(\sqrt{q_\sigma}\vee\sqrt{\rho_n}-\sqrt{p_\sigma})^2(q_\sigma\vee\rho_n+p_\sigma)}{q_\sigma^2\vee\rho_n^2}\cdot p_\sigma\\
&\le 8\int_{\mathcal{G}_2}\Big\|\frac{\nabla p_\sigma}{p_\sigma}\Big\|_2^2\frac{\Big((\sqrt{q_\sigma}-\sqrt{p_\sigma})^2+(\sqrt{\rho_n}-\sqrt{p_\sigma})^2\Big)(q_\sigma\vee\rho_n+p_\sigma)}{q_\sigma^2\vee\rho_n^2}\cdot p_\sigma\\
&\le 8\int_{\mathcal{G}_2}\Big\|\frac{\nabla p_\sigma}{p_\sigma}\Big\|_2^2\Big(2(\sqrt{q_\sigma}-\sqrt{p_\sigma})^2+\frac{(\sqrt{\rho_n}-\sqrt{p_\sigma})^2(q_\sigma\vee\rho_n+p_\sigma)}{q_\sigma^2\vee\rho_n^2}\cdot p_\sigma\Big)\\
&\hspace{6cm}\text{(by } q_\sigma\vee\rho_n\ge\rho_n\ge p_\sigma\ge 0)\\
&\le 8\int_{\mathcal{G}_2}\Big\|\frac{\nabla p_\sigma}{p_\sigma}\Big\|_2^2\Big(2(\sqrt{q_\sigma}-\sqrt{p_\sigma})^2+\frac{(\rho_n+p_\sigma)(q_\sigma\vee\rho_n+p_\sigma)}{q_\sigma^2\vee\rho_n^2}\cdot p_\sigma\Big)\\
&\le 16\int_{\mathcal{G}_2}\Big\|\frac{\nabla p_\sigma}{p_\sigma}\Big\|_2^2\Big((\sqrt{q_\sigma}-\sqrt{p_\sigma})^2+2p_\sigma\Big)\hspace{1cm}\text{(by } q_\sigma\vee\rho_n\ge\rho_n\ge p_\sigma\ge 0)\\
&\le \frac{32}{\sigma^2}\log\Big(\frac{(2\pi\sigma^2)^{-d/2}}{\rho_n}\Big)\int_{\mathcal{G}_2}(\sqrt{q_\sigma}-\sqrt{p_\sigma})^2+32\int_{\mathcal{G}_2}\Big\|\frac{\nabla p_\sigma}{p_\sigma}\Big\|_2^2 p_\sigma.\hspace{0.5cm}\text{(by Lemma 5)}
\end{aligned}
$$

**Bounding (II-b)**

$$
\begin{aligned}
\text{(II-b)} &= 2\int_{\mathcal{G}_2}\frac{\|\nabla p_\sigma-\nabla q_\sigma\|_2^2}{q_\sigma^2\vee\rho_n^2}\cdot p_\sigma\\
&\le 2\int_{\mathcal{G}_2}\frac{\|\nabla p_\sigma-\nabla q_\sigma\|_2^2}{q_\sigma\vee\rho_n}\hspace{2cm}\text{(by } q_\sigma\ge\rho_n>p_\sigma)\\
&\le 4\int_{\mathcal{G}_2}\frac{\|\nabla p_\sigma-\nabla q_\sigma\|_2^2}{q_\sigma\vee\rho_n+p_\sigma}.\hspace{3cm}\text{(20)}
\end{aligned}
$$

**Combining Case 1 and 2**

**Bounding (I-a) + (II-a)**

$$
\begin{aligned}
\text{(I-a)}+\text{(II-a)} &\le \frac{32}{\sigma^2}\log\Big(\frac{(2\pi\sigma^2)^{-d/2}}{\rho_n}\Big)\int_{\mathbb{R}^d}(\sqrt{q_\sigma}-\sqrt{p_\sigma})^2+32\int_{\mathcal{G}_2}\Big\|\frac{\nabla p_\sigma}{p_\sigma}\Big\|_2^2 p_\sigma\\
&\le \frac{32}{\sigma^2}\log\Big(\frac{(2\pi\sigma^2)^{-d/2}}{\rho_n}\Big)\mathsf{H}^2(P_\sigma,Q_\sigma)+32\int_{\mathcal{G}_2}\Big\|\frac{\nabla p_\sigma}{p_\sigma}\Big\|_2^2 p_\sigma.\hspace{1cm}\text{(21)}
\end{aligned}
$$

**Bounding (I-b) + (II-b)**

$$
\text{(I-b)}+\text{(II-b)}\le 4\int_{\mathbb{R}^d}\frac{\|\nabla p_\sigma-\nabla q_\sigma\|_2^2}{q_\sigma\vee\rho_n+p_\sigma}.
$$

For $1\le j\le d$ and $k\ge 0$, let

$$
\Delta_{j,k}^2 := \int_{\mathbb{R}^d}\frac{(\partial_j^k p_\sigma(\boldsymbol{x})-\partial_j^k q_\sigma(\boldsymbol{x}))^2}{p_\sigma+q_\sigma\vee\rho_n}\,\mathrm{d}\boldsymbol{x},\quad\text{with }\partial_j^k p_\sigma:=\frac{\partial^k}{\partial x_j^k}p_\sigma.
$$

Then, we have

$$
\text{(I-b)}+\text{(II-b)}\le \int_{\mathbb{R}^d}\frac{\|\nabla p_\sigma-\nabla q_\sigma\|_2^2}{p_\sigma+q_\sigma\vee\rho_n}=4\sum_{j=1}^d\Delta_{j,1}^2.\hspace{1cm}\text{(22)}
$$

**(1) Bounding $\Delta_{j,0}^2$.**

$$\begin{aligned}
\Delta_{j,0}^2 &= \int_{\mathbb{R}^d} \frac{(p_\sigma(\boldsymbol{x}) - q_\sigma(\boldsymbol{x}))^2}{p_\sigma(\boldsymbol{x}) + q_\sigma(\boldsymbol{x}) \vee \rho_n} \mathrm{d}\boldsymbol{x} \\
&= \int_{\mathbb{R}^d} \frac{(\sqrt{p_\sigma(\boldsymbol{x})} - \sqrt{q_\sigma(\boldsymbol{x})})^2 (\sqrt{p_\sigma(\boldsymbol{x})} + \sqrt{q_\sigma(\boldsymbol{x})})^2}{p_\sigma(\boldsymbol{x}) + q_\sigma(\boldsymbol{x}) \vee \rho_n} \mathrm{d}\boldsymbol{x} \\
&\leq 2 \int_{\mathbb{R}^d} \frac{(\sqrt{p_\sigma(\boldsymbol{x})} - \sqrt{q_\sigma(\boldsymbol{x})})^2 (p_\sigma(\boldsymbol{x}) + q_\sigma(\boldsymbol{x}))}{p_\sigma(\boldsymbol{x}) + q_\sigma(\boldsymbol{x}) \vee \rho_n} \mathrm{d}\boldsymbol{x} \\
&\leq 2 \int_{\mathbb{R}^d} (\sqrt{p_\sigma(\boldsymbol{x})} - \sqrt{q_\sigma(\boldsymbol{x})})^2 \mathrm{d}\boldsymbol{x}. \\
&\leq 2\mathsf{H}^2(P_\sigma, Q_\sigma).
\end{aligned} \tag{23}$$

**(2) Bounding $\Delta_{j,k}^2$ for $k \geq 2$.**

Note that

$$\Delta_{j,k}^2 = \int_{\mathbb{R}^d} \frac{(\partial_j^k p_\sigma(\boldsymbol{x}) - \partial_j^k q_\sigma(\boldsymbol{x}))^2}{p_\sigma(\boldsymbol{x}) + q_\sigma(\boldsymbol{x}) \vee \rho_n} \mathrm{d}\boldsymbol{x} \leq \frac{1}{\rho_n} \int_{\mathbb{R}^d} (\partial_j^k p_\sigma(\boldsymbol{x}) - \partial_j^k q_\sigma(\boldsymbol{x}))^2 \mathrm{d}\boldsymbol{x}.$$

By Lemma 4, we have

$$\Delta_{j,k}^2 \leq \frac{4}{\rho_n (2\pi\sigma^2)^{d/2}} \sigma^{-2k+1} \inf_{a \geq \sqrt{2k-1}} \left\{ a^{2k} \mathsf{H}^2(P_\sigma, Q_\sigma) + \sqrt{\frac{2}{\pi}} a^{2k-1} e^{-a^2} \right\}. \tag{24}$$

**(3) Bounding $\Delta_{j,1}^2$.**

$$\begin{aligned}
\Delta_{j,k}^2 &= \int_{\mathbb{R}^d} \frac{(\partial_j^k p_\sigma(\boldsymbol{x}) - \partial_j^k q_\sigma(\boldsymbol{x}))^2}{p_\sigma(\boldsymbol{x}) + q_\sigma(\boldsymbol{x}) \vee \rho_n} \mathrm{d}\boldsymbol{x} \\
&= \int_{\mathbb{R}^d} \frac{\partial_j^k (p_\sigma(\boldsymbol{x}) - q_\sigma(\boldsymbol{x})) \partial_j^k (p_\sigma(\boldsymbol{x}) - q_\sigma(\boldsymbol{x}))}{p_\sigma(\boldsymbol{x}) + q_\sigma(\boldsymbol{x}) \vee \rho_n} \mathrm{d}\boldsymbol{x} \\
&= - \int_{\mathbb{R}^d} \partial_j^{k-1} (p_\sigma(\boldsymbol{x}) - q_\sigma(\boldsymbol{x})) \partial_j^k (p_\sigma(\boldsymbol{x}) - q_\sigma(\boldsymbol{x})) \partial_j \left( \frac{1}{p_\sigma(\boldsymbol{x}) + q_\sigma(\boldsymbol{x}) \vee \rho_n} \right) \mathrm{d}\boldsymbol{x} \\
&\quad - \int_{\mathbb{R}^d} \frac{\partial_j^{k-1} (p_\sigma(\boldsymbol{x}) - q_\sigma(\boldsymbol{x})) \partial_j^{k+1} (p_\sigma(\boldsymbol{x}) - q_\sigma(\boldsymbol{x}))}{p_\sigma(\boldsymbol{x}) + q_\sigma(\boldsymbol{x}) \vee \rho_n} \mathrm{d}\boldsymbol{x}.
\end{aligned}$$

Note that

$$\begin{aligned}
\left| \partial_j \left( \frac{1}{p_\sigma(\boldsymbol{x}) + q_\sigma(\boldsymbol{x}) \vee \rho_n} \right) \right| &\leq \left| \frac{-\partial_j p_\sigma(\boldsymbol{x}) - \partial_j q_\sigma(\boldsymbol{x})}{(p_\sigma(\boldsymbol{x}) + q_\sigma(\boldsymbol{x}) \vee \rho_n)^2} \right| \\
&\leq \frac{\|\nabla p_\sigma(\boldsymbol{x})\|_2 + \|\nabla q_\sigma(\boldsymbol{x})\|_2}{p_\sigma(\boldsymbol{x}) + q_\sigma(\boldsymbol{x}) \vee \rho_n} \frac{1}{p_\sigma(\boldsymbol{x}) + q_\sigma(\boldsymbol{x}) \vee \rho_n} \\
&\leq \frac{1}{p_\sigma(\boldsymbol{x}) + q_\sigma(\boldsymbol{x}) \vee \rho_n} \frac{2\sqrt{2}}{\sigma} \sqrt{\log \left( \frac{(2\pi\sigma^2)^{-d/2}}{\rho_n} \right)}. \\
&\quad \text{(by Lemma 5 and } q_\sigma \geq \rho_n\text{)}
\end{aligned}$$

Therefore,

$$
\begin{aligned}
\Delta_{j,k}^2 \leq\; & -\int_{\mathbb{R}^d} \partial_j^{k-1}(p_\sigma(\boldsymbol{x}) - q_\sigma(\boldsymbol{x})) \partial_j^k (p_\sigma(\boldsymbol{x}) - q_\sigma(\boldsymbol{x})) \partial_j \Big( \frac{1}{p_\sigma(\boldsymbol{x}) + q_\sigma(\boldsymbol{x}) \vee \rho_n} \Big) \mathrm{d}\boldsymbol{x} \\
& - \int_{\mathbb{R}^d} \frac{\partial_j^{k-1}(p_\sigma(\boldsymbol{x}) - q_\sigma(\boldsymbol{x})) \partial_j^{k+1}(p_\sigma(\boldsymbol{x}) - q_\sigma(\boldsymbol{x}))}{p_\sigma(\boldsymbol{x}) + q_\sigma(\boldsymbol{x}) \vee \rho_n} \mathrm{d}\boldsymbol{x} \\
\leq\; & \frac{2\sqrt{2}}{\sigma} \sqrt{\log\Big( \frac{(2\pi\sigma^2)^{-d/2}}{\rho_n} \Big)} \int_{\mathbb{R}^d} \frac{|\partial_j^{k-1}(p_\sigma(\boldsymbol{x}) - q_\sigma(\boldsymbol{x}))| \cdot |\partial_j^k(p_\sigma(\boldsymbol{x}) - q_\sigma(\boldsymbol{x}))|}{p_\sigma(\boldsymbol{x}) + q_\sigma(\boldsymbol{x}) \vee \rho_n} \mathrm{d}\boldsymbol{x} \\
& + \int_{\mathbb{R}^d} \frac{|\partial_j^{k-1}(p_\sigma(\boldsymbol{x}) - q_\sigma(\boldsymbol{x}))| \cdot |\partial_j^{k+1}(p_\sigma(\boldsymbol{x}) - q_\sigma(\boldsymbol{x}))|}{p_\sigma(\boldsymbol{x}) + q_\sigma(\boldsymbol{x}) \vee \rho_n} \mathrm{d}\boldsymbol{x} \\
\leq\; & \frac{2\sqrt{2}}{\sigma} \sqrt{\log\Big( \frac{(2\pi\sigma^2)^{-d/2}}{\rho_n} \Big)} \sqrt{\int_{\mathbb{R}^d} \frac{(\partial_j^{k-1}(p_\sigma(\boldsymbol{x}) - q_\sigma(\boldsymbol{x})))^2}{p_\sigma(\boldsymbol{x}) + q_\sigma(\boldsymbol{x}) \vee \rho_n} \mathrm{d}\boldsymbol{x}} \cdot \sqrt{\int_{\mathbb{R}^d} \frac{(\partial_j^k(p_\sigma(\boldsymbol{x}) - q_\sigma(\boldsymbol{x})))^2}{p_\sigma(\boldsymbol{x}) + q_\sigma(\boldsymbol{x}) \vee \rho_n} \mathrm{d}\boldsymbol{x}} \\
& + \sqrt{\int_{\mathbb{R}^d} \frac{(\partial_j^{k-1}(p_\sigma(\boldsymbol{x}) - q_\sigma(\boldsymbol{x})))^2}{p_\sigma(\boldsymbol{x}) + q_\sigma(\boldsymbol{x}) \vee \rho_n} \mathrm{d}\boldsymbol{x}} \cdot \sqrt{\int_{\mathbb{R}^d} \frac{(\partial_j^{k+1}(p_\sigma(\boldsymbol{x}) - q_\sigma(\boldsymbol{x})))^2}{p_\sigma(\boldsymbol{x}) + q_\sigma(\boldsymbol{x}) \vee \rho_n} \mathrm{d}\boldsymbol{x}} \\
=\; & \frac{2\sqrt{2}}{\sigma} \sqrt{\log\Big( \frac{(2\pi\sigma^2)^{-d/2}}{\rho_n} \Big)} \Delta_{j,k-1}\Delta_{j,k} + \Delta_{j,k-1}\Delta_{j,k+1}.
\end{aligned}
$$

Divide both sides of the above inequality by $\Delta_{j,k-1}\Delta_{i,k}$ and denote by

$$
C_{\rho,\sigma,d} := \frac{2\sqrt{2}}{\sigma} \sqrt{\log\Big( \frac{(2\pi\sigma^2)^{-d/2}}{\rho_n} \Big)},
$$

we obtain

$$
\frac{\Delta_{j,k}}{\Delta_{j,k-1}} \leq C_{\rho,\sigma,d} + \frac{\Delta_{j,k+1}}{\Delta_{j,k}}, \quad \text{for all } k \geq 1. \tag{25}
$$

- Suppose first that there exist an integer $1 \leq k \leq k_0$ such that $\Delta_{i,k+1} \leq \beta\Delta_{i,k}$. Then applying Eq. (25) recursively for $1, \cdots, k$, we obtain

$$
\frac{\Delta_{i,1}}{\Delta_{i,0}} \leq kC_{\rho,\sigma,d} + \beta.
$$

Then we have

$$
\begin{aligned}
\Delta_{j,1} &\leq \Big( kC_{\rho,\sigma,d} + \beta \Big) \Delta_{j,0} \\
&\leq \sqrt{2}\Big( kC_{\rho,\sigma,d} + \beta \Big) \mathsf{H}(P_\sigma, Q_\sigma) && \text{(by Eq. (23))} \\
&\leq \sqrt{2}\Big( k_0 C_{\rho,\sigma,d} + \beta \Big) \mathsf{H}(P_\sigma, Q_\sigma). && (26)
\end{aligned}
$$

- On the other hand, suppose that $\Delta_{j,k+1} > \beta\Delta_{j,k}$ for every integer $1 \leq k \leq k_0$. Then, by Eq. (25), we have

$$
\frac{\Delta_{j,k}}{\Delta_{j,k-1}} \leq C_{\rho,\sigma,d} + \frac{\Delta_{j,k+1}}{\Delta_{j,k}} \leq \Big( 1 + \frac{C_{\rho,\sigma,d}}{\beta} \Big) \frac{\Delta_{j,k+1}}{\Delta_{j,k}} \quad \text{for every } k = 1, \ldots, k_0.
$$

Recursively applying the above inequality we obtain,

$$
\frac{\Delta_{j,1}}{\Delta_{j,0}} \leq \Big( 1 + \frac{C_{\rho,\sigma,d}}{\beta} \Big)^k \frac{\Delta_{j,k+1}}{\Delta_{j,k}} \quad \text{for every } k = 0, \ldots, k_0.
$$

Taking the geometric mean of the above inequality for $k = 1, \ldots, k_0$, we obtain

$$
\begin{aligned}
\frac{\Delta_{j,1}}{\Delta_{j,0}} &\leq \Big( \prod_{k=0}^{k_0} \Big( 1 + \frac{C_{\rho,\sigma,d}}{\beta} \Big)^k \frac{\Delta_{i,k+1}}{\Delta_{i,k}} \Big)^{1/(k_0+1)} \\
&= \Big( 1 + \frac{C_{\rho,\sigma,d}}{\beta} \Big)^{k_0/2} \Delta_{j,k_0+1}^{1/(k_0+1)} \Delta_{j,0}^{-1/(k_0+1)},
\end{aligned}
$$

which implies that

$$\Delta_{j,1} \leq \left(1 + \frac{C_{\rho,\sigma,d}}{\beta}\right)^{k_0/2} \Delta_{j,k_0+1}^{1/(k_0+1)} \Delta_{j,0}^{k_0/(k_0+1)}.$$

Therefore, with $k = k_0 + 1$ and Eq. (24) we have

$$\Delta_{j,1} \leq \left(1 + \frac{C_{\rho,\sigma,d}}{\beta}\right)^{\frac{k_0}{2}} \left(\frac{4(2\pi\sigma^2)^{-d/2}}{\rho_n}\sigma^{-2k_0-1} \inf_{a \geq \sqrt{2k_0+1}} \left\{ a^{2k_0+2}\mathsf{H}^2(P_\sigma, Q_\sigma)\right.\right.$$

$$\left.\left. + a^{2k_0+1}e^{-a^2}\right\}\right)^{\frac{1}{2k_0+2}} \left(2\mathsf{H}^2(P_\sigma, Q_\sigma)\right)^{\frac{k_0}{2k_0+2}}$$

$$\leq \frac{1}{2}\left(\frac{(2\pi\sigma^2)^{-\frac{d}{2}}}{\rho_n}\right)^{\frac{1}{2k_0+2}} \left(1 + \frac{C_{\rho,\sigma,d}}{\beta}\right)^{\frac{k_0}{2}} \sigma^{-\frac{2k_0+1}{2k_0+2}} \inf_{a \geq \sqrt{2k_0+1}} \left(a^{2k_0+2}\mathsf{H}^2(P_\sigma, Q_\sigma)\right.$$

$$\left. + a^{2k_0+1}e^{-a^2}\right)^{\frac{1}{2k_0+2}} \mathsf{H}^{\frac{k_0}{k_0+1}}(P_\sigma \| Q_\sigma)$$

$$= \frac{1}{2}\left(\frac{(2\pi\sigma^2)^{-\frac{d}{2}}}{\rho_n}\right)^{\frac{1}{2k_0+2}} \left(1 + \frac{C_{\rho,\sigma,d}}{\beta}\right)^{\frac{k_0}{2}} \sigma^{-\frac{2k_0+1}{2k_0+2}} \inf_{a \geq \sqrt{2k_0+1}} a\left(\mathsf{H}^2(P_\sigma, Q_\sigma)\right.$$

$$\left. + e^{-a^2}\right)^{\frac{1}{2k_0+2}} \mathsf{H}^{\frac{k_0}{k_0+1}}(P_\sigma \| Q_\sigma) \tag{27}$$

**Choose $\beta, k_0, a$:**

- Choose $\beta = k_0 C_{\rho,\sigma,d}$, Eq. (26) becomes

$$\Delta_{j,1} \leq 2\sqrt{2}k_0 C_{\rho,\sigma,d}\mathsf{H}(P_\sigma, Q_\sigma). \tag{28}$$

and the term in Eq. (27),

$$\left(1 + \frac{C_{\rho,\sigma,d}}{\beta}\right)^{\frac{k_0}{2}} = \left(1 + \frac{1}{k_0}\right)^{\frac{k_0}{2}} \leq \sqrt{e}. \tag{29}$$

- Choose $k_0$ so that we have $\left(\frac{(2\pi\sigma^2)^{-\frac{d}{2}}}{\rho_n}\right)^{\frac{1}{2k_0+2}} \leq \sqrt{e}$ and thereby the term in Eq. (27) will be $\left(\frac{(2\pi\sigma^2)^{-\frac{d}{2}}}{\rho_n}\right)^{\frac{1}{2k_0+2}}\left(1 + \frac{C_{\rho,\sigma,d}}{\beta}\right)^{\frac{k_0}{2}} \leq e$. This requires that

$$\left(\frac{(2\pi\sigma^2)^{-d/2}}{\rho_n}\right)^{\frac{1}{2k_0+2}} = \exp\left(\frac{1}{2k_0+2}\log\left(\frac{(2\pi\sigma^2)^{-d/2}}{\rho_n}\right)\right) \leq \sqrt{e}. \tag{30}$$

Hence, for all $n \geq 1$, we can choose $k_0 \geq 1$ to be the smaller integer such that

$$\log\left(\frac{(2\pi\sigma^2)^{-d/2}}{\rho_n}\right) + 1 \leq k_0 \leq \log\left(\frac{(2\pi\sigma^2)^{-d/2}}{\rho_n}\right).$$

Consider the term in Eq. (28), we obtain,

$$2\sqrt{2}k_0 C_{\rho,\sigma,\sigma} \leq \frac{8}{\sigma}\left(\log\left(\frac{(2\pi\sigma^2)^{-d/2}}{\rho_n}\right)\right)^{3/2}$$

which gives that

$$\Delta_{k,1} \leq 2\sqrt{2}k_0 C_{\rho,\sigma,d} \leq \frac{8}{\sigma}\left(\log\left(\frac{(2\pi\sigma^2)^{-d/2}}{\rho_n}\right)\right)^{3/2}. \tag{31}$$

- Choose $a^2 = \max\{2k_0 + 1, -\log\left(\mathsf{H}^2(P_\sigma, Q_\sigma)\right)\}$. Notice that $a \geq 1$ and

$$e^{-a^2} \leq \mathsf{H}^2(P_\sigma, Q_\sigma).$$

Consider the term in Eq. (27), we have

$$\left(\mathsf{H}^2(P_\sigma, Q_\sigma) + e^{-a^2}\right)^{\frac{1}{2k_0+2}} \le \left(\mathsf{H}^2(P_\sigma, Q_\sigma) + \mathsf{H}^2(P_\sigma, Q_\sigma)\right)^{\frac{1}{2(k_0+1)}} \le 2\mathsf{H}^{\frac{1}{k_0+1}}(P_\sigma, Q_\sigma).$$
(32)

Combine Eq. (29), (30) and (32), we obtain that Eq. (27) is upper-bounded by

$$\Delta_{j,1} \le ea\sigma^{-\frac{2k_0+1}{2k_0+2}} \mathsf{H}(P_\sigma, Q_\sigma) \le ea \max\{\sigma^{-1}, 1\}\mathsf{H}(P_\sigma, Q_\sigma)$$
(33)

Combining Eq. (31) and (33) and $k_0 = \lfloor \log\left(\frac{(2\pi\sigma^2)^{-d/2}}{\rho_n}\right)\rfloor$, we obtain

$$\Delta_{j,1} \le \max\left\{\frac{8}{\sigma}\log^{3/2}\left(\frac{(2\pi\sigma^2)^{-d/2}}{\rho_n}\right), ea, ea\sigma^{-1}\right\}\mathsf{H}(P_\sigma, Q_\sigma).$$

Hence, by Eq. (20) we obtain

$$\text{(I-b)} + \text{(II-b)} \le 4\sum_{j=1}^d \Delta_{j,1}^2 \le 4d \max\left\{\frac{64}{\sigma^2}\log^3\left(\frac{(2\pi\sigma^2)^{-d/2}}{\rho_n}\right), e^2a^2, \frac{e^2a^2}{\sigma^2}\right\}\mathsf{H}^2(P_\sigma, Q_\sigma).$$
(34)

Therefore, by combining (I-a), (I-b), (II-a), (II-b), i.e., Eq. (21) and (34), we obtain

$$\int_{\mathbb{R}^d}\left\|\frac{\nabla p_\sigma(\boldsymbol{x})}{p_\sigma(\boldsymbol{x})} - \frac{\nabla q_\sigma(\boldsymbol{x})}{q_\sigma(\boldsymbol{x}) \vee \rho_n}\right\|_2^2 p_\sigma(\boldsymbol{x})\mathrm{d}\boldsymbol{x}$$

$$\le 4d \max\left\{\frac{64}{\sigma^2}\log^3\left(\frac{(2\pi\sigma^2)^{-d/2}}{\rho_n}\right), e^2\left(2\log\left(\frac{(2\pi\sigma^2)^{-d/2}}{\rho_n}\right) + 1\right), \frac{e^2}{\sigma^2}\log\left(\mathsf{H}^{-2}(P_\sigma, Q_\sigma)\right)\right\}\mathsf{H}^2(P_\sigma, Q_\sigma)$$

$$+ \frac{32}{\sigma^2}\log\left(\frac{(2\pi\sigma^2)^{-d/2}}{\rho_n}\right)\mathsf{H}^2(P_\sigma, Q_\sigma) + 32\int_{\mathcal{G}_2}\left\|\frac{\nabla p_\sigma(\boldsymbol{x})}{p_\sigma(\boldsymbol{x})}\right\|_2^2 p_\sigma(\boldsymbol{x})\mathrm{d}\boldsymbol{x}$$

$$\le \frac{288d}{\sigma^2}\max\left\{\log^3\left(\frac{(2\pi\sigma^2)^{-d/2}}{\rho_n}\right), \frac{1}{2}\log\left(\mathsf{H}^{-2}(P_\sigma, Q_\sigma)\right)\right\}\mathsf{H}^2(P_\sigma, Q_\sigma) + 32\int_{\mathcal{G}_2}\left\|\frac{\nabla p_\sigma(\boldsymbol{x})}{p_\sigma(\boldsymbol{x})}\right\|_2^2 p_\sigma(\boldsymbol{x})\mathrm{d}\boldsymbol{x}.$$

$$\square$$

# B Convergence of Smoothed Sub-Gaussian Distribution in KL Divergence

## B.1 Rényi divergence and Rényi mutual information

**Definition 2** (Rényi Divergence and Rényi Mutual Information [51]). *Assume random variables* $(\boldsymbol{X}, \boldsymbol{Y})$ *have joint distribution* $P_{\boldsymbol{XY}}$*. For any* $\lambda \in \mathbb{R} \setminus \{0, 1\}$*, the Rényi divergence of order* $\lambda$ *between probability distributions* $P$ *and* $Q$ *is defined as*

$$D_\lambda(P \| Q) := \frac{1}{\lambda - 1} \log \left( \mathbb{E}_Q \left[ \left( \frac{\mathrm{d}P}{\mathrm{d}Q} \right)^\lambda \right] \right).$$

*The Rényi Mutual Information of order* $\lambda$ *are defined as*

$$I_\lambda(\boldsymbol{X}; \boldsymbol{Y}) := D_\lambda(P_{\boldsymbol{X}, \boldsymbol{Y}} \| P_{\boldsymbol{X}} \otimes P_{\boldsymbol{Y}}),$$

*where* $P_{\boldsymbol{X}}, P_{\boldsymbol{Y}}$ *to denote the marginal distribution with respect to* $\boldsymbol{X}, \boldsymbol{Y}$*, and* $P_{\boldsymbol{X}} \otimes P_{\boldsymbol{Y}}$ *denotes the joint distribution of* $(\boldsymbol{X}', \boldsymbol{Y}')$ *where* $\boldsymbol{X}' \sim P_{\boldsymbol{X}}, \boldsymbol{Y}' \sim P_{\boldsymbol{Y}}$ *are independent to each other.*

The following lemma is a restatement of [42, Lemma 5], in which we explicitly demonstrate the dependence of the bound on the Gaussian parameter $\sigma$ and relieve the exponential dependence of $\sigma$.

**Lemma 6.** *Fix* $\sigma > 0$*, let* $\boldsymbol{X} \sim P$*,* $\boldsymbol{Z} \sim \mathcal{N}(0, \sigma^2 \boldsymbol{I}_d)$*,* $\boldsymbol{X} \perp \boldsymbol{Z}$ *and* $\boldsymbol{Y} = \boldsymbol{X} + \boldsymbol{Z}$*. Fix* $1 < \lambda < 2$*. If* $P$ *is a* $\alpha$*-sub-Gaussian distribution, we have,*

$$I_\lambda(\boldsymbol{X}; \boldsymbol{Y}) \leq \frac{1}{\lambda - 1} \left( \log \left( \frac{C_d}{(2 - \lambda)^{d/2}} \right) + d \log \frac{\alpha}{\sigma} \right). \tag{35}$$

*for some* $C_d > 0$ *depends only on* $d$*.*

*Proof.* By the definition of Rényi divergence Definition 2, we have

$$
\begin{aligned}
I_\lambda(\boldsymbol{X}; \boldsymbol{Y}) &= \frac{1}{\lambda - 1} \log \left( \mathbb{E}_{P_{\boldsymbol{X}} \otimes P_{\boldsymbol{Y}}} \left[ \left( \frac{\mathrm{d}P_{\boldsymbol{X}, \boldsymbol{Y}}}{\mathrm{d}(P_{\boldsymbol{X}} \otimes P_{\boldsymbol{Y}})} \right)^\lambda \right] \right) \\
&= \frac{1}{\lambda - 1} \log \left( \mathbb{E}_{P_{\boldsymbol{X}} \otimes P_{\boldsymbol{Y}}} \left[ \left( \frac{\mathrm{d}P_{\boldsymbol{Y}|\boldsymbol{X}}}{\mathrm{d}P_{\boldsymbol{Y}}} \right)^\lambda \right] \right) \\
&= \frac{1}{\lambda - 1} \log \left( C \mathbb{E}_{P_{\boldsymbol{X}} \otimes P_{\boldsymbol{Y}}} \left[ \frac{\varphi_\sigma^\lambda(\boldsymbol{Y} - \boldsymbol{X})}{\left( \mathbb{E}_{\boldsymbol{X}}[\varphi_\sigma(\boldsymbol{Y} - \boldsymbol{X})] \right)^\lambda} \right] \right), \\
&\qquad\qquad\qquad\qquad\qquad\qquad (\text{by } \boldsymbol{Y} = \boldsymbol{X} + \sigma \boldsymbol{Z}, \boldsymbol{Z} \sim \mathcal{N}(0, \boldsymbol{I}_d))
\end{aligned}
$$

for some positive constant $C > 0$. Therefore, we only need to upper-bound $\mathbb{E}_{\boldsymbol{X}} \left[ \int_{\mathbb{R}^d} \frac{\varphi_\sigma^\lambda(\boldsymbol{Y} - \boldsymbol{X})}{(\mathbb{E}_{\boldsymbol{X}}[\varphi_\sigma(\boldsymbol{Y} - \boldsymbol{X})])^\lambda} \mathrm{d}y \right]$ for sub-Gaussian distributions.

Decompose $\mathbb{R}^d = \bigcup_i c_i$ as a union of cubes of diameter $2\sigma$. For any $\boldsymbol{X} \in c_i$, we have

$$\mathbb{E}_{\boldsymbol{X}} \left[ \exp \left( -\frac{\|\boldsymbol{Y} - \boldsymbol{X}\|_2^2}{2\sigma^2} \right) \right] \geq \Pr(\boldsymbol{X} \in c_i) \cdot \mathbb{E}_{\boldsymbol{X}} \left[ \exp \left( -\frac{\|\boldsymbol{Y} - \boldsymbol{X}\|_2^2}{2\sigma^2} \right) \Big| \boldsymbol{X} \in c_i \right].$$

Fix any (non-random) $\boldsymbol{X}' \in c_i$, by $\|\boldsymbol{X} - \boldsymbol{X}'\|_2 \leq 2\sigma$ for all $\boldsymbol{X}, \boldsymbol{X}' \in c_i$, we have

$$
\begin{aligned}
\|\boldsymbol{Y} - \boldsymbol{X}\|_2^2 &\leq \left( \|\boldsymbol{Y} - \boldsymbol{X}'\|_2 + \|\boldsymbol{X}' - \boldsymbol{X}\|_2 \right)^2 \\
&\leq \|\boldsymbol{Y} - \boldsymbol{X}'\|_2^2 + 4\sigma \|\boldsymbol{Y} - \boldsymbol{X}'\|_2 + 4\sigma^2 \\
&= \frac{3}{2} \|\boldsymbol{Y} - \boldsymbol{X}'\|_2^2 - \frac{1}{2} (\|\boldsymbol{Y} - \boldsymbol{X}'\|_2 + 4\sigma)^2 + 12\sigma^2 \\
&\leq \frac{3}{2} \|\boldsymbol{Y} - \boldsymbol{X}'\|_2^2 + 12\sigma^2.
\end{aligned}
$$

Then, for any $\boldsymbol{X}, \boldsymbol{X}' \in c_i$, we obtain

$$
\begin{aligned}
\mathbb{E}_{\boldsymbol{X}} \Big[ \exp\Big( -\frac{\|\boldsymbol{Y} - \boldsymbol{X}\|_2^2}{2\sigma^2} \Big) \Big] &\geq \Pr(\boldsymbol{X} \in c_i) \cdot \mathbb{E}_{\boldsymbol{X}} \Big[ \exp\Big( -\frac{3\|\boldsymbol{Y} - \boldsymbol{X}'\|_2^2}{4\sigma^2} - \frac{6\sigma^2}{\sigma^2} \Big) \Big| \boldsymbol{X} \in c_i \Big] \\
&= \exp(-6) \cdot \Pr(\boldsymbol{X} \in c_i) \cdot \mathbb{E}_{\boldsymbol{X}} \Big[ \exp\Big( -\frac{3\|\boldsymbol{Y} - \boldsymbol{X}'\|_2^2}{4\sigma^2} \Big) \Big] \\
&= \exp(-6) \cdot \Pr(\boldsymbol{X} \in c_i) \cdot \exp\Big( -\frac{3\|\boldsymbol{Y} - \boldsymbol{X}'\|_2^2}{4\sigma^2} \Big), \qquad (36)
\end{aligned}
$$

which indicates that

$$
\begin{aligned}
\frac{\exp\Big( -\frac{\lambda \|\boldsymbol{Y} - \boldsymbol{X}\|_2^2}{2\sigma^2} \Big)}{(\mathbb{E}_{\boldsymbol{X}}[\varphi_\sigma(\boldsymbol{Y} - \boldsymbol{X})])^{\lambda-1}} &\leq \exp\Big( 6(\lambda - 1) \Big) \Pr(\boldsymbol{X} \in c_i)^{1-\lambda} \exp\Big( -\frac{(3-\lambda)\|\boldsymbol{Y} - \boldsymbol{X}\|_2^2}{4\sigma^2} \Big) \\
&\leq \exp(6) \Pr(\boldsymbol{X} \in c_i)^{1-\lambda} \exp\Big( -\frac{\|\boldsymbol{Y} - \boldsymbol{X}\|_2^2}{4\sigma^2} \Big). \qquad (1 \leq \lambda \leq 2)
\end{aligned}
$$

Therefore, for any $\boldsymbol{X} \in \mathbb{R}^d$ we have

$$
\begin{aligned}
\int_{\mathbb{R}^d} \frac{\exp\Big( -\frac{\lambda \|\boldsymbol{Y} - \boldsymbol{X}\|_2^2}{2\sigma^2} \Big)}{(\mathbb{E}_{\boldsymbol{X}}[\varphi_\sigma(\boldsymbol{Y} - \boldsymbol{X})])^{\lambda-1}} \mathrm{d}y &\leq \int_{\mathbb{R}^d} \exp(6) \Pr(\boldsymbol{X} \in c_i)^{1-\lambda} \exp\Big( -\frac{\|\boldsymbol{Y} - \boldsymbol{X}\|_2^2}{4\sigma^2} \Big) \mathrm{d}y \\
&= \exp(6) \Pr(\boldsymbol{X} \in c_i)^{1-\lambda} \int_{\mathbb{R}^d} \exp\Big( -\frac{\|\boldsymbol{Y} - \boldsymbol{X}\|_2^2}{4\sigma^2} \Big) \mathrm{d}y \\
&\leq \exp(6)(4\pi\sigma^2)^{d/2} \Pr(\boldsymbol{X} \in c_i)^{1-\lambda}.
\end{aligned}
$$

Taking expectation over $\boldsymbol{X}$, we obtain

$$
\begin{aligned}
\mathbb{E}_{\boldsymbol{X}} \Big[ \int_{\mathbb{R}^d} \frac{\varphi_\sigma^\lambda(\boldsymbol{Y} - \boldsymbol{X})}{(\mathbb{E}_{\boldsymbol{X}}[\varphi_\sigma(\boldsymbol{Y} - \boldsymbol{X})])^{\lambda-1}} \mathrm{d}y \Big] &\leq \sum_i \Pr(\boldsymbol{X} \in c_i) \cdot \int_{\mathbb{R}^d} \frac{\varphi_\sigma^\lambda(\boldsymbol{Y} - \boldsymbol{X})}{(\mathbb{E}_{\boldsymbol{X}}[\varphi_\sigma(\boldsymbol{Y} - \boldsymbol{X})])^{\lambda-1}} \mathrm{d}y \\
&= \sum_i \Pr(\boldsymbol{X} \in c_i) \cdot \int_{\mathbb{R}^d} \frac{(2\pi\sigma^2)^{-d/2} \exp\Big( -\frac{\lambda\|\boldsymbol{Y} - \boldsymbol{X}\|_2^2}{2\sigma^2} \Big)}{(\mathbb{E}_{\boldsymbol{X}}[\varphi_\sigma(\boldsymbol{Y} - \boldsymbol{X})])^{\lambda-1}} \mathrm{d}y \\
&\leq \exp(6) 2^{d/2} \sum_i \Pr(\boldsymbol{X} \in c_i)^{2-\lambda}.
\end{aligned}
$$

Let $\mathcal{C}_r := \{c_i | (r-1)\sigma \leq \|s_i\|_2 < r\sigma\}$ denote the set of cubes whose centers $s_i$ belong to $\{(r-1)\sigma \leq \|s_i\|_2 < r\sigma\}$. Then we have $|\mathcal{C}_r| = C_d r^{d-1}$. We further let $P_{\mathcal{C}_r} := \sum_{c_i \in \mathcal{C}_r} \Pr(\boldsymbol{X} \in c_i)$.

If $P$ is $\alpha$-sub-Gaussian, we have for all $c_i \in \mathcal{C}_r$,

$$
\begin{aligned}
\Pr(\boldsymbol{X} \in c_i) &\leq \Pr(\|\boldsymbol{X} - s_i\|_2 \leq \sigma) && \text{(by } c_i \text{ is a cube of diameter } 2\sigma) \\
&\leq \Pr(|\|s_i\|_2 - \sigma| \leq \|\boldsymbol{X}\|_2 \leq \|s_i\|_2 + \sigma) \\
&\leq \Pr(|\|s_i\|_2 - \sigma| \leq \|\boldsymbol{X}\|_2) \\
&\leq C \exp\Big( -\frac{r^2\sigma^2}{\alpha^2} \Big), && \text{(by } \{(r-1)\sigma \leq \|s_i\|_2 < r\sigma\} \text{ and } P \text{ is } \alpha\text{-sub-Gaussian)}
\end{aligned}
$$

which gives that

$$\sum_{i=1}^{\infty} \Pr(\boldsymbol{X} \in c_i)^{2-\lambda} = \sum_{r=1}^{\infty} \sum_{c_i \in \mathcal{C}_r} \Pr(\boldsymbol{X} \in c_i)^{2-\lambda} \leq \sum_{r=1}^{\infty} C |\mathcal{C}_r| \exp\left(-\frac{(2-\lambda)r^2\sigma^2}{\alpha^2}\right)$$

$$\leq \sum_{r=1}^{\infty} C_d r^{d-1} \exp\left(-\frac{(2-\lambda)r^2\sigma^2}{\alpha^2}\right)$$

$$\leq C_d \int_0^{\infty} r^{d-1} e^{-\frac{(2-\lambda)\sigma^2}{2\alpha^2}r^2} \, \mathrm{d}r$$

$$= C_d \Gamma(d/2) \left(\frac{(2-\lambda)\sigma^2}{\alpha^2}\right)^{-d/2}$$

$$\text{(by } \int_0^{\infty} r^d \exp(-ar^2)\mathrm{d}r = \frac{\Gamma(\frac{d+1}{2})}{2a^{\frac{d+1}{2}}} \text{ with } a > 0, \Gamma(d) := (d-1)!\text{)}$$

$$= C_d'' \left(\frac{\alpha^2}{(2-\lambda)\sigma^2}\right)^{d/2}.$$

Hence, we obtain for all $1 < \lambda \leq 2$,

$$I_\lambda(\boldsymbol{X};\boldsymbol{Y}) \leq \frac{1}{\lambda-1} \log\left(C_d'' 2^{d/2} \exp(6) \sum_i \Pr(\boldsymbol{X} \in c_i)^{2-\lambda}\right)$$

$$\leq \frac{1}{\lambda-1} \log\left(C_d''' \left(\frac{\alpha^2}{(2-\lambda)\sigma^2}\right)^{d/2}\right)$$

$$= \frac{1}{\lambda-1}\left(\log\left(\frac{C_d'''}{(2-\lambda)^{d/2}}\right) + d\log\frac{\alpha}{\sigma}\right).$$

$\square$

## B.2   Proof of Lemma 3

The following lemma is a restatement of [42, Theorem 3], in which we explicitly demonstrate the dependence of the bound on the sub-Gaussian parameter $\alpha$ while removing the dependence on the Gaussian smoothing parameter $\sigma$.

**Lemma 3.** (Empirical Convergence of Gaussian Smoothed Sub-Gaussian Distributions in KL-Divergence) *Given* $d \geq 1, n \geq 3, \sigma > 0$. *Suppose that* $P$ *is a* $d$-dimensional $\alpha$-sub-Gaussian *distribution. Let* $P_\sigma = P * \mathcal{N}(0, \sigma^2 \boldsymbol{I}_d)$ *and* $\hat{P}$ *be the empirical measure of an i.i.d. sample of size* $n$ *drawn from* $P$ *and* $\hat{P}_\sigma = \hat{P} * \mathcal{N}(0, \sigma^2 \boldsymbol{I}_d)$. *Then we have*

$$\mathbb{E}_{P^{\otimes n}}\left[\mathrm{KL}(\hat{P}_\sigma \| P_\sigma^*)\right] \leq C_d\left(\frac{\alpha}{\sigma}\right)^d \frac{\log^{d/2} n}{n}.$$

*Proof.* Let $\boldsymbol{X} \sim P, \boldsymbol{Z} \sim \mathcal{N}(0, \sigma^2 \boldsymbol{I}_d), \boldsymbol{X} \perp \boldsymbol{Z}$ and $\boldsymbol{Y} = \boldsymbol{X} + \boldsymbol{Z}$. Then, we have $\boldsymbol{Y}|\boldsymbol{X} \sim \mathcal{N}(\boldsymbol{X}, \sigma^2 \boldsymbol{I}_d)$, which indicates that $P_{\boldsymbol{Y}|\boldsymbol{X}} \cdot \hat{P} \sim \hat{P} * \mathcal{N}(0, \sigma^2 \boldsymbol{I}_d)$. Therefore, adopting [42, Lemma 4] and Lemma 6, we obtain that for any $1 < \lambda < 2$,

$$\mathbb{E}_{P^{\otimes n}}\left[\mathrm{KL}(\hat{P}_\sigma \| P_\sigma)\right] \leq \frac{1}{\lambda-1} \log(1 + \exp((\lambda-1)(I_\lambda(\boldsymbol{X};\boldsymbol{Y}) - \log n)))$$

$$\leq \frac{1}{\lambda-1} \log\left(1 + \exp\left(\log\left(\frac{C_d}{(2-\lambda)^{d/2}} \cdot \left(\frac{\alpha}{\sigma}\right)^d\right) - (\lambda-1)\log n\right)\right)$$

$$\text{(by Lemma 6)}$$

$$= \frac{1}{\lambda-1} \log\left(1 + \frac{C_d n^{-(\lambda-1)}}{(2-\lambda)^{d/2}} \cdot \left(\frac{\alpha}{\sigma}\right)^d\right)$$

$$\leq \frac{C_d}{(\lambda-1)n^{\lambda-1}(2-\lambda)^{d/2}} \left(\frac{\alpha}{\sigma}\right)^d. \qquad \text{(by } \log(1+x) \leq x, \forall x > 0\text{)}$$

Choosing $\lambda = 2 - \frac{1}{\log n}$, by $n \geq 3$ we have $1 < \lambda < 2$ and

$$n^{\lambda - 1} = n^{-\frac{1}{\log n} + 1} = n \cdot \exp\left(-\log n \cdot \frac{1}{\log n}\right) = \frac{n}{e}. \tag{37}$$

We have

$$\mathbb{E}_{P^{\otimes n}}\left[\mathrm{KL}(\hat{P}_\sigma \| P_\sigma)\right] \leq \frac{C_d e (\log n)^{d/2}}{(1 - 1/\log n)n}\left(\frac{\alpha}{\sigma}\right)^d \leq C_d'\left(\frac{\alpha}{\sigma}\right)^d \frac{\log^{d/2} n}{n}.$$

$\square$

# C   Score Estimation by Regularized Empirical Score Functions

## C.1   Sub-Gaussian tail bound for score functions

The following Lemma follows a similar proof from [30, Lemma 5], while our results do not require
the assumption that the parameters satisfy $\sigma \leq \alpha$:

**Lemma 7** (Sub-Gaussian Tail Bounds for Score Functions). *Given a $\alpha$-sub-Gaussian distribution
$P$, let $P_\sigma := P * \mathcal{N}(0, \sigma^2 \boldsymbol{I}_d)$ with density function $p_\sigma$. Fix $0 < \rho \leq (2\pi\sigma^2)^{-d/2} e^{-1}$ let $\mathcal{G} := \{\boldsymbol{x} \in \mathbb{R}^d : p_\sigma(\boldsymbol{x}) \leq \rho\}$. Then,*

$$\int_{\mathcal{G}} \left\| \frac{\nabla p_\sigma(\boldsymbol{x})}{p_\sigma(\boldsymbol{x})} \right\|_2^2 p_\sigma(\boldsymbol{x}) \mathrm{d}\boldsymbol{x} \leq \frac{2\rho}{\sigma^2} \log\left( \frac{(2\pi\sigma^2)^{-\frac{d}{2}}}{\rho} \right) \left( 32(\alpha^2 + \sigma^2) \log n \right)^{\frac{d}{2}} + \frac{2d^{3/2}}{n^2\sigma^2}.$$

*Proof.* For some $A > 0$, set $\mathcal{A} = \mu + [-A, A]^d$, where $\mu = \mathbb{E}_{\boldsymbol{X} \sim P_\sigma}[\boldsymbol{X}]$. Then we have

$$\int_{\mathcal{G}} \left\| \frac{\nabla p_\sigma(\boldsymbol{x})}{p_\sigma(\boldsymbol{x})} \right\|_2^2 p_\sigma(\boldsymbol{x}) \mathrm{d}\boldsymbol{x}$$

$$= \int_{\mathcal{G} \cap \mathcal{A}} \left\| \frac{\nabla p_\sigma(\boldsymbol{x})}{p_\sigma(\boldsymbol{x})} \right\|_2^2 p_\sigma(\boldsymbol{x}) \mathrm{d}\boldsymbol{x} + \int_{\mathcal{G} \cap \mathcal{A}^c} \left\| \frac{\nabla p_\sigma(\boldsymbol{x})}{p_\sigma(\boldsymbol{x})} \right\|_2^2 p_\sigma(\boldsymbol{x}) \mathrm{d}\boldsymbol{x}$$

$$\leq \underbrace{\int_{\mathcal{A}} \left\| \frac{\nabla p_\sigma}{p_\sigma} \right\|_2^2 p_\sigma(\boldsymbol{x}) \, \mathbb{1}\{p_\sigma(\boldsymbol{x}) \leq \rho\} \mathrm{d}\boldsymbol{x}}_{:= \text{(I)}} + \underbrace{\int_{\mathcal{A}^c} \left\| \frac{\nabla p_\sigma}{p_\sigma} \right\|_2^2 p_\sigma(\boldsymbol{x}) \, \mathbb{1}\{p_\sigma(\boldsymbol{x}) \leq \rho\} \mathrm{d}\boldsymbol{x}}_{:= \text{(II)}}.$$

Then by the sub-Gaussian tail bound of $P_\sigma$ with parameter $\sqrt{\alpha^2 + \sigma^2}$,

$$\Pr[\boldsymbol{X} \notin \mathcal{A}] = \int_{\mathcal{A}^c} p_\sigma(\boldsymbol{x}) \mathrm{d}\boldsymbol{x} \leq 2d \exp\left( -\frac{A^2}{2(\alpha^2 + \sigma^2)} \right). \tag{38}$$

Let $\boldsymbol{X}_0 \sim P, \boldsymbol{X} \sim P_\sigma, \boldsymbol{Z} \sim \mathcal{N}(0, \boldsymbol{I}_d)$ and we have $\boldsymbol{X}|\boldsymbol{X}_0 \sim \mathcal{N}(\boldsymbol{X}_0, \sigma^2 \boldsymbol{I}_d)$.

$$\text{(II)} \leq \int_{\mathcal{A}^c} \left\| \frac{\nabla p_\sigma(\boldsymbol{x})}{p_\sigma(\boldsymbol{x})} \right\|_2^2 p_\sigma(\boldsymbol{x}) \mathrm{d}\boldsymbol{x} \tag{39}$$

$$= \int_{\mathcal{A}^c} \left\| \frac{1}{\sigma^2} \mathbb{E}[\boldsymbol{X}_0 - \boldsymbol{X}|\boldsymbol{X} = \boldsymbol{x}] \right\|_2^2 p_\sigma(\boldsymbol{x}) \mathrm{d}\boldsymbol{x} \qquad \text{(by Tweedie's formula Eq. (15))}$$

$$= \frac{1}{\sigma^2} \mathbb{E}\left[ \|\mathbb{E}[\boldsymbol{Z}|\boldsymbol{X}]\|_2^2 \, \mathbb{1}\{\boldsymbol{X} \notin \mathcal{A}\} \right] \qquad \text{(by } \boldsymbol{X} = \boldsymbol{X}_0 + \sigma\boldsymbol{Z})$$

$$\leq \frac{1}{\sigma^2} \mathbb{E}\left[ \mathbb{E}[\|\boldsymbol{Z}\|_2^2|\boldsymbol{X}] \, \mathbb{1}\{\boldsymbol{X} \notin \mathcal{A}\} \right] \qquad \text{(by Jensen's inequality)}$$

$$\leq \frac{1}{\sigma^2} \sqrt{\mathbb{E}[\|\boldsymbol{Z}\|_2^4] \Pr[\boldsymbol{X} \notin \mathcal{A}]} \qquad \text{(by Cauchy-Schswarz)}$$

$$\leq \frac{1}{\sigma^2} \sqrt{(2d + d^2) 2d \exp\left( -\frac{A^2}{2(\alpha^2 + \sigma^2)} \right)} \qquad \text{(by Eq. (38) and } \mathbb{E}[\|\boldsymbol{Z}\|_2^4] = 2d + d^2)$$

$$\leq \frac{2d^{3/2}}{\sigma^2} \sqrt{\exp\left( -\frac{A^2}{2(\alpha^2 + \sigma^2)} \right)}. \tag{40}$$

Let $A = \sqrt{8(\alpha^2 + \sigma^2) \log n}$, then

$$\Pr[\boldsymbol{X} \notin \mathcal{A}] \leq 2d \exp\left( -\frac{8(\alpha^2 + \sigma^2) \log n}{2(\alpha^2 + \sigma^2)} \right) = 2dn^{-4},$$

which gives that

$$\text{(II)} \leq \frac{2d^{3/2}}{n^2\sigma^2}. \tag{41}$$

By Lemma 5 and notice that $x \log(\frac{c}{x})$ is monotonously increasing on $[0, \frac{c}{e}]$, we have

$$\text{(I)} \leq \int_{\mathcal{A}} \frac{4}{\sigma^2} \log\Big(\frac{(2\pi\sigma^2)^{-d/2}}{p_\sigma(\boldsymbol{x})}\Big) p_\sigma(\boldsymbol{x}) \, \mathbb{1}\{p_\sigma(\boldsymbol{x}) \leq \rho\} \mathrm{d}\boldsymbol{x} \qquad \text{(by Lemma 5)}$$

$$\leq \frac{4}{\sigma^2} \int_{\mathcal{A}} \log\Big(\frac{(2\pi\sigma^2)^{-d/2}}{\rho}\Big) \rho \mathrm{d}\boldsymbol{x} \qquad \text{(by } p_\sigma(\boldsymbol{x}) \leq \rho \leq (2\pi\sigma^2)^{-d/2}e^{-1})$$

$$= \frac{4\rho}{\sigma^2} \log\Big(\frac{(2\pi\sigma^2)^{-d/2}}{\rho}\Big) (2A)^d$$

$$= \frac{4\rho}{\sigma^2} \log\Big(\frac{(2\pi\sigma^2)^{-d/2}}{\rho}\Big) \Big(32(\alpha^2 + \sigma^2)\log n\Big)^{d/2}. \qquad (42)$$

Combine Eq. (41) and (42) we obtain

$$\int_{\mathcal{G}} \Big\|\frac{\nabla p_\sigma(\boldsymbol{x})}{p_\sigma(\boldsymbol{x})}\Big\|_2^2 p_\sigma(\boldsymbol{x}) \mathrm{d}\boldsymbol{x} \leq \frac{2\rho}{\sigma^2} \log\Big(\frac{(2\pi\sigma^2)^{-d/2}}{\rho}\Big)\Big(32(\alpha^2 + \sigma^2)\log n\Big)^{d/2} + \frac{2d^{3/2}}{n^2\sigma^2}.$$

$\square$

### C.2 Score estimation error for sub-Gaussian distributions

**Lemma 1** *For any $d \geq 1, n \geq 3$, Let $P$ be a $\alpha$-sub-Gaussian distribution on $\mathbb{R}^d$ and $\hat{P}$ be its empirical distribution associated to a sample $\{\boldsymbol{x}^{(i)}\}_{i=1}^n$. For any $\sigma \gtrsim \alpha n^{-1/d}\log^{1/2} n$, let $P_\sigma = P * \mathcal{N}(0, \sigma^2 \boldsymbol{I}_d), \hat{P}_\sigma^{(n)} = \hat{P}^{(n)} * \mathcal{N}(0, \sigma^2 \boldsymbol{I}_d)$ with density functions $p_\sigma, \hat{p}_\sigma : \mathbb{R}^d \to \mathbb{R}_+$. Fix $0 < \rho_n \leq (2\pi\sigma^2)^{-d/2}e^{-1}n^{-1}$, then we have*

$$\mathbb{E}_{\{\boldsymbol{x}^{(i)}\}_{i=1}^n}\Big[\int_{\mathbb{R}^d} \Big\|\frac{\nabla p_\sigma(\boldsymbol{x})}{p_\sigma(\boldsymbol{x})} - \frac{\nabla \hat{p}_\sigma(\boldsymbol{x})}{\hat{p}_\sigma(\boldsymbol{x}) \vee \rho_n}\Big\|_2^2 p_\sigma(\boldsymbol{x})\mathrm{d}\boldsymbol{x}\Big] \lesssim \sigma^{-d-2}(\sigma^d + \alpha^d)\log^3\Big(\frac{(2\pi\sigma^2)^{-\frac{d}{2}}}{\rho_n}\Big)\frac{\log^{d/2} n}{n}.$$

*Proof.* By Lemma 2, we have

$$\int_{\mathbb{R}^d} \Big\|\frac{\nabla p_\sigma(\boldsymbol{x})}{p_\sigma(\boldsymbol{x})} - \frac{\nabla \hat{p}_\sigma(\boldsymbol{x})}{\hat{p}_\sigma(\boldsymbol{x}) \vee \rho_n}\Big\|_2^2 p_\sigma(\boldsymbol{x})\mathrm{d}\boldsymbol{x}$$

$$\leq Cd\sigma^{-2}\Big(\log^3\Big(\frac{(2\pi\sigma^2)^{-d/2}}{\rho_n}\Big) \vee \log\big(\mathsf{H}^{-2}(P_\sigma, Q_\sigma)\big)\Big)\mathsf{H}^2(P_\sigma, \hat{P}_\sigma) + 32\int_{\mathcal{G}_2}\Big\|\frac{\nabla p_\sigma(\boldsymbol{x})}{p_\sigma(\boldsymbol{x})}\Big\|_2^2 p_\sigma(\boldsymbol{x})\mathrm{d}\boldsymbol{x}.$$

By Lemma 3, for all $\sigma > 0, n \geq 3, d \geq 1$,

$$\mathbb{E}_{P^{\otimes n}}[\mathsf{H}^2(P_\sigma, \hat{P}_\sigma)] \leq \mathbb{E}_{P^{\otimes n}}[\mathrm{KL}(P_\sigma \| \hat{P}_\sigma)] \leq C_d\Big(\frac{\alpha}{\sigma}\Big)^d \frac{\log^{d/2} n}{n}.$$

Notice that $x \mapsto x \log x^{-1}$ is concave and by Jensen's inequality we have

$$\mathbb{E}_{P^{\otimes n}}\Big[\log\big(\mathsf{H}^{-2}(P_\sigma, \hat{P}_\sigma^{(n)})\big)\mathsf{H}^2(P_\sigma, \hat{P}_\sigma^{(n)})\Big] \leq \log\Big(\frac{1}{\mathbb{E}_{P^{\otimes n}}\big[\mathsf{H}^2(P_\sigma, \hat{P}_\sigma^{(n)})\big]}\Big)\mathbb{E}_{P^{\otimes n}}\big[\mathsf{H}^2(P_\sigma, \hat{P}_\sigma^{(n)})\big].$$

Note that $x \mapsto x \log(x^{-1})$ is increasing in $(0, e^{-1})$. If $C_d \alpha^d \sigma^{-d} n^{-1}\log^{d/2} n \leq e^{-1}$, which can be satisfied when $(\alpha/\sigma)^d \lesssim n\log^{-d/2} n$, i.e., $\sigma \gtrsim \alpha n^{-1/d}\log^{1/2} n$. Therefore, we obtain

$$\log\Big(\mathbb{E}_{P^{\otimes n}}\big[\mathsf{H}^{-2}(P_\sigma, \hat{P}_\sigma^{(n)})\big]\Big)\mathbb{E}_{P^{\otimes n}}\big[\mathsf{H}^2(P_\sigma, \hat{P}_\sigma^{(n)})\big]$$

$$\leq \log\Big(\frac{n}{C_d\log^{d/2} n} \cdot \Big(\frac{\sigma}{\alpha}\Big)^d\Big)C_d\Big(\frac{\alpha}{\sigma}\Big)^d\frac{\log^{d/2} n}{n}$$

$$= \Big(\log(C_d^{-1}n\log^{-d/2} n)\Big)C_d\Big(\frac{\alpha}{\sigma}\Big)^d\frac{\log^{d/2} n}{n} + C_d\Big(\frac{\alpha}{\sigma}\Big)^d\log\Big(\frac{\sigma}{\alpha}\Big)^d\frac{\log^{d/2} n}{n}$$

$$\leq C_d'\log n\Big(\frac{\alpha}{\sigma}\Big)^d\frac{\log^{d/2} n}{n} + \frac{1}{2}C_d\frac{\log^{d/2} n}{n} \qquad (\tfrac{1}{x}\log(x) \leq 1/2, \forall x > 0)$$

$$\leq C_d'\Big(\frac{\alpha}{\sigma}\Big)^d\frac{\log^{d/2+1} n}{n}.$$

Hence, for any $d \geq 1, n \geq 3, \sigma \gtrsim \alpha n^{-1/d} \log^{1/2} n$,

$$\mathbb{E}_{\{\boldsymbol{x}^{(i)}\}_{i=1}^n \sim P^{\otimes n}} \left[ \int_{\mathbb{R}^d} \left\| \frac{\nabla p_\sigma(\boldsymbol{x})}{p_\sigma(\boldsymbol{x})} - \frac{\nabla \hat{p}_\sigma(\boldsymbol{x})}{\hat{p}_\sigma(\boldsymbol{x}) \vee \rho_n} \right\|_2^2 p_\sigma(\boldsymbol{x}) \mathrm{d}\boldsymbol{x} \right]$$

$$\leq C d \sigma^{-2} \left( \log^3 \left( \frac{(2\pi\sigma^2)^{-d/2}}{\rho_n} \right) \mathbb{E}_{P^{\otimes n}} [\mathsf{H}^2(P_\sigma, \hat{P}_\sigma^{(n)})] + \mathbb{E}_{P^{\otimes n}} \left[ \log \left( \mathsf{H}^{-2}(P_\sigma, \hat{P}_\sigma^{(n)}) \right) \mathsf{H}^2(P_\sigma, \hat{P}_\sigma^{(n)}) \right] \right)$$

$$+ 32 \int_{\mathcal{G}_2} \left\| \frac{\nabla p_\sigma(\boldsymbol{x})}{p_\sigma(\boldsymbol{x})} \right\|_2^2 p_\sigma(\boldsymbol{x}) \mathrm{d}\boldsymbol{x}$$

$$\lesssim \sigma^{-2} \left( \frac{\alpha}{\sigma} \right)^d \left( \log^3 \left( \frac{(2\pi\sigma^2)^{-d/2}}{\rho_n} \right) + \log n \right) \frac{\log^{d/2} n}{n} + \int_{\mathcal{G}_2} \left\| \frac{\nabla p_\sigma(\boldsymbol{x})}{p_\sigma(\boldsymbol{x})} \right\|_2^2 p_\sigma(\boldsymbol{x}) \mathrm{d}\boldsymbol{x}$$

$$\lesssim \sigma^{-2} \left( \frac{\alpha}{\sigma} \right)^d \log^3 \left( \frac{(2\pi\sigma^2)^{-d/2}}{\rho_n} \right) \frac{\log^{d/2} n}{n} + \int_{\mathcal{G}_2} \left\| \frac{\nabla p_\sigma(\boldsymbol{x})}{p_\sigma(\boldsymbol{x})} \right\|_2^2 p_\sigma(\boldsymbol{x}) \mathrm{d}\boldsymbol{x}.$$
$$\text{(by } 0 < \rho_n \leq (2\pi\sigma^2)^{-d/2} e^{-1} n^{-1})$$

Notice that $x \log \left( \frac{(2\pi\sigma^2)^{-d/2}}{x} \right)$ is nondecreasing on $x \in [0, (2\pi\sigma^2)^{-d/2} e^{-1}]$. By sub-Gaussian tail bound Lemma 7, for all $d \geq 1, n \geq 3, \sigma \gtrsim \alpha n^{-1/d} \log^{1/2} n$ and $0 < \rho_n \leq (2\pi\sigma^2)^{-d/2} e^{-1} n^{-1}$, we obtain

$$\mathbb{E}_{\{\boldsymbol{x}^{(i)}\}_{i=1}^n \sim P^{\otimes n}} \left[ \int_{\mathbb{R}^d} \left\| \frac{\nabla p_\sigma(\boldsymbol{x})}{p_\sigma(\boldsymbol{x})} - \frac{\nabla \hat{p}_\sigma(\boldsymbol{x})}{\hat{p}_\sigma(\boldsymbol{x}) \vee \rho_n} \right\|_2^2 p_\sigma(\boldsymbol{x}) \mathrm{d}\boldsymbol{x} \right]$$

$$\lesssim \alpha^d \sigma^{-d-2} \log^3 \left( \frac{(2\pi\sigma^2)^{-d/2}}{\rho_n} \right) \frac{\log^{d/2} n}{n} + \frac{\rho_n}{\sigma^2} \log \left( \frac{(2\pi\sigma^2)^{-d/2}}{\rho_n} \right) \left( (\alpha^2 + \sigma^2) \log n \right)^{d/2} + \frac{d^{3/2}}{n^2 \sigma^2}$$

$$\lesssim \alpha^d \sigma^{-d-2} \log^3 \left( \frac{(2\pi\sigma^2)^{-d/2}}{\rho_n} \right) \frac{\log^{d/2} n}{n} + \sigma^{-d-2} (\log n + 1) (\alpha^2 + \sigma^2)^{d/2} \frac{\log^{d/2} n}{n}$$
$$\text{(by } 0 < \rho_n \leq (2\pi\sigma^2)^{-d/2} e^{-1} n^{-1})$$

$$\lesssim \alpha^d \sigma^{-d-2} \log^3 \left( \frac{(2\pi\sigma^2)^{-d/2}}{\rho_n} \right) \frac{\log^{d/2} n}{n} + \sigma^{-d-2} (\alpha^d + \sigma^d) \frac{\log^{d/2+1} n}{n}$$

$$\lesssim \sigma^{-d-2} (\alpha^d + \sigma^d) \log^3 \left( \frac{(2\pi\sigma^2)^{-d/2}}{\rho_n} \right) \frac{\log^{d/2} n}{n}.$$

$$\square$$

## C.3  Score estimations along OU process by regularized empirical score functions

**Theorem 4.** *Suppose the target distribution $P_0$ satisfies Assumption 1 and let $\hat{P}_0$ be the empirical distribution associated to a sample $\{\boldsymbol{x}^{(i)}\}_{i=1}^n$. For any $d \geq 1, n \geq 3$ and any $\frac{1}{2}\alpha^2 n^{-2/d} \log n < t_0 \leq 1$ and $T = n^{\mathcal{O}(1)}$, let $\{\boldsymbol{x}_t\}_{t \in [t_0, T]}$ be the solutions of the process Eq. (2) with density function $p_t : \mathbb{R}^d \to \mathbb{R}_+$. Let $\hat{p}_t(\boldsymbol{y}) = \sum_{i=1}^n \varphi_{\sigma_t}(\boldsymbol{y} - e^{-t}\boldsymbol{x}^{(i)})$ be the empirical density function. Let $\rho_{n,t} = (2\pi(1 - e^{-2t}))^{-d/2} e^{-1} n^{-1}$. Then,*

$$\mathbb{E}_{\{\boldsymbol{x}^{(i)}\}_{i=1}^n \sim P^{\otimes n}} \left[ \int_{t=t_0}^T \int_{\mathbb{R}^d} \left\| \frac{\nabla p_t(\boldsymbol{x}_t)}{p_t(\boldsymbol{x}_t)} - \frac{\nabla \hat{p}_t(\boldsymbol{x}_t)}{\hat{p}_t(\boldsymbol{x}_t) \vee \rho_{n,t}} \right\|_2^2 p_t(\boldsymbol{x}_t) \mathrm{d}\boldsymbol{x}_t \mathrm{d}t \right] \lesssim \alpha^d t_0^{-d/2} n^{-1} \log^{\frac{d}{2}+4} n.$$

*Proof.* For OU process, we have $\boldsymbol{X}_t = e^{-t}\boldsymbol{X}_0 + \sqrt{1 - e^{-2t}}\boldsymbol{Z}, \boldsymbol{Z} \sim \mathcal{N}(0, \boldsymbol{I}_d)$. With Assumption 1, $e^{-t}\boldsymbol{X}_0$ is $e^{-t}\alpha$-sub-Gaussian. To use Lemma 1, we need $\sigma_t^2 = 1 - \exp(-2t) \gtrsim \alpha^2 n^{-2/d} \log n$, i.e., $t \gtrsim -\log \left( 1 - \frac{\alpha^2 \log n}{n^{2/d}} \right)$ and $\alpha^2 n^{-2/d} \log n \lesssim 1$. Notice that $T \geq t_0 > \frac{1}{2}\alpha^2 n^{-2/d} \log n$, we have for

all $t \in [t_0, T]$,

$$t > \frac{1}{2}\alpha^2 n^{-2/d} \log n \geq \frac{1}{2} \log\left(1 + \frac{\alpha^2 \log n}{n^{2/d}}\right) = -\frac{1}{2} \log\left(\frac{1}{1 + \frac{\alpha^2 \log n}{n^{2/d}}}\right)$$

$$= -\frac{1}{2} \log\left(1 - \frac{\alpha^2 \log n}{n^{2/d}} \cdot \frac{n^{2/d}}{n^{2/d} + \alpha^2 \log n}\right)$$

$$= -\frac{1}{2} \log\left(1 - \frac{\alpha^2 \log n}{n^{2/d}} + \frac{\alpha^4 \log^2 n}{n^{2/d}(n^{2/d} + \alpha^2 \log n)}\right)$$

$$\geq -\frac{1}{2} \log\left(1 - \frac{\alpha^2 \log n}{n^{2/d}} + \frac{\alpha^4 \log^2 n}{2n^{2/d}(n^{2/d} \wedge \alpha^2 \log n)}\right)$$

$$\gtrsim -\frac{1}{2} \log\left(1 - \frac{\alpha^2 \log n}{n^{2/d}} + \frac{\alpha^4 \log^2 n}{2n^{2/d}\alpha^2 \log n}\right) \qquad \text{(by } \alpha^2 n^{-2/d} \log n \lesssim 1\text{)}$$

$$= -\frac{1}{2} \log\left(1 - \frac{1}{2}\frac{\alpha^2 \log n}{n^{2/d}}\right),$$

which gives that

$$1 - \exp(-2t) \gtrsim \alpha^2 n^{2/d} \log n$$

and it follows from Lemma 1 and $\rho_{n,t} = (2\pi(1 - e^{-2t}))^{-d/2} e^{-1} n^{-1}$ that

$$\mathbb{E}_{\{\boldsymbol{x}^{(i)}\}_{i=1}^n \sim P^{\otimes n}}\left[\int_{\mathbb{R}^d} \left\|\frac{\nabla p_t(\boldsymbol{x}_t)}{p_t(\boldsymbol{x}_t)} - \frac{\nabla \hat{p}_t(\boldsymbol{x}_t)}{\hat{p}_t(\boldsymbol{x}_t) \vee \rho_{n,t}}\right\|_2^2 p_t(\boldsymbol{x}_t)\mathrm{d}\boldsymbol{x}_t\right] \lesssim e^{-dt}(\alpha^d \sigma_t^{-d} \vee 1)\sigma^{-2}\frac{\log^{d/2+3} n}{n}$$

**Case 1:** When $\alpha \geq 1$, we always have $\alpha \geq \sigma_t, \forall t > 0$.

$$\mathbb{E}_{\{\boldsymbol{x}^{(i)}\}_{i=1}^n \sim P^{\otimes n}}\left[\int_{\mathbb{R}^d} \left\|\frac{\nabla p_t(\boldsymbol{x}_t)}{p_t(\boldsymbol{x}_t)} - \frac{\nabla \hat{p}_t(\boldsymbol{x}_t)}{\hat{p}_t(\boldsymbol{x}_t) \vee \rho_{n,t}}\right\|_2^2 p_t(\boldsymbol{x}_t)\mathrm{d}\boldsymbol{x}_t\right]$$

$$\lesssim e^{-dt}\alpha^d(1 - e^{-2t})^{-d/2-1}\frac{\log^{d/2+3} n}{n} \leq \alpha^d\left(\frac{e^{-2t}}{1 - e^{-2t}}\right)^{d/2}\frac{1}{1 - e^{-2t}}\frac{\log^{d/2+3} n}{n}.$$

Note that

$$\frac{\exp(-2t)}{1 - \exp(-2t)} \leq \frac{1}{t} \wedge \frac{1}{t^2}, \quad \text{for any } t > 0. \tag{43}$$

To see this, let $f(t) := 1 - \exp(-2t) - t\exp(-2t), \forall t \in [0, \infty)$. Since $f(0) = 0$ and we have

$$f'(t) = 2\exp(-2t) - \exp(-2t) + 2t\exp(-2t) = \exp(-2t) + 2t\exp(-2t) > 0, \quad \text{for any } t \geq 0,$$

which indicates that $f(t) \geq f(0) = 0$ for any $t > 0$ and validate Eq. (43). Then, for any $T \geq t_0 > 0$, we have

$$\int_{t_0}^T \left(\frac{e^{-2t}}{1 - e^{-2t}}\right)^{d/2}\frac{1}{1 - e^{-2t}}\mathrm{d}t \leq \int_{t_0}^T t^{-d/2}(t^{-1} + 1)\mathrm{d}t \qquad \text{(by Eq. (43))}$$

$$= \int_{t_0}^T t^{-d/2-1}\mathrm{d}t + \int_{t_0}^T t^{-d/2}\mathrm{d}t$$

$$= -\frac{2}{d}t^{-d/2}\Big|_{t=t_0}^{t=T} - \frac{2}{d+2}t^{-d/2+1}\Big|_{t=t_0}^{t=T}$$

$$\leq \frac{2}{d}t_0^{-d/2} + \frac{2}{d+2}t_0^{-d/2+1} \lesssim t_0^{-d/2},$$

which gives that

$$\mathbb{E}_{\{\boldsymbol{x}^{(i)}\}_{i=1}^n \sim P^{\otimes n}}\left[\int_{t=t_0}^T \int_{\mathbb{R}^d} \left\|\frac{\nabla p_t(\boldsymbol{x}_t)}{p_t(\boldsymbol{x}_t)} - \frac{\nabla \hat{p}_t(\boldsymbol{x}_t)}{\hat{p}_t(\boldsymbol{x}_t) \vee \rho_{n,t}}\right\|_2^2 p_t(\boldsymbol{x}_t)\mathrm{d}\boldsymbol{x}_t\mathrm{d}t\right]$$

$$\lesssim \alpha^d \frac{\log^{\frac{d}{2}+3} n}{n}\int_{t=t_0}^T \left(\frac{e^{-2t}}{1 - e^{-2t}}\right)^{d/2}\frac{1}{1 - e^{-2t}}\mathrm{d}t \lesssim \alpha^d t_0^{-d/2}n^{-1}\log^{\frac{d}{2}+3} n.$$

**Case 2:** When $0 < \alpha < 1$.

**Low noise region:** When $\alpha > \sigma_t = \sqrt{1 - \exp(-2t)}$, which indicates that $t_0 < t < -\frac{1}{2}\log(1-\alpha^2)$, it follows from Case 1 that

$$
\mathbb{E}_{\{\boldsymbol{x}^{(i)}\}_{i=1}^n \sim P^{\otimes n}} \left[ \int_{t=t_0}^{-\frac{1}{2}\log(1-\alpha^2)} \int_{\mathbb{R}^d} \left\| \frac{\nabla p_t(\boldsymbol{x}_t)}{p_t(\boldsymbol{x}_t)} - \frac{\nabla \hat{p}_t(\boldsymbol{x}_t)}{\hat{p}_t(\boldsymbol{x}_t) \vee \rho_{n,t}} \right\|_2^2 p_t(\boldsymbol{x}_t) \mathrm{d}\boldsymbol{x}_t \mathrm{d}t \right]
$$

$$
\lesssim \alpha^d t_0^{-d/2} n^{-1} \log^{\frac{d}{2}+3} n \le t_0^{-d/2} n^{-1} \log^{\frac{d}{2}+3} n.
$$

**High noise region:** When $\alpha \le \sigma_t$, which indicates that $-\frac{1}{2}\log(1-\alpha^2) \le t \le T$, we have

$$
\mathbb{E}_{\{\boldsymbol{x}^{(i)}\}_{i=1}^n \sim P^{\otimes n}} \left[ \int_{\mathbb{R}^d} \left\| \frac{\nabla p_t(\boldsymbol{x}_t)}{p_t(\boldsymbol{x}_t)} - \frac{\nabla \hat{p}_t(\boldsymbol{x}_t)}{\hat{p}_t(\boldsymbol{x}_t) \vee \rho_{n,t}} \right\|_2^2 p_t(\boldsymbol{x}_t) \mathrm{d}\boldsymbol{x}_t \right] \lesssim e^{-dt}(1 - e^{-2t})^{-1} \frac{\log^{d/2+3} n}{n},
$$

which implies that

$$
\mathbb{E}_{\{\boldsymbol{x}^{(i)}\}_{i=1}^n \sim P^{\otimes n}} \left[ \int_{t=-\frac{1}{2}\log(1-\alpha^2)}^{T} \int_{\mathbb{R}^d} \left\| \frac{\nabla p_t(\boldsymbol{x}_t)}{p_t(\boldsymbol{x}_t)} - \frac{\nabla \hat{p}_t(\boldsymbol{x}_t)}{\hat{p}_t(\boldsymbol{x}_t) \vee \rho_{n,t}} \right\|_2^2 p_t(\boldsymbol{x}_t) \mathrm{d}\boldsymbol{x}_t \mathrm{d}t \right]
$$

$$
\lesssim \frac{\log^{d/2+3} n}{n} \int_{t=-\frac{1}{2}\log(1-\alpha^2)}^{T} \frac{e^{-dt}}{1 - e^{-2t}} \mathrm{d}t \le \frac{\log^{d/2+3} n}{n} \int_{t=-\frac{1}{2}\log(1-\alpha^2)}^{T} \frac{e^{-t}}{1 - e^{-2t}} \mathrm{d}t
$$

$$
= \frac{\log^{d/2+3} n}{2n} \int_{t=-\frac{1}{2}\log(1-\alpha^2)}^{T} \frac{1}{e^t - e^{-t}} \mathrm{d}t \le \frac{\log^{d/2+3} n}{2n} \int_{t=-\frac{1}{2}\log(1-\alpha^2)}^{T} \frac{1}{e^t - 1} \mathrm{d}t
$$

$$
\le \frac{\log^{d/2+3} n}{2n} \int_{t=-\frac{1}{2}\log(1-\alpha^2)}^{T} \frac{1}{t} \mathrm{d}t \qquad\qquad \text{(by } \tfrac{1}{\exp(t)-1} \le t^{-1}, \forall t > 0\text{)}
$$

$$
\lesssim \frac{\log^{d/2+4} n}{n}.
$$

$\square$

### C.4 Score estimation along Brownian motion process

**Brownian motion process** Consider the Brownian motion process as the forward process of diffusion models:

$$
\mathrm{d}\boldsymbol{X}_t = \mathrm{d}\boldsymbol{B}_t \quad (0 \le t \le T), \quad \boldsymbol{X}_0 \sim P_0. \tag{44}
$$

It has an explicit solution

$$
\boldsymbol{X}_t = \boldsymbol{X}_0 + \sqrt{t}\boldsymbol{Z} \quad (0 \le t \le T), \quad \boldsymbol{Z} \sim \mathcal{N}(0, \boldsymbol{I}_d) \perp \boldsymbol{X}_0, \tag{45}
$$

which implies that $\boldsymbol{X}_t | \boldsymbol{X}_0 \sim \mathcal{N}(\boldsymbol{X}_0, t\boldsymbol{I}_d)$. The reverse Brownian motion process is given by

$$
\mathrm{d}Y_t = \nabla \log p_{T-t}(Y_t) \mathrm{d}t + \mathrm{d}\boldsymbol{B}_t \quad (0 \le t \le T), \quad Y_0 \sim P_T. \tag{46}
$$

**Corollary 3.** *Suppose the target distribution $P_0$ satisfies Assumption 1 and let $\hat{P}_0$ be the empirical distribution associated to a sample $\{\boldsymbol{x}^{(i)}\}_{i=1}^n$. For any $d \ge 1, n \ge 3$ and any $\frac{1}{2}\alpha^2 n^{-2/d} \log n \le t_0 \le 1$ and $T = n^{\mathcal{O}(1)}$, let $\{\boldsymbol{x}_t\}_{t \in [t_0, T]}$ be the solutions of the process Eq. (44) with density function $p_t : \mathbb{R}^d \to \mathbb{R}_+$. Let $\hat{p}_t(\boldsymbol{y}) = \sum_{i=1}^n \varphi_{\sigma_t}(\boldsymbol{y} - m_t \boldsymbol{x}^{(i)})$ be the empirical density function with $\sigma_t = \sqrt{t}, m_t = 1$. Let $\rho_{n,t} = (2\pi t)^{-d/2} e^{-1} n^{-1}$. Then, we have*

$$
\mathbb{E}_{\{\boldsymbol{x}^{(i)}\}_{i=1}^n \sim P^{\otimes n}} \left[ \int_{t=t_0}^{T} \int_{\mathbb{R}^d} \left\| \frac{\nabla p_t(\boldsymbol{x}_t)}{p_t(\boldsymbol{x}_t)} - \frac{\nabla \hat{p}_t(\boldsymbol{x}_t)}{\hat{p}_t(\boldsymbol{x}_t) \vee \rho_{n,t}} \right\|_2^2 p_t(\boldsymbol{x}_t) \mathrm{d}\boldsymbol{x}_t \mathrm{d}t \right] \le C_d \alpha^d t_0^{-d/2} n^{-1} \log^{\frac{d}{2}+4}.
$$

*Proof.* For Brownian process, we have $\boldsymbol{X}_t = \boldsymbol{X}_0 + \sqrt{t}\boldsymbol{Z}, \boldsymbol{Z} \sim \mathcal{N}(0, \boldsymbol{I}_d)$. With Assumption 1, $\boldsymbol{X}_0$ is $\alpha$-sub-Gaussian. For any $d \ge 1, n \ge 3$ and any $T \ge t_0 \ge \frac{1}{2}\alpha^2 n^{-2/d} \log n$, by Lemma 1, we have

$$
\mathbb{E}_{\{\boldsymbol{x}^{(i)}\}_{i=1}^n \sim P^{\otimes n}} \left[ \int_{\mathbb{R}^d} \left\| \frac{\nabla p_t(\boldsymbol{x}_t)}{p_t(\boldsymbol{x}_t)} - \frac{\nabla \hat{p}_t(\boldsymbol{x}_t)}{\hat{p}_t(\boldsymbol{x}_t) \vee \rho_{n,t}} \right\|_2^2 p_t(\boldsymbol{x}_t) \mathrm{d}\boldsymbol{x}_t \right] \le C_d \left( \alpha^d t^{-d/2} \vee 1 \right) t^{-1} \frac{\log^{d/2+4} n}{n}.
$$

**Low noise region:** when $\alpha > \sqrt{t}$,

For any $0 < t_0 \leq \alpha^2$, we have

$$\int_{t_0}^{\alpha^2} t^{-d/2-1}\mathrm{d}t = -\frac{2}{d}t^{-d/2}\Big|_{t=t_0}^{t=\alpha^2} = \frac{2}{d}\Big(-\alpha^{-d} + t_0^{-d/2}\Big) \leq \frac{2}{d}t_0^{-d/2}.$$

Therefore, we obtain

$$\mathbb{E}_{\{\boldsymbol{x}^{(i)}\}_{i=1}^n \sim P^{\otimes n}}\left[\int_{t=t_0}^{\alpha^2}\int_{\mathbb{R}^d}\Big\|\frac{\nabla p_t(\boldsymbol{x}_t)}{p_t(\boldsymbol{x}_t)} - \frac{\nabla \hat{p}_t(\boldsymbol{x}_t)}{\hat{p}_t(\boldsymbol{x}_t) \vee \rho_{n,t}}\Big\|_2^2 p_t(\boldsymbol{x}_t)\mathrm{d}\boldsymbol{x}_t\mathrm{d}t\right]$$

$$\lesssim \alpha^d \frac{\log^{\frac{d}{2}+3} n}{n}\int_{t=t_0}^{\alpha^2} t^{-d/2-1}\mathrm{d}t \leq C_d'\alpha^d t_0^{-d/2} n^{-1}\log^{\frac{d}{2}+3} n.$$

**High noise region:** when $\alpha \leq \sqrt{t}$,

$$\mathbb{E}_{\{\boldsymbol{x}^{(i)}\}_{i=1}^n \sim P^{\otimes n}}\left[\int_{t=\alpha^2}^{T}\int_{\mathbb{R}^d}\Big\|\frac{\nabla p_t(\boldsymbol{x}_t)}{p_t(\boldsymbol{x}_t)} - \frac{\nabla \hat{p}_t(\boldsymbol{x}_t)}{\hat{p}_t(\boldsymbol{x}_t) \vee \rho_{n,t}}\Big\|_2^2 p_t(\boldsymbol{x}_t)\mathrm{d}\boldsymbol{x}_t\mathrm{d}t\right]$$

$$\lesssim \frac{\log^{\frac{d}{2}+3} n}{n}\int_{t=\alpha^2}^{T} t^{-1}\mathrm{d}t$$

$$= \frac{\log^{\frac{d}{2}+3} n}{n}\Big(\log T - 2\log \alpha\Big) \lesssim \frac{\log^{\frac{d}{2}+4} n}{n}$$

$\square$

# D  Score Approximation by Deep Neural Networks

## D.1  Deep ReLU Neural Networks

We follow the notation used in [38] for ReLU neural networks. For a function $\phi \in \mathcal{NN}(\#\text{input} = d; \text{widthvec} = [N_1, N_2, \ldots, N_L]; \#\text{output} = 1)$, if we set $N_0 = d$ and $N_{L+1} = 1$. Then, $\phi$ can be represented in a form of function compositions as:

$$\phi = \mathcal{L}_L \circ \text{ReLU} \circ \mathcal{L}_{L-1} \circ \text{ReLU} \circ \cdots \circ \mathcal{L}_1 \circ \text{ReLU} \circ \mathcal{L}_0,$$

where $\text{ReLU} : \mathbb{R} \to \mathbb{R}$ denote the rectified linear unit, i.e. $\text{ReLU}(x) = \max\{0, x\}$ and $\mathcal{L}_i$ is the $i$-th affine linear transform in $\phi$ with weight matrix $\boldsymbol{W}_i \in \mathbb{R}^{N_{i+1} \times N_i}$ and bias vector $\boldsymbol{b}_i \in \mathbb{R}^{N_{i+1}}$ , i.e.

$$\boldsymbol{u}_{i+1} = \boldsymbol{W}_i \cdot \tilde{\boldsymbol{u}}_i + \boldsymbol{b}_i := \mathcal{L}_i(\tilde{\boldsymbol{u}}_i), \quad \text{for } i = 0, 1, \ldots, L$$

and $\tilde{\boldsymbol{u}}_0 = \boldsymbol{x} \in \mathbb{R}^d, \tilde{\boldsymbol{u}}_i = \text{ReLU}(\boldsymbol{u}_i)$ for $i = 1, 2, \ldots, L..$ We say that a neural network (architecture) with width $N$ and depth $L$ if the maximum width of any hidden layer in the network is at most $N$, and the total number of hidden layers does not exceed $L$.

For simplicity of notation, we write $\|\cdot\|_\infty$ as $|\cdot|$ throughout this section.

## D.2  Neural network approximation for Gaussian kernel density estimators

### D.2.1  The main result

**Assumption 4.** *Given a sample $\{\boldsymbol{x}^{(i)}\}_{i=1}^n$ of size $n$, there exist $s \in \mathbb{N}_+, 0 < \epsilon < 1 \le \alpha$ such that*

$$\sup_{i \in [n]} |\boldsymbol{x}^{(i)}| \le \sqrt{2\alpha s \log(\epsilon^{-1})}.$$

**Lemma 8** (Approximation of Gaussian Kernel Density Estimator). *Given a data set $\{\boldsymbol{x}^{(i)}\}_{i=1}^n$, let $m_t := \exp(-t), \sigma_t := \sqrt{1 - \exp(-2t)}$ for any $t \in [t_0, \infty)$. For any $\boldsymbol{y} \in \mathbb{R}^d, t \in [t_0, \infty)$, the Gaussian kernel density estimators is given by*

$$f_{kde}(\boldsymbol{y}, t) := \frac{1}{n} \sum_{i=1}^n \exp\left(-\frac{\|\boldsymbol{y} - m_t \boldsymbol{x}^{(i)}\|_2^2}{2\sigma_t^2}\right). \tag{47}$$

*Fix any $0 < \epsilon < t_0 \le 1/2$, there exist $N, L, s \in \mathbb{N}_+$ such that $N^{-2} L^{-2} \le \epsilon$ and Assumption 4 holds. Then, there exists a function $\phi_{kde}$ implemented by a ReLU DNN with width $\le \mathcal{O}\big(s^{6d+3} N^3 \log_2(N) \vee s^{6d+3} \log^3(\epsilon^{-1})\big)$ and depth $\le \mathcal{O}\big(L^3 \log_2(L) \vee s^2 \log^2(\epsilon^{-1})\big)$ such that*

$$\left|\phi_{kde}(\boldsymbol{y}, t) - f_{kde}(\boldsymbol{y}, t)\right| \lesssim \alpha^{3s} (2s)! s^{3d+9s+1} \log^{9s}(\epsilon^{-1}) \epsilon^s, \quad \text{for any } \boldsymbol{y} \in \mathbb{R}^d, t \in [t_0, \infty),$$

*and $0 \le \phi_{kde}(\boldsymbol{y}, t) \lesssim 1$.*

### D.2.2  Proof of Lemma 8

We decompose $\mathbb{R}^d = \mathcal{B} \cup \overline{\mathcal{B}}$, where

$$\mathcal{B} := \{\boldsymbol{y} \in \mathbb{R}^d : |\boldsymbol{y}| \le 2\sqrt{2\alpha s \log(\epsilon^{-1})}\}, \tag{48}$$

$$\overline{\mathcal{B}} := \{\boldsymbol{y} \in \mathbb{R}^d : |\boldsymbol{y}| > 2\sqrt{2\alpha s \log(\epsilon^{-1})}\}. \tag{49}$$

We approximate $f_{\text{kde}}$ on $\boldsymbol{y} \in \overline{\mathcal{B}}, t \in [t_0, \infty)$ in **Part I** and $\boldsymbol{y} \in \mathcal{B}, t \in [t_0, \infty)$ in **Part II**.

**Part I: Approximating $f_{\text{kde}}$ on $\overline{\mathcal{B}}$**

Fix any $N, L, s \in \mathbb{N}_+$, here we aim to approximate $f_{\text{kde}}$ for $\boldsymbol{y} \in \overline{\mathcal{B}}$. By Assumption 4, we have

$$\frac{\|\boldsymbol{y} - m_t \boldsymbol{x}^{(i)}\|_2^2}{2\sigma_t^2} > \frac{(\sqrt{2\alpha s \log(\epsilon^{-1})})^2}{2\sigma_t^2} \ge \frac{2s \log(\epsilon^{-1})}{2} = s \log(\epsilon^{-1}), \quad \forall i \in [n], \tag{50}$$

which implies that

$$f_{\text{kde}}(\boldsymbol{y}, t) < \exp\big(-s \log(\epsilon^{-1})\big) = \epsilon^s, \quad \text{for any } \boldsymbol{y} \in \overline{\mathcal{B}}, t \in (0, \infty). \tag{51}$$

Therefore, $f_{\text{kde}}(\boldsymbol{y}, t), \forall \boldsymbol{y} \in \overline{\mathcal{B}}_{\boldsymbol{y}}, t \in [t_0, \infty)$ can be well approximated with an error within $\epsilon$ by simply setting the output of the neural network to be zero. Therefore, we only need to consider the approximation error of a neural network for $\boldsymbol{y} \in \mathcal{B}, t \in [t_0, \infty)$.

**Part II: Approximating $f_{\text{kde}}$ on $\mathcal{B}$**

In the following, we prove the neural network approximation results for Gaussian kernel density estimator Eq. (47) for $\boldsymbol{y} \in \mathcal{B}, t \in [t_0, \infty)$. Given $\{\boldsymbol{x}^{(i)}\}_{i=1}^n$, for any $\boldsymbol{y} \in \mathcal{B}, t \in [t_0, \infty)$, denote by

$$h^{(i)} := \frac{\|\boldsymbol{y} - m_t \boldsymbol{x}^{(i)}\|_2^2}{2\sigma_t^2}, \quad \text{for any } i \in [n]. \tag{52}$$

Then we have

$$0 \le h^{(i)} \le \frac{2\|\boldsymbol{y}\|_2^2 + 2m_t\|\boldsymbol{x}^{(i)}\|_2^2}{2\sigma_t^2} = 10\sigma_t^{-2}\alpha s \log(\epsilon^{-1}) \le 10t_0^{-1}\alpha s \log(\epsilon^{-1}). \tag{53}$$

**Step 1: Domain decomposition**

Let $\phi_{\tilde{h}}$ be defined as in Eq. (138). By Eq. (142), we can find a universal constant $\widetilde{C}_d$ depended only on $d$ such that

$$0 \le \phi_{\tilde{h}}(\boldsymbol{y}, t) \le \widetilde{C}_d t_0^{-1} \alpha s \log(\epsilon^{-1}), \quad \text{for any } \boldsymbol{y} \in \mathcal{B}, t \in [t_0, \infty]. \tag{54}$$

Set $K = N^4 L^4$, where $\delta \in (0, \widetilde{C}_d t_0^{-1} \alpha s \log(\epsilon^{-1})/K)$. Let $\Omega([0, \widetilde{C}_d t_0^{-1} \alpha s \log(\epsilon^{-1})], K, \delta)$ partition $[0, \widetilde{C}_d t_0^{-1} \alpha s \log(\epsilon^{-1})]$ into $K$ cubes $Q_\beta$ for $\beta \in \{0, 1, \ldots, K-1\}$, where

$$Q_\beta := \Big\{ h \in \mathbb{R} : h \in \Big[\tfrac{\beta \widetilde{C}_d t_0^{-1} \alpha s \log(\epsilon^{-1})}{K}, \tfrac{(\beta+1)\widetilde{C}_d t_0^{-1} \alpha s \log(\epsilon^{-1})}{K} - \delta \cdot \mathbb{1}_{\{\beta \le K-2\}}\Big]\Big\}. \tag{55}$$

For each $\beta$, we define

$$\tilde{h}_\beta := \frac{\beta \widetilde{C}_d t_0^{-1} \alpha s \log(\epsilon^{-1})}{K}, \quad \beta \in \{0, 1, \ldots, K-1\}. \tag{56}$$

Clearly, $[0, \widetilde{C}_d t_0^{-1} \alpha s \log(\epsilon^{-1})] = \Omega([0, \widetilde{C}_d t_0^{-1} \alpha s \log(\epsilon^{-1})], K, \delta) \bigcup \big( \cup_{\beta \in \{0,1,\cdots,K-1\}} Q_\beta \big)$ and $\tilde{h}_\beta$ is the vertex of $Q_\beta$ with minimum $\| \cdot \|_1$-norm.

**Step 2: Taylor expansion of the Gaussian density kernel estimators**

For all $\{h^{(i}(\boldsymbol{y}, t)\}_{i=1}^n$, denote by

$$\tilde{h}(\boldsymbol{y}, t) := \Big( \frac{1}{n} \sum_{i=1}^n \big(h^{(i)}(\boldsymbol{y}, t)\big)^s \Big)^{1/s}. \tag{57}$$

Clearly, we have $\tilde{h}(\boldsymbol{y}, t) \in [0, 10t_0^{-1}\alpha s \log(\epsilon^{-1})]$ for any $\boldsymbol{y} \in \mathcal{B}, t \in [t_0, \infty)$. By Eq. (141),

$$\big|\phi_{\tilde{h}}(\boldsymbol{y}, t) - \tilde{h}(\boldsymbol{y}, t)\big| \lesssim \alpha^2 s! s^{2s+2} \log^{2s+2}(\epsilon^{-1})\epsilon^s.$$

We choose $\tilde{h}_\beta$ for $\beta \in \{0, 1, \ldots, K-1\}$ such that for some $\boldsymbol{y}, t$, we have

$$\big|\phi_{\tilde{h}}(\boldsymbol{y}, t) - \tilde{h}_\beta(\boldsymbol{y}, t)\big| \le \frac{\widetilde{C}_d t_0^{-1} \alpha s \log(\epsilon^{-1})}{K}. \tag{58}$$

In what follow, we use $\tilde{h} \equiv \tilde{h}(\boldsymbol{y}, t)$ and $\tilde{h}_\beta \equiv \tilde{h}_\beta(\boldsymbol{y}, t)$. For any $\boldsymbol{y} \in \mathcal{B}, t \in [t_0, \infty)$, we have

$$\big|\tilde{h} - \tilde{h}_\beta\big| \le \underbrace{\big|\tilde{h} - \phi_{\tilde{h}}(\boldsymbol{y}, t)\big|}_{\lesssim \text{ Eq. (141)}} + \big|\phi_{\tilde{h}}(\boldsymbol{y}, t) - \tilde{h}_\beta\big|$$

$$\lesssim \alpha^2 s! s^{2s+2} \log^{2s+2}(\epsilon^{-1})\epsilon^s + \frac{t_0^{-1}\alpha s \log(\epsilon^{-1})}{K}$$

$$= \alpha^2 s! s^{2s+2} \log^{2s+2}(\epsilon^{-1})\epsilon^s + \frac{t_0^{-1}\alpha s \log(\epsilon^{-1})}{N^4 L^4}$$

$$\le \alpha^2 s! s^{2s+2} \log^{2s+2}(\epsilon^{-1})\epsilon^s + \alpha s \log(\epsilon^{-1})\epsilon^s \qquad (\text{by } N^{-2}L^{-2} \le \epsilon \le t_0)$$

$$\lesssim \alpha^2 s! s^{2s+2} \log^{2s+2}(\epsilon^{-1})\epsilon^s. \tag{59}$$

The Taylor expansion of the Gaussian kernel density estimator at $\tilde{h}_\beta$ up to order $s-1$ is given by

$$\frac{1}{n}\sum_{i=1}^{n}\exp\left(-\frac{\|\boldsymbol{y}-m_t\boldsymbol{x}^{(i)}\|^2}{2\sigma_t^2}\right)$$

$$=\frac{1}{n}\sum_{i=1}^{n}\exp(-h^{(i)})=\frac{1}{n}\sum_{i=1}^{n}\left\{\sum_{k=0}^{s-1}\frac{(-1)^k\exp(-\tilde{h}_\beta)}{k!}\left(h^{(i)}-\tilde{h}_\beta\right)^k+\frac{(-1)^s\exp(-\theta^{(i)})}{s!}\left(h^{(i)}-\tilde{h}_\beta\right)^s\right\}$$

$$=\sum_{k=0}^{s-1}\frac{(-1)^k\exp(-\tilde{h}_\beta)}{k!}\frac{1}{n}\sum_{i=1}^{n}\left(h^{(i)}-\tilde{h}_\beta\right)^k+\frac{1}{n}\sum_{i=1}^{n}\frac{(-1)^s\exp(-\theta^{(i)})}{s!}\left(h^{(i)}-\tilde{h}_\beta\right)^s, \tag{60}$$

for some real number $\theta^{(i)}$ that is between $h^{(i)}$ and $\tilde{h}_\beta$. By Minkowski's inequality, for all $s \geq k \geq 0$,

$$\left(\frac{1}{n}\sum_{i=1}^{n}\left(h^{(i)}\right)^k\right)^{1/k}\leq\left(\frac{1}{n}\sum_{i=1}^{n}\left(h^{(i)}\right)^s\right)^{1/s}=\tilde{h}. \tag{61}$$

Eq. (59) and (61) give that

$$\left|\frac{1}{n}\sum_{i=1}^{n}\frac{(-1)^s\exp(-\theta^{(i)})}{s!}\left(h^{(i)}-\tilde{h}_\beta\right)^s\right|$$

$$\leq\left|\frac{1}{n}\sum_{i=1}^{n}\frac{(-1)^s}{s!}\sum_{k=0}^{s}\frac{s!}{k!}\left(h^{(i)}\right)^k(-\tilde{h}_\beta)^{s-k}\right| \qquad\qquad \text{(by } (x-y)^s=\sum_{k=0}^{s}\frac{s!}{k!}x^k(-y)^{s-k})$$

$$=\left|\frac{(-1)^s}{s!}\sum_{k=0}^{s}\frac{s!}{k!}(-\tilde{h}_\beta)^{s-k}\left[\underbrace{\frac{1}{n}\sum_{i=1}^{n}\left(h^{(i)}\right)^k}_{\leq\tilde{h}^k\text{ by Eq. (61)}}\right]\right|\leq\left|\frac{(-1)^s}{s!}\sum_{k=0}^{s}\frac{s!}{k!}(-\tilde{h}_\beta)^{s-k}\tilde{h}^k\right|=\left|\frac{(-1)^s}{s!}(\tilde{h}-\tilde{h}_\beta)^s\right|$$

$$\leq\frac{1}{s!}\left|\tilde{h}-\tilde{h}_\beta\right|^s\lesssim\frac{1}{s!}\alpha^2 s! s^{2s+2}\log^{2s+2}(\epsilon^{-1})\epsilon^s=\alpha^2 s^{2s+2}\log^{2s+2}(\epsilon^{-1})\epsilon^s. \tag{62}$$

Denote by $\boldsymbol{a}:=[a_1,a_2,a_3,a_4]^\top\in\mathbb{N}_+^4$ and

$$C_{\boldsymbol{x}}^{\boldsymbol{\nu}_2,\boldsymbol{\nu}_3}:=\frac{1}{n}\sum_{i=1}^{n}\left(\boldsymbol{x}^{(i)}\right)^{\boldsymbol{\nu}_2+2\boldsymbol{\nu}_3}. \tag{63}$$

we have

$$\frac{1}{n}\sum_{i=1}^{n}\left(h^{(i)}-\tilde{h}_\beta\right)^k$$

$$=\frac{1}{n}\sum_{i=1}^{n}\left(\frac{\|\boldsymbol{y}-m_t\boldsymbol{x}^{(i)}\|^2}{2\sigma_t^2}-\tilde{h}_\beta\right)^k$$

$$=\frac{1}{2^k\sigma_t^{2k}}\frac{1}{n}\sum_{i=1}^{n}\left(\|\boldsymbol{y}-m_t\boldsymbol{x}^{(i)}\|^2-2\sigma_t^2\tilde{h}_\beta\right)^k$$

$$=\frac{1}{2^k\sigma_t^{2k}}\frac{1}{n}\sum_{i=1}^{n}\sum_{\|\boldsymbol{a}\|_1=k}\frac{k!}{\boldsymbol{a}!}\|\boldsymbol{y}\|^{2a_1}\times(-2m_t\boldsymbol{y}^\top\boldsymbol{x}^{(i)})^{a_2}\times m_t^{2a_3}\|\boldsymbol{x}^{(i)}\|^{2a_3}\times(-2\sigma_t^2\tilde{h}_\beta)^{a_4}$$

$$=\sum_{\|\boldsymbol{a}\|_1=k}\frac{k!(-2)^{a_2+a_4}m_t^{a_2+2a_3}}{\boldsymbol{a}!2^k\sigma_t^{2k-2a_4}}\tilde{h}_\beta^{a_4}\sum_{\|\boldsymbol{\nu}_1\|_1=a_1}\frac{a_1!}{\boldsymbol{\nu}_1!}\boldsymbol{y}^{2\boldsymbol{\nu}_1}\sum_{\|\boldsymbol{\nu}_2\|_1=a_2}\frac{a_2!}{\boldsymbol{\nu}_2!}\boldsymbol{y}^{\boldsymbol{\nu}_2}\frac{1}{n}\sum_{i=1}^{n}\left(\boldsymbol{x}^{(i)}\right)^{\boldsymbol{\nu}_2}\sum_{\|\boldsymbol{\nu}_3\|_1=a_3}\frac{a_3!}{\boldsymbol{\nu}_3!}\left(\boldsymbol{x}^{(i)}\right)^{2\boldsymbol{\nu}_3}$$

$$=\sum_{\|\boldsymbol{a}\|_1=k}\frac{k!(-2)^{a_2+a_4}m_t^{a_2+2a_3}}{a_4!2^k\sigma_t^{2k-2a_4}}\tilde{h}_\beta^{a_4}\sum_{\|\boldsymbol{\nu}_1\|_1=a_1}\frac{1}{\boldsymbol{\nu}_1!}\boldsymbol{y}^{2\boldsymbol{\nu}_1}\sum_{\|\boldsymbol{\nu}_2\|_1=a_2}\frac{1}{\boldsymbol{\nu}_2!}\boldsymbol{y}^{\boldsymbol{\nu}_2}\sum_{\|\boldsymbol{\nu}_3\|_1=a_3}\frac{1}{\boldsymbol{\nu}_3!}\left(\frac{1}{n}\sum_{i=1}^{n}\left(\boldsymbol{x}^{(i)}\right)^{\boldsymbol{\nu}_2+2\boldsymbol{\nu}_3}\right)$$

$$=\sum_{\|\boldsymbol{\nu}_1\|_1+\|\boldsymbol{\nu}_2\|_1+\|\boldsymbol{\nu}_3\|_1+a_4=k}\frac{C_{\boldsymbol{x}}^{\boldsymbol{\nu}_2,\boldsymbol{\nu}_3}k!(-1)^{\|\boldsymbol{\nu}_2\|_1+a_4}m_t^{\|\boldsymbol{\nu}_2\|_1+2\|\boldsymbol{\nu}_3\|_1}}{a_4!2^{k-\|\boldsymbol{\nu}_2\|_1-a_4}\sigma_t^{2(k-a_4)}\boldsymbol{\nu}_1!\boldsymbol{\nu}_2!\boldsymbol{\nu}_3!}\left\{\tilde{h}_\beta^{a_4}\times\boldsymbol{y}^{2\boldsymbol{\nu}_1+\boldsymbol{\nu}_2}\right\}. \tag{64}$$

Denote by $\tilde{\boldsymbol{\nu}} := [\boldsymbol{\nu}_1, \boldsymbol{\nu}_2, \boldsymbol{\nu}_3] \in \mathbb{N}_+^{3d}$, we obtain

$$
\sum_{k=0}^{s-1} \frac{(-1)^k \exp(-\tilde{h}_\beta)}{k!} \frac{1}{n} \sum_{i=1}^{n} \left(h^{(i)} - \tilde{h}_\beta\right)^k
$$

$$
= \sum_{k=0}^{s-1} \frac{(-1)^k \exp(-\tilde{h}_\beta)}{k!} \sum_{\|\boldsymbol{\nu}_1\|_1 + \|\boldsymbol{\nu}_2\|_1 + \|\boldsymbol{\nu}_3\|_1 + a_4 = k} \frac{C_{\boldsymbol{x}}^{\boldsymbol{\nu}_2, \boldsymbol{\nu}_3} k! (-1)^{\|\boldsymbol{\nu}_2\|_1 + a_4} m_t^{\|\boldsymbol{\nu}_2\|_1 + 2\|\boldsymbol{\nu}_3\|_1}}{2^{k - \|\boldsymbol{\nu}_2\|_1 - a_4} \sigma_t^{2(k - a_4)} \boldsymbol{\nu}_1! \boldsymbol{\nu}_2! \boldsymbol{\nu}_3! a_4!} \left\{ \tilde{h}_\beta^{a_4} \times \boldsymbol{y}^{2\boldsymbol{\nu}_1 + \boldsymbol{\nu}_2} \right\}
$$

$$
= \exp(-\tilde{h}_\beta) \times \sum_{k=0}^{s-1} \sum_{\|\tilde{\boldsymbol{\nu}}\|_1 + a_4 = k} \frac{C_{\boldsymbol{x}}^{\boldsymbol{\nu}_2, \boldsymbol{\nu}_3} m_t^{\|\boldsymbol{\nu}_2\|_1 + 2\|\boldsymbol{\nu}_3\|_1}}{(-2)^{k - \|\boldsymbol{\nu}_2\|_1 - a_4} \sigma_t^{2(k - a_4)} \boldsymbol{\nu}_1! \boldsymbol{\nu}_2! \boldsymbol{\nu}_3! a_4!} \left\{ \tilde{h}_\beta^{a_4} \times \boldsymbol{y}^{2\boldsymbol{\nu}_1 + \boldsymbol{\nu}_2} \right\}. \tag{65}
$$

Combine Eq. (60) and (65) gives that

$$
\frac{1}{n} \sum_{i=1}^{n} \exp\left(-\frac{\|\boldsymbol{y} - m_t \boldsymbol{x}^{(i)}\|^2}{2\sigma_t^2}\right)
$$

$$
= \sum_{k=0}^{s-1} \frac{(-1)^k \exp(-\tilde{h}_\beta)}{k!} \frac{1}{n} \sum_{i=1}^{n} \left(h^{(i)} - \tilde{h}_\beta\right)^k + \frac{1}{n} \sum_{i=1}^{n} \frac{(-1)^s \exp(-\theta^{(i)})}{s!} \left(h^{(i)} - \tilde{h}_\beta\right)^s
$$

$$
= \exp(-\tilde{h}_\beta) \sum_{k=0}^{s-1} \sum_{\|\tilde{\boldsymbol{\nu}}\|_1 + a_4 = k} \frac{C_{\boldsymbol{x}}^{\boldsymbol{\nu}_2, \boldsymbol{\nu}_3} m_t^{\|\boldsymbol{\nu}_2\|_1 + 2\|\boldsymbol{\nu}_3\|_1}}{(-2)^{k - \|\boldsymbol{\nu}_2\|_1 - a_4} \sigma_t^{2(k - a_4)} \boldsymbol{\nu}_1! \boldsymbol{\nu}_2! \boldsymbol{\nu}_3! a_4!} \tilde{h}_\beta^{a_4} \boldsymbol{y}^{2\boldsymbol{\nu}_1 + \boldsymbol{\nu}_2}
$$

$$
+ \frac{1}{n} \sum_{i=1}^{n} \frac{(-1)^s \exp(-\theta^{(i)})}{s!} \left(h^{(i)} - \tilde{h}_\beta\right)^s,
$$

where the second term is bounded by Eq. (62):

$$
\left| \frac{1}{n} \sum_{i=1}^{n} \frac{(-1)^s \exp(-\theta^{(i)})}{s!} \left(h^{(i)} - \tilde{h}_\beta\right)^s \right| \lesssim \alpha^2 s! s^{2s+2} \log^{2s+2}(\epsilon^{-1}) \epsilon^s.
$$

In what follows, we aim to construct a ReLU DNN $\phi_{\text{kde}}$ to approximate the first term.

### Step 3: Construction of the ReLU DNN $\phi_{\text{kde}}$

For each $\tilde{\boldsymbol{\nu}} \in \mathbb{N}^{3d}, a_4 \in \mathbb{N}$ such that $\|\tilde{\boldsymbol{\nu}}\|_1 + a_4 \leq s - 1$, we have $0 \leq \|\boldsymbol{\nu}_2\|_1 + 2\|\boldsymbol{\nu}_3\|_1 \leq 2(s-1), 2\|\boldsymbol{\nu}_1\|_1 + \|\boldsymbol{\nu}_2\|_1 \leq 2(s-1)$. For each $k = 0, 1, \ldots, s-1$, we have $0 \leq k - a_4 \leq s - 1$. Then, by Propositions 6 and 7, there exist

$$
\phi_{\tilde{h}_\beta}^{a_4} \in \mathcal{NN}\left(\text{width} \lesssim s^{3d+2} N^2 \log_2(N) \vee s^{3d+2} \log^3(\epsilon^{-1}); \text{depth} \lesssim L^2 \log_2(L) \vee s^2 \log^2(\epsilon^{-1})\right) \tag{66}
$$

$$
\phi_{\tilde{h}_\beta}^{\exp} \in \mathcal{NN}\left(\text{width} \lesssim s^{3d+2} N^3 \log_2(N) \vee s^{3d+2} \log^3(\epsilon^{-1}); \text{depth} \lesssim L^3 \log_2(L) \vee s^2 \log^2(\epsilon^{-1})\right) \tag{67}
$$

such that

$$
\left| \phi_{\tilde{h}_\beta}^{a_4}(\boldsymbol{y}, t) - \tilde{h}_\beta^{a_4}(\boldsymbol{y}, t) \right| \leq \alpha^{2s} s^{2s} \log^{2s}(\epsilon^{-1}) N^{-4s} L^{-4s}, \quad \forall \boldsymbol{y} \in \mathcal{B}, \forall t \in [t_0, \infty), \tag{68}
$$

$$
\left| \phi_{\tilde{h}_\beta}^{\exp}(\boldsymbol{y}, t) - \exp(-\tilde{h}_\beta)(\boldsymbol{y}, t) \right| \leq N^{-4s} L^{-4s}, \quad \forall \boldsymbol{y} \in \mathcal{B}, \forall t \in [t_0, \infty), \tag{69}
$$

and

$$
0 \leq \phi_{\tilde{h}_\beta}^{a_4}(\boldsymbol{y}, t) \lesssim t_0^{-a_4} \alpha^{a_4} s^{a_4} \log^{a_4}(\epsilon^{-1}), \tag{70}
$$

$$
0 \leq \phi_{\tilde{h}_\beta}^{\exp}(\boldsymbol{y}, t) \leq 1. \tag{71}
$$

Moreover, by Eq. (132), (133) and (139), for each $\tilde{\boldsymbol{\nu}} \in \mathbb{N}^{3d}, a_4 \in \mathbb{N}$ such that $\|\tilde{\boldsymbol{\nu}}\|_1 + a_4 \leq k, k = 0, 1, \ldots, s-1$, there exists

$$
\phi_1^{\tilde{\boldsymbol{\nu}}, a_4} \in \mathcal{NN}\left(\text{width} \lesssim s^3 N \log_2(N) \vee s^3 \log^3(\epsilon^{-1}); \text{depth} \lesssim s^2 L \log_2(L) \vee s^2 \log^2(\epsilon^{-1})\right). \tag{72}
$$

such that for any $\boldsymbol{y} \in \mathcal{B}, t \in [t_0, \infty)$,

$$\left| \phi_1^{\tilde{\boldsymbol{\nu}}, a_4}(\boldsymbol{y}, t) - m_t^{\|\boldsymbol{\nu}_2\|_1 + 2\|\boldsymbol{\nu}_3\|_1} \sigma_t^{-2(k-a_4)} \boldsymbol{y}^{2\boldsymbol{\nu}_1 + \boldsymbol{\nu}_2} C_{\boldsymbol{x}}^{\boldsymbol{\nu}_2, \boldsymbol{\nu}_3} \right| \lesssim \alpha^{2s}(2s)! s^{8s} \log^{8s}(\epsilon^{-1}) \epsilon^{2s}, \quad (73)$$

and

$$0 \le \phi_1^{\tilde{\boldsymbol{\nu}}, a_4}(\boldsymbol{y}, t) \lesssim t_0^{-s} \alpha^s s^s \log^s(\epsilon^{-1}). \tag{74}$$

By Lemma 13 and Eq. (70) and (74), there exists

$$\phi_{\text{multi}}^{(3)} \in \mathcal{NN}\left(\text{width} \le 9(N+1)+1, \text{depth} \le 14sL\right), \tag{75}$$

such that

$$\left| \phi_{\text{multi}}^{(3)}\left(\phi_{\tilde{h}_\beta}^{a_4}(\boldsymbol{y}, t), \phi_1^{\tilde{\boldsymbol{\nu}}, a_4}(\boldsymbol{y}, t)\right) - \phi_{\tilde{h}_\beta}^{a_4}(\boldsymbol{y}, t) \times \phi_1^{\tilde{\boldsymbol{\nu}}, a_4}(\boldsymbol{y}, t) \right|$$
$$\le 6 t_0^{-k} \alpha^k s^k \log^k(\epsilon^{-1}) \cdot t_0^{-s} \alpha^s s^s \log^s(\epsilon^{-1})(N+1)^{-14sL}$$
$$\lesssim t_0^{-2s} \alpha^{2s} s^{2s} \log^{2s}(\epsilon^{-1}) \epsilon^{4s} \qquad \text{(by Eq. (131))}$$
$$\le \alpha^{2s} s^{2s} \log^{2s}(\epsilon^{-1}) \epsilon^{2s}. \tag{76}$$

For each $\tilde{\boldsymbol{\nu}} \in \mathbb{N}^{3d}, a_4 \in \mathbb{N}$ such that $\|\tilde{\boldsymbol{\nu}}\|_1 + a_4 \le k, k = 0, 1, \ldots, s-1$, define

$$\phi_2^{\tilde{\boldsymbol{\nu}}, a_4}(\boldsymbol{y}, t) := \phi_{\text{abs}}\left(\phi_{\text{multi}}^{(3)}\left(\phi_{\tilde{h}_\beta}^{a_4}(\boldsymbol{y}, t), \phi_1^{\tilde{\boldsymbol{\nu}}, a_4}(\boldsymbol{y}, t)\right)\right), \quad \text{for any } \boldsymbol{y} \in \mathcal{B}, t \in [t_0, \infty), \tag{77}$$

where $\phi_{\text{abs}}(x) := \text{ReLU}(x) + \text{ReLU}(-x) = |x|$, for any $x \in \mathbb{R}$. By the size of $\phi_{\tilde{h}_\beta}^{a_4}, \phi_1^{\tilde{\boldsymbol{\nu}}, a_4}, \phi_{\text{multi}}^{(3)}, \phi_{\text{abs}}$, we have

$$\phi_2^{\tilde{\boldsymbol{\nu}}, a_4} \in \mathcal{NN}\left(\text{width} \lesssim s^{3d+2} N^2 \log_2(N) \vee s^{3d+2} \log^3(\epsilon^{-1}); \text{depth} \lesssim L^2 \log_2(L) \vee s^2 \log^2(\epsilon^{-1})\right) \tag{78}$$

and for any $\boldsymbol{y} \in \mathcal{B}, t \in [t_0, \infty)$,

$$\left| \phi_2^{\tilde{\boldsymbol{\nu}}, a_4}(\boldsymbol{y}, t) - \tilde{h}_\beta^{a_4} \times m_t^{\|\boldsymbol{\nu}_2\|_1 + 2\|\boldsymbol{\nu}_3\|_1} \sigma_t^{-2(k-a_4)} \boldsymbol{y}^{2\boldsymbol{\nu}_1 + \boldsymbol{\nu}_2} C_{\boldsymbol{x}}^{\boldsymbol{\nu}_2, \boldsymbol{\nu}_3} \right|$$
$$\le \underbrace{\left| \phi_{\text{multi}}^{(3)}\left(\phi_{\tilde{h}_\beta}^{a_4}(\boldsymbol{y}, t), \phi_1^{\tilde{\boldsymbol{\nu}}, a_4}(\boldsymbol{y}, t)\right) - \phi_{\tilde{h}_\beta}^{a_4}(\boldsymbol{y}, t) \times \phi_1^{\tilde{\boldsymbol{\nu}}, a_4}(\boldsymbol{y}, t) \right|}_{\lesssim \text{ Eq. (76)}} + \underbrace{\left| \phi_{\tilde{h}_\beta}^{a_4}(\boldsymbol{y}, t) - \tilde{h}_\beta^{a_4} \right|}_{\lesssim \text{ Eq. (68)}} \cdot \underbrace{\left| \phi_1^{\tilde{\boldsymbol{\nu}}, a_4}(\boldsymbol{y}, t) \right|}_{\lesssim \text{ Eq. (74)}}$$
$$+ \left| \tilde{h}_\beta^{a_4} \right| \cdot \underbrace{\left| \phi_1^{\tilde{\boldsymbol{\nu}}, a_4}(\boldsymbol{y}, t) - m_t^{\|\boldsymbol{\nu}_2\|_1 + 2\|\boldsymbol{\nu}_3\|_1} \sigma_t^{-2(k-a_4)} \boldsymbol{y}^{2\boldsymbol{\nu}_1 + \boldsymbol{\nu}_2} C_{\boldsymbol{x}}^{\boldsymbol{\nu}_2, \boldsymbol{\nu}_3} \right|}_{\lesssim \text{ Eq. (73)}}$$
$$\lesssim \alpha^{2s} s^{2s} \log^{2s}(\epsilon^{-1}) \epsilon^{2s} + \alpha^{2s} s^{2s} \log^{2s}(\epsilon^{-1}) N^{-4s} L^{-4s} \cdot t_0^{-s} \alpha^s s^s \log^s(\epsilon^{-1})$$
$$+ t_0^{-s} \alpha^s s^s \log^s(\epsilon^{-1}) \cdot \alpha^{2s}(2s)! s^{8s} \log^{8s}(\epsilon^{-1}) \epsilon^{2s}$$
$$\lesssim \alpha^{2s}(3s)! s^{9s} \log^{9s}(\epsilon^{-1}) \epsilon^s, \tag{79}$$

which gives that

$$0 \le \phi_2^{\tilde{\boldsymbol{\nu}}, a_4}(\boldsymbol{y}, t) \lesssim \left| \tilde{h}_\beta^{a_4} \times m_t^{\|\boldsymbol{\nu}_2\|_1 + 2\|\boldsymbol{\nu}_3\|_1} \sigma_t^{-2(k-a_4)} \boldsymbol{y}^{2\boldsymbol{\nu}_1 + \boldsymbol{\nu}_2} C_{\boldsymbol{x}}^{\boldsymbol{\nu}_2, \boldsymbol{\nu}_3} \right| + \alpha^{2s}(2s)! s^{9s} \log^{9s}(\epsilon^{-1}) \epsilon^s$$
$$\lesssim t_0^{-a_4 - k + a_4} \cdot \left(s \log(\epsilon^{-1})\right)^{\|\boldsymbol{\nu}_1\|_1 + \|\boldsymbol{\nu}_2\|_1 + \|\boldsymbol{\nu}_3\|_1 + a_4}$$
$$= t_0^{-(s-1)} s^{s-1} \log^{s-1}(\epsilon^{-1}). \tag{80}$$

Similar to Eq. (134), for any $\tilde{\boldsymbol{\nu}} \in \mathbb{N}_+^{3d}, a_4 \in \mathbb{N}_+$, we have

$$\sum_{\|\tilde{\boldsymbol{\nu}}\|_1 + a_4 = k, \tilde{\boldsymbol{\nu}} \in \mathbb{N}^{3d}, a_4 \in \mathbb{N}} 1 = \binom{k+3d}{3d} = \frac{(k+3d)!}{(3d)! k!} \le (k+1)^{3d},$$

which implies that for any $s \in \mathbb{N}_+$,

$$\sum_{k=0}^{s-1} \sum_{\|\tilde{\boldsymbol{\alpha}}\|_1 + a_4 = k} 1 \le \sum_{k=0}^{s-1} (k+1)^{3d} \le s \cdot (s-1+1)^{3d} = s^{3d+1}. \tag{81}$$

With Eq. (80) and (81) and the face that $\|\boldsymbol{\nu}_1\|_1 + \|\boldsymbol{\nu}_2\|_1 + \|\boldsymbol{\nu}_3\|_1 + a_4 \leq s - 1$, we have

$$
0 \leq \sum_{k=0}^{s-1} \sum_{\|\tilde{\boldsymbol{\nu}}\|_1 + a_4 = k} \frac{2^{-(k-\|\boldsymbol{\nu}\|_1 - a_4)}}{\boldsymbol{\nu}_1! \boldsymbol{\nu}_2! \boldsymbol{\nu}_3! a_4!} \phi_2^{\tilde{\boldsymbol{\nu}}, a_4}(\boldsymbol{y}, t) \lesssim s^{3d+1} t_0^{-(s-1)} \alpha^{s-1} s^{s-1} \log^{s-1}(\epsilon^{-1})
$$

$$
= t_0^{-(s-1)} \alpha^{s-1} s^{3d+s} \log^{s-1}(\epsilon^{-1}). \tag{82}
$$

Again, by Lemma 13 and Eq. (71) and (82), there exists

$$
\phi_{\text{multi}}^{(4)} \in \mathcal{NN}\big(\text{width} \leq 9(N+1)+1, \text{depth} \leq 7sL\big), \tag{83}
$$

such that for any $x \in [0, 1]$ (c.f. Eq. (71)) and $y \in [0, t_0^{-(s-1)} \alpha^{s-1} s^{3d+s} \log^{s-1}(\epsilon^{-1})]$ (c.f. Eq. (82)),

$$
\left| \phi_{\text{multi}}^{(4)}\left( \phi_{\tilde{h}_\beta}^{\exp}(\boldsymbol{y}, t), \sum_{k=0}^{s-1} \sum_{\|\tilde{\boldsymbol{\nu}}\|_1 + a_4 = k} \frac{2^{-(k-\|\boldsymbol{\nu}\|_1 - a_4)}}{\boldsymbol{\nu}_1! \boldsymbol{\nu}_2! \boldsymbol{\nu}_3! a_4!} \phi_2^{\tilde{\boldsymbol{\nu}}, a_4}(\boldsymbol{y}, t) \right) \right.
$$

$$
\left. - \phi_{\tilde{h}_\beta}^{\exp}(\boldsymbol{y}, t) \times \sum_{k=0}^{s-1} \sum_{\|\tilde{\boldsymbol{\nu}}\|_1 + a_4 = k} \frac{2^{-(k-\|\boldsymbol{\nu}\|_1 - a_4)}}{\boldsymbol{\nu}_1! \boldsymbol{\nu}_2! \boldsymbol{\nu}_3! a_4!} \phi_2^{\tilde{\boldsymbol{\nu}}, a_4}(\boldsymbol{y}, t) \right|
$$

$$
\lesssim t_0^{-(s-1)} \alpha^{s-1} s^{3d+s} \log^{s-1}(\epsilon^{-1})(N+1)^{-7sL}. \tag{84}
$$

Given $\phi_{\tilde{h}_\beta}^{\exp}, \phi_2^{\tilde{\boldsymbol{\nu}}, a_4}, \phi_{\text{multi}}^{(4)}$ above, for any $\boldsymbol{y} \in \mathbb{R}^d, t \in [t_0, \infty)$, we define

$$
\phi_{\text{kde}}(\boldsymbol{y}, t) := \phi_{\text{abs}}\left( \phi_{\text{multi}}^{(4)}\left( \phi_{\tilde{h}_\beta}^{\exp}(\boldsymbol{y}, t), \sum_{k=0}^{s-1} \sum_{\|\tilde{\boldsymbol{\nu}}\|_1 + a_4 = k} \frac{2^{-(k-\|\boldsymbol{\nu}\|_1 - a_4)}}{\boldsymbol{\nu}_1! \boldsymbol{\nu}_2! \boldsymbol{\nu}_3! a_4!} \phi_2^{\tilde{\boldsymbol{\nu}}, a_4}(\boldsymbol{y}, t) \right) \right), \tag{85}
$$

By Eq. (67), (78), (81), (83), (85) and (111), we obtain that

$$
\phi_{\text{kde}} \in \mathcal{NN}\big(\text{width} \lesssim s^{6d+3} N^3 \log_2(N) \vee s^{6d+3} \log^3(\epsilon^{-1}); \text{depth} \lesssim L^3 \log_2(L) \vee s^2 \log^2(\epsilon^{-1})\big). \tag{86}
$$

**Step 4: Approximation error.**

For any $\boldsymbol{y} \in \mathcal{B}$ and any $t \in [t_0, \infty)$,

$$\left| \phi_{\mathrm{kde}}(\boldsymbol{y}, t) - \sum_{k=0}^{s-1} \frac{(-1)^k \exp(-\tilde{h}_\beta)}{k!} \frac{1}{n} \sum_{i=1}^{n} \left(h^{(i)} - \tilde{h}_\beta\right)^k \right|$$

$$\leq \left| \phi_{\mathrm{multi}}^{(4)}\left( \phi_{\tilde{h}_\beta}^{\exp}(\boldsymbol{y}, t), \sum_{k=0}^{s-1} \sum_{\|\tilde{\boldsymbol{\nu}}\|_1 + a_4 = k} \frac{2^{-(k-\|\boldsymbol{\nu}\|_1 - a_4)}}{\boldsymbol{\nu}_1! \boldsymbol{\nu}_2! \boldsymbol{\nu}_3! a_4!} \phi_2^{\tilde{\boldsymbol{\nu}}, a_4}(\boldsymbol{y}, t) \right) \right.$$

$$\left. - \exp(-\tilde{h}_\beta) \times \sum_{k=0}^{s-1} \sum_{\|\tilde{\boldsymbol{\nu}}\|_1 + a_4 = k} \frac{(-2)^{-(k-\|\boldsymbol{\nu}_2\|_1 - a_4)}}{\boldsymbol{\nu}_1! \boldsymbol{\nu}_2! \boldsymbol{\nu}_3! a_4!} \tilde{h}_\beta^{a_4} m_t^{\|\boldsymbol{\nu}_2\|_1 + 2\|\boldsymbol{\nu}_3\|_1} \sigma_t^{-2(k-a_4)} \boldsymbol{y}^{2\boldsymbol{\nu}_1 + \boldsymbol{\nu}_2} C_{\boldsymbol{x}}^{\boldsymbol{\nu}_2, \boldsymbol{\nu}_3} \right|$$

$$\leq \left| \phi_{\mathrm{multi}}^{(4)}\left( \phi_{\tilde{h}_\beta}^{\exp}(\boldsymbol{y}, t), \sum_{k=0}^{s-1} \sum_{\|\tilde{\boldsymbol{\nu}}\|_1 + a_4 = k} \frac{2^{-(k-\|\boldsymbol{\nu}\|_1 - a_4)}}{\boldsymbol{\nu}_1! \boldsymbol{\nu}_2! \boldsymbol{\nu}_3! a_4!} \phi_2^{\tilde{\boldsymbol{\nu}}, a_4}(\boldsymbol{y}, t) \right) \right.$$

$$\left. - \phi_{\tilde{h}_\beta}^{\exp}(\boldsymbol{y}, t) \times \sum_{k=0}^{s-1} \sum_{\|\tilde{\boldsymbol{\nu}}\|_1 + a_4 = k} \frac{2^{-(k-\|\boldsymbol{\nu}\|_1 - a_4)}}{\boldsymbol{\nu}_1! \boldsymbol{\nu}_2! \boldsymbol{\nu}_3! a_4!} \phi_2^{\tilde{\boldsymbol{\nu}}, a_4}(\boldsymbol{y}, t) C_{\boldsymbol{x}}^{\boldsymbol{\nu}_2, \boldsymbol{\nu}_3} \right| \qquad (\lesssim \text{Eq. (84)})$$

$$+ \underbrace{\left| \phi_{\tilde{h}_\beta}^{\exp}(\boldsymbol{y}, t) - \exp(-\tilde{h}_\beta) \right|}_{\leq \text{ Eq. (69)}} \cdot \underbrace{\sum_{k=0}^{s-1} \sum_{\|\tilde{\boldsymbol{\nu}}\|_1 + a_4 = k} \frac{2^{-(k-\|\boldsymbol{\nu}\|_1 - a_4)}}{\boldsymbol{\nu}_1! \boldsymbol{\nu}_2! \boldsymbol{\nu}_3! a_4!} \left| \phi_2^{\tilde{\boldsymbol{\nu}}, a_4}(\boldsymbol{y}, t) \right|}_{\lesssim \text{ Eq. (82)}}$$

$$+ \left| \exp(-\tilde{h}_\beta) \right| \cdot \sum_{k=0}^{s-1} \sum_{\|\tilde{\boldsymbol{\nu}}\|_1 + a_4 = k} \frac{2^{-(k-\|\boldsymbol{\nu}\|_1 - a_4)}}{\boldsymbol{\nu}_1! \boldsymbol{\nu}_2! \boldsymbol{\nu}_3! a_4!} \underbrace{\left| \phi_2^{\tilde{\boldsymbol{\nu}}, a_4}(\boldsymbol{y}, t) - \tilde{h}_\beta^{a_4} m_t^{\|\boldsymbol{\nu}_2\|_1 + 2\|\boldsymbol{\nu}_3\|_1} \sigma_t^{-2(k-a_4)} \boldsymbol{y}^{2\boldsymbol{\nu}_1 + \boldsymbol{\nu}_2} C_{\boldsymbol{x}}^{\boldsymbol{\nu}_2, \boldsymbol{\nu}_3} \right|}_{\leq \text{ Eq. (79)}}$$

$$\lesssim t_0^{-(s-1)} \alpha^{s-1} s^{3d+s} \log^{s-1}(\epsilon^{-1})(N+1)^{-7sL} + N^{-4s} L^{-4s} \cdot t_0^{-(s-1)} \alpha^{s-1} s^{3d+s} \log^{s-1}(\epsilon^{-1})$$

$$+ s^{3d+1} \cdot (2s)! \alpha^{3s} s^{9s} \log^{9s}(\epsilon^{-1}) \epsilon^s$$

$$\lesssim \alpha^{3s}(2s)! s^{3d+9s+1} \log^{9s}(\epsilon^{-1}) \epsilon^s. \tag{87}$$

Therefore, for any $\boldsymbol{y} \in \mathcal{B}, t \in [t_0, \infty)$,

$$\left| \phi_{\mathrm{kde}}(\boldsymbol{y}, t) - f_{\mathrm{kde}}(\boldsymbol{y}, t) \right|$$

$$= \underbrace{\left| \phi_{\mathrm{kde}}(\boldsymbol{y}, t) - \sum_{k=0}^{s-1} \frac{(-1)^k \exp(-\tilde{h}_\beta)}{k!} \frac{1}{n} \sum_{i=1}^{n} \left(h^{(i)} - \tilde{h}_\beta\right)^k \right|}_{\lesssim \text{ Eq. (87)}} + \underbrace{\left| \frac{1}{n} \sum_{i=1}^{n} \frac{(-1)^s \exp(-\theta^{(i)})}{s!} \left(h^{(i)} - \tilde{h}_\beta\right)^s \right|}_{\lesssim \text{ Eq. (62)}}$$

$$\tag{88}$$

$$\lesssim \alpha^{3s}(2s)! s^{3d+9s+1} \log^{9s}(\epsilon^{-1}) \epsilon^s + \alpha^2 s^{2s+2} \log^{2s+2}(\epsilon^{-1}) \epsilon^s$$

$$\lesssim \alpha^{3s}(2s)! s^{3d+9s+1} \log^{9s}(\epsilon^{-1}) \epsilon^s, \tag{89}$$

which gives that

$$0 \leq \phi_{\mathrm{kde}}(\boldsymbol{y}, t) \lesssim \left| f_{\mathrm{kde}}(\boldsymbol{y}, t) \right| + \alpha^{3s}(2s)! s^{3d+9s+1} \log^{9s}(\epsilon^{-1}) \epsilon^s \lesssim 1. \tag{90}$$

**Combine Part I and II**

Combining the approximating results from Part I (c.f. Eq. (51)) and II (c.f. Eq. (89)), we obtain that

$$\left| \phi_{\mathrm{kde}}(\boldsymbol{y}, t) - f_{\mathrm{kde}}(\boldsymbol{y}, t) \right| \lesssim \alpha^{3s}(2s)! s^{3d+9s+1} \log^{9s}(\epsilon^{-1}) \epsilon^s, \quad \text{for any } \boldsymbol{y} \in \mathbb{R}^d, t \in [t_0, \infty), \quad (91)$$

and

$$0 \leq \phi_{\mathrm{kde}}(\boldsymbol{y}, t) \lesssim 1.$$

Therefore, we finish the proof.

### D.2.3 Neural network approximations for basic functions

In the section, we give the approximation results used in approximating the Gaussian kernel density estimator functions:

- Approximating $m_t, \sigma_t^2$ by $\phi_m, \phi_{\sigma^2}$, respectively (c.f. Lemma 9).
- Approximating $m_t^k$ by $\phi_m^k$ (c.f. Proposition 1).
- Approximating $\sigma_t^{-2k}$ by $\phi_{1/\sigma^2}^k$ (c.f. Proposition 2).
- Approximating $\boldsymbol{y}^{\boldsymbol{\nu}}$ by $\phi_{\mathrm{poly}}^{\boldsymbol{\nu}}$ (c.f. Proposition 3).
- Approximating $\tilde{h}(\boldsymbol{y}, t)$ by $\phi_{\tilde{h}}$ (c.f. Proposition 4).
- Approximating $\tilde{h}_\beta(\boldsymbol{y}, t)$ by $\phi_{\tilde{h}_\beta}$ (c.f. Proposition 5).
- Approximating $\tilde{h}_\beta^k(\boldsymbol{y}, t)$ by $\phi_{\tilde{h}_\beta}^k$ (c.f. Proposition 6).
- Approximating $\exp(-\tilde{h}_\beta(\boldsymbol{y}, t))$ by $\phi_{\tilde{h}_\beta}^{\mathrm{exp}}$ (c.f. Proposition 7).

We give detailed derivations for the sizes of neural networks and approximation errors for approximating each of the above functions as below.

### D.2.4 Approximations of $m_t$ and $\sigma_t^2$ for OU process

**Lemma 9** (Approximate $m_t, \sigma_t^2$ for OU process). *For all $t \in [0, \infty)$, let $m_t := \exp(-t)$ and $\sigma_t := \sqrt{1 - \exp(-2t)}$. For any $N, L, s \in \mathbb{N}_+$ and $0 < \epsilon < 1$ such that $N^{-2}L^{-2} \leq \epsilon$, there exist some functions $\phi_m, \phi_{\sigma^2}$ implemented by some ReLU DNNs with width $48s^2(N+1)\log_2(8N)$ and depth $18s^2(L+2)\log_2(4L) + 2$ such that for all $t \in [0, \log(\epsilon^{-1})]$,*

$$|\phi_m(t) - m_t| \lesssim s^s \log^s(\epsilon^{-1})\epsilon^s,$$
$$|\phi_{\sigma^2}(t) - \sigma_t^2| \lesssim s^s \log^s(\epsilon^{-1})\epsilon^s.$$

*Proof.* For OU process, we have $m_t = \exp(-t)$ and $\sigma_t^2 = 1 - \exp(-2t)$ for all $t \in [0, \infty]$. Therefore, both $m_t, \sigma_t^2$ can be approximated by some ReLU DNNs that well approximate the exponential function $\exp(-t)$.

**Step 1:** $t \in (s\log(\epsilon^{-1}), \infty)$.

For $t > s\log(\epsilon^{-1})$, we have

$$\exp(-t) < \exp\left(-s\log(\epsilon^{-1})\right) = \epsilon^s,$$

which indicates that $\exp(-t), \forall t > s\log(\epsilon^{-1})$ can be well approximated with an error within $\epsilon$ by simply setting the output of the neural network to be zero. Therefore, we only need to consider the approximation error of a neural network for $t \in [0, s\log(\epsilon^{-1})]$.

**Step 2:** $t \in [0, s\log(\epsilon^{-1})]$.

Notice that for any $N, L \in \mathbb{N}_+$ such that $N^{-2}L^{-2} \leq \epsilon$, we have

$$N^{-2}L^{-2} \leq \epsilon \leq \frac{1}{\log(\epsilon^{-1})} \leq \frac{1}{s\log(\epsilon^{-1})}.$$

Then, it follows from Lemma 16 that there exists a function $\phi$ implemented by a ReLU DNN with width $48s^2(N+1)\log_2(8N)$ and depth $18s^2(L+2)\log_2(4L)+2$ such that for all $t \in [0, s\log(\epsilon^{-1})]$,

$$|\phi(t) - \exp(-t)| \leq \left(45s + s^s\log^s(\epsilon^{-1}) + 4\right)N^{-2s}L^{-2s} \lesssim s^s\log^s(\epsilon^{-1})N^{-2s}L^{-2s} \leq s^s\log^s(\epsilon^{-1})\epsilon^s.$$

Therefore, there exists a function $\phi$ implemented by a ReLU DNN with width $48s^2(N+1)\log_2(8N)$ and depth $18s^2(L+2)\log_2(4L) + 2$ such that for all $t \in [0, \infty)$,

$$|\phi(t) - \exp(-t)| \leq s^s\log^s(\epsilon^{-1})\epsilon^s.$$

Hence, there exists
$$\phi_m \in \mathcal{NN}\big(\text{width} \leq 48s^2(N+1)\log_2(8N); \text{depth} \leq 18s^2(L+2)\log_2\big(4L\big)+2\big)$$
such that for all $t \in [0,\infty)$,
$$|\phi_m(t) - m_t| \leq s^s \log^s(\epsilon^{-1})\epsilon^s.$$

Similar, by letting $\tilde{t} = 2t$, there exist
$$\tilde{\phi}_{\sigma^2} \in \mathcal{NN}\big(\text{width} \leq 48s^2(N+1)\log_2(8N); \text{depth} \leq 18s^2(L+2)\log_2\big(4L\big)+2\big)$$
such that for all $\tilde{t} \in [0,\infty)$,

$$|\tilde{\phi}_{\sigma^2}(\tilde{t}) - \exp(-\tilde{t})| \leq s^s \log^s(\epsilon^{-1})\epsilon^s.$$

Define $\phi_{\sigma^2}(t) := \tilde{\phi}_{\sigma^2}(2t)$. Clear, we have
$$\phi_{\sigma^2} \in \mathcal{NN}\big(\text{width} \leq 48s^2(N+1)\log_2(8N); \text{depth} \leq 18s^2(L+2)\log_2\big(4L\big)+2\big)$$
and
$$|\phi_{\sigma^2}(t) - \exp(-2t)| = |\tilde{\phi}_{\sigma^2}(\tilde{t}) - \exp(-\tilde{t})| \leq s^s \log^s(\epsilon^{-1})\epsilon^s.$$
$\square$

**Proposition 1** (Approximating $m_t^k$ by $\phi_m^k$)**.** *For any $k,s \in \mathbb{N}_+$ with $k \leq s$ and $0 < \epsilon < 1$. Let $m_t := \exp(-t), \forall t \in [0,\infty)$. There exist $N, L \in \mathbb{N}_+$ with $N^{-2}L^{-2} \leq \epsilon$, and*
$$\phi_m^k \in \mathcal{NN}\big(\text{width} \lesssim s^2 N \log_2(N); \text{depth} \lesssim s^2 L \log_2(L)\big) \tag{92}$$
*such that*
$$\left|\phi_m^k(t) - m_t^k\right| \lesssim s^s \log^s(\epsilon^{-1})\epsilon^s, \quad \text{for any } t \in [0,\infty]. \tag{93}$$
*and*
$$0 \leq \phi_m(t) \lesssim 1 + s^s \log^s(\epsilon^{-1})\epsilon^s \lesssim 1. \tag{94}$$

*Proof.* Since $m_t^k = \exp\big(-kt\big)$. By Lemma 9, there exist
$$\tilde{\phi}_m \in \mathcal{NN}\big(\text{width} \lesssim s^2 N \log_2(N); \text{depth} \lesssim s^2 L \log_2(L)\big)$$
such that
$$\left|\tilde{\phi}_m(t) - m_t^{\|\boldsymbol{\nu}_1\|_1 + 2\|\boldsymbol{\nu}_2\|_1}\right| \leq s^s \log^s(\epsilon^{-1})\epsilon^s, \quad \text{for any } t \in [0,\infty].$$
Then, we define
$$\phi_m(t) := \text{ReLU}\big(\tilde{\phi}_m(t)\big) + \text{ReLU}\big(-\tilde{\phi}_m(t)\big) \tag{95}$$
and clearly, we have
$$\phi_m \in \mathcal{NN}\big(\text{width} \lesssim s^2 N \log_2(N); \text{depth} \lesssim s^2(L\log_2(L)\big)$$
and by the fact that $|x| = \text{ReLU}(x) + \text{ReLU}(-x)$,
$$\left|\phi_m(t) - m_t^{\|\boldsymbol{\nu}_1\|_1 + 2\|\boldsymbol{\nu}_2\|_1}\right| \leq \left|\tilde{\phi}_m(t) - m_t^{\|\boldsymbol{\nu}_1\|_1 + 2\|\boldsymbol{\nu}_2\|_1}\right| \lesssim s^s \log^s(\epsilon^{-1})\epsilon^s, \quad \text{for any } t \in [0,\infty].$$
Therefore,
$$0 \leq \phi_m(t) \lesssim 1 + s^s \log^s(\epsilon^{-1})\epsilon^s \lesssim 1, \quad \text{for any } t \in [0,\infty).$$
$\square$

**Proposition 2** (Approximating $\sigma_t^{-2k}$)**.** *For any $k,s \in \mathbb{N}_+$ with $k \leq s$ and $0 < \epsilon < t_0 < 1/2$. Let $\sigma_t := \sqrt{1 - \exp(-2t)}, \forall t \in [t_0, \infty)$. There exist $N, L \in \mathbb{N}_+$ with $N^{-2}L^{-2} \leq \epsilon$, and*
$$\phi_{1/\sigma^2}^k \in \mathcal{NN}\big(\text{width} \lesssim s^3 N \log_2(N) \vee s^3 \log^3(\epsilon^{-1});$$
$$\text{depth} \lesssim s^2 L \sqrt{\log(\epsilon^{-1})} \log_2(L\sqrt{\log(\epsilon^{-1})}) \vee s^2 \log^2(\epsilon^{-1})\big),$$
*such that*
$$\left|\phi_{1/\sigma^2}^k(t) - \sigma_t^{-2k}\right| \lesssim k! s^{3s} \log^{3s}(\epsilon^{-1})\epsilon^s, \quad \text{for any } t \in [t_0, \infty),$$
*and*
$$0 \leq \phi_{1/\sigma^2}^k(t) \lesssim t_0^{-k}.$$

*Proof.* When $k = 0$, $\frac{1}{\sigma_t^{2k}} \equiv 1$, which is trivial. In what follows, we focus on $k \geq 1$.

Note that $\sigma_t = \sqrt{1 - \exp(-2t)}$ for all $t \in [0, \infty)$, we have

$$\sigma_t^{2k} = \left(1 - \exp(-2t)\right)^k = \sum_{r=0}^{k} \frac{(-1)^r k!}{r!} \exp(-2rt)$$

For each $k = 1, \ldots, s$ and each $r = 0, 1, \ldots, k$, by Lemma 9 there exists

$$\phi_{\exp} \in \mathcal{NN}\left(\text{width} \lesssim (3s)^2 N \log_2(N); \text{depth} \lesssim (3s)^2 L \log_2(L)\right)$$

such that

$$\left|\phi_{\exp}(t) - \exp(-2rt)\right| \lesssim s^{3s} \log^{3s}(\epsilon^{-1})\epsilon^{3s}, \quad \text{for any } t \in [0, \infty). \tag{96}$$

Define

$$\phi_{\sigma^2}^k(t) := \sum_{r=0}^{k} \frac{(-1)^r k!}{r!} \phi_{\exp}(t), \quad \text{for any } t \in [t_0, \infty).$$

Clearly,

$$\phi_{\sigma^2}^k \in \mathcal{NN}\left(\text{width} \lesssim s^3 N \log_2(N); \text{depth} \lesssim s^2(L \log_2(L))\right) \tag{97}$$

and

$$
\begin{aligned}
\left|\phi_{\sigma^2}^k(t) - \sigma_t^{2k}\right| &\leq \sum_{r=0}^{k} \frac{k!}{r!} \left|\phi_{\exp}(t) - \exp(-2rt)\right| \\
&\lesssim ek! s^{3s} \log^{3s}(\epsilon^{-1})\epsilon^{3s} \qquad \text{(by Eq. (96) and } \sum_{r=0}^{k} \frac{1}{r!} \leq e) \\
&\lesssim k! s^{3s} \log^{3s}(\epsilon^{-1})\epsilon^{3s}. \tag{98}
\end{aligned}
$$

Note that for any $0 < t_0 \leq 1/2$,

$$\sigma_t^2 = 1 - \exp(-2t) \geq 1 - \exp(-2t_0) \geq t_0, \quad \text{for any } t \in [t_0, \infty], \tag{99}$$

which gives that

$$
\begin{aligned}
\phi_{\sigma^2}^k(t) &\geq \sigma_t^{2k} - k! s^{3s} \log^{3s}(\epsilon^{-1})\epsilon^{3s} \geq t_0^k - k! s^{3s} \log^{3s}(\epsilon^{-1})\epsilon^{3s} \gtrsim t_0^k, \\
\phi_{\sigma^2}^k(t) &\leq \sigma_t^{2k} + k! s^{3s} \log^{3s}(\epsilon^{-1})\epsilon^{3s} \leq 1 + k! s^{3s} \log^{3s}(\epsilon^{-1})\epsilon^{3s} \lesssim 1. \tag{100}
\end{aligned}
$$

Recall that $0 < \epsilon \leq t_0$. By Lemma 23, there exists

$$\phi_{\text{rec}} \in \mathcal{NN}\left(\text{width} \lesssim s^3 \log^3(\epsilon^{-1}); \text{depth} \lesssim s^2 \log^2(\epsilon^{-1})\right), \tag{101}$$

such that for any $x \in [t_0^k, 1] \subseteq [\epsilon^s, \epsilon^{-s}]$ and $x' \in \mathbb{R}$,

$$\left|\phi_{\text{rec}}(x') - \frac{1}{x}\right| \leq \epsilon^s + \frac{|x' - x|}{\epsilon^{2s}}. \tag{102}$$

For each $k = 1, \ldots, s$, define

$$\phi_{1/\sigma^2}^k(t) = \text{ReLU}\left(\phi_{\text{rec}}\left(\phi_{\sigma^2}^k(t)\right)\right) + \text{ReLU}\left(-\phi_{\text{rec}}\left(\phi_{\sigma^2}^k(t)\right)\right), \quad \text{for any } t \in [0, \infty). \tag{103}$$

Recall the fact that $|x| = \text{ReLU}(x) + \text{ReLU}(-x)$, we have $\phi_{1/\sigma^2}^k(t) \geq 0$ for any $t \in [0, \infty)$ and

$$
\begin{aligned}
\left|\phi_{1/\sigma^2}^k(t) - \sigma_t^{-2k}\right| &= \left|\left|\phi_{\text{rec}}\left(\phi_{\sigma^2}^k(t)\right)\right| - \sigma_t^{-2k}\right| \\
&\leq \left|\phi_{\text{rec}}\left(\phi_{\sigma^2}^k(t)\right) - \sigma_t^{-2k}\right| \\
&\leq \epsilon^s + \epsilon^{-2s} \underbrace{\left|\phi_{\sigma^2}^k(t) - \sigma_t^{2k}\right|}_{\leq \text{Eq. (98)}} \\
&\lesssim \epsilon^s + k! s^{3s} \log^{3s}(\epsilon^{-1})\epsilon^{3s-2s} \\
&\lesssim k! s^{3s} \log^{3s}(\epsilon^{-1})\epsilon^s, \tag{104}
\end{aligned}
$$

which gives that

$$0 \le \phi^k_{1/\sigma^2}(t) \lesssim \sigma_t^{-2k} + k!s^{3s}\log^{3s}(\epsilon^{-1})\epsilon^s \lesssim t_0^{-k}. \tag{105}$$

Moreover, Eq. (97) and (101) indicates that

$$\phi^k_{1/\sigma^2} \in \mathcal{NN}\big(\text{width} \lesssim s^3 N \log_2(N) \vee s^3 \log^3(\epsilon^{-1}); \text{depth} \lesssim s^2 L \log_2(L) \vee s^2 \log^2(\epsilon^{-1})\big). \tag{106}$$

$\square$

**Proposition 3** (Approximating $\boldsymbol{y^\nu}$)**.** *Given* $k, s \in \mathbb{N}_+, \boldsymbol{\nu} \in \mathbb{N}^d$ *with* $\|\boldsymbol{\nu}\|_1 \le k \le s$ *and* $0 < \epsilon < 1$. *Let*

$$\boldsymbol{y} \in \mathcal{B} := [-2\sqrt{2\alpha s \log(\epsilon^{-1})}, 2\sqrt{2\alpha s \log(\epsilon^{-1})}]^d.$$

*There exist* $N, L \in \mathbb{N}_+$ *with* $N^{-2}L^{-2} \le \epsilon$, *and*

$$\phi^{\boldsymbol{\nu}}_{poly} \in \mathcal{NN}\big(\text{width} \le 9(N+1) + k - 1; \text{depth} \le 7s(k-1)L\big) \tag{107}$$

*such that*

$$\big|\phi^{\boldsymbol{\nu}}_{poly}(\boldsymbol{y}) - \boldsymbol{y^\nu}\big| \lesssim k\alpha^{k/2}s^{k/2}\log^{k/2}(\epsilon^{-1})(N+1)^{-7sL}, \quad \text{for any } \boldsymbol{y} \in \mathcal{B}. \tag{108}$$

*and*

$$\big|\phi^{\boldsymbol{\nu}}_{poly}(\boldsymbol{y})\big| \lesssim k\alpha^{k/2}s^{k/2}\log^{k/2}(\epsilon^{-1}). \tag{109}$$

*Proof.* By Proposition 9, there exists

$$\phi^{\boldsymbol{\nu}}_{\text{poly}} \in \mathcal{NN}\big(\text{width} \le 9(N+1) + k - 1; \text{depth} \le 7s(k-1)L\big)$$

such that

$$\big|\phi^{\boldsymbol{\nu}}_{\text{poly}}(\boldsymbol{y}) - \boldsymbol{y^\nu}\big| \le 30k\big(2\sqrt{2\alpha s \log(\epsilon^{-1})}\big)^k (N+1)^{-7sL}$$
$$\lesssim k\alpha^{k/2}s^{k/2}\log^{k/2}(\epsilon^{-1})(N+1)^{-7sL}, \quad \text{for any } \boldsymbol{y} \in \mathcal{B}.$$

which gives that

$$\big|\phi^{\boldsymbol{\nu}}_{\text{poly}}(\boldsymbol{y})\big| \lesssim |\boldsymbol{y^\nu}| + k\alpha^{k/2}s^{k/2}\log^{k/2}(\epsilon^{-1})(N+1)^{-7sL}$$
$$\lesssim k\alpha^{k/2}s^{k/2}\log^{k/2}(\epsilon^{-1}) + k\alpha^{k/2}s^{k/2}\log^{k/2}(\epsilon^{-1})(N+1)^{-7sL}$$
$$\lesssim k\alpha^{k/2}s^{k/2}\log^{k/2}(\epsilon^{-1}).$$

$\square$

**Proposition 4** (Approximating $\tilde{h}(\boldsymbol{y}, t)$)**.** *Given* $s \in \mathbb{N}_+$ *and* $\tilde{h}$ *is defined in Eq. (57):*

$$\tilde{h}(\boldsymbol{y}, t) := \Big(\frac{1}{n}\sum_{i=1}^n \big(h^{(i)}(\boldsymbol{y}, t)\big)^s\Big)^{1/s}.$$

*For any* $0 < \epsilon < 1$, *There exist* $N, L \in \mathbb{N}_+$ *with* $N^{-2}L^{-2} \le \epsilon$, *and*

$$\phi_{\tilde{h}} \in \mathcal{NN}\big(\text{width} \lesssim s^{3d+2}N\log_2(N) \vee s^{3d+2}\log^3(\epsilon^{-1}); \text{depth} \lesssim s^2 L \log_2(L) \vee s^2 \log^2(\epsilon^{-1})\big)$$

*such that*

$$\big|\phi_{\tilde{h}}(\boldsymbol{y}, t) - \tilde{h}(\boldsymbol{y}, t)\big| \lesssim \alpha^2 s!s^{2s+2}\log^{2s+2}(\epsilon^{-1})\epsilon^s, \quad \text{for any } \boldsymbol{y} \in \mathcal{B}, \text{ and } t \in [t_0, \infty),$$

*and*

$$0 \le \phi_{\tilde{h}}(\boldsymbol{y}, t) \lesssim t_0^{-1}\alpha s \log(\epsilon^{-1}).$$

*Proof.* **Step 1: Taylor expansion of $\tilde{h}(\boldsymbol{y}, t)$.** Recall from Eq. (63) that $C^{\boldsymbol{\nu_2}, \boldsymbol{\nu_3}}_{\boldsymbol{x}} :=$ $\frac{1}{n}\sum_{i=1}^n \big(\boldsymbol{x}^{(i)}\big)^{\boldsymbol{\nu_2}+2\boldsymbol{\nu_3}}$ and $\tilde{\boldsymbol{\nu}} := [\boldsymbol{\nu_1}, \boldsymbol{\nu_2}, \boldsymbol{\nu_2}] \in \mathbb{N}^{3d}_+$. Similar to the derivation for Eq. (65), we can obtain

$$\tilde{h}(\boldsymbol{y}, t) := \Big(\frac{1}{n}\sum_{i=1}^n \big(h^{(i)}(\boldsymbol{y}, t)\big)^s\Big)^{1/s} = \Bigg(\sum_{\|\tilde{\boldsymbol{\nu}}\|_1 = s} \frac{C^{\boldsymbol{\nu_2}, \boldsymbol{\nu_3}}_{\boldsymbol{x}} s! m_t^{\|\boldsymbol{\nu_2}\|_1 + 2\|\boldsymbol{\nu_3}\|_1}}{(-2)^{s-\|\boldsymbol{\nu_2}\|_1}\sigma_t^{2s}\boldsymbol{\nu_1}!\boldsymbol{\nu_2}!\boldsymbol{\nu_3}!}\boldsymbol{y}^{2\boldsymbol{\nu_1}+\boldsymbol{\nu_2}}\Bigg)^{1/s} \tag{110}$$

**Step 2: Approximating each base function.** Notice that

$$|x| = \text{ReLU}(x) + \text{ReLU}(-x), \quad \text{for any } x \in \mathbb{R}.$$

Therefore, we define $\phi_{\text{abs}}$ to approximate $|x|$ by

$$\phi_{\text{abs}}(x) := \text{ReLU}(x) + \text{ReLU}(-x), \quad \text{for any } x \in \mathbb{R},$$

and we have

$$\phi_{\text{abs}} \in \mathcal{N}\mathcal{N}\big(\text{width} = 2, \text{depth} = 1\big). \tag{111}$$

For each $\tilde{\boldsymbol{\nu}} \in \mathbb{N}^{3d}$ such that $\|\tilde{\boldsymbol{\nu}}\|_1 \leq s$, we have $0 \leq \|\boldsymbol{\nu}_2\|_1 + 2\|\boldsymbol{\nu}_3\|_1 \leq 2s$ and $2\|\boldsymbol{\nu}_1\|_1 + \|\boldsymbol{\nu}_2\|_1 \leq 2s$. By Propositions 1 to 3, there exist

$$\phi_m^{\|\boldsymbol{\nu}_2\|_1 + 2\|\boldsymbol{\nu}_3\|_1} \in \mathcal{N}\mathcal{N}\big(\text{width} \lesssim s^2 N \log_2(N); \text{depth} \lesssim s^2(L \log_2(L)), \tag{112}$$

$$\phi_{1/\sigma^2}^s \in \mathcal{N}\mathcal{N}\big(\text{width} \lesssim s^3 N \log_2(N) \vee s^3 \log^3(\epsilon^{-1});$$

$$\text{depth} \lesssim s^2 L \log_2(L) \vee s^2 \log^2(\epsilon^{-1})\big), \tag{113}$$

$$\phi_{\text{poly}}^{2\boldsymbol{\nu}_1 + \boldsymbol{\nu}_2} \in \mathcal{N}\mathcal{N}\big(\text{width} \leq 9(N+1) + 2s - 1; \text{depth} \leq 7s(2s-1)L\big) \tag{114}$$

such that

$$\left|\phi_m^{\|\boldsymbol{\nu}_2\|_1 + 2\|\boldsymbol{\nu}_3\|_1}(t) - m_t^{\|\boldsymbol{\nu}_2\|_1 + 2\|\boldsymbol{\nu}_3\|_1}\right| \lesssim s^{2s} \log^{2s}(\epsilon^{-1}) \epsilon^{2s}, \quad \text{for any } t \in [0, \infty], \tag{115}$$

$$\left|\phi_{1/\sigma^2}^s(t) - \sigma_t^{-2s}\right| \lesssim s! s^{3s} \log^{3s}(\epsilon^{-1}) \epsilon^s, \quad \text{for any } t \in [t_0, \infty), \tag{116}$$

$$\left|\phi_{\text{poly}}^{2\boldsymbol{\nu}_1 + \boldsymbol{\nu}_2}(\boldsymbol{y}) - \boldsymbol{y}^{2\boldsymbol{\nu}_1 + \boldsymbol{\nu}_2}\right| \lesssim \alpha^s s^s \log^s(\epsilon^{-1})(N+1)^{-7sL}, \quad \text{for any } \boldsymbol{y} \in \mathcal{B}. \tag{117}$$

and

$$0 \leq \phi_m^{\|\boldsymbol{\nu}_2\|_1 + 2\|\boldsymbol{\nu}_3\|_1}(t) \lesssim 1, \tag{118}$$

$$0 \leq \phi_{1/\sigma^2}^s(t) \lesssim t_0^{-s}. \tag{119}$$

Fix $\{\boldsymbol{x}^{(i)}\}_{i=1}^n$, for any $\boldsymbol{y} \in \mathbb{R}^d, t \in [t_0, \infty)$, define

$$\phi_{\text{poly}}^{\tilde{\boldsymbol{\nu}}}(\boldsymbol{y}) := C_{\boldsymbol{x}}^{\boldsymbol{\nu}_2, \boldsymbol{\nu}_3} \phi_{\text{poly}}^{2\boldsymbol{\nu}_1 + \boldsymbol{\nu}_2}(\boldsymbol{y}), \tag{120}$$

where $C_{\boldsymbol{x}}^{\boldsymbol{\nu}_2, \boldsymbol{\nu}_3} := \frac{1}{n} \sum_{i=1}^n \big(\boldsymbol{x}^{(i)}\big)^{\boldsymbol{\nu}_2 + 2\boldsymbol{\nu}_3}$. Clearly, we have

$$\phi_{\text{poly}}^{\tilde{\boldsymbol{\nu}}} \in \mathcal{N}\mathcal{N}\big(\text{width} \leq 9(N+1) + 2s - 1; \text{depth} \leq 7s(2s-1)L\big). \tag{121}$$

Recall from Assumption 4 that $\sup_{i \in [n]} |\boldsymbol{x}^{(i)}| \leq \sqrt{2\alpha s \log(\epsilon^{-1})}$,

$$\left|\phi_{\text{poly}}^{\tilde{\boldsymbol{\nu}}}(\boldsymbol{y}) - C_{\boldsymbol{x}}^{\boldsymbol{\nu}_2, \boldsymbol{\nu}_3} \boldsymbol{y}^{2\boldsymbol{\nu}_1 + \boldsymbol{\nu}_2}\right| \leq C_{\boldsymbol{x}}^{\boldsymbol{\nu}_2, \boldsymbol{\nu}_3} \left|\phi_{\text{poly}}^{2\boldsymbol{\nu}_1 + \boldsymbol{\nu}_2}(\boldsymbol{y}) - \boldsymbol{y}^{2\boldsymbol{\nu}_1 + \boldsymbol{\nu}_2}\right|$$

$$\lesssim \alpha^{2s} s^{2s} \log^{2s}(\epsilon^{-1})(N+1)^{-7sL}, \quad \text{for any } \boldsymbol{y} \in \mathcal{B}. \tag{122}$$

which gives that

$$\left|\phi_{\text{poly}}^{\tilde{\boldsymbol{\nu}}}(\boldsymbol{y})\right| \lesssim \left|C_{\boldsymbol{x}}^{\boldsymbol{\nu}_2, \boldsymbol{\nu}_3} \boldsymbol{y}^{2\boldsymbol{\nu}_1 + \boldsymbol{\nu}_2}\right| + \alpha^{2s} s^{2s} \log^{2s}(\epsilon^{-1})(N+1)^{-7sL}$$

$$\lesssim \big(\sqrt{\alpha s \log(\epsilon^{-1})}\big)^{2\|\boldsymbol{\nu}_1\|_1 + 2\|\boldsymbol{\nu}_2\|_1 + \|\boldsymbol{\nu}_3\|_1} = \alpha^s s^s \log^s(\epsilon^{-1}). \tag{123}$$

By Lemma 13 and Eq. (119) and (123), there exists

$$\phi_{\text{multi}}^{(1)} \in \mathcal{N}\mathcal{N}\big(\text{width} \leq 9(N+1) + 1, \text{depth} \leq 7sL\big), \tag{124}$$

such that

$$\left|\phi_{\text{multi}}^{(1)}\big(\phi_{1/\sigma^2}^s(t), \phi_{\text{poly}}^{\tilde{\boldsymbol{\nu}}}(\boldsymbol{y})\big) - \phi_{1/\sigma^2}^s(t) \cdot \phi_{\text{poly}}^{2\boldsymbol{\nu}_1 + \boldsymbol{\nu}_2}(\boldsymbol{y}) C_{\boldsymbol{x}}^{\boldsymbol{\nu}_2, \boldsymbol{\nu}_3}\right| \lesssim t_0^{-s} \alpha^s s^s \log^s(\epsilon^{-1})(N+1)^{-7sL}. \tag{125}$$

Therefore, for any $\tilde{\nu} \in \mathbb{N}^{3d}$ such that $\|\tilde{\nu}\|_1 = \|\nu_1\|_1 + \|\nu_2\|_1 + \|\nu_3\|_1 \leq s$,

$$\left| \phi_{\text{multi}}^{(1)}\big(\phi_{1/\sigma^2}^s(t), \phi_{\text{poly}}^{\tilde{\nu}}(y)\big) - \sigma_t^{-2s} y^{2\nu_1+\nu_2} C_x^{\nu_2,\nu_3} \right|$$

$$\leq \underbrace{\left| \phi_{\text{multi}}^{(1)}\big(\phi_{1/\sigma^2}^s(t), \phi_{\text{poly}}^{\tilde{\nu}}(y)\big) - \phi_{1/\sigma^2}^s(t) \cdot \phi_{\text{poly}}^{\tilde{\nu}}(y) \right|}_{\lesssim \text{ Eq. (125)}} + \underbrace{\left| \phi_{1/\sigma^2}^s(t) - \sigma_t^{-2s} \right|}_{\lesssim \text{ Eq. (116)}} \cdot \underbrace{\left| \phi_{\text{poly}}^{\tilde{\nu}}(y) \right|}_{\lesssim \text{ Eq. (123)}}$$

$$+ \sigma_t^{-2s} \underbrace{\left| \phi_{\text{poly}}^{\tilde{\nu}}(y) - y^{2\nu_1+\nu_2} C_x^{\nu_2,\nu_3} \right|}_{\lesssim \text{ Eq. (122)}}$$

$$\lesssim t_0^{-s} \alpha^s s^s \log^s(\epsilon^{-1})(N+1)^{-7sL} + s! s^{3s} \log^{3s}(\epsilon^{-1}) \epsilon^s \cdot \alpha^s s^s \log^s(\epsilon^{-1})$$

$$+ t_0^{-s} \cdot \alpha^{2s} s^{2s} \log^{2s}(\epsilon^{-1})(N+1)^{-7sL}$$

$$\lesssim \alpha^s s^s \log^s(\epsilon^{-1}) \epsilon^s + s! t_0^{-s} \alpha^s s^{4s} \log^{4s}(\epsilon^{-1}) \epsilon^{2s} + \alpha^{2s} s^{2s} \log^{2s}(\epsilon^{-1}) \epsilon^s \qquad \text{(by Eq. (131))}$$

$$\lesssim \alpha^s s! s^{4s} \log^{4s}(\epsilon^{-1}) \epsilon^s, \qquad\qquad\qquad (126)$$

which gives that

$$\left| \phi_{\text{multi}}^{(1)}\big(\phi_{1/\sigma^2}^s(t), \phi_{\text{poly}}^{\tilde{\nu}}(y)\big) \right| \lesssim \left| \sigma_t^{-2s} y^{2\nu_1+\nu_2} C_x^{\nu_2,\nu_3} \right| + \alpha^s s! s^{4s} \log^{4s}(\epsilon^{-1}) \epsilon^s$$

$$\lesssim t_0^{-s} \alpha^s s^s \log^s(\epsilon^{-1}) + \alpha^s s! s^{4s} \log^{4s}(\epsilon^{-1}) \epsilon^s$$

$$\lesssim t_0^{-s} \alpha^s s^s \log^s(\epsilon^{-1}). \qquad\qquad (127)$$

Again, by Lemma 13 and Eq. (118) and (127), we have

$$\phi_{\text{multi}}^{(2)} \in \mathcal{NN}\big(\text{width} \leq 9(N+1)+1, \text{depth} \leq 7sL\big) \qquad (128)$$

such that

$$\left| \phi_{\text{multi}}^{(2)}\big(\phi_m^{\|\nu_2\|_1+2\|\nu_3\|_1}(t), \phi_{\text{multi}}^{(1)}\big(\phi_{1/\sigma^2}^s(t), \phi_{\text{poly}}^{\tilde{\nu}}(y)\big)\big) \right.$$

$$\left. - \phi_m^{\|\nu_2\|_1+2\|\nu_3\|_1}(t) \cdot \phi_{\text{multi}}^{(1)}\big(\phi_{1/\sigma^2}^s(t), \phi_{\text{poly}}^{\tilde{\nu}}(y)\big) \right|$$

$$\lesssim t_0^{-s} \alpha^s s^s \log^s(\epsilon^{-1})(N+1)^{-7sL}. \qquad\qquad (129)$$

**Step 3: Construct the whole neural network.**

Given $\phi_{\text{abs}}, \phi_m^{\|\nu_2\|_1+2\|\nu_3\|_1}, \phi_{1/\sigma^2}^s, \phi_{\text{poly}}^{\tilde{\nu}}, \phi_{\text{multi}}^{(1)}, \phi_{\text{multi}}^{(2)}$ above, for all $y \in \mathcal{B}, t \in [t_0, \infty)$, we define $\phi_{\tilde{h}}$ by

$$\phi_{\tilde{h}, \tilde{\nu}}(y, t) := \phi_{\text{abs}}\left( \phi_{\text{multi}}^{(2)}\big(\phi_m^{\|\nu_2\|_1+2\|\nu_3\|_1}(t), \phi_{\text{multi}}^{(1)}\big(\phi_{1/\sigma^2}^s(t), \phi_{\text{poly}}^{\tilde{\nu}}(y)\big)\big) \right). \qquad (130)$$

For any $N, L, s \in \mathbb{N}_+$,

$$(N+1)^{-7sL} = (N+1)^{-4sL}(N+1)^{-3sL} \leq N^{-4sL} 2^{-3sL} = N^{-4s}(2^{\frac{3}{4}L})^{-4s} < N^{-4s} L^{-4s} \leq \epsilon^{2s}. \qquad (131)$$

Then, we obtain

$$
\left| \phi_{\tilde{h}, \tilde{\boldsymbol{\nu}}}(\boldsymbol{y}, t) - m_t^{\|\boldsymbol{\nu}_2\|_1 + 2\|\boldsymbol{\nu}_3\|_1} \sigma_t^{-2s} \boldsymbol{y}^{2\boldsymbol{\nu}_1 + \boldsymbol{\nu}_2} C_{\boldsymbol{x}}^{\boldsymbol{\nu}_2, \boldsymbol{\nu}_3} \right|
$$

$$
\leq \left| \phi_{\mathrm{multi}}^{(2)} \left( \phi_m^{\|\boldsymbol{\nu}_2\|_1 + 2\|\boldsymbol{\nu}_3\|_1}(t), \phi_{\mathrm{multi}}^{(1)} \left( \phi_{1/\sigma^2}^s(t), \phi_{\mathrm{poly}}^{\tilde{\boldsymbol{\nu}}}(\boldsymbol{y}) \right) \right) - m_t^{\|\boldsymbol{\nu}_2\|_1 + 2\|\boldsymbol{\nu}_3\|_1} \sigma_t^{-2s} \boldsymbol{y}^{2\boldsymbol{\nu}_1 + \boldsymbol{\nu}_2} C_{\boldsymbol{x}}^{\boldsymbol{\nu}_2, \boldsymbol{\nu}_3} \right|
$$
$$
\text{(by } \phi_{\mathrm{abs}}(\cdot) = |\cdot| )
$$

$$
\leq \underbrace{\left| \phi_{\mathrm{multi}}^{(2)} \left( \phi_m^{\|\boldsymbol{\nu}_2\|_1 + 2\|\boldsymbol{\nu}_3\|_1}(t), \phi_{\mathrm{multi}}^{(1)} \left( \phi_{1/\sigma^2}^s(t), \phi_{\mathrm{poly}}^{\tilde{\boldsymbol{\nu}}}(\boldsymbol{y}) \right) \right) - \phi_m^{\|\boldsymbol{\nu}_2\|_1 + 2\|\boldsymbol{\nu}_3\|_1}(t) \cdot \phi_{\mathrm{multi}}^{(1)} \left( \phi_{1/\sigma^2}^s(t), \phi_{\mathrm{poly}}^{\tilde{\boldsymbol{\nu}}}(\boldsymbol{y}) \right) \right|}_{\lesssim \text{ Eq. (129)}}
$$

$$
+ \underbrace{\left| \phi_m^{\|\boldsymbol{\nu}_2\|_1 + 2\|\boldsymbol{\nu}_3\|_1}(t) - m_t^{\|\boldsymbol{\nu}_2\|_1 + 2\|\boldsymbol{\nu}_3\|_1} \right|}_{\leq \text{Eq. (115)}} \cdot \underbrace{\left| \phi_{\mathrm{multi}}^{(1)} \left( \phi_{1/\sigma^2}^s(t), \phi_{\mathrm{poly}}^{\tilde{\boldsymbol{\nu}}}(\boldsymbol{y}) \right) \right|}_{\lesssim \text{ Eq. (127)}}
$$

$$
+ m_t^{\|\boldsymbol{\nu}_2\|_1 + 2\|\boldsymbol{\nu}_3\|_1} \underbrace{\left| \phi_{\mathrm{multi}}^{(1)} \left( \phi_{1/\sigma^2}^s(t), \phi_{\mathrm{poly}}^{\tilde{\boldsymbol{\nu}}}(\boldsymbol{y}) \right) - \sigma_t^{-2s} \boldsymbol{y}^{2\boldsymbol{\nu}_1 + \boldsymbol{\nu}_2} C_{\boldsymbol{x}}^{\boldsymbol{\nu}_2, \boldsymbol{\nu}_3} \right|}_{\leq \text{Eq. (126)}}
$$

$$
\lesssim t_0^{-s} \alpha^s s^s \log^s(\epsilon^{-1})(N+1)^{-7sL} + s^{2s} \log^{2s}(\epsilon^{-1}) \epsilon^{2s} \cdot t_0^{-s} \alpha^s s^s \log^s(\epsilon^{-1}) + \alpha^s s! s^{4s} \log^{4s}(\epsilon^{-1}) \epsilon^s
$$
$$
\leq \alpha^s s^s \log^s(\epsilon^{-1}) \epsilon^s + \alpha^s s^{3s} \log^{3s}(\epsilon^{-1}) \epsilon^s + \alpha^s s! s^{4s} \log^{4s}(\epsilon^{-1}) \epsilon^s \qquad \text{(by } \epsilon \leq t_0 \text{ and Eq. (131))}
$$
$$
\lesssim \alpha^s s! s^{4s} \log^{4s}(\epsilon^{-1}) \epsilon^s. \tag{132}
$$

Therefore, for $\tilde{\boldsymbol{\nu}} \in \mathbb{N}_+^{3d}$ such that $\|\tilde{\boldsymbol{\nu}}\|_1 = s$ and $0 < t_0 \leq 1/2$, for any $\boldsymbol{y} \in \mathcal{B}, t \in [t_0, \infty)$,

$$
0 \leq \phi_{\tilde{h}, \tilde{\boldsymbol{\nu}}}(\boldsymbol{y}, t) \lesssim m_t^{2\|\boldsymbol{\nu}_2\|_1 + \|\boldsymbol{\nu}_3\|_1} \sigma_t^{-2s} \boldsymbol{y}^{2\boldsymbol{\nu}_1 + \boldsymbol{\nu}_2} C_{\boldsymbol{x}}^{\boldsymbol{\nu}_2, \boldsymbol{\nu}_3} + \alpha^s s! s^{4s} \log^{4s}(\epsilon^{-1}) \epsilon^s
$$
$$
\lesssim t_0^{-s} \alpha^s s^s \log^s(\epsilon^{-1}). \tag{133}
$$

Notice that

$$
(s/d + 1)^{d-1} \leq \sum_{\|\tilde{\boldsymbol{\nu}}\|_1 = s, \tilde{\boldsymbol{\nu}} \in \mathbb{N}^{3d}} 1 = \binom{s + 3d - 1}{3d - 1} \leq (s+1)^{3d-1}, \tag{134}
$$

which gives that

$$
0 \leq \sum_{\|\tilde{\boldsymbol{\nu}}\|_1 = s} \frac{s!}{2^{s - \|\boldsymbol{\nu}_2\|_1} \boldsymbol{\nu}_1! \boldsymbol{\nu}_2! \boldsymbol{\nu}_3!} \phi_{\tilde{h}, \tilde{\boldsymbol{\nu}}}(\boldsymbol{y}, t)
$$
$$
\lesssim \sum_{\|\tilde{\boldsymbol{\nu}}\|_1 = s} \frac{2^{-(s - \|\boldsymbol{\nu}\|_1)} s!}{\boldsymbol{\nu}_1! \boldsymbol{\nu}_2! \boldsymbol{\nu}_3!} \left( \frac{\left| m_t^{\|\boldsymbol{\nu}_2\|_1 + 2\|\boldsymbol{\nu}_3\|_1} \boldsymbol{y}^{2\boldsymbol{\nu}_1 + \boldsymbol{\nu}_2} C_{\boldsymbol{x}}^{\boldsymbol{\nu}_2, \boldsymbol{\nu}_3} \right|}{\sigma_t^{2s}} + \alpha^s s! s^{4s} \log^{4s}(\epsilon^{-1}) \epsilon^s \right)
$$
$$
\lesssim t^{-s} \alpha^s (s+1)^{3d-1} s! s^s \log^s(\epsilon^{-1}). \tag{135}
$$

Additionally, by Lemma 17, there exists

$$
\phi_{\mathrm{root}}^s \in \mathcal{NN}\left( \text{width} \leq 48(2s)^2(N+1) \log_2(8N), \text{depth} \leq 18(2s)^2(L+2) \log_2(4L) + 2 \right), \tag{136}
$$

and for any $k \in \mathbb{N}_+$ with $k \leq s$ and for any $x \in [0, t_0^{-s} \alpha^s (s+1)^{3d-1} s! s^s \log^s(\epsilon^{-1})]$,

$$
\left| \phi_{\mathrm{root}}^s(x) - x^{1/s} \right| \leq (90s + 5)\left( t_0^{-s} \alpha^s (s+1)^{3d-1} s! s^s \log^s(\epsilon^{-1}) \right)^{\frac{1}{2s}} N^{-4s} L^{-4s}, \quad \text{(by Lemma 17)}
$$
$$
\leq 95 t_0^{-1/2} \alpha^{1/2} (s+1)^{\frac{3d-1}{2s}} s^2 \log^{1/2}(\epsilon^{-1}) \epsilon^{2s} \qquad \text{(by } (s!)^{\frac{1}{2s}} \leq (s^s)^{\frac{1}{2s}} = \sqrt{s})
$$
$$
\lesssim t_0^{-1/2} \alpha^{1/2} s^2 \log^{1/2}(\epsilon^{-1}) \epsilon^{2s}. \tag{137}
$$

For any $\boldsymbol{y} \in \mathbb{R}^d, t \in [t_0, \infty)$, we define $\phi_{\tilde{h}}$ to approximate $\tilde{h}$ as below

$$
\phi_{\tilde{h}}(\boldsymbol{y}, t) := \phi_{\mathrm{root}}\left( \sum_{\|\tilde{\boldsymbol{\nu}}\|_1 = s} \frac{2^{-(s - \|\boldsymbol{\nu}_2\|_1)} s!}{\boldsymbol{\nu}_1! \boldsymbol{\nu}_2! \boldsymbol{\nu}_3!} \phi_{\tilde{h}, \tilde{\boldsymbol{\nu}}}(\boldsymbol{y}, t) \right), \tag{138}
$$

*i) Network size.* With the sizes of $\phi_{\mathrm{abs}}, \phi_m^{\|\boldsymbol{\nu}_2\|_1 + 2\|\boldsymbol{\nu}_3\|_1}, \phi_{1/\sigma^2}^s, \phi_{\mathrm{poly}}^{\tilde{\boldsymbol{\nu}}}, \phi_{\mathrm{multi}}^{(1)}, \phi_{\mathrm{multi}}^{(2)}$ (Eq. (111) to (113), (121), (124) and (128)), for each $\tilde{\boldsymbol{\nu}} \in \mathbb{N}_+^{3d}$ such that $\|\tilde{\boldsymbol{\nu}}\|_1 = s$,

$$\phi_{\tilde{h}, \tilde{\boldsymbol{\nu}}} \in \mathcal{NN}\left(\text{width} \lesssim s^3 N \log_2(N) \vee s^3 \log^3(\epsilon^{-1}); \text{depth} \lesssim s^2 L \log_2(L) \vee s^2 \log^2(\epsilon^{-1})\right). \quad (139)$$

With Eq. (134), (138) and (139), we have

$$\phi_{\tilde{h}} \in \mathcal{NN}\left(\text{width} \lesssim s^{3d+2} N \log_2(N) \vee s^{3d+2} \log^3(\epsilon^{-1}); \text{depth} \lesssim s^2 L \log_2(L) \vee s^2 \log^2(\epsilon^{-1})\right). \quad (140)$$

*ii) Approximation error.*

With $|a^{1/k} - b^{1/k}| \le \frac{1}{k} \max\{a^{1/k-1}, b^{1/k-1}\}|a - b|$, we obtain

$$\left|\phi_{\tilde{h}}(\boldsymbol{y}, t) - \tilde{h}\right|$$

$$\le \underbrace{\left|\phi_{\mathrm{root}}^s\left(\sum_{\|\tilde{\boldsymbol{\nu}}\|_1 = s} \frac{2^{-(s-\|\boldsymbol{\nu}_2\|_1)} s!}{\boldsymbol{\nu}_1! \boldsymbol{\nu}_2! \boldsymbol{\nu}_3!} \phi_{\tilde{h}, \tilde{\boldsymbol{\nu}}}(\boldsymbol{y}, t)\right) - \left(\sum_{\|\tilde{\boldsymbol{\nu}}\|_1 = s} \frac{2^{-(s-\|\boldsymbol{\nu}_2\|_1)} s!}{\boldsymbol{\nu}_1! \boldsymbol{\nu}_2! \boldsymbol{\nu}_3!} \phi_{\tilde{h}, \tilde{\boldsymbol{\nu}}}(\boldsymbol{y}, t)\right)^{1/s}\right|}_{\le \text{Eq. (137)}}$$

$$+ \left|\left(\sum_{\|\tilde{\boldsymbol{\nu}}\|_1 = s} \frac{2^{-(s-\|\boldsymbol{\nu}_2\|_1)} s!}{\boldsymbol{\nu}_1! \boldsymbol{\nu}_2! \boldsymbol{\nu}_3!} \phi_{\tilde{h}, \tilde{\boldsymbol{\nu}}}(\boldsymbol{y}, t)\right)^{1/s} - \left(\sum_{\|\tilde{\boldsymbol{\nu}}\|_1 = s} \frac{s! m_t^{\|\boldsymbol{\nu}_2\|_1 + 2\|\boldsymbol{\nu}_3\|_1}}{(-2)^{s-\|\boldsymbol{\nu}_2\|_1} \sigma_t^{2s} \boldsymbol{\nu}_1! \boldsymbol{\nu}_2! \boldsymbol{\nu}_3!} \boldsymbol{y}^{2\boldsymbol{\nu}_1 + \boldsymbol{\nu}_2} C_{\boldsymbol{x}}^{\boldsymbol{\nu}_2, \boldsymbol{\nu}_3}\right)^{1/s}\right|$$

$$\lesssim t_0^{-1/2} \alpha^{1/2} s^2 \log^{1/2}(\epsilon^{-1}) \epsilon^{2s}$$

$$+ \frac{1}{s}\left(t_0^{-s} \alpha^{2s} (s+1)^{3d-1} s! s^{2s} \log^{2s}(\epsilon^{-1})\right)^{\frac{1}{s}-1} \left|\sum_{\|\tilde{\boldsymbol{\nu}}\|_1 = s} \frac{2^{-(s-\|\boldsymbol{\nu}_2\|_1)} s!}{\boldsymbol{\nu}_1! \boldsymbol{\nu}_2! \boldsymbol{\nu}_3!}\left(\phi_{\tilde{h}, \tilde{\boldsymbol{\nu}}}(\boldsymbol{y}, t) - \frac{m_t^{\|\boldsymbol{\nu}_2\|_1 + 2\|\boldsymbol{\nu}_3\|_1} \boldsymbol{y}^{2\boldsymbol{\nu}_1 + \boldsymbol{\nu}_2} C_{\boldsymbol{x}}^{\boldsymbol{\nu}_2, \boldsymbol{\nu}_3}}{\sigma_t^{2s}}\right)\right|$$

$$\le t_0^{-1/2} \alpha^{1/2} s^2 \log^{1/2}(\epsilon^{-1}) \epsilon^{2s}$$

$$+ \frac{1}{s}\left(t^{-s} \alpha^s (s+1)^{3d-1} s! s^s \log^s(\epsilon^{-1})\right)^{\frac{1}{s}-1} \sum_{\|\tilde{\boldsymbol{\nu}}\|_1 = s} \frac{2^{-(s-\|\boldsymbol{\nu}_2\|_1)} s!}{\boldsymbol{\nu}_1! \boldsymbol{\nu}_2! \boldsymbol{\nu}_3!} \underbrace{\left|\phi_{\tilde{h}, \tilde{\boldsymbol{\nu}}}(\boldsymbol{y}, t) - \frac{m_t^{\|\boldsymbol{\nu}_2\|_1 + 2\|\boldsymbol{\nu}_3\|_1} \boldsymbol{y}^{2\boldsymbol{\nu}_1 + \boldsymbol{\nu}_2} C_{\boldsymbol{x}}^{\boldsymbol{\nu}_2, \boldsymbol{\nu}_3}}{\sigma_t^{2s}}\right|}_{\le \text{Eq. (132)}}$$

$$\lesssim t_0^{-1/2} \alpha^{1/2} s^2 \log^{1/2}(\epsilon^{-1}) \epsilon^{2s}$$

$$+ \frac{1}{s}\left(t_0^{-s} (s+1)^{3d-1} \alpha^{2s} s! s^{2s} \log^{2s}(\epsilon^{-1})\right)^{\frac{1}{s}-1} \cdot (s+1)^{3d-1} s! \cdot \alpha^{2s} s! s^{4s} \log^{4s}(\epsilon^{-1}) \epsilon^s$$

$$= t_0^{-1/2} \alpha^{1/2} s^2 \log^{1/2}(\epsilon^{-1}) \epsilon^{2s} + (s+1)^{\frac{3d-1}{s}} t_0^{s-1} \alpha^2 s! s^{2s+2} \log^{2s+2}(\epsilon^{-1}) \epsilon^s$$

$$\lesssim t_0^{-1/2} \alpha^{1/2} s^2 \log^{1/2}(\epsilon^{-1}) \epsilon^{2s} + \alpha^2 s! s^{2s+2} \log^{2s+2}(\epsilon^{-1}) \epsilon^s$$

$$\qquad\qquad\qquad\qquad\qquad\qquad\qquad\qquad \left(\text{by } (s+1)^{\frac{3d-1}{s}} \le e^{(3d-1)}, \forall s \in \mathbb{N}_+\right)$$

$$\lesssim \alpha^2 s! s^{2s+2} \log^{2s+2}(\epsilon^{-1}) \epsilon^s, \qquad (141)$$

which implies that

$$0 \le \phi_{\tilde{h}}(\boldsymbol{y}, t) \lesssim |\tilde{h}| + \alpha^2 s! s^{2s+2} \log^{2s+2}(\epsilon^{-1}) \epsilon^s$$

$$\lesssim t_0^{-1} \alpha s \log(\epsilon^{-1}) + \alpha^2 s! s^{2s+2} \log^{2s+2}(\epsilon^{-1}) \epsilon^s$$

$$\lesssim t_0^{-1} \alpha s \log(\epsilon^{-1}), \quad \text{for any } \boldsymbol{y} \in \mathcal{B}, t \in [t_0, \infty]. \qquad (142)$$

$\square$

**Proposition 5** (Approximating $\tilde{h}_\beta(\boldsymbol{y}, t)$). *Given $s \in \mathbb{N}_+, 0 < \epsilon \le t_0 \le 1/2$, set $K = N^4 L^4$. Let $\tilde{C}_d$ is defined in Eq. (54) and $\tilde{h}_\beta$ be defined in Eq. (56), i.e.,*

$$\tilde{h}_\beta := \frac{\beta \widetilde{C}_d t_0^{-1} \alpha s \log(\epsilon^{-1})}{K}, \quad \beta \in \{0, 1, \dots, K-1\}.$$

*Let $\phi_{\tilde{h}}$ be defined in Eq. (138). Then, there exist $N, L \in \mathbb{N}_+$ with $N^{-2} L^{-2} \le \epsilon$, and*

$$\phi_{\tilde{h}_\beta} \in \mathcal{NN}\left(\text{width} \lesssim s^{3d+2} N^2 \vee s^{3d+2} \log^3(\epsilon^{-1}); \text{depth} \lesssim L^2 \vee s^2 \log^2(\epsilon^{-1})\right)$$

*such that for any $\boldsymbol{y} \in \mathcal{B}, t \in [t_0, \infty)$,*

$$\phi_{\tilde{h}_\beta}(\boldsymbol{y}, t) = \tilde{h}_\beta, \quad \beta \in \{0, 1, \dots, K-1\}.$$

*and*

$$\left| \phi_{\tilde{h}_\beta}(\boldsymbol{y}, t) - \phi_{\tilde{h}}(\boldsymbol{y}, t) \right| \lesssim \alpha s \log(\epsilon^{-1}) N^{-2s} L^{-2s}.$$

*Proof.* For any $\boldsymbol{y} \in \mathcal{B}, t \in [t_0, \infty)$, we define

$$\bar{\phi}_{\tilde{h}}(\boldsymbol{y}, t) := \frac{\phi_{\tilde{h}}(\boldsymbol{y}, t)}{\widetilde{C}_d t_0^{-1} \alpha s \log(\epsilon^{-1})}, \tag{143}$$

which indicates that $0 \leq \bar{\phi}_{\tilde{h}}(\boldsymbol{y}, t) \leq 1$. For $K = N^4 L^4$, by Proposition 12, there exists a ReLU DNN

$$\phi_{\text{step}} \in \mathcal{NN}\left( \text{width} \leq 4N^2 + 3; \text{depth} \leq 4L^2 + 5 \right)$$

such that

$$\phi_{\text{step}}(\bar{\phi}_{\tilde{h}}(\boldsymbol{y}, t)) = k, \text{ if } \phi_{\tilde{h}}(\boldsymbol{y}, t) \in \left[ \frac{k\widetilde{C}_d \alpha s \log(\epsilon^{-1})}{Kt_0}, \frac{(k+1)\widetilde{C}_d \alpha s \log(\epsilon^{-1})}{Kt_0} - \delta \cdot \mathbb{1}_{k \leq K-2} \right], \quad k = 0, 1, \dots, K-1,$$

where $\delta \in (0, \widetilde{C}_d t_0^{-1} \alpha s \log(\epsilon^{-1})/K)$. Define $\phi_{\tilde{h}_\beta}$ by

$$\phi_{\tilde{h}_\beta}(\boldsymbol{y}, t) := \frac{\widetilde{C}_d t_0^{-1} \alpha s \log(\epsilon^{-1})}{K} \phi_{\text{step}}\left( \frac{\phi_{\tilde{h}}(\boldsymbol{y}, t)}{\widetilde{C}_d t_0^{-1} \alpha s \log(\epsilon^{-1})} \right), \quad \text{for any } \boldsymbol{y} \in \mathcal{B}, t \in [t_0, \infty). \tag{144}$$

Clearly, by the size of $\phi_{\tilde{h}}$ (c.f. Eq. (140)) and $\phi_{\text{step}}$, we have

$$\phi_{\tilde{h}_\beta} \in \mathcal{NN}\left( \text{width} \lesssim s^{3d+2} N^2 \vee s^{3d+2} \log^3(\epsilon^{-1}); \text{depth} \lesssim L^2 \vee s^2 \log^2(\epsilon^{-1}) \right) \tag{145}$$

and for any $\boldsymbol{y} \in \mathcal{B}$ and any $t \in [t_0, \infty)$,

$$\phi_{\tilde{h}_\beta}(\boldsymbol{y}, t) = \frac{\beta \widetilde{C}_d t_0^{-1} \alpha s \log(\epsilon^{-1})}{K} = \tilde{h}_\beta, \quad \text{for } \beta \in \{0, 1, \dots, K-1\}, \tag{146}$$

such that

$$\left| \phi_{\tilde{h}_\beta}(\boldsymbol{y}, t) - \phi_{\tilde{h}}(\boldsymbol{y}, t) \right| = \left| \tilde{h}_\beta - \phi_{\tilde{h}}(\boldsymbol{y}, t) \right| \lesssim \frac{\alpha s \log(\epsilon^{-1})}{Kt_0} = \frac{\alpha s \log(\epsilon^{-1})}{N^4 L^4 t_0} \lesssim \alpha s \log(\epsilon^{-1}) N^{-2s} L^{-2s}.$$

$\square$

**Proposition 6** (Approximating $\tilde{h}_\beta^k(\boldsymbol{y}, t)$ )**.** *Under the same settings of Proposition 5. Given $k \in \mathbb{N}, k \leq s$, there exist $N, L \in \mathbb{N}_+$ with $N^{-2} L^{-2} \leq \epsilon$, and*

$$\phi_{\tilde{h}_\beta}^k \in \mathcal{NN}\left( \text{width} \lesssim s^{3d+2} N^2 \log_2(N) \vee s^{3d+2} \log^3(\epsilon^{-1}); \text{depth} \lesssim L^2 \log_2(L) \vee s^2 \log^2(\epsilon^{-1}) \right).$$

*such that*

$$\left| \phi_{\tilde{h}_\beta}^k(\boldsymbol{y}, t) - \tilde{h}_\beta^k(\boldsymbol{y}, t) \right| \leq \alpha^s s^s \log^s(\epsilon^{-1}) N^{-2s} L^{-2s}, \quad \text{for any } \boldsymbol{y} \in \mathcal{B}, t \in [t_0, \infty),$$

*and*

$$0 \leq \phi_{\tilde{h}_\beta}^k(\boldsymbol{y}, t) \lesssim t_0^{-k} \alpha^k s^k \log^k(\epsilon^{-1}).$$

*Proof.* For $\beta \in \{0, 1, \dots, K-1\}$ and $k = 0, 1, \dots, s$, we define

$$\xi_\beta^k := \frac{1}{\widetilde{C}_d^k t_0^{-k} \alpha^k s^k \log^k(\epsilon^{-1})} \left( \frac{\beta \widetilde{C}_d t_0^{-1} \alpha s \log(\epsilon^{-1})}{K} \right)^k. \tag{147}$$

With $K = N^4 L^4$, we have $\xi_\beta^k \in [0, 1]$ for any $\beta \in \{0, 1, \dots, K-1\}$ and $k = 0, 1, \dots, s$. By Proposition 13, there exists a ReLU DNN

$$\phi_{\text{point}}^k \in \mathcal{NN}\left( \text{width} \leq 16s(N^2 + 1)\log_2(8N^2); \text{depth} \leq 5(L^2 + 2)\log_2(4L^2) \right)$$

such that for any fixed $k \in \{0, 1, \ldots, s\}$, we have

$$\left| \phi_{\text{point}}^k(\beta) - \xi_\beta^k \right| \leq N^{-4s} L^{-4s}, \quad \text{for each } \beta = 0, 1, \ldots, K - 1,$$
$$0 \leq \phi_{\text{point}}^k(\beta) \leq 1.$$

We define

$$\phi_{\tilde{h}_\beta}^k(\boldsymbol{y}, t) := \widetilde{C}_d^k t_0^{-k} \alpha^k s^k \log^k(\epsilon^{-1}) \phi_{\text{point}}^k \left( \frac{K \phi_{\tilde{h}_\beta}(\boldsymbol{y}, t)}{\widetilde{C}_d t_0^{-1} \alpha s \log(\epsilon^{-1})} \right), \quad \forall \boldsymbol{y} \in \mathbb{R}^d, \forall t \in [t_0, \infty). \quad (148)$$

Clearly, by the size of $\phi_{\tilde{h}_\beta}$ (c.f. Eq. (145)) and $\phi_{\text{point}}^k$, we have

$$\phi_{\tilde{h}_\beta}^k \in \mathcal{NN}\left( \text{width} \lesssim s N^2 \log_2(N) \vee s^{3d+2} \log^3(\epsilon^{-1}); \text{depth} \lesssim L^2 \log_2(L) \vee s^2 \log^2(\epsilon^{-1}) \right).$$

Then for all $\tilde{h}_\beta = \frac{\beta \widetilde{C}_d t_0^{-1} \alpha s \log(\epsilon^{-1})}{K}, \beta \in \{0, 1, \ldots, K - 1\}$, we have

$$\phi_{\tilde{h}_\beta}^k(\boldsymbol{y}, t) \in \left[ 0, \widetilde{C}_d^k t_0^{-k} \alpha^k s^k \log^k(\epsilon^{-1}) \right] \quad (149)$$

and for any $k = 0, 1, \ldots, s$ and $\beta \in \{0, 1, \ldots, K - 1\}$,

$$\begin{aligned}
\left| \phi_{\tilde{h}_\beta}^k(\boldsymbol{y}, t) - \tilde{h}_\beta^k(\boldsymbol{y}, t) \right| &= \left| \phi_{\tilde{h}_\beta}^k \left( \frac{\beta \widetilde{C}_d t_0^{-1} \alpha s \log(\epsilon^{-1})}{K} \right) - \left( \frac{\beta \widetilde{C}_d t_0^{-1} \alpha s \log(\epsilon^{-1})}{K} \right)^k \right| \\
&= \left| \widetilde{C}_d^k t_0^{-k} \alpha^k s^k \log^k(\epsilon^{-1}) \phi_{\text{point}}^k(\beta) - \widetilde{C}_d^k t_0^{-k} \alpha^k s^k \log^k(\epsilon^{-1}) \xi_\beta^k \right| \\
&\leq \widetilde{C}_d^k t_0^{-k} \alpha^k s^k \log^k(\epsilon^{-1}) \left| \phi_{\text{point}}^k(\beta) - \xi_\beta^k \right| \\
&\lesssim t_0^{-k} \alpha^k s^k \log^k(\epsilon^{-1}) N^{-4s} L^{-4s} \\
&\leq \alpha^k s^k \log^k(\epsilon^{-1}) N^{-2s} L^{-2s}. \quad \text{(by } N^{-2} L^{-2} \leq \epsilon \leq t_0\text{)}
\end{aligned}$$

$\square$

**Proposition 7** (Approximating $\exp(-\tilde{h}_\beta(\boldsymbol{y}, t))$)**.** *Under the settings of Proposition 5. Then, there exist $N, L \in \mathbb{N}_+$ with $N^{-2} L^{-2} \leq \epsilon$, and*

$$\phi_{\tilde{h}_\beta}^{\exp} \in \mathcal{NN}\left( \text{width} \lesssim s^{3d+2} N^2 \log_2(N) \vee s^{3d+2} \log^3(\epsilon^{-1}); \text{depth} \lesssim L^2 \log_2(L) \vee s^2 \log^2(\epsilon^{-1}) \right)$$

*such that*

$$\left| \phi_{\tilde{h}_\beta}^{\exp}(\boldsymbol{y}, t) - \exp(-\tilde{h}_\beta(\boldsymbol{y}, t)) \right| \leq N^{-4s} L^{-4s}, \quad \text{for any } \boldsymbol{y} \in \mathcal{B}, t \in [t_0, \infty),$$

*and*

$$0 \leq \phi_{\tilde{h}_\beta}^{\exp}(\boldsymbol{y}, t) \leq 1.$$

*Proof.* Similar to approximate $\tilde{h}_\beta^k$, for $\beta \in \{0, 1, \ldots, K - 1\}$, we define

$$\xi_\beta^{\exp} := \exp\left( -\frac{\beta \widetilde{C}_d t_0^{-1} \alpha s \log(\epsilon^{-1})}{K} \right). \quad (150)$$

Then we have $\xi_\beta \in [0, 1]$ for any $\beta \in \{0, 1, \ldots, K - 1\}$. Again, by Proposition 13, there exists a ReLU DNN

$$\phi_{\text{point}}^{\exp} \in \mathcal{NN}\left( \text{width} \leq 16s(N^2 + 1) \log_2(8N^2); \text{depth} \leq 5(L^2 + 2) \log_2(4L^2) \right)$$

such that

$$\left| \phi_{\text{point}}^{\exp}(\beta) - \xi_\beta^{\exp} \right| \leq N^{-4s} L^{-4s}, \quad \text{for each } \beta = 0, 1, \ldots, K - 1,$$
$$0 \leq \phi_{\text{point}}^{\exp}(\beta) \leq 1.$$

We define

$$\phi_{\tilde{h}_\beta}^{\exp}(\boldsymbol{y}, t) := \phi_{\text{point}}^{\exp}\left(\frac{K\phi_{\tilde{h}_\beta}(\boldsymbol{y}, t)}{\widetilde{C}_d t_0^{-1}\alpha s \log(\epsilon^{-1})}\right), \quad \text{for any } \boldsymbol{y} \in \mathbb{R}^d, t \in [t_0, \infty). \tag{151}$$

By the size of $\phi_{\tilde{h}_\beta}$ (c.f. Eq. (145)) and $\phi_{\text{point}}^{\exp}$, we have

$$\phi_{\tilde{h}_\beta}^{\exp} \in \mathcal{NN}\left(\text{width} \lesssim s^{3d+2} N^2 \log_2(N) \vee s^{3d+2} \log^3(\epsilon^{-1}); \text{depth} \lesssim L^2 \log_2(L) \vee s^2 \log^2(\epsilon^{-1})\right).$$

Then for all $\tilde{h}_\beta = \frac{\beta \widetilde{C}_d t_0^{-1} \alpha s \log(\epsilon^{-1})}{K}, \beta \in \{0, 1, \ldots, K-1\}$, we have

$$\phi_{\tilde{h}_\beta}^{\exp}(\tilde{h}_\beta) \in [0, 1],$$

and for each $\beta \in \{0, 1, \cdots, K-1\}$,

$$\begin{aligned}
\left|\phi_{\tilde{h}_\beta}^{\exp}(\boldsymbol{y}, t) - \exp(-\tilde{h}_\beta(\boldsymbol{y}, t))\right| &= \left|\phi_{\tilde{h}_\beta}^{\exp}\left(\frac{\beta \widetilde{C}_d t_0^{-1}\alpha s \log(\epsilon^{-1})}{K}\right) - \exp\left(-\frac{\beta \widetilde{C}_d t_0^{-1}\alpha s \log(\epsilon^{-1})}{K}\right)\right| \\
&= \left|\phi_{\text{point}}^{\exp}(\beta) - \xi_\beta^{\exp}\right| \\
&\leq N^{-4s} L^{-4s}.
\end{aligned}$$

$\square$

## D.3 Neural network approximations for regularized empirical score functions

Recall that the regularized empirical score function at time $t$ is given by

$$\begin{aligned}
f_{\text{score}}^{\text{kde}}(\boldsymbol{x}_t, t) &:= \frac{\nabla \hat{p}_t(\boldsymbol{x}_t)}{\hat{p}_t(\boldsymbol{x}_t) \vee \rho_{n,t}} = \frac{(2\pi\sigma_t^2)^{-d/2} \frac{1}{n} \sum_{i=1}^n \exp\left(-\frac{\|\boldsymbol{x}_t - m_t \boldsymbol{x}^{(i)}\|_2^2}{2\sigma_t^2}\right) \frac{-(\boldsymbol{x}_t - m_t \boldsymbol{x}^{(i)})}{\sigma_t^2}}{(2\pi\sigma_t^2)^{-d/2} \frac{1}{n} \sum_{i=1}^n \exp\left(-\frac{\|\boldsymbol{x}_t - m_t \boldsymbol{x}^{(i)}\|_2^2}{2\sigma_t^2}\right) \vee \rho_{n,t}} \\
&= -\frac{1}{\sigma_t^2} \frac{\frac{1}{n}\sum_{i=1}^n \exp\left(-\frac{\|\boldsymbol{x}_t - m_t \boldsymbol{x}^{(i)}\|_2^2}{2\sigma_t^2}\right)(\boldsymbol{x}_t - m_t \boldsymbol{x}^{(i)})}{\frac{1}{n}\sum_{i=1}^n \exp\left(-\frac{\|\boldsymbol{x}_t - m_t \boldsymbol{x}^{(i)}\|_2^2}{2\sigma_t^2}\right) \vee e^{-1} n^{-1}},
\end{aligned}$$

where $\rho_{n,t} = (2\pi\sigma^2)^{-d/2} e^{-1} n^{-1}$.

### D.3.1 Approximation of regularized empirical score functions in $L^\infty$-norm

**Lemma 10** ($L^\infty$-Approximation of Regularized Empirical Score Functions). *Given a set of sample* $\{\boldsymbol{x}^{(i)}\}_{i=1}^n$, *for any* $\boldsymbol{y} \in \mathbb{R}^d, t \in [t_0, \infty)$, *let* $m_t := \exp(-t), \sigma_t := \sqrt{1 - \exp(-2t)}$. *Fix* $\rho_{n,t} := (2\pi\sigma_t^2)^{-d/2} e^{-1} n^{-1}$, *and* $0 < t_0 \leq 1/2$, *let* $N, L, s \in \mathbb{N}_+$ *such that* $N^{-2} L^{-2} \leq \epsilon$ *and* $0 < \epsilon \leq t_0 \wedge n^{-1/s}$. *Suppose Assumption 4 holds. Then, there exists a function* $\phi_{\text{score}}$ *implemented by a ReLU DNN with width* $\leq \mathcal{O}\left(s^{6d+3} N^3 \log_2(N) \vee s^{6d+3} \log^3(\epsilon^{-1})\right)$ *and depth* $\leq \mathcal{O}\left(L^3 \log_2(L) \vee s^2 \log^2(\epsilon^{-1})\right)$ *such that*

$$\left|\phi_{\text{score}}(\boldsymbol{y}, t) - f_{\text{score}}^{\text{kde}}(\boldsymbol{y}, t)\right| \lesssim \alpha^{18s+\frac{1}{2}}(12s)! s^{3d+54s+\frac{3}{2}} \log^{54s+\frac{1}{2}}(\epsilon^{-1})\epsilon^s, \quad \forall \boldsymbol{y} \in \mathbb{R}^d, \forall t \in [t_0, \infty),$$

*and we have* $|\phi_{\text{score}}(\boldsymbol{y}, t)| \lesssim \sigma_t^{-1} \sqrt{\log n}$.

**Corollary 4.** *Under the same conditions of Lemma 10, let* $\alpha^2 n^{-2/d} \log n \leq t_0 \leq 1/2$. *Fix* $k \in \mathbb{N}_+$ *such that* $k \geq d/2$. *Then, there exists a ReLU DNN with width* $\leq \mathcal{O}(n^{\frac{3}{2k}} \log_2 n)$ *and depth* $\leq \mathcal{O}(\log^2 n)$ *such that*

$$\left|\phi_{\text{score}}(\boldsymbol{y}, t) - f_{\text{score}}^{\text{kde}}(\boldsymbol{y}, t)\right| \lesssim \alpha^{18k+\frac{1}{2}} n^{-1} \log^{54k+\frac{1}{2}} n, \quad \forall \boldsymbol{y} \in \mathbb{R}^d, \forall t \in [t_0, \infty),$$

*and we have* $|\phi_{\text{score}}(\boldsymbol{y}, t)| \lesssim \sigma_t^{-1} \sqrt{\log n}$.

*Proof.* Apply Lemma 10 with $\epsilon = n^{-1/k}, s = k$ and $N = \lceil n^{1/(2k)} \rceil, L = 1$ such that $N^{-2} L^{-2} \leq \epsilon \leq t_0 \wedge n^{-1/s}$ hold and we complete the proof. $\square$

### D.3.2 Proof of Lemma 10

For any $\boldsymbol{y} \in \mathbb{R}^d, t \in [t_0, \infty)$, recall that the empirical score functions is given by

$$
\begin{aligned}
f_{\text{score}}^{\text{kde}}(\boldsymbol{y}, t) &= \frac{1}{\sigma_t^2} \frac{m_t \times \frac{1}{n} \sum_{i=1}^n \exp\left(-\frac{\|\boldsymbol{y} - m_t \boldsymbol{x}^{(i)}\|_2^2}{2\sigma_t^2}\right) \boldsymbol{x}^{(i)} - \boldsymbol{y} \times \frac{1}{n} \sum_{i=1}^n \exp\left(-\frac{\|\boldsymbol{y} - m_t \boldsymbol{x}^{(i)}\|_2^2}{2\sigma_t^2}\right)}{\frac{1}{n} \sum_{i=1}^n \exp\left(-\frac{\|\boldsymbol{y} - m_t \boldsymbol{x}^{(i)}\|_2^2}{2\sigma_t^2}\right) \vee e^{-1} n^{-1}} \\
&= \frac{1}{\sigma_t^2} \frac{m_t \times f_{\text{kde}}^{(3)}(\boldsymbol{y}, t) - \boldsymbol{y} \times f_{\text{kde}}^{(2)}(\boldsymbol{y}, t)}{f_{\text{kde}}^{(1)}(\boldsymbol{y}, t) \vee e^{-1} n^{-1}} \\
&= \frac{1}{\sigma_t^2} \frac{f_{\text{score}}^{(2)}(\boldsymbol{y}, t)}{f_{\text{score}}^{(1)}(\boldsymbol{y}, t)},
\end{aligned}
\tag{152}
$$

where we denote

$$
f_{\text{kde}}^{(1)}(\boldsymbol{y}, t), f_{\text{kde}}^{(2)}(\boldsymbol{y}, t) := \frac{1}{n} \sum_{i=1}^n \exp\left(-\frac{\|\boldsymbol{y} - m_t \boldsymbol{x}^{(i)}\|_2^2}{2\sigma_t^2}\right),
\tag{153}
$$

$$
f_{\text{kde}}^{(3)}(\boldsymbol{y}, t) := \frac{1}{n} \sum_{i=1}^n \exp\left(-\frac{\|\boldsymbol{y} - m_t \boldsymbol{x}^{(i)}\|_2^2}{2\sigma_t^2}\right) \boldsymbol{x}^{(i)}.
\tag{154}
$$

$$
f_{\text{score}}^{(1)}(\boldsymbol{y}, t) := f_{\text{kde}}^{(1)}(\boldsymbol{y}, t) \vee e^{-1} n^{-1},
\tag{155}
$$

$$
f_{\text{score}}^{(2)}(\boldsymbol{y}, t) := m_t \times f_{\text{kde}}^{(3)}(\boldsymbol{y}, t) - \boldsymbol{y} \times f_{\text{kde}}^{(2)}(\boldsymbol{y}, t).
\tag{156}
$$

Similar to Eq. (48) and (49), we decompose $\mathbb{R}^d = \tilde{\mathcal{B}} \cup \overline{\tilde{\mathcal{B}}}$, where

$$
\tilde{\mathcal{B}} := \{\boldsymbol{y} \in \mathbb{R}^d : |\boldsymbol{y}| \le 3\sqrt{2\alpha s \log(\epsilon^{-1})}\},
\tag{157}
$$

$$
\overline{\tilde{\mathcal{B}}} := \{\boldsymbol{y} \in \mathbb{R}^d : |\boldsymbol{y}| > 3\sqrt{2\alpha s \log(\epsilon^{-1})}\}.
\tag{158}
$$

We approximate $f_{\text{kde}}$ on $\boldsymbol{y} \in \overline{\tilde{\mathcal{B}}}, t \in [t_0, \infty)$ in **Part I** and $\boldsymbol{y} \in \tilde{\mathcal{B}}, t \in [t_0, \infty)$ in **Part II**.

### Part I: Approximating $f_{\text{score}}^{\text{kde}}$ on $\overline{\tilde{\mathcal{B}}}$

For any $\boldsymbol{y} \in \overline{\tilde{\mathcal{B}}}, t \in [t_0, \infty), 0 \le \epsilon \le \exp(-\frac{1}{4\alpha})$, similar to the derivations for Eq. (51) that, by Assumption 4, we have

$$
\frac{\|\boldsymbol{y} - m_t \boldsymbol{x}^{(i)}\|_2^2}{2\sigma_t^2} > \frac{(2\sqrt{2\alpha s \log(\epsilon^{-1})})^2}{2\sigma_t^2} \ge \frac{8\alpha s \log(\epsilon^{-1})}{2} = 4\alpha s \log(\epsilon^{-1}) \ge 1,
$$

which gives

$$
\exp\left(-\frac{\|\boldsymbol{y} - m_t \boldsymbol{x}^{(i)}\|_2^2}{2\sigma_t^2}\right) \le \epsilon^{4s}, \quad \text{for all } i = 1, \dots, n.
$$

Noting that the function $x \mapsto \exp(-x^2/2)x$ is monotonously decreasing in $[1, \infty)$, then

$$
\begin{aligned}
\left|f_{\text{score}}^{\text{kde}}(\boldsymbol{y}, t)\right| &= \left| \frac{\frac{1}{\sigma_t^2} \frac{1}{n} \sum_{i=1}^n \exp\left(-\frac{\|\boldsymbol{y} - m_t \boldsymbol{x}^{(i)}\|_2^2}{2\sigma_t^2}\right)(m_t \boldsymbol{x}^{(i)} - \boldsymbol{y})}{\frac{1}{n} \sum_{i=1}^n \exp\left(-\frac{\|\boldsymbol{y} - m_t \boldsymbol{x}^{(i)}\|_2^2}{2\sigma_t^2}\right) \vee e^{-1} n^{-1}} \right| \\
&\le \frac{en}{\sigma_t} \frac{1}{n} \sum_{i=1}^n \exp\left(-\frac{\|\boldsymbol{y} - m_t \boldsymbol{x}^{(i)}\|_2^2}{2\sigma_t^2}\right) \frac{|m_t \boldsymbol{x}^{(i)} - \boldsymbol{y}|}{\sigma_t} \\
&\le \frac{2en}{\sigma_t} \sqrt{\alpha s \log(\epsilon^{-1})} \epsilon^{4s} \\
&\quad (\exp(-x^2/2)x \text{ is monotonously decreasing in } [2\sqrt{\alpha s \log(\epsilon^{-1})}, \infty)) \\
&\le 2e\sqrt{\alpha s \log(\epsilon^{-1})} \epsilon^{2s}.
\end{aligned}
\tag{159}
$$

### Part II: Approximating $f_{\text{score}}^{\text{kde}}$ on $\tilde{\mathcal{B}}$

**Step 1: Approximating $f_{\text{score}}^{(1)}$ by $\phi_{\text{score}}^{(1)}$**

Notice that

$$x = \text{ReLU}(x) - \text{ReLU}(-x), \quad \text{and } |x| = \text{ReLU}(x) + \text{ReLU}(-x).$$

Then, for any $x, y \in \mathbb{R}$, we have

$$
\begin{aligned}
\max\{x, y\} &= \frac{x + y + |x - y|}{2} \\
&= \frac{1}{2}\big(\text{ReLU}(x + y) - \text{ReLU}(-x - y) + \text{ReLU}(x - y) + \text{ReLU}(-x + y)\big),
\end{aligned}
$$

which indicates that

$$\phi_{\max} \in \mathcal{NN}\big(\text{width} = 4; \text{depth} = 1\big), \tag{160}$$

and

$$\phi_{\max}(x, y) = \max\{x, y\}, \quad \text{for any } x, y \in \mathbb{R}.$$

By Lemma 8, there exists

$$
\begin{aligned}
\phi_{\text{kde}}^{(1)} \in \mathcal{NN}\big(&\text{width} \lesssim (3s)^{6d+3} N^3 \log_2(N) \vee (3s)^{6d+3} \log^3(\epsilon^{-1}); \\
&\text{depth} \lesssim L^3 \log_2(L) \vee s^2 \log^2(\epsilon^{-1})\big).
\end{aligned}
\tag{161}
$$

such that

$$\big|\phi_{\text{kde}}^{(1)}(\boldsymbol{y}, t) - f_{\text{kde}}^{(1)}(\boldsymbol{y}, t)\big| \lesssim \alpha^{18s}(12s)! s^{3d+54s+1} \log^{54s}(\epsilon^{-1}) \epsilon^{6s}, \tag{162}$$

and we have

$$0 \le \phi_{\text{kde}}^{(1)}(\boldsymbol{y}, t) \lesssim 1 \tag{163}$$

For any $\boldsymbol{y} \in \mathbb{R}^d, t \in [t_0, \infty)$, we define

$$\phi_{\text{score}}^{(1)}(\boldsymbol{y}, t) := \phi_{\max}\big(\phi_{\text{kde}}^{(1)}(\boldsymbol{y}, t), e^{-1} n^{-1}\big). \tag{164}$$

By Eq. (160) and (161), we have

$$
\begin{aligned}
\phi_{\text{score}}^{(1)} \in \mathcal{NN}\big(&\text{width} \lesssim s^{6d+3} N^3 \log_2(N) \vee s^{6d+3} \log^3(\epsilon^{-1}); \\
&\text{depth} \lesssim L^3 \log_2(L) \vee s^2 \log^2(\epsilon^{-1})\big),
\end{aligned}
\tag{165}
$$

and for any $\boldsymbol{y} \in \tilde{\mathcal{B}}, t \in [t_0, \infty)$, we have

$$\big|\phi_{\text{score}}^{(1)}(\boldsymbol{y}, t) - f_{\text{score}}^{(1)}(\boldsymbol{y}, t)\big| \le \big|\phi_{\text{kde}}^{(1)}(\boldsymbol{y}, t) - f_{\text{kde}}^{(1)}(\boldsymbol{y}, t)\big| \lesssim \alpha^{18s}(12s)! s^{3d+54s+1} \log^{54s}(\epsilon^{-1}) \epsilon^{6s}, \tag{166}$$

which implies that for any $\boldsymbol{y} \in \tilde{\mathcal{B}}, t \in [t_0, \infty)$,

$$e^{-1} n^{-1} \le \phi_{\text{score}}^{(1)}(\boldsymbol{y}, t) \lesssim |f_{\text{kde}}^{(1)}(\boldsymbol{y}, t)| + \alpha^{18s}(12s)! s^{3d+54s+1} \log^{54s}(\epsilon^{-1}) \epsilon^{6s} \lesssim 1. \tag{167}$$

**Step 2: Approximating $f_{\text{score}}^{(2)}$ by $\phi_{\text{score}}^{(2)}$**

Again, by Lemma 8, there exists

$$
\begin{aligned}
\phi_{\text{kde}}^{(2)} \in \mathcal{NN}\big(&\text{width} \lesssim s^{6d+3} N^3 \log_2(N) \vee s^{6d+3} \log^3(\epsilon^{-1}); \\
&\text{depth} \lesssim L^3 \log_2(L) \vee s^2 \log^2(\epsilon^{-1})\big).
\end{aligned}
\tag{168}
$$

such that

$$\big|\phi_{\text{kde}}^{(2)}(\boldsymbol{y}, t) - f_{\text{kde}}^{(2)}(\boldsymbol{y}, t)\big| \lesssim \alpha^{9s}(6s)! s^{3d+27s+2} \log^{27s}(\epsilon^{-1}) \epsilon^{3s}, \tag{169}$$

and we have

$$0 \le \phi_{\text{kde}}^{(2)}(\boldsymbol{y}, t) \lesssim 1. \tag{170}$$

By Lemma 13, for each $j = 1, \ldots, d$, there exists

$$\phi_{\text{multi}, j}^{(5)} \in \mathcal{NN}\big(\text{width} \le 9(N + 1) + 1, \text{depth} \le 7sL\big),$$

such that for any $y_j \in \left[0, 3\sqrt{2\alpha s \log(\epsilon^{-1})}\right], \forall h \in [0,1]$,

$$\left|\phi_{\mathrm{multi},j}^{(5)}(y_j, h) - y_j h\right| \leq \sqrt{\alpha s \log(\epsilon^{-1})}(N+1)^{-7sL}.$$

For all $\boldsymbol{y} \in \mathbb{R}^d, h \in \mathbb{R}$, we define

$$\phi_{\mathrm{multi}}^{(5)}(\boldsymbol{y}, h) := \left[\phi_{\mathrm{multi},1}^{(5)}(y_1, h), \cdots, \phi_{\mathrm{multi},d}^{(5)}(y_d, h)\right]. \tag{171}$$

Then, we have

$$\phi_{\mathrm{multi}}^{(5)} \in \mathcal{NN}\big(\mathrm{width} \leq 9d(N+1) + d, \mathrm{depth} \leq 14sL\big), \tag{172}$$

and

$$\left|\phi_{\mathrm{multi}}^{(5)}\big(\boldsymbol{y}, \phi_{\mathrm{kde}}^{(2)}(\boldsymbol{y}, t)\big) - \boldsymbol{y}\phi_{\mathrm{kde}}^{(2)}(\boldsymbol{y}, t)\right| \lesssim \sqrt{\alpha s \log(\epsilon^{-1})}(N+1)^{-14sL}, \tag{173}$$

which gives that

$$\left|\phi_{\mathrm{multi}}^{(5)}\big(\boldsymbol{y}, \phi_{\mathrm{kde}}^{(2)}(\boldsymbol{y}, t)\big)\right| \lesssim |\boldsymbol{y}| \cdot \left|\phi_{\mathrm{kde}}^{(2)}(\boldsymbol{y}, t)\right| + \sqrt{\alpha s \log(\epsilon^{-1})}(N+1)^{-14sL} \lesssim \sqrt{\alpha s \log(\epsilon^{-1})}. \tag{174}$$

For any $\boldsymbol{y} \in \tilde{\mathcal{B}}, t \in [t_0, \infty)$, it follows that

$$\begin{aligned}
&\left|\phi_{\mathrm{multi}}^{(5)}\big(\boldsymbol{y}, \phi_{\mathrm{kde}}^{(2)}(\boldsymbol{y}, t)\big) - \boldsymbol{y}f_{\mathrm{kde}}^{(2)}(\boldsymbol{y}, t)\right| \\
&\leq \underbrace{\left|\phi_{\mathrm{multi}}^{(5)}\big(\boldsymbol{y}, \phi_{\mathrm{kde}}^{(2)}(\boldsymbol{y}, t)\big) - \boldsymbol{y}\phi_{\mathrm{kde}}^{(2)}(\boldsymbol{y}, t)\right|}_{\lesssim \text{ Eq. (173)}} + |\boldsymbol{y}| \cdot \underbrace{\left|\phi_{\mathrm{kde}}^{(2)}(\boldsymbol{y}, t) - \phi_{\mathrm{kde}}^{(2)}(\boldsymbol{y}, t)\right|}_{\lesssim \text{ Eq. (169)}} \\
&\lesssim \sqrt{\alpha s \log(\epsilon^{-1})}(N+1)^{-14sL} + \alpha^{9s+1/2}(6s)!s^{3d+27s+3/2}\log^{27s+1/2}(\epsilon^{-1})\epsilon^{3s} \\
&\lesssim \alpha^{9s+1/2}(6s)!s^{3d+27s+3/2}\log^{27s+1/2}(\epsilon^{-1})\epsilon^{3s}.
\end{aligned} \tag{175}$$

Moreover, similar to the derivations of Eq. (65), we can obtain

$$\begin{aligned}
f_{\mathrm{kde}}^{(3)}(\boldsymbol{y}, t) &:= \frac{1}{n}\sum_{i=1}^{n}\exp\Big(-\frac{\|\boldsymbol{y} - m_t\boldsymbol{x}^{(i)}\|_2^2}{2\sigma_t^2}\Big)\boldsymbol{x}^{(i)} \\
&= \exp(-\tilde{h}_\beta)\sum_{k=0}^{s-1}\sum_{\|\tilde{\boldsymbol{\nu}}\|_1 + a_4 = k}\frac{(-2)^{-(k-\|\boldsymbol{\nu}_2\|_1 - a_4)}m_t^{\|\boldsymbol{\nu}_2\|_1 + 2\|\boldsymbol{\nu}_3\|_1}}{\sigma_t^{2(k-a_4)}\boldsymbol{\nu}_1!\boldsymbol{\nu}_2!\boldsymbol{\nu}_3!a_4!}\tilde{h}_\beta^{a_4}\boldsymbol{y}^{2\boldsymbol{\nu}_1 + \boldsymbol{\nu}_2} \cdot \frac{1}{n}\sum_{i=1}^{n}\big(\boldsymbol{x}^{(i)}\big)^{\boldsymbol{\nu}_2 + 2\boldsymbol{\nu}_3} \cdot \boldsymbol{x}^{(i)}.
\end{aligned}$$

For each $\tilde{\boldsymbol{\nu}} \in \mathbb{N}^{3d}, a_4 \in \mathbb{N}$ such that $\|\tilde{\boldsymbol{\nu}}\|_1 + a_4 \leq s - 1$, denote by

$$\tilde{C}_{\boldsymbol{x}}^{\boldsymbol{\nu}_2, \boldsymbol{\nu}_3} := \frac{1}{n}\sum_{i=1}^{n}\big(\boldsymbol{x}^{(i)}\big)^{\boldsymbol{\nu}_2 + 2\boldsymbol{\nu}_3} \cdot \boldsymbol{x}^{(i)} \in \mathbb{R}^d. \tag{176}$$

Following a similar derivation for $\phi_{\mathrm{kde}}$ (i.e., Eq. (86) and (91)) in Appendix D.2, we can obtain that there exists

$$\begin{aligned}
\phi_{\mathrm{kde}}^{(3)} \in \mathcal{NN}\big(&\mathrm{width} \lesssim s^{6d+3}N^3\log_2(N) \vee s^{6d+3}\log^3(\epsilon^{-1}); \\
&\mathrm{depth} \lesssim L^3\log_2(L) \vee s^2\log^2(\epsilon^{-1})\big),
\end{aligned} \tag{177}$$

such that for any $\boldsymbol{y} \in \tilde{\mathcal{B}}, t \in [t_0, \infty)$,

$$\left|\phi_{\mathrm{kde}}^{(3)}(\boldsymbol{y}, t) - f_{\mathrm{kde}}^{(3)}(\boldsymbol{y}, t)\right| \lesssim \alpha^{9s}(6s)!s^{3d+27s+1}\log^{27s}(\epsilon^{-1})\epsilon^{3s}, \tag{178}$$

and we have

$$\begin{aligned}
\left|\phi_{\mathrm{kde}}^{(3)}(\boldsymbol{y}, t)\right| &\lesssim \left|f_{\mathrm{kde}}^{(3)}(\boldsymbol{y}, t)\right| + \alpha^{9s}(6s)!s^{3d+27s+1}\log^{27s}(\epsilon^{-1})\epsilon^{3s} \\
&\lesssim \sqrt{\alpha s \log(\epsilon^{-1})} + \alpha^{9s}(6s)!s^{3d+27s+1}\log^{27s}(\epsilon^{-1})\epsilon^{3s} \lesssim \sqrt{\alpha s \log(\epsilon^{-1})}.
\end{aligned} \tag{179}$$

By Lemma 9, there exists

$$\phi_m \in \mathcal{NN}\big(\mathrm{width} \lesssim s^2(N+1)\log_2(N); \mathrm{depth} \lesssim s^2 L\log_2(L)\big) \tag{180}$$

such that for any $t \in [0, \infty)$,

$$|\phi_m(t) - m_t| \lesssim s^{3s} \log^{3s}(\epsilon^{-1})\epsilon^{3s}, \tag{181}$$

and

$$0 \le \phi_m(t) \lesssim 1. \tag{182}$$

Similar to Eq. (172), there exists

$$\phi_{\text{multi}}^{(6)} \in \mathcal{NN}\big(\text{width} \le 9d(N+1) + d, \text{depth} \le 14sL\big), \tag{183}$$

such that for any $\phi_m(t) \in [0, 1], \phi_{\text{kde}}^{(2)}(\boldsymbol{y}, t) \in [-\sqrt{\alpha s \log(\epsilon^{-1})}, \sqrt{\alpha s \log(\epsilon^{-1})}]^d$,

$$\big|\phi_{\text{multi}}^{(6)}(\phi_m(t), \phi_{\text{kde}}^{(3)}(\boldsymbol{y}, t)) - \phi_m(t)\phi_{\text{kde}}^{(3)}(\boldsymbol{y}, t)\big| \lesssim \sqrt{\alpha s \log(\epsilon^{-1})}(N+1)^{-14sL}. \tag{184}$$

It follows that

$$\begin{aligned}
&\big|\phi_{\text{multi}}^{(6)}(\phi_m(t), \phi_{\text{kde}}^{(3)}(\boldsymbol{y}, t)) - m_t f_{\text{kde}}^{(3)}(\boldsymbol{y}, t)\big| \\
&\le \underbrace{\big|\phi_{\text{multi}}^{(6)}(\phi_m(t), \phi_{\text{kde}}^{(3)}(\boldsymbol{y}, t)) - \phi_m(t)\phi_{\text{kde}}^{(3)}(\boldsymbol{y}, t)\big|}_{\lesssim \text{ Eq. (184)}} + \underbrace{\big|\phi_m(t) - m_t\big|}_{\lesssim \text{ Eq. (181)}} \cdot \underbrace{\big|\phi_{\text{kde}}^{(3)}(\boldsymbol{y}, t)\big|}_{\lesssim \text{ Eq. (179)}} \\
&\quad + m_t \underbrace{\big|\phi_{\text{kde}}^{(3)}(\boldsymbol{y}, t) - f_{\text{kde}}^{(3)}(\boldsymbol{y}, t)\big|}_{\lesssim \text{ Eq. (178)}} \\
&\lesssim \sqrt{\alpha s \log(\epsilon^{-1})}(N+1)^{-14sL} + s^{3s} \log^{3s}(\epsilon^{-1})\epsilon^{3s} \cdot \sqrt{\alpha s \log(\epsilon^{-1})} \\
&\quad + \alpha^{9s}(6s)! s^{3d+27s+1} \log^{27s}(\epsilon^{-1})\epsilon^{3s} \\
&\lesssim \alpha^{9s}(6s)! s^{3d+27s+1} \log^{27s}(\epsilon^{-1})\epsilon^{3s}. \tag{185}
\end{aligned}$$

With Eq. (168), (172), (177), (180) and (183), we define

$$\phi_{\text{score}}^{(2)}(\boldsymbol{y}, t) := \phi_{\text{multi}}^{(6)}\big(\phi_m(t), \phi_{\text{kde}}^{(3)}(\boldsymbol{y}, t)\big) - \phi_{\text{multi}}^{(5)}\big(\boldsymbol{y}, \phi_{\text{kde}}^{(2)}(\boldsymbol{y}, t)\big), \text{ for any } \boldsymbol{y} \in \mathbb{R}^d, t \in [0, \infty). \tag{186}$$

By the sizes of $\phi_{\text{kde}}^{(2)}, \phi_{\text{kde}}^{(3)}, \phi_m, \phi_{\text{multi}}^{(5)}, \phi_{\text{multi}}^{(6)}$, we have

$$\begin{aligned}
\phi_{\text{score}}^{(2)} \in \mathcal{NN}\big(&\text{width} \lesssim s^{6d+3}N^3 \log_2(N) \vee s^{6d+3} \log^3(\epsilon^{-1}); \\
&\text{depth} \lesssim L^3 \log_2(L) \vee s^2 \log^2(\epsilon^{-1})\big). \tag{187}
\end{aligned}$$

Moreover, for any $\boldsymbol{y} \in \tilde{\mathcal{B}}, t \in [t_0, \infty)$,

$$\begin{aligned}
&\big|\phi_{\text{score}}^{(2)}(\boldsymbol{y}, t) - f_{\text{score}}^{(2)}(\boldsymbol{y}, t)\big| \\
&= \big|\phi_{\text{multi}}^{(6)}\big(\phi_m(t), \phi_{\text{kde}}^{(3)}(\boldsymbol{y}, t)\big) - \phi_{\text{multi}}^{(5)}\big(\boldsymbol{y}, \phi_{\text{kde}}^{(2)}(\boldsymbol{y}, t)\big) - m_t \times f_{\text{kde}}^{(3)}(\boldsymbol{y}, t) + \boldsymbol{y} \times f_{\text{kde}}^{(2)}(\boldsymbol{y}, t)\big| \\
&\le \underbrace{\big|\phi_{\text{multi}}^{(6)}\big(\phi_m(t), \phi_{\text{kde}}^{(3)}(\boldsymbol{y}, t)\big) - m_t \times f_{\text{kde}}^{(3)}(\boldsymbol{y}, t)\big|}_{\lesssim \text{ Eq. (185)}} + \underbrace{\big|\phi_{\text{multi}}^{(5)}\big(\boldsymbol{y}, \phi_{\text{kde}}^{(2)}(\boldsymbol{y}, t)\big) - \boldsymbol{y} \times f_{\text{kde}}^{(2)}(\boldsymbol{y}, t)\big|}_{\lesssim \text{ Eq. (175)}} \\
&\lesssim \alpha^{9s}(6s)! s^{3d+27s+1} \log^{27s}(\epsilon^{-1})\epsilon^{3s} + \alpha^{9s+1/2}(6s)! s^{3d+27s+3/2} \log^{27s+1/2}(\epsilon^{-1})\epsilon^{3s} \\
&\lesssim \alpha^{9s+1/2}(6s)! s^{3d+27s+3/2} \log^{27s+1/2}(\epsilon^{-1})\epsilon^{3s}, \tag{188}
\end{aligned}$$

which implies that

$$\begin{aligned}
\big|\phi_{\text{score}}^{(2)}(\boldsymbol{y}, t)\big| &\lesssim \big|f_{\text{score}}^{(2)}(\boldsymbol{y}, t)\big| + \alpha^{9s+1/2}(6s)! s^{3d+27s+3/2} \log^{27s+1/2}(\epsilon^{-1})\epsilon^{3s} \\
&\le m_t\big|f_{\text{kde}}^{(3)}(\boldsymbol{y}, t)\big| + |\boldsymbol{y}| \cdot \big|f_{\text{kde}}^{(2)}(\boldsymbol{y}, t)\big| + \alpha^{9s+1/2}(6s)! s^{3d+27s+3/2} \log^{27s+1/2}(\epsilon^{-1})\epsilon^{3s} \\
&\lesssim \sqrt{\alpha s \log(\epsilon^{-1})}. \tag{189}
\end{aligned}$$

**Step 3: Construction of the neural network $\phi_{\text{score}}$.**

Recall from Eq. (167) that $e^{-1}n^{-1} \le \phi_{\text{score}}^{(1)}(\boldsymbol{y}, t) \lesssim 1$ for any $\boldsymbol{y} \in \tilde{\mathcal{B}}, t \in [t_0, \infty)$ and $0 < \epsilon \le t_0 \wedge n^{-1/s}$. By Lemma 23, there exists

$$\phi_{\text{rec}} \in \mathcal{NN}\big(\text{width} \lesssim s^3 \log^3(\epsilon^{-1}); \text{depth} \lesssim s^2 \log^2(\epsilon^{-1})\big), \tag{190}$$

such that for any $n^{-1} \lesssim \phi_{\text{score}}^{(1)}(\boldsymbol{y}, t) \lesssim 1$ and $x' \in \mathbb{R}$,

$$\left| \phi_{\text{rec}}\big(\phi_{\text{score}}^{(1)}(\boldsymbol{y}, t)\big) - \frac{1}{f_{\text{score}}^{(1)}(\boldsymbol{y}, t)} \right| \leq \epsilon^{2s} + \epsilon^{-4s} \underbrace{\left| \phi_{\text{score}}^{(1)}(\boldsymbol{y}, t) - f_{\text{score}}^{(1)}(\boldsymbol{y}, t) \right|}_{\lesssim \text{ Eq. (166)}}$$

$$\lesssim \epsilon^{2s} + \alpha^{18s}(12s)! s^{3d+54s+1} \log^{54s}(\epsilon^{-1}) \epsilon^{2s}$$

$$\lesssim \alpha^{18s}(12s)! s^{3d+54s+1} \log^{54s}(\epsilon^{-1}) \epsilon^{2s}, \tag{191}$$

which implies that

$$\left| \phi_{\text{rec}}\big(\phi_{\text{score}}^{(1)}(\boldsymbol{y}, t)\big) \right| \lesssim \frac{1}{\left| f_{\text{score}}^{(1)}(\boldsymbol{y}, t) \right|} + \alpha^{18s}(12s)! s^{3d+54s+1} \log^{54s}(\epsilon^{-1}) \epsilon^{2s} \qquad \text{(by Eq. (167))}$$

$$\lesssim n + \alpha^{18s}(12s)! s^{3d+54s+1} \log^{54s}(\epsilon^{-1}) \epsilon^{2s} \lesssim n. \tag{192}$$

By Lemma 13 and Eq. (189) and (192), there exists

$$\phi_{\text{multi}}^{(7)} \in \mathcal{NN}\big(\text{width} \leq 9(N+1)+1, \text{depth} \leq 14sL\big), \tag{193}$$

such that

$$\left| \phi_{\text{multi}}^{(7)}\Big(\phi_{\text{score}}^{(2)}(\boldsymbol{y}, t), \phi_{\text{rec}}\big(\phi_{\text{score}}^{(1)}(\boldsymbol{y}, t)\big)\Big) - \phi_{\text{score}}^{(2)}(\boldsymbol{y}, t) \times \phi_{\text{rec}}\big(\phi_{\text{score}}^{(1)}(\boldsymbol{y}, t)\big) \right|$$

$$\lesssim n\sqrt{\alpha s \log(\epsilon^{-1})}(N+1)^{-14sL} \leq n\sqrt{\alpha s \log(\epsilon^{-1})}\epsilon^{4s} \leq \sqrt{\alpha s \log(\epsilon^{-1})}\epsilon^{3s}, \tag{194}$$

which indicates that

$$\left| \phi_{\text{multi}}^{(7)}\Big(\phi_{\text{score}}^{(2)}(\boldsymbol{y}, t), \phi_{\text{rec}}\big(\phi_{\text{score}}^{(1)}(\boldsymbol{y}, t)\big)\Big) - \phi_{\text{score}}^{(2)}(\boldsymbol{y}, t), \times\phi_{\text{rec}}\big(\phi_{\text{score}}^{(1)}(\boldsymbol{y}, t)\big) \right|$$

$$\lesssim \underbrace{\left| \phi_{\text{score}}^{(2)}(\boldsymbol{y}, t) \right|}_{\lesssim \text{ Eq. (189)}} \cdot \underbrace{\left| \phi_{\text{rec}}\big(\phi_{\text{score}}^{(1)}(\boldsymbol{y}, t)\big) \right|}_{\lesssim \text{ Eq. (192)}} + \sqrt{\alpha s \log(\epsilon^{-1})}\epsilon^{s}$$

$$\lesssim n\sqrt{\alpha s \log(\epsilon^{-1})} + \sqrt{\alpha s \log(\epsilon^{-1})}\epsilon^{3s} \lesssim n\sqrt{\alpha s \log(\epsilon^{-1})} \tag{195}$$

By Proposition 2, there exists

$$\phi_{1/\sigma^2} \in \mathcal{NN}\big(\text{width} \lesssim s^3 N \log_2(N) \vee s^3 \log^3(\epsilon^{-1});$$

$$\text{depth} \lesssim s^2 L \sqrt{\log(\epsilon^{-1})} \log_2(L\sqrt{\log(\epsilon^{-1})}) \vee s^2 \log^2(\epsilon^{-1})\big), \tag{196}$$

such that

$$\left| \phi_{1/\sigma^2}(t) - \sigma_t^{-2} \right| \lesssim (2s)! s^{6s} \log^{6s}(\epsilon^{-1}) \epsilon^{2s}, \quad \text{for any } t \in [t_0, \infty), \tag{197}$$

and

$$0 \leq \phi_{1/\sigma^2}(t) \lesssim t_0^{-1}. \tag{198}$$

By Lemma 13 and Eq. (195) and (198),

$$\phi_{\text{multi}}^{(8)} \in \mathcal{NN}\big(\text{width} \leq 9(N+1)+1, \text{depth} \leq 14sL\big), \tag{199}$$

such that

$$\left| \phi_{\text{multi}}^{(8)}\Big(\phi_{1/\sigma^2}(t), \phi_{\text{multi}}^{(7)}\Big(\phi_{\text{score}}^{(2)}(\boldsymbol{y}, t), \phi_{\text{rec}}\big(\phi_{\text{score}}^{(1)}(\boldsymbol{y}, t)\big)\Big)\Big) \right.$$

$$\left. - \phi_{1/\sigma^2}(t) \times \phi_{\text{multi}}^{(7)}\Big(\phi_{\text{score}}^{(2)}(\boldsymbol{y}, t), \phi_{\text{rec}}\big(\phi_{\text{score}}^{(1)}(\boldsymbol{y}, t)\big)\Big) \right|$$

$$\lesssim t_0^{-1} n\sqrt{\alpha s \log(\epsilon^{-1})}(N+1)^{-14sL} \leq t_0^{-1} n\sqrt{\alpha s \log(\epsilon^{-1})}\epsilon^{4s} \leq \sqrt{\alpha s \log(\epsilon^{-1})}\epsilon^{2s}. \tag{200}$$

With Eq. (165), (187), (190), (193), (196) and (199), for any $\boldsymbol{y} \in \mathbb{R}^d, t \in [t_0, \infty)$, we define

$$\phi_{\text{score}}(\boldsymbol{y}, t) := \phi_{\text{multi}}^{(8)}\Big(\phi_{1/\sigma^2}(t), \phi_{\text{multi}}^{(7)}\Big(\phi_{\text{score}}^{(2)}(\boldsymbol{y}, t), \phi_{\text{rec}}\big(\phi_{\text{score}}^{(1)}(\boldsymbol{y}, t)\big)\Big)\Big). \tag{201}$$

By the sizes of $\phi_{\text{score}}^{(1)}, \phi_{\text{score}}^{(2)}, \phi_{\text{rec}}, \phi_{1/\sigma^2}(t), \phi_{\text{multi}}^{(7)}, \phi_{\text{multi}}^{(8)}$, we have

$$\phi_{\text{score}} \in \mathcal{NN}\big(\text{width} \lesssim s^{6d+3} N^3 \log_2(N) \vee s^{6d+3} \log^3(\epsilon^{-1});$$

$$\text{depth} \lesssim L^3 \log_2(L) \vee s^2 \log^2(\epsilon^{-1})\big) \tag{202}$$

**Step 4: Approximation error of $\phi_{\text{score}}$.**

For any $\boldsymbol{y} \in \tilde{\mathcal{B}}, t \in [t_0, \infty)$,

$$\left| \phi_{\text{score}}(\boldsymbol{y}, t) - f_{\text{score}}^{\text{kde}}(\boldsymbol{y}, t) \right|$$

$$\leq \left| \phi_{\text{multi}}^{(8)}\left( \phi_{1/\sigma^2}(t), \phi_{\text{multi}}^{(7)}\left( \phi_{\text{score}}^{(2)}(\boldsymbol{y}, t), \phi_{\text{rec}}\left(\phi_{\text{score}}^{(1)}(\boldsymbol{y}, t)\right) \right) \right) \right.$$
$$\left. - \phi_{1/\sigma^2}(t) \times \phi_{\text{multi}}^{(7)}\left( \phi_{\text{score}}^{(2)}(\boldsymbol{y}, t), \phi_{\text{rec}}\left(\phi_{\text{score}}^{(1)}(\boldsymbol{y}, t)\right) \right) \right| \qquad \text{(by Eq. (200))}$$

$$+ \underbrace{\left| \phi_{1/\sigma^2}(t) - \frac{1}{\sigma_t^2} \right|}_{\lesssim \text{ Eq. (197)}} \cdot \underbrace{\left| \phi_{\text{multi}}^{(7)}\left( \phi_{\text{score}}^{(2)}(\boldsymbol{y}, t), \phi_{\text{rec}}\left(\phi_{\text{score}}^{(1)}(\boldsymbol{y}, t)\right) \right) \right|}_{\lesssim \text{ Eq. (195)}}$$

$$+ \frac{1}{\sigma_t^2} \underbrace{\left| \phi_{\text{multi}}^{(7)}\left( \phi_{\text{score}}^{(2)}(\boldsymbol{y}, t), \phi_{\text{rec}}\left(\phi_{\text{score}}^{(1)}(\boldsymbol{y}, t)\right) \right) - \phi_{\text{score}}^{(2)}(\boldsymbol{y}, t) \times \phi_{\text{rec}}\left(\phi_{\text{score}}^{(1)}(\boldsymbol{y}, t)\right) \right|}_{\lesssim \text{ Eq. (194)}}$$

$$+ \frac{1}{\sigma_t^2} \underbrace{\left| \phi_{\text{score}}^{(2)}(\boldsymbol{y}, t) - f_{\text{score}}^{(2)}(\boldsymbol{y}, t) \right|}_{\lesssim \text{ Eq. (188)}} \cdot \underbrace{\left| \phi_{\text{rec}}\left(\phi_{\text{score}}^{(1)}(\boldsymbol{y}, t)\right) \right|}_{\leq \text{ Eq. (192)}}$$

$$+ \frac{1}{\sigma_t^2} \left| f_{\text{score}}^{(2)}(\boldsymbol{y}, t) \right| \cdot \underbrace{\left| \phi_{\text{rec}}\left(\phi_{\text{score}}^{(1)}(\boldsymbol{y}, t)\right) - \frac{1}{f_{\text{score}}^{(1)}(\boldsymbol{y}, t)} \right|}_{\lesssim \text{ Eq. (191)}}$$

$$\lesssim \sqrt{\alpha s \log(\epsilon^{-1})} \epsilon^{2s} + (2s)! s^{6s} \log^{6s}(\epsilon^{-1}) \epsilon^{2s} \cdot n \sqrt{\alpha s \log(\epsilon^{-1})} + t_0^{-1} \sqrt{\alpha s \log(\epsilon^{-1})} \epsilon^{3s}$$
$$+ n t_0^{-1} \cdot \alpha^{9s+1/2} (6s)! s^{3d+27s+3/2} \log^{27s+1/2}(\epsilon^{-1}) \epsilon^{3s}$$
$$+ t_0^{-1} \sqrt{\alpha s \log(\epsilon^{-1})} \cdot \alpha^{18s} (12s)! s^{3d+54s+1} \log^{54s}(\epsilon^{-1}) \epsilon^{2s}$$
$$\lesssim \alpha^{18s+1/2} (12s)! s^{3d+54s+3/2} \log^{54s+1/2}(\epsilon^{-1}) \epsilon^s. \qquad (203)$$

By Lemma 5 and $\rho_{n,t} = (2\pi\sigma_t)^{-d/2} e^{-1} n^{-1}$, we have for any $\boldsymbol{y} \in \mathbb{R}^d, t \in [t_0, \infty)$,

$$|f_{\text{score}}^{\text{kde}}(\boldsymbol{y}, t)| = \left| \frac{\nabla \hat{p}_t(\boldsymbol{y})}{\hat{p}_t(\boldsymbol{y}) \vee \rho_n} \right| \leq \left\| \frac{\nabla \hat{p}_t(\boldsymbol{y})}{\hat{p}_t(\boldsymbol{y}) \vee \rho_n} \right\|_2 \leq \frac{\sqrt{2}}{\sigma_t} \sqrt{\log\left( \frac{(2\pi\sigma_t^2)^{-d/2}}{\rho_{n,t}} \right)}$$

$$= \frac{\sqrt{2}}{\sigma_t} \sqrt{\log n + 1} \lesssim \sigma_t^{-1} \sqrt{\log n}, \qquad (204)$$

which gives that

$$\left| \phi_{\text{score}}(\boldsymbol{y}, t) \right| \lesssim \left| f_{\text{score}}^{\text{kde}}(\boldsymbol{y}, t) \right| + \alpha^{18s+1/2} (12s)! s^{3d+54s+3/2} \log^{54s+1/2}(\epsilon^{-1}) \epsilon^s \lesssim \sigma_t^{-1} \sqrt{\log n}.$$

Therefore, we have completed the proof of Lemma 10.

### D.3.3 Approximation of regularized empirical score functions for sub-Gaussian distributions

**Lemma 11.** *Suppose that $P$ satisfies Assumption 1. For any $d, n \in \mathbb{N}_+, 0 < t_0 \leq 1/2$, fix $\rho_{n,t} := (2\pi\sigma_t^2)^{-d/2}e^{-1}n^{-1}$. Let $N, L, s \in \mathbb{N}_+, \epsilon \in \mathbb{R}_+$ such that $N^{-2}L^{-2} \leq \epsilon \leq t_0 \wedge n^{-1/s}$. Let $m_t := \exp(-t), \sigma_t := \sqrt{1 - \exp(-2t)}$ for any $t \in [t_0, \infty)$. Then, there exists a function $\phi_{score}$ implemented by a ReLU DNN with width $\leq \mathcal{O}\big(s^{6d+3}N^3 \log_2(N) \vee s^{6d+3} \log^3(\epsilon^{-1})\big)$ and depth $\leq \mathcal{O}\big(L^3 \log_2(L) \vee s^2 \log^2(\epsilon^{-1})\big)$ such that*

$$\mathbb{E}_{\{\boldsymbol{x}^{(i)}\}_{i=1}^n \sim P^{\otimes n}}\left[\int_{\mathbb{R}^d} \left\|\frac{\nabla \hat{p}_t(\boldsymbol{y})}{\hat{p}_t(\boldsymbol{y}) \vee \rho_{n,t}} - \phi_{score}(\boldsymbol{y}, t)\right\|_2^2 \mathrm{d}\boldsymbol{y}\right] \lesssim \alpha^{74s}((12s)!)^2 s^{6d+108s+3} \log^{108s+1}(\epsilon^{-1})\epsilon^{2s},$$

*and we have $\|\phi_{score}(\cdot, t)\|_\infty \lesssim \sigma_t^{-1}\sqrt{\log n}$.*

*Proof.* For some $A > 0$, set $\mathcal{A} = \mu + [-A, A]^d$, where $\mu = \mathbb{E}_{X \sim P_0}[X]$. With loss of generality, we assume that $\mathbb{E}_{X \sim P_0}[X] = 0$. Our results can be easily applied to $\mathbb{E}_{X \sim P_0}[X] \neq 0$. By the sub-Gaussian tail bound of $P_0$ with parameter $\alpha$,

$$\Pr[X \notin \mathcal{A}] = \int_{\mathcal{A}^c} p_t(\boldsymbol{x})\mathrm{d}\boldsymbol{x} \leq 2d\exp\left(-\frac{A^2}{2\alpha^2}\right). \tag{205}$$

Let $A = \mathcal{O}(\sqrt{\alpha^2 s \log n})$, then

$$\Pr[X \notin \mathcal{A}] \leq 2d\exp\left(-\frac{\mathcal{O}(\alpha^2 s \log n)}{2\alpha^2}\right) \leq 2d\exp(-\mathcal{O}(s \log n)) = 2dn^{-\mathcal{O}(s)},$$

By Lemma 5 and $\rho_{n,t} = (2\pi\sigma_t)^{-d/2}e^{-1}n^{-1}$, we have for any $\boldsymbol{y} \in \mathbb{R}^d, t \in [t_0, \infty)$,

$$\left\|\frac{\nabla \hat{p}_t(\boldsymbol{y})}{\hat{p}_t(\boldsymbol{y}) \vee \rho_n}\right\|_2^2 \leq \frac{2}{\sigma_t^2} \log\left(\frac{(2\pi\sigma_t^2)^{-d/2}}{\rho_{n,t}}\right) = \frac{2}{\sigma_t^2}(\log n + 1) \lesssim \frac{1}{\sigma_t^2} \log n. \tag{206}$$

By Lemma 10, there exists a function

$$\phi_{score} \in \mathcal{NN}\big(\text{width} \lesssim s^{6d+3}N^3 \log_2(N) \vee s^{6d+3} \log^3(\epsilon^{-1});$$
$$\text{depth} \lesssim L^3 \log_2(L) \vee s^2 \log^2(\epsilon^{-1})\big)$$

such that for any $\sup_{i \in [n]} |\boldsymbol{x}^{(i)}| \leq A = \sqrt{\mathcal{O}(\alpha^2 s \log n)}$,

$$\left|\phi_{score}(\boldsymbol{y}, t) - f_{score}^{kde}(\boldsymbol{y}, t)\right| \lesssim \alpha^{37s}(12s)!s^{3d+54s+\frac{3}{2}} \log^{54s+\frac{1}{2}}(\epsilon^{-1})\epsilon^s, \quad \forall \boldsymbol{y} \in \mathbb{R}^d, \forall t \in [t_0, \infty).$$

Thus, as we have shown in Eq. (204), for all $\boldsymbol{y} \in \mathbb{R}^d, t \in [t_0, \infty)$, we have $|\phi_{score}(\boldsymbol{y}, t)| \leq \|\phi_{score}(\boldsymbol{y}, t)\|_2 \lesssim \sigma_t^{-1}\sqrt{\log n}$, which gives that

$$\|\phi_{score}(\boldsymbol{y}, t)\|_2^2 \lesssim \sigma_t^{-2} \log n. \tag{207}$$

With Eq. (206) and (207),

$$\int_{\mathcal{A}^c} \int_{\mathbb{R}^d} \left\|\frac{\nabla \hat{p}_t(\boldsymbol{y})}{\hat{p}_t(\boldsymbol{y}) \vee \rho_{n,t}} - \phi_{score}(\boldsymbol{y}, t)\right\|_2^2 \mathrm{d}\boldsymbol{y}p(\boldsymbol{x})\mathrm{d}\boldsymbol{x}$$
$$\leq 2\int_{\mathcal{A}^c} \left(\left\|\frac{\nabla \hat{p}_t(\boldsymbol{y})}{\hat{p}_t(\boldsymbol{y}) \vee \rho_{n,t}}\right\|_2^2 + \|\phi_{score}(\boldsymbol{y}, t)\|_2^2\right)p(\boldsymbol{x})\mathrm{d}\boldsymbol{x}$$
$$\lesssim n^{-\mathcal{O}(s)}\big(\sigma_t^{-2} \log n + \sigma_t^{-2} \log n\big) \lesssim n^{-\mathcal{O}(s)} \log n. \tag{208}$$

On the other hand, for $\boldsymbol{x} \in \mathcal{A}$, we have

$$\int_{\mathcal{A}} \int_{\mathbb{R}^d} \left[\left\|\frac{\nabla \hat{p}_t(\boldsymbol{y})}{\hat{p}_t(\boldsymbol{y}) \vee \rho_{n,t}} - \phi_{score}(\boldsymbol{y}, t)\right\|_2^2\right]p_0(\boldsymbol{x})\mathrm{d}\boldsymbol{x}$$
$$\lesssim \int_{\mathcal{A}^c} d\left(\alpha^{37s}(12s)!s^{3d+54s+\frac{3}{2}} \log^{54s+\frac{1}{2}}(\epsilon^{-1})\epsilon^s\right)^2 p(\boldsymbol{x})\mathrm{d}\boldsymbol{x}$$
$$\lesssim \alpha^{74s}((12s)!)^2 s^{6d+108s+3} \log^{108s+1}(\epsilon^{-1})\epsilon^{2s}. \tag{209}$$

Combine Eq. (208) and (209), we obtain

$$\mathbb{E}_{\{\boldsymbol{x}^{(i)}\}_{i=1}^n \sim P^{\otimes n}}\left[\int_{\mathbb{R}^d}\left\|\frac{\nabla\hat{p}_t(\boldsymbol{y})}{\hat{p}_t(\boldsymbol{y})\vee\rho_{n,t}} - \phi_{\text{score}}(\boldsymbol{y},t)\right\|_2^2 \mathrm{d}\boldsymbol{y}\right]$$
$$\lesssim \alpha^{74s}((12s)!)^2 s^{6d+108s+3}\log^{108s+1}(\epsilon^{-1})\epsilon^{2s}. \tag{210}$$

$\square$

**Lemma 12** ($L^2$-Approximation of Regularized Empirical Score Functions for Sub-Gaussian Distributions)**.** *Suppose that $P$ satisfies Assumption 1. For any $d, n \in \mathbb{N}_+$, fix $\rho_{n,t} := (2\pi\sigma_t^2)^{-d/2}e^{-1}n^{-1}$ and $n^{-2/d} \leq t_0 \leq 1/2$. Let $m_t := \exp(-t), \sigma_t := \sqrt{1-\exp(-2t)}$ for any $t \in [t_0,\infty)$. Fix $k \in \mathbb{N}_+$ with $k \geq d/2$. Then, there exists a ReLU DNN $\phi_{score}$ with width $\leq \mathcal{O}\big(n^{\frac{3}{2k}}\log_2 n\big)$ and depth $\leq \mathcal{O}\big(\log^2 n\big)$ such that*

$$\mathbb{E}_{\{\boldsymbol{x}^{(i)}\}_{i=1}^n \sim P^{\otimes n}}\left[\int_{\mathbb{R}^d}\left\|\frac{\nabla\hat{p}_t(\boldsymbol{y})}{\hat{p}_t(\boldsymbol{y})\vee\rho_{n,t}} - \phi_{score}(\boldsymbol{y},t)\right\|_2^2\mathrm{d}\boldsymbol{y}\right] \lesssim \alpha^{74k}\log^{108k+1}(n)n^{-1},$$

*and we have $\|\phi_{score}(\cdot,t)\|_\infty \lesssim \sigma_t^{-1}\sqrt{\log n}$.*

*Proof.* Fix $k \in \mathbb{N}_+$ and $k \geq d/2$, apply Lemma 11 with $\epsilon = n^{-1/k}, s = k$ and $N = \lceil n^{1/(2k)}\rceil, L = 1$ such that $N^{-2}L^{-2} \leq \epsilon \leq t_0 \wedge n^{-1/s}$ hold and we complete the proof. $\square$

### D.4 Score approximation errors by deep ReLU neural networks

We are now able to prove the score approximation error bounds of deep RelU neural networks for sub-Gaussian distributions by combining Theorem 1 and Corollary 12:

**Theorem 1** (Neural Network Score Approximation for Sub-Gaussian Distributions) *Suppose that $P_0$ satisfies Assumption 1. For any $1 \leq d \lesssim \sqrt{\log n}, n \geq 3$ and any $\frac{1}{2}\alpha^2 n^{-2/d}\log n < t_0 \leq 1$ and $T = n^{\mathcal{O}(1)}$, let $\{\boldsymbol{x}_t\}_{t\in[t_0,T]}$ be the solutions of the process Eq. (2) with density function $p_t : \mathbb{R}^d \to \mathbb{R}_+$. Fix $k \in \mathbb{N}_+$ with $d/2 \leq k \lesssim \frac{\log n}{\log\log n}$. Then, there exists a ReLU DNN $\phi_{score}$ with width $\leq \mathcal{O}\big(n^{\frac{3}{2k}}\log_2 n\big)$ and depth $\leq \mathcal{O}\big(\log^2 n\big)$ such that*

$$\mathbb{E}_{\{\boldsymbol{x}^{(i)}\}_{i=1}^n \sim P_0^{\otimes n}}\left[\mathbb{E}_{\boldsymbol{x}_t\sim P_t}\left[\|\nabla\log p_t(\boldsymbol{x}_t) - \phi_{score}(\boldsymbol{x}_t,t)\|_2^2\right]\right] \lesssim \sigma_t^{-d-2}\big(\sigma_t^d + \alpha^d\big)\frac{\log^{d/2+3}n}{n},$$

*and we have $\|\phi_{score}(\cdot,t)\|_\infty \lesssim \sigma_t^{-1}\sqrt{\log n}$. Moreover, let $T = n^{\mathcal{O}(1)}$, we have*

$$\mathbb{E}_{\{\boldsymbol{x}^{(i)}\}_{i=1}^n \sim P_0^{\otimes n}}\left[\int_{t=t_0}^T \mathbb{E}_{\boldsymbol{x}_t\sim P_t}\left[\|\nabla\log p_t(\boldsymbol{x}_t) - \phi_{score}(\boldsymbol{x}_t,t)\|_2^2\right]\mathrm{d}t\right] \lesssim \alpha^d t_0^{-d/2}n^{-1}\log^{\frac{d}{2}+4}n.$$

*Proof.* By Lemma 1, with $\rho_{n,t} = (2\pi\sigma_t^2)^{-d/2}e^{-1}n^{-1}$, we have

$$\mathbb{E}_{\{\boldsymbol{x}^{(i)}\}_{i=1}^n \sim P^{\otimes n}}\left[\mathbb{E}_{\boldsymbol{x}_t\sim P_t}\left[\left\|\frac{\nabla p_t(\boldsymbol{x}_t)}{p_t(\boldsymbol{x}_t)} - \frac{\nabla\hat{p}_t(\boldsymbol{x}_t)}{\hat{p}_t(\boldsymbol{x}_t)\vee\rho_{n,t}}\right\|_2^2\right]\right]$$
$$\lesssim \sigma_t^{-d-2}\big(\sigma_t^d + \alpha^d\big)\log^3\left(\frac{(2\pi\sigma_t^2)^{-\frac{d}{2}}}{\rho_n}\right)\frac{\log^{d/2}n}{n}$$
$$\lesssim \sigma_t^{-d-2}\big(\sigma_t^d + \alpha^d\big)\frac{\log^{d/2+3}n}{n}$$

Therefore, together with $\phi_{\mathrm{score}}$ from Lemma 12, we obtain

$$\mathbb{E}_{\{\boldsymbol{x}^{(i)}\}_{i=1}^n \sim P_0^{\otimes n}}\left[\mathbb{E}_{\boldsymbol{x}_t \sim P_t}\left[\|\nabla \log p_t(\boldsymbol{x}_t) - \phi_{\mathrm{score}}(\boldsymbol{x}_t, t)\|_2^2\right]\right]$$

$$\leq \underbrace{\mathbb{E}_{\{\boldsymbol{x}^{(i)}\}_{i=1}^n \sim P_0^{\otimes n}}\left[2\mathbb{E}_{\boldsymbol{x}_t \sim P_t}\left[\left\|\frac{\nabla p_t(\boldsymbol{x}_t)}{p_t(\boldsymbol{x}_t)} - \frac{\nabla \hat{p}_t(\boldsymbol{x}_t)}{\hat{p}_t(\boldsymbol{x}_t) \vee \rho_{n,t}}\right\|_2^2\right]\right]}_{\lesssim \text{ Lemma 1}}$$

$$+ \underbrace{\mathbb{E}_{\{\boldsymbol{x}^{(i)}\}_{i=1}^n \sim P_0^{\otimes n}}\left[2\mathbb{E}_{\boldsymbol{x}_t \sim P_t}\left[\left\|\frac{\nabla \hat{p}_t(\boldsymbol{x}_t)}{\hat{p}_t(\boldsymbol{x}_t) \vee \rho_{n,t}} - \phi_{\mathrm{score}}(\boldsymbol{x}_t, t)\right\|_2^2\right]\right]}_{\lesssim \text{ Lemma 12}}$$

$$\lesssim \sigma_t^{-d-2}\left(\sigma_t^d + \alpha^d\right)\frac{\log^{d/2+3} n}{n} + \alpha^{74k}\frac{\log^{108k+1}(n)}{n}$$

$$\lesssim \sigma_t^{-d-2}\left(\sigma_t^d + \alpha^d\right)\frac{\log^{d/2+3} n}{n},$$

where the last inequality follows from the fact that its second term will be dominated by the first term. To see this, let's fix $d/2 \leq k \leq \frac{(1+2/d)\log n + \log\log n - 2\log\alpha}{74\log\alpha + 108\log\log n}$, then we have

$$74k\log\alpha + 108k\log n \leq (1 + 2/d)\log n + \log\log n - 2\log\alpha,$$

which gives

$$\alpha^{74k}\log^{108k+1}(n) \leq \alpha^{-2}n^{1+\frac{2}{d}}\log^2 n$$

$$\lesssim \sigma_t^{-d-2}\alpha^d\frac{\log^{d/2+3} n}{n} \qquad (\text{by } \alpha^2 n^{-2/d}\log n \leq t_0 \leq \sigma_t^2)$$

$$\leq \sigma_t^{-d-2}\left(\sigma_t^d + \alpha^d\right)\frac{\log^{d/2+3} n}{n}.$$

Notice that we also need

$$\frac{d}{2} \leq \frac{(1+2/d)\log n + \log\log n - 2\log\alpha}{74\log\alpha + 108\log\log n},$$

which implies that $d \lesssim \sqrt{\log n}$. Assuming that $\alpha$ is a universal constant, we have verified that to ensure our results hold, we require $d/2 \leq k \lesssim \frac{\log n + \log\log n}{\log\log n} \lesssim \frac{\log n}{\log\log n}$ for $1 \leq d \lesssim \sqrt{\log n}$.

Moreover, following the same proof for Theorem 4, we obtain

$$\mathbb{E}_{\{\boldsymbol{x}^{(i)}\}_{i=1}^n \sim P_0^{\otimes n}}\left[\int_{t=t_0}^T \mathbb{E}_{\boldsymbol{x}_t \sim P_t}\left[\|\nabla \log p_t(\boldsymbol{x}_t) - \phi_{\mathrm{score}}(\boldsymbol{x}_t, t)\|_2^2\right]\mathrm{d}t\right]$$

$$\lesssim \int_{t=t_0}^T \sigma_t^{-d-2}\left(\sigma_t^d + \alpha^d\right)\frac{\log^{d/2+3} n}{n}\mathrm{d}t \lesssim \alpha^d t_0^{-d/2} n^{-1}\log^{\frac{d}{2}+4} n.$$

Thus, we complete the proof. $\qquad\qquad\square$

## D.5 Auxiliary approximation lemmas

**Proposition 8** (Approximate Square Function on $[a, b]$). *For any $N, L \in \mathbb{N}_+$ and $a, b \in \mathbb{R}$ with $a < b$, there exists a function $\phi$ implemented by a ReLU DNN with width $3N + 1$ and depth $L$ such that*

$$|\phi(x) - x^2| \leq N^{-L}, \quad \text{for any } x \in [a, b].$$

*Proof.* For any $x \in [a, b]$, let $\tilde{x} = \frac{x-a}{b-a}$, which implies that $\tilde{x} \in [0, 1]$. By Lemma 18, there exists a function $\phi$ implemented by a ReLU DNN with width $3N$ and depth $L$ such that

$$|\phi(\tilde{x}) - \tilde{x}^2| \leq N^{-L},$$

which gives that

$$\left|(b-a)^2 \phi\left(\frac{x-a}{b-a}\right) + 2ax - a^2 - x^2\right| \leq (b-a)^2 N^{-L}.$$

For any $x \in \mathbb{R}$, define

$$\tilde{\phi}(x) := (b-a)^2 \phi\left(\frac{x-a}{b-a}\right) + 2a \cdot \text{ReLU}(x + |a|) - a^2 - 2a|a|.$$

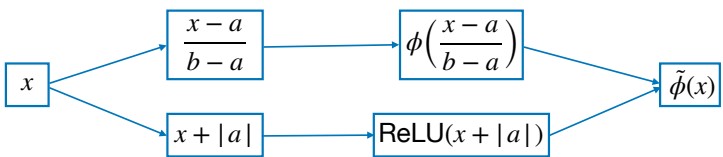

Figure 1: An illustration of the network architecture implementing $\tilde{\phi}$ for approximating $x^2$ on $[a, b]$.

By $\phi \in \mathcal{NN}(\text{width} \leq 3N; \text{depth} \leq L)$ from Lemma 18, the network in Proposition 8 has width $\leq 3N + 1$ and depth $\leq L + 2$. It follows that $\tilde{\phi}$ can be implemented by a ReLU DNN with width $3N + 1$ and depth $L$, since the two hidden layers have the identify function as their activation functions.

Since $x + |a| \geq 0$ for any $x \in [a, b]$, which indicates that

$$\tilde{\phi}(x) = (b-a)^2 \phi\left(\frac{x-a}{b-a}\right) + 2ax - a^2, \quad \text{for any } x \in [a, b].$$

Therefore,

$$|\tilde{\phi}(x) - x^2| \leq (b-a)^2 N^{-L}, \quad \text{for any } x \in [a, b].$$

$\square$

**Lemma 13** (Approximate $xy$ on $[a_1, b_1] \times [a_2, b_2]$). *For any $N, L \in \mathbb{N}_+$ and $a_1, a_2, b_1, b_2 \in \mathbb{R}$ with $a_1 < b_1$ and $a_2 < b_2$, there exists a function $\phi$ implemented by a ReLU DNN with width $9N + 1$ and depth $L$ such that*

$$|\phi(x, y) - xy| \leq 6(b_1 - a_1)(b_2 - a_2)N^{-L}, \quad \text{for any } x \in [a_1, b_1], y \in [a_2, b_2].$$

*Proof.* For any $x \in [a_1, b_1], y \in [a_2, b_2]$, let $\tilde{x} = \frac{x-a_1}{b_1-a_1}, \tilde{y} = \frac{y-a_2}{b_2-a_2}$, which implies that $\tilde{x}, \tilde{y} \in [0, 1]$. By Lemma 19, there exists a function $\tilde{\phi}$ implemented by a ReLU DNN with width $9N$ and depth $L$ such that

$$\left|\tilde{\phi}(\tilde{x}, \tilde{y}) - \tilde{x}\tilde{y}\right| \leq 6N^{-L},$$

which gives that

$$\left|(b_1 - a_1)(b_2 - a_2)\tilde{\phi}\left(\frac{x-a_1}{b_1-a_1}, \frac{y-a_2}{b_2-a_2}\right) - xy + a_1 y + a_2 x - a_1 a_2\right| \leq 6(b_1 - a_1)(b_2 - a_2)N^{-L}.$$

For any $x \in \mathbb{R}$, define

$$\phi(x, y) := (b_1 - a_1)(b_2 - a_2)\tilde{\phi}\Big(\frac{x - a_1}{b_1 - a_1}, \frac{y - a_2}{b_2 - a_2}\Big) + a_1 x + a_2 y - a_1 a_2. \qquad (211)$$

By construction, $\tilde{\phi}$ can be implemented by a ReLU DNN with width $3N + 1$ and depth $L$. Therefore, for all $x \in [a_1, b_1], y \in [a_2, b_2]$,

$$|\phi(x, y) - xy| \le 6(b_1 - a_1)(b_2 - a_2)N^{-L}.$$

$\square$

**Lemma 14** (Approximate Monomial Function on $[-R, R]^k$). *For any $N, L, k, s \in \mathbb{N}_+$ with $s \ge k \ge 2$ and $R \in \mathbb{R}_+$, there exists a function $\phi$ implemented by a ReLU DNN with width $9(N + 1) + k - 1$ and depth $7sL(k - 1)$ such that*

$$|\phi(\boldsymbol{x}) - x_1 x_2 \cdots x_k| \le 30(k - 1)R^k(N + 1)^{-7sL}, \quad \text{for any } \boldsymbol{x} = [x_1, x_2, \ldots, x_k]^\top \in [-R, R]^k.$$

*Proof.* By Lemma 13, there exists a function $\phi_{\text{multi}}$ implemented by a ReLU DNN with width $9(N + 1) + 1$ and depth $7sL$ such that

$$|\phi_{\text{multi}}(x, y) - xy| \le 6(b_1 - a_1)(b_2 - a_2)(N + 1)^{-7sL} \quad \text{for any } x \in [a_1, b_1], y \in [a_1, b_2].$$

For all $x_1, x_2 \in [-R, R]$, define $\phi_1 : [-R, R]^2 \to [-R^2, R^2]$, which can be implemented by a ReLU DNN with width $9(N + 1) + 1$ and depth $7sL$ such that

$$|\phi_1(x_1, x_2) - x_1 x_2| \le 6(2R)^2(N + 1)^{-7sL} < 30R^2(N + 1)^{-7sL}. \qquad (212)$$

Next, we construct a sequence of functions $\phi_i : [-R, R]^{i+1} \to [-R, R]$ for $i \in \{1, 2, \ldots, k - 1\}$ by induction such that

  (i) $\phi_i$ can be implemented by a ReLU DNN with width $9(N + 1) + i$ and depth $7sLi$ for each $i \in \{1, 2, \ldots, k - 1\}$.

 (ii) For any $i \in \{1, 2, \ldots, k - 1\}$ and $x_1, x_2, \ldots, x_{i+1} \in [-R, R]$, it holds that

$$|\phi_i(x_1, \ldots, x_{i+1}) - x_1 x_2 \cdots x_{i+1}| \le 30i R^{i+1}(N + 1)^{-7sL}. \qquad (213)$$

First, let us consider the case $i = 1$, it is obvious that the two required conditions are true:

  (i) $9(N + 1) + i = 9(N + 1) + 1$ and $7sLi = 7sL$ if $i = 1$;

 (ii) Eq. (212) implies Eq. (213) for $i = 1$.

Now assume $\phi_i$ has been defined; we then define

$$\phi_{i+1}(x_1, \ldots, x_{i+2}) := \phi_{\text{multi}}\big(\phi_i(x_1, \ldots, x_{i+1}), x_{i+2}\big) \quad \text{for any } x_1, \ldots, x_{i+2} \in \mathbb{R}.$$

Note that $\phi_i \in \mathcal{NN}(\text{width} \le 9(N + 1) + i; \text{depth} \le 7sLi)$ and $\phi_{\text{multi}} \in \mathcal{NN}(\text{width} \le 9(N + 1) + 1; \text{depth} \le 7kL)$. Then $\phi_{i+1}$ can be implemented via a ReLU DNN with width

$$\max\{9(N + 1) + i + 1, 9(N + 1) + 1\} = 9(N + 1) + (i + 1)$$

and depth $7sLi + 7sL = 7sL(i + 1)$. By the hypothesis of induction, we have

$$\Big|\phi_i(x_1, \ldots, x_{i+1}) - x_1 x_2 \ldots x_{i+1}\Big| \le 30i R^{i+1}(N + 1)^{-7sL}.$$

Recall the fact that

$$30i R^{i+1}(N + 1)^{-7sL} \le 30k R^{i+1} 2^{-7s} \le 30k R^{i+1}\frac{2^{-7}}{s} < 0.25 R^{i+1}$$

for any $N, L, k, s \in \mathbb{N}_+$ with $s \geq k$ and $i \in \{1, 2, \ldots, k-1\}$. By $x_1 x_2 \cdots x_{i+1} \in [-R^{i+1}, R^{i+1}]$, it follows that

$$\phi_i(x_1, \ldots, x_{i+1}) \in [-1.25R^{i+1}, 1.25R^{i+1}], \quad \text{for any } x_1, \ldots, x_{i+1} \in [-R, R].$$

Therefore, by Eq. (211) and (213), we have

$$\begin{aligned}
&\left| \phi_{i+1}(x_1, \ldots, x_{i+2}) - x_1 x_2 \ldots x_{i+2} \right| \\
&= \left| \phi_1\big(\phi_i(x_1, \ldots, x_{i+1}), \mathrm{ReLU}(x_{i+2})\big) - x_1 x_2 \cdots x_{i+2} \right| \\
&\leq \left| \phi_1\big(\phi_i(x_1, \ldots, x_{i+1}), x_{i+2}\big) - \phi_i(x_1, \ldots, x_{i+1}) x_{i+2} \right| + \left| \phi_i(x_1, \ldots, x_{i+1}) x_{i+2} - x_1 x_2 \ldots x_{i+2} \right| \\
&\leq 6 \times 2.5 R^{i+1} \times 2R \times (N+1)^{-7sL} + 30i R^{i+1}(N+1)^{-7sL} \cdot |x_{i+2}| \qquad \text{(by Lemma 13)} \\
&\leq 30 R^{i+2}(N+1)^{-7sL} + 30i R^{i+1}(N+1)^{-7sL} \cdot R \\
&= 30(i+1) R^{i+2}(N+1)^{-7sL},
\end{aligned}$$

for any $x_1, x_2, \ldots, x_{i+2} \in [-R, R]$, which means we finish the process of induction. Now let $\phi := \phi_{k-1}$, by the principle of induction, we have

$$\left| \phi(x_1, \ldots, x_k) - x_1 x_2 \cdots x_k \right| \leq 30(k-1) R^k (N+1)^{-7sL}, \quad \text{for any } x_1, \ldots, x_k \in [-R, R].$$

So $\phi$ is the desired function implemented by a ReLU DNN with width $9(N+1) + k - 1$ and depth $7sL(k-1)$. $\qquad \square$

**Proposition 9** (Approximate Multivariate Polynomials on $[-R, R]^d$)**.** *Assume* $P(\boldsymbol{x}) = \boldsymbol{x}^{\boldsymbol{\nu}} = x_1^{\nu_1} x_2^{\nu_2} \cdots x_d^{\nu_d}$ *for* $\boldsymbol{\nu} \in \mathbb{N}^d$ *with* $\|\boldsymbol{\nu}\|_1 \leq k \in \mathbb{N}_+$*. For any* $N, L, s \in \mathbb{N}_+$ *with* $s \geq k$ *and* $R \in \mathbb{R}_+$*, there exists a function* $\phi$ *implemented by a ReLU DNN with width* $9(N+1) + k - 1$ *and depth* $7s(k-1)L$ *such that*

$$|\phi(\boldsymbol{x}) - P(\boldsymbol{x})| \leq 30k R^k (N+1)^{-7sL}, \quad \text{for any } \boldsymbol{x} \in [-R, R]^d.$$

*Proof.* The case $k = 1$ is trivial, so we assume that $k \geq 2$. Set $\tilde{k} = \|\boldsymbol{\nu}\|_1 \leq k$, denote by $\boldsymbol{\nu} = [\nu_1, \nu_2, \ldots, \nu_d]^\top$ and let $[z_1, \ldots, z_{\tilde{k}}]^\top \in \mathbb{R}^{\tilde{k}}$ be the vector such that

$$z_l = x_j, \quad \text{if } \sum_{i=1}^{j-1} \nu_i < l \leq \sum_{i=1}^{l} \nu_i, \quad \text{for } j = 1, \ldots, d.$$

That is,

$$[z_1, z_2, \ldots, z_{\tilde{k}}]^\top = [\overbrace{x_1, \ldots, x_1}^{\nu_1 \text{ times}}, \overbrace{x_2, \ldots, x_2}^{\nu_2 \text{ times}}, \ldots, \overbrace{x_d, \ldots, x_d}^{\nu_d \text{ times}}] \in \mathbb{R}^{\tilde{k}}.$$

Then, we have $P(\boldsymbol{x}) = \boldsymbol{x}^{\boldsymbol{\nu}} = z_1 z_2 \ldots z_{\tilde{k}}$.

We construct the target ReLU DNN in two steps:

**Step 1:** There exists an affine linear map $\phi_{\mathrm{lin}} : \mathbb{R}^d \to \mathbb{R}^k$ that duplicates $\boldsymbol{x}$ to form a new vector $[z_1, z_2, \ldots, z_{\tilde{k}}, 1, \ldots, 1]^\top \in \mathbb{R}^k$, i.e.,

$$\phi_{\mathrm{lin}}(\boldsymbol{x}) = [z_1, z_2, \ldots, z_{\tilde{k}}, 1, \ldots, 1]^\top \in \mathbb{R}^k.$$

**Step 2:** By Lemma 14, there exists a function $\phi_{\mathrm{mon}} : \mathbb{R}^k \to \mathbb{R}$ implemented by a ReLU DNN with width $9(N+1) + k - 1$ and depth $7sL(k-1)$ such that

$$|\phi_{\mathrm{mon}}(\boldsymbol{x}) - \boldsymbol{x}^{\boldsymbol{\nu}}| \leq 30(k-1) R^k (N+1)^{-7sL}.$$

For any $\boldsymbol{x} \in [-R, R]^d$, define

$$\phi(\boldsymbol{x}) := \phi_{\mathrm{mon}}\big(\phi_{\mathrm{lin}}(\boldsymbol{x})\big). \tag{214}$$

Clearly, we have

$$\phi \in \mathcal{NN}\big(\text{width} \leq 9(N+1) + k - 1; \text{depth} \leq 7s(k-1)L\big)$$

and

$$\begin{aligned}
|\phi(\boldsymbol{x}) - P(\boldsymbol{x})| &= |\phi(\boldsymbol{x}) - \boldsymbol{x}^{\boldsymbol{\nu}}| \\
&= \left|\phi_{\mathrm{mon}}\big(\phi_{\mathrm{lin}}(\boldsymbol{x})\big) - x_1^{\nu_1} x_2^{\nu_2} \cdots x_d^{\nu_d}\right| \\
&= \left|\phi_{\mathrm{mon}}(z_1, z_2, \ldots, z_{\tilde{k}}, 1, \ldots, 1) - z_1 z_2 \ldots z_{\tilde{k}}\right| \\
&\leq 30(k-1)R^k(N+1)^{-7sL} \qquad \text{(by Lemma 14)} \\
&\leq 30kR^k(N+1)^{-7sL},
\end{aligned}$$

for any $x_1, x_2, \ldots, x_d \in [-R, R]$. $\qquad\qquad\qquad\qquad\qquad\qquad\qquad\qquad\qquad\square$

**Proposition 10** (Approximate One-Dimensional Step Function on $[a, b]$)**.** *For any $N, L, d \in \mathbb{N}_+$, $a, b \in \mathbb{R}$ with $a < b$, and $\delta \in (0, \frac{b-a}{3K}]$ with $K = \lfloor N^{1/d} \rfloor^2 \lfloor L^{2/d} \rfloor$, there exists a one-dimensional function $\phi$ implemented by a ReLU DNN with width $4\lfloor N^{1/d} \rfloor + 3$ and depth $4L + 5$ such that*

$$\phi(x) = k, \quad \text{if } x \in \left[a + \tfrac{k(b-a)}{K}, a + \tfrac{(k+1)(b-a)}{K} - \delta \cdot \mathbb{1}_{\{k \leq K-2\}}\right], \quad \text{for } k = 0, 1, \ldots, K-1.$$

*Proof.* Let $\tilde{x} = \frac{x-a}{b-a}$ for all $x \in \left[a + \frac{k(b-a)}{K}, a + \frac{(k+1)(b-a)}{K} - \delta \cdot \mathbb{1}_{\{k \leq K-2\}}\right], k = 0, 1, \ldots, K-1$, which gives

$$\tilde{x} \in \left[\tfrac{k}{K}, \tfrac{k+1}{K} - \tfrac{\delta}{b-a} \cdot \mathbb{1}_{\{k \leq K-2\}}\right], \quad \text{for } k = 0, 1, \ldots, K-1.$$

Then, by Proposition 12, there exists a one-dimensional function $\tilde{\phi}$ implemented by a ReLU DNN with width $4\lfloor N^{1/d} \rfloor + 3$ and depth $4L + 5$ such that

$$\tilde{\phi}(\tilde{x}) = k, \quad \text{for all } \tilde{x} \in \left[\tfrac{k}{K}, \tfrac{k+1}{K} - \tfrac{\delta}{b-a} \cdot \mathbb{1}_{\{k \leq K-2\}}\right], \quad \text{for } k = 0, 1, \ldots, K-1.$$

Let $\phi(x) = \tilde{\phi}\big(\frac{x-a}{b-a}\big)$. Then, $\phi$ can be implemented by a ReLU DNN with the same sizes as a ReLU DNN for implementing $\tilde{\phi}$ and we have

$$\phi(x) = k, \quad \text{if } x \in \left[a + \tfrac{k(b-a)}{K}, a + \tfrac{(k+1)(b-a)}{K} - \delta \cdot \mathbb{1}_{\{k \leq K-2\}}\right], \quad \text{for } k = 0, 1, \ldots, K-1.$$

$$\square$$

**Definition 3** (Modulus of Continuity [38])**.** *For any $a, b \in \mathbb{R}$ with $b > a$. The modulus of continuity of a continuous function $f \in C([a, b]^d)$ is defined as*

$$\omega_f(r) := \sup\left\{|f(\boldsymbol{x}) - f(\boldsymbol{y})| : \|\boldsymbol{x} - \boldsymbol{y}\|_2 \leq r, \boldsymbol{x}, \boldsymbol{y} \in [a, b]^d\right\}, \quad \text{for any } r \geq 0.$$

**Definition 4** (Trifling Region [38])**.** *Given any $K \in \mathbb{N}^+$ and $\delta \in (0, \frac{1}{K})$, define a trifling region $\Omega([0, 1]^d, K, \delta)$ of $[0, 1]^d$ as*

$$\Omega([0, 1]^d, K, \delta) := \bigcup_{i=1}^d \left\{\boldsymbol{x} = [x_1, x_2, \ldots, x_d]^\top \in [0, 1]^d : x_i \in \bigcup_{k=1}^{K-1} \big(\tfrac{k}{K} - \delta, \tfrac{k}{K}\big)\right\}.$$

*In particular, $\Omega([0, 1]^d, K, \delta) = \emptyset$ if $K = 1$.*

The following Lemma extends the approximation results of [38, Lemma 3.3] from $[0, 1]$ to $[0, R], \forall R > 0$:

**Lemma 15** (Approximate Trifling Regions on $[0, R]$)**.** *Given any $\varepsilon > 0, K, R \in \mathbb{N}_+$, and $\delta \in (0, \frac{1}{3K}]$, assume $f \in C([0, R])$ and $g : \mathbb{R} \to \mathbb{R}$ is a general function with*

$$|g(x) - f(x)| \leq \varepsilon, \quad \text{for any } \boldsymbol{x} \in [0, R] \setminus \Omega([0, R], K, \delta). \tag{215}$$

*i.e., $g(x) \in \mathcal{B}(f(x), \varepsilon)$ for any $x \in [0, R] \setminus \Omega([0, R], K, \delta)$. Then,*

$$|\phi(x) - f(x)| \leq \varepsilon + \omega_f(\delta), \quad \text{for any } x \in [0, R],$$

*where*

$$\phi(x) := \mathrm{mid}\big(g(x - \delta), g(x), g(x + \delta)\big), \quad \text{for any } x \in \mathbb{R}.$$

*Proof.* Divide $[0, R]$ into $KR$ small intervals denoted by $\mathcal{I}_k = [\frac{k}{K}, \frac{k+1}{K}]$ for $k = 0, 1, \ldots, KR - 1$. For each $k \in \{0, 1, \cdots, KR - 1\}$, we further divide $\mathcal{I}_k$ into four small closed intervals as

$$\mathcal{I}_k = \mathcal{I}_{k,1} \cup \mathcal{I}_{k,2} \cup \mathcal{I}_{k,3} \cup \mathcal{I}_{k,4},$$

where $\mathcal{I}_{k,1} = [\frac{k}{K}, \frac{k}{K} + \delta], \mathcal{I}_{k,2} = [\frac{k}{K} + \delta, \frac{k+1}{K} - 2\delta], \mathcal{I}_{k,3} = [\frac{k+1}{K} - 2\delta, \frac{k+1}{K} - \delta], \mathcal{I}_{k,4} = [\frac{k}{K} - \delta, \frac{k+1}{K}]$. Following a similar proof in [38, Lemma 3.3], we have for all $x \in \mathcal{I}_{k,1}, \mathcal{I}_{k,2}, \mathcal{I}_{k,3}, \mathcal{I}_{k,4}$,

$$\mathrm{mid}\big(g(x - \delta), g(x), g(x + \delta)\big) \in \mathcal{B}(f(x), \varepsilon + \omega_f(\delta)).$$

By $[0, R] = \bigcup_{k=0}^{KR-1} \big(\cup_{j=1}^4 \mathcal{I}_{k,j}\big)$, which implies that

$$\mathrm{mid}\big(g(x - \delta), g(x), g(x + \delta)\big) \in \mathcal{B}(f(x), \varepsilon + \omega_f(\delta)), \quad \text{for all } x \in [0, R].$$

Therefore, we have

$$|\phi(x) - f(x)| \leq \varepsilon + \omega_f(\delta), \quad \text{for any } x \in [0, R].$$

$\square$

While the technique of approximating the univariate exponential function $\exp(-x)$ using ReLU neural networks has been explored in recent works, e.g., [22, 43], we develop a new proof strategy tailored to our setting, which enables separate control of the approximation rates in terms of the network's width and depth.

**Lemma 16** (Approximation of $\exp(-x)$ on [0, R]). *For any $N, L \in \mathbb{N}_+$ and $R > 0$ such that $N^{-2}L^{-2} \leq R^{-1}$, there exists a function $\phi$ implemented by a ReLU DNN with width $48s^2(N + 1)\log_2(8N)$ and depth $18s^2(L + 2)\log_2(4L) + 2$ such that*

$$|\phi(x) - \exp(-x)| \leq \big(45s + R^s + 4\big)N^{-2s}L^{-2s}, \quad \text{for any } x \in [0, R].$$

*Proof.* Set $K = N^2L^2$ and $\delta \in (0, 1/K)$. Let $\Omega([0, R], K, \delta)$ partition $[0, R]$ into $K$ intervals $\mathcal{I}_\beta$ for $\beta \in \{0, 1, \ldots, K - 1\}$. For each $\beta$, we define $x_\beta := \frac{\beta R}{K}$ and

$$\mathcal{I}_\beta := \Big\{x \in \mathbb{R} : x \in \big[\tfrac{\beta R}{K}, \tfrac{(\beta+1)R}{K} - \delta \cdot \mathbb{1}_{\{\beta \leq K-2\}}\big]\Big\}.$$

Clearly, we have $[0, R] = \Omega([0, R], K, \delta) \bigcup \big(\cup_{\beta \in \{0,1,\cdots,K-1\}} \mathcal{I}_\beta\big)$ and $x_\beta$ is the vertex of $\mathcal{I}_\beta$ with minimum $\|\cdot\|_1$-norm.

**Step 1: Approximation on non-trifling regions: $x \in [0, R] \setminus \Omega([0, R], K, \delta)$.**

**Approximate $x_\beta$.**

By Proposition 12, there exists a ReLU DNN

$$\phi_{\text{step}} \in \mathcal{NN}\big(\text{width} \leq 4N + 3; \text{depth} \leq 4L + 5\big)$$

such that

$$\phi_{\text{step}}(x/R) = k, \quad \text{if } x \in \big[\tfrac{kR}{K}, \tfrac{(k+1)R}{K} - \delta \cdot \mathbb{1}_{k \leq K-2}\big], \quad \text{for } k = 0, 1, \ldots, K - 1.$$

Define a ReLU DNN $\tilde{\phi}_{\text{step}}$ by

$$\tilde{\phi}_{\text{step}}(x) := \tfrac{R}{K}\phi_{\text{step}}(x/R), \quad \text{for any } x \in [0, R].$$

Obviously, we have

$$\tilde{\phi}_{\text{step}} \in \mathcal{NN}\big(\text{width} \leq 4N + 3; \text{depth} \leq 4L + 5\big) \tag{216}$$

and

$$\tilde{\phi}_{\text{step}}(x) = \frac{\beta R}{K} = x_\beta, \quad \text{for } \beta \in \{0, 1, \ldots, K - 1\}.$$

By Taylor expansion of $\exp(-x)$ at $x_\beta$ up to order $s - 1$, we have:

$$\exp(-x) = \underbrace{\sum_{k=0}^{s-1} \frac{(-1)^k \exp(-x_\beta)}{k!}\big(x - x_\beta\big)^k}_{=:F_1} + \underbrace{\frac{(-1)^s \exp(-\theta)}{s!}\big(x - x_\beta\big)^s}_{=:F_2}.$$

for some real number $\theta$ that is between $t$ and $x_\beta$.

For all $x \in [0, R] \setminus \Omega([0, R], K, \delta)$, the magnitude of $F_2$ can be bounded by

$$|F_2| := \left| \frac{(-1)^s \exp(-\theta)}{s!} (x - x_\beta)^s \right| \leq \frac{1}{s!} (x - x_\beta)^s \leq \frac{1}{s!} \left( \frac{R}{K} \right)^s \leq \frac{1}{s!} R^s N^{-2s} L^{-2s}. \quad (217)$$

Therefore, we only need to construct a ReLU DNN to approximate $F_1$.

**Approximate** $\exp(-x_\beta)$**.** For $\beta \in \{0, 1, \ldots, K-1\}$, define

$$\xi_\beta := \exp\left( -\frac{\beta R}{K} \right).$$

Then we have $\xi_\beta \in [0, 1]$ for all $\beta \in \{0, 1, \ldots, K-1\}$. With $K = N^2 L^2$, by Proposition 13, there exists a ReLU DNN

$$\phi_{\text{point}} \in \mathcal{NN}\left( \text{width} \leq 16s(N+1)\log_2(8N); \text{depth} \leq 5(L+2)\log_2(4L) \right)$$

such that

$$|\phi_{\text{point}}(\beta) - \xi_\beta| \leq N^{-2s} L^{-2s}, \quad \text{for } \beta = 0, 1, \ldots, K-1,$$
$$0 \leq \phi_{\text{point}}(\beta) \leq 1.$$

For any $x \in \mathbb{R}$, define

$$\tilde{\phi}_{\text{point}}(x) := \phi_{\text{point}}\left( \frac{xK}{R} \right).$$

Obviously, we also have

$$\tilde{\phi}_{\text{point}} \in \mathcal{NN}\left( \text{width} \leq 16s(N+1)\log_2(8N); \text{depth} \leq 5(L+2)\log_2(4L) \right). \quad (218)$$

Then for all $x_\beta = \frac{\beta R}{K}, \beta \in \{0, 1, \ldots, K-1\}$, we have $0 \leq \tilde{\phi}_{\text{point}}(x_\beta) \leq 1$ and

$$\begin{aligned}
\left| \tilde{\phi}_{\text{point}}(x_\beta) - \exp(-x_\beta) \right| &= \left| \tilde{\phi}_{\text{point}}\left( \frac{\beta R}{K} \right) - \exp\left( -\frac{\beta R}{K} \right) \right| \\
&= \left| \phi_{\text{point}}(\beta) - \xi_\beta \right| \\
&\leq N^{-2s} L^{-2s}. \quad (219)
\end{aligned}$$

**Approximate** $(x - x_\beta)^k$**.** Let $\tilde{x} := x - x_\beta$, then $\tilde{x} \in [0, \frac{R}{K}] \subseteq [0, 1]$. By Proposition 11, there exists a ReLU DNN

$$\phi_{\text{poly}} \in \mathcal{NN}\left( \text{width} \leq 9(N+1) + s - 1, \text{depth} \leq 7s(s-1)L \right) \quad (220)$$

such that for all $0 \leq k \leq s$,

$$\left| \phi_{\text{poly}}(\tilde{x}) - \tilde{x}^k \right| \leq 9s(N+1)^{-7sL}. \quad (221)$$

Note that $0 \leq \tilde{x} \leq R/K$, which gives that $\tilde{t}^k \in [0, 1]$. For $0 \leq k \leq s$,

$$9s(N+1)^{-7sL} \leq 9s2^{-7s} \leq 0.1.$$

With Eq. (221) we have for all $x \in [0, R] \setminus \Omega([0, R], K, \delta)$,

$$\phi_{\text{poly}}(\tilde{x}) = \phi_{\text{poly}}(x - x_\beta) \in [-0.1, 1.1].$$

**Approximate** $\exp(-x)$ **for** $x \in [0, R \setminus \Omega([0, R], K, \delta)$**.** By Lemma 13, there exists a ReLU DNN

$$\phi_{\text{multi}} \in \mathcal{NN}\left( \text{width} \leq 9(N+1) + 1; \text{depth} \leq 2s(L+1) \right) \quad (222)$$

such that for any $x \in [a_1, b_1], y \in [a_2, b_2]$,

$$|\phi_{\text{multi}}(x, y) - xy| \leq 6(b_1 - a_1)(b_2 - a_2)(N+1)^{-2s(L+1)}.$$

For all $x \in [0, \infty)$, we construct a neural network of the form:

$$\tilde{\phi}_{\exp}(x) := \phi_{\text{multi}}\Big( \tilde{\phi}_{\text{point}}\big(\tilde{\phi}_{\text{step}}(x)\big), \sum_{k=0}^{s-1} \frac{1}{k!} \phi_{\text{poly}}\big(x - \tilde{\phi}_{\text{step}}(x)\big) \Big) \qquad (223)$$

By Eq. (216), (218), (220) and (222), it is easy to verify that $\tilde{\phi}_{\exp}$ can be implemented by a ReLU DNN with width

$$\max\Big\{ 4N + 3, 16s(N+1)\log_2(8N), s \cdot \big(9(N+1) + s - 1\big), 9(N+1) + 1 \Big\} \leq 16s^2(N+1)\log_2(8N),$$

and depth

$$\begin{aligned}
& 4L + 5 + 5(L+2)\log_2\big(4L\big) + 7s^2 L + 2s(L+1) + 3 \\
& \leq 4(L+2) + 5(L+2)\log_2\big(4L\big) + 7s^2(L+2) + 2s^2(L+2) \\
& \leq 18s^2(L+2)\log_2\big(4L\big).
\end{aligned}$$

Therefore, we have

$$\tilde{\phi}_{\exp} \in \mathcal{NN}\big(\text{width} \leq 16s^2(N+1)\log_2(8N); \text{depth} \leq 18s^2(L+2)\log_2\big(4L\big)\big). \qquad (224)$$

Fix $\beta \in \{0, 1, \ldots, K-1\}$, for all $x \in \mathcal{I}_\beta$, we have

$$\Big| \tilde{\phi}_{\exp}(x) - \exp(-x) \Big|$$

$$\leq \Bigg| \phi_{\text{multi}}\Big( \tilde{\phi}_{\text{point}}\big(\tilde{\phi}_{\text{step}}(x)\big), \sum_{k=0}^{s-1} \frac{1}{k!} \phi_{\text{poly}}\big(x - \tilde{\phi}_{\text{step}}(x)\big) \Big) - \sum_{k=0}^{s-1} \frac{(-1)^k \exp(-x_\beta)}{k!}\big(x - x_\beta\big)^k \Bigg| + |F_2|$$

$$\leq \underbrace{\Bigg| \phi_{\text{multi}}\Big( \tilde{\phi}_{\text{point}}\big(\tilde{\phi}_{\text{step}}(x)\big), \sum_{k=0}^{s-1} \frac{1}{k!} \phi_{\text{poly}}\big(x - \tilde{\phi}_{\text{step}}(x)\big) \Big) - \tilde{\phi}_{\text{point}}\big(\tilde{\phi}_{\text{step}}(x)\big) \times \sum_{k=0}^{s-1} \frac{1}{k!} \phi_{\text{poly}}\big(x - \tilde{\phi}_{\text{step}}(x)\big) \Bigg|}_{=:F_{1,1}}$$

$$+ \underbrace{\Bigg| \tilde{\phi}_{\text{point}}\big(\tilde{\phi}_{\text{step}}(x)\big) \times \sum_{k=0}^{s-1} \frac{1}{k!} \phi_{\text{poly}}\big(x - \tilde{\phi}_{\text{step}}(x)\big) - \exp(-x_\beta) \times \sum_{k=0}^{s-1} \frac{1}{k!} \phi_{\text{poly}}\big(x - \tilde{\phi}_{\text{step}}(x)\big) \Bigg|}_{=:F_{1,2}}$$

$$+ \underbrace{\Bigg| \exp(-x_\beta) \times \sum_{k=0}^{s-1} \frac{1}{k!} \phi_{\text{poly}}\big(x - \tilde{\phi}_{\text{step}}(x)\big) - \exp(-x_\beta) \sum_{k=0}^{s-1} \frac{(-1)^k}{k!}\big(x - x_\beta\big)^k \Bigg|}_{=:F_{1,3}} + \frac{1}{s!} R^s N^{-2s} L^{-2s}.$$

**Bounding $F_{1,1}$.** For all $x \in [0, R \setminus \Omega([0, R], K, \delta)$, we have $\tilde{\phi}_{\text{point}}\big(\tilde{\phi}_{\text{step}}(x)\big) \in [0, 1]$ and

$$-0.3 < -e \times 0.1 \leq \sum_{k=0}^{s-1} \frac{-0.1}{k!} \leq \sum_{k=0}^{s-1} \frac{1}{k!}\phi_{\text{poly}}(t - x_\beta) \leq \sum_{k=0}^{s-1} \frac{1.1}{k!} \leq e \times 1.1 < 3.$$

Therefore,
$$F_{1,1} \leq 6s\big(3 - (-0.3)\big)(N+1)^{-2s(L+1)} < 20s(N+1)^{-2s(L+1)}. \qquad (225)$$

**Bounding $F_{1,2}$.**

$$F_{1,2} \leq \underbrace{\Big| \tilde{\phi}_{\text{point}}\big(\tilde{\phi}_{\text{step}}(x)\big) - \exp(-x_\beta) \Big|}_{\leq \text{Eq. (219)}} \cdot \Bigg| \sum_{k=0}^{s-1} \frac{1}{k!} \phi_{\text{poly}}\big(x - \tilde{\phi}_{\text{step}}(x)\big) \Bigg| \leq 3N^{-2s}L^{-2s}. \qquad (226)$$

**Bounding $F_{1,3}$.**

$$F_{1,3} \leq \left| \exp(-x_\beta) \right| \cdot \left| \sum_{k=0}^{s-1} \frac{1}{k!} \phi_{\text{poly}}\big(x - \tilde{\phi}_{\text{step}}(x)\big) - \sum_{k=0}^{s-1} \frac{(-1)^k}{k!} \big(x - x_\beta\big)^k \right|$$

$$\leq \sum_{k=0}^{s-1} \frac{1}{k!} \underbrace{\left| \phi_{\text{poly},k}\big(x - \tilde{\phi}_{\text{step}}(x)\big) - \big(x - x_\beta\big)^k \right|}_{\leq \text{Eq. (221)}}$$

$$\leq \sum_{k=0}^{s-1} \frac{1}{k!} 9s(N+1)^{-7sL} = 9es(N+1)^{-7sL} < 25s(N+1)^{-7sL}. \tag{227}$$

Combine Eq. (217) and (225) to (227), we obtain

$$|\tilde{\phi}_{\text{exp}}(x) - \exp(-t)| \leq 20s(N+1)^{-2s(L+1)} + 3N^{-2s}L^{-2s} + 25s(N+1)^{-7sL} + \frac{1}{s!}R^s N^{-2s} L^{-2s}$$

$$\leq 20s(N+1)^{-2s(L+1)} + 25s(N+1)^{-7sL} + (3 + R^s)N^{-2s}L^{-2s}.$$

Note that for any $N, L, s \in \mathbb{N}_+$,

$$(N+1)^{-7sL} \leq (N+1)^{-2s(L+1)} \leq (N+1)^{-2s}2^{-2sL} \leq N^{-2s}L^{-2s},$$

which gives that

$$|\tilde{\phi}_{\text{exp}}(x) - \exp(-x)| \leq \big(45s + R^s + 3\big)N^{-2s}L^{-2s}. \tag{228}$$

**Step 2: Approximation on the whole regions: $x \in [0, R]$).**

For all $x \in \mathbb{R}$, define $\phi_{\text{exp}}$ by:

$$\phi_{\text{exp}}(x) := \text{mid}\big(\tilde{\phi}_{\text{exp}}(x - \delta), \tilde{\phi}_{\text{exp}}(x), \tilde{\phi}_{\text{exp}}(x + \delta)\big).$$

Then, by Lemma 15, we have for all $\delta \in (0, \frac{R}{3K}]$,

$$|\phi_{\text{exp}}(x) - \exp(-x)| \leq \big(45s + R^s + 3\big)N^{-2s}L^{-2s} + \omega_{\exp(-x)}(\delta), \quad \text{for any } x \in [0, R],$$

where $\omega_{\exp(-x)}(\delta)$ is defined as

$$\omega_{\exp(-x)}(\delta) := \sup\left\{ |\exp(-x) - \exp(-y)| : \|x - y\|_2 \leq \delta, x, y \in [0, \lceil\sqrt{R}\rceil]^2 \right\}.$$

Choose $\delta \in (0, \frac{R}{3K}]$ such that

$$\omega_{\exp(-x)}(\delta) \leq N^{-2s}L^{-2s},$$

which gives that for all $x \in [0, R]$,

$$|\phi_{\text{exp}}(x) - \exp(-x)| \leq \big(45s + R^s + 3\big)N^{-2s}L^{-2s} + N^{-2s}L^{-2s} = \big(45s + R^s + 4\big)N^{-2s}L^{-2s}.$$

To determine the size of the network for implementing $\phi_{\text{exp}}$, note that from Eq. (224), we have

$$\tilde{\phi}_{\text{exp}}(\cdot - \delta), \tilde{\phi}_{\text{exp}}(\cdot), \tilde{\phi}_{\text{exp}}(\cdot + \delta) \in \mathcal{NN}\big(\text{width} \leq 16s^2(N+1)\log_2(8N); \text{depth} \leq 18s^2(L+2)\log_2\big(4L\big)\big).$$

Then, we have

$$\phi_{\text{exp}} \in \mathcal{NN}\big(\#\text{input} = 1; \text{width} \leq 48s^2(N+1)\log_2(8N); \text{depth} \leq 18s^2(L+2)\log_2\big(4L\big) \#\text{output} = 3\big). \tag{229}$$

Recall that $\text{mid}(\cdot, \cdot, \cdot) \in \mathcal{NN}(\text{width} \leq 14; \text{depth} \leq 2)$ by Lemma 22. Therefore, $\phi_{\text{exp}} = \text{mid}(\cdot, \cdot, \cdot) \circ \tilde{\phi}_{\text{exp}}$ can be implemented by a ReLU DNN with width

$$\max\big\{48s^2(N+1)\log_2(8N), 14\big\} = 48s^2(N+1)\log_2(8N)$$

and depth

$$18s^2(L+2)\log_2(4L) + 2.$$

$\square$

**Lemma 17** (Approximate $k$-th Root Function on $[0, R]$). *For any $N, L, k \in \mathbb{N}_+$ and $R > 0$, there exists a function $\phi$ implemented by a ReLU DNN with width $48s^2(N+1)\log_2(8N)$ and depth $18s^2(L+2)\log_2(4L) + 2$ such that*

$$|\phi(x) - x^{1/k}| \leq (45s+5)R^{1/k}N^{-2s}L^{-2s}, \quad \text{for any } x \in [0, R].$$

*Proof.* Let $\tilde{x} = x/R \in [0, 1]$, we have $x^{1/k} = R^{1/k}\tilde{x}^{1/k}$ for any $x \in [0, R]$.

**Step 1: Decompose the domain.**

Set $K = N^2L^2$ and $\delta \in (0, 1/K)$. Let $\Omega([0,1], K, \delta)$ partition $[0, 1]$ into $K$ intervals $\mathcal{I}_\beta$ for $\beta \in \{0, 1, \ldots, K-1\}$. For each $\beta$, we define $x_\beta := \frac{\beta}{K}$ and

$$\mathcal{I}_\beta := \left\{ x \in \mathbb{R} : x \in \left[ \tfrac{\beta}{K}, \tfrac{(\beta+1)R}{K} - \delta \cdot \mathbb{1}_{\{\beta \leq K-2\}} \right] \right\}.$$

Clearly, we have $[0, ] = \Omega([0,], K, \delta) \bigcup \left( \cup_{\beta \in \{0,1,\cdots,K-1\}} \mathcal{I}_\beta \right)$ and $x_\beta$ is the vertex of $\mathcal{I}_\beta$ with minimum $\|\cdot\|_1$-norm.

**Step 2: Taylor expansion of $\tilde{x}^{1/k}$.**

By Taylor expansion of $\tilde{x}^{1/k}$ at $x_\beta$ up to order $s-1$, we have:

$$\tilde{x}^{1/k} = \underbrace{\sum_{r=0}^{s-1}(-1)^r \binom{1/k}{r}(1-\tilde{x})^r}_{=:F_1} + \underbrace{(-1)^s\theta^s\binom{1/k}{s}(1-\tilde{x})^s}_{=:F_2}.$$

for some real number $\theta$ that is between $t$ and $x_\beta$.

**Step 3: Approximation error and the size of the network**

Following a similar proof for Lemma 16, we can obtain that for any $s, k \in \mathbb{N}_+$, there exists a function $\tilde{\phi}$ implemented by a ReLU DNN with width $48s^2(N+1)\log_2(8N)$ and depth $18s^2(L+2)\log_2(4L)$ such that

$$|\tilde{\phi}(\tilde{x}) - \tilde{x}^{1/k}| \leq (45s+5)N^{-2s}L^{-2s}, \quad \text{for any } \tilde{x} \in [0, 1].$$

For any $x \in [0, R]$, define $\phi$ by

$$\phi(x) := R^{1/k}\tilde{\phi}(x/R).$$

Clearly, we have

$$\phi \in \mathcal{NN}\left(\text{width} \leq 48s^2(N+1)\log_2(8N); \text{depth} \leq 18s^2(L+2)\log_2(4L)\right)$$

such that

$$\begin{aligned}
\left|\phi(x) - x^{1/k}\right| &= R^{1/k}\left|\tilde{\phi}(x/R) - (x/R)^{1/k}\right| \\
&\leq (45s+5)R^{1/k}N^{-2s}L^{-2s}, \quad \text{for any } x \in [0, R].
\end{aligned}$$

$\square$

### D.6 Existing approximation results

**Lemma 18** (Approximate Square Function on $[0, 1]$ [38]). *For any $N, L \in \mathbb{N}_+$, there exists a function $\phi$ implemented by a ReLU DNN with width $3N$ and depth $L$ such that*

$$|\phi(x) - x^2| \leq N^{-L}, \quad \text{for any } x \in [0, 1].$$

**Lemma 19** (Approximate $xy$ on $[0, 1]$ [38]). *For any $N, L \in \mathbb{N}_+$, there exists a function $\phi$ implemented by a ReLU DNN with width $9N$ and depth $L$ such that*

$$|\phi(x, y) - xy| \leq 6N^{-L}, \quad \text{for any } x, y \in [0, 1].$$

**Lemma 20** (Approximate $xy$ on $[a, b]$ [38]). *For any $N, L \in \mathbb{N}_+$ and $a, b \in \mathbb{R}$ with $a < b$, there exists a function $\phi$ implemented by a ReLU DNN with width $9N + 1$ and depth $L$ such that*

$$|\phi(x, y) - xy| \leq 6(b - a)^2 N^{-L}, \quad \text{for any } x, y \in [a, b].$$

**Lemma 21** (Approximate Monomial Function on $[0, 1]$ [38]). *For any $N, L, k \in \mathbb{N}_+$ with $k \geq 2$, there exists a function $\phi$ implemented by a ReLU DNN with width $9(N + 1) + k - 1$ and depth $7kL(k - 1)$ such that*

$$|\phi(\boldsymbol{x}) - x_1 x_2 \cdots x_k| \leq 9(k - 1)(N + 1)^{-7kL}, \quad \text{for any } \boldsymbol{x} = [x_1, x_2, \ldots, x_k]^\top \in [0, 1]^k.$$

**Proposition 11** (Approximate Multivariate Polynomials on $[0, 1]^d$ [38]). *Assume $P(\boldsymbol{x}) = \boldsymbol{x}^{\boldsymbol{\nu}} = x_1^{\nu_1} x_2^{\nu_2} \cdots x_d^{\nu_d}$ for $\boldsymbol{\nu} \in \mathbb{N}^d$ with $\|\boldsymbol{\nu}\|_1 \leq k \in \mathbb{N}_+$. For any $N, L \in \mathbb{N}_+$, there exists a function $\phi$ implemented by a ReLU DNN with width $9(N + 1) + k - 1$ and depth $7k^2 L$ such that*

$$|\phi(\boldsymbol{x}) - P(\boldsymbol{x})| \leq 9k(N + 1)^{-7kL}, \quad \text{for any } \boldsymbol{x} \in [0, 1]^d.$$

Our goal is to construct a step function $\Psi$ mapping $\boldsymbol{x} \in Q_{\boldsymbol{\beta}}$ to $\boldsymbol{x}_{\boldsymbol{\beta}} = \frac{\boldsymbol{\beta}}{K}$ for any $\boldsymbol{\beta} \in \{0, 1, \ldots, K - 1\}^d$. We only need to approximate one-dimensional step functions, because in the multidimensional case, we can simply set $\Psi(\boldsymbol{x}) = [\psi(x_1), \psi(x_2), \ldots, \psi(x_d)]^\top$, where $\psi$ is a one-dimensional step function.

**Proposition 12** (Approximate One-Dimensional Step Function on $[0, 1]$ [38]). *For any $N, L, d \in \mathbb{N}_+$ and $\delta \in (0, \frac{1}{3K}]$ with $K = \lfloor N^{1/d} \rfloor^2 \lfloor L^{2/d} \rfloor$, there exists a one-dimensional function $\phi$ implemented by a ReLU DNN with width $4\lfloor N^{1/d} \rfloor + 3$ and depth $4L + 5$ such that*

$$\phi(x) = k, \quad \text{if } x \in \left[\frac{k}{K}, \frac{k + 1}{K} - \delta \cdot \mathbb{1}_{\{k \leq K - 2\}}\right], \quad \text{for } k = 0, 1, \ldots, K - 1.$$

**Proposition 13** (Point Fitting on $[0, 1]$ [38]). *Given any $N, L, d \in \mathbb{N}_+$ and $\xi_i \in [0, 1]$ for $i = 0, 1, \ldots, N^2 L^2 - 1$, there exists a function $\phi$ implemented by a ReLU DNN with width $16s(N + 1) \log_2(8N)$ and depth $5(L + 2) \log_2(4L)$ such that*

*(1) $|\phi(i) - \xi_i| \leq N^{-2s} L^{-2s}$ for $i = 0, 1, \ldots, N^2 L^2 - 1$;*

*(2) $0 \leq \phi(x) \leq 1$ for any $x \in \mathbb{R}$.*

**Lemma 22** (Approximate Middle Value Function [38]). *The middle value function $\text{mid}(x_1, x_2, x_3)$ can be implemented by a ReLU DNN with width $14$ and depth $2$.*

**Lemma 23** (Approximating the Reciprocal Function [22]). *For any $0 < \epsilon < 1$, there exists a function $\phi$ implemented by a ReLU DNN*

$$\phi \in \mathcal{NN}\left(\text{width} \lesssim \log^3(\epsilon^{-1}); \text{depth} \lesssim \log^2(\epsilon^{-1})\right)$$

*such that*

$$\left|\phi(x') - \frac{1}{x}\right| \leq \epsilon + \frac{|x' - x|}{\epsilon^2}, \quad \text{for all } x \in [\epsilon, \epsilon^{-1}] \text{ and } x' \in \mathbb{R}.$$

# E  Neural Network Score Estimation and Distribution Estimation

## E.1  Score estimation errors by deep neural networks

**Theorem 2** (Neural Network Score Estimation for Sub-Gaussian Distributions) *Assume that the conditions of Theorem 1 hold. For $3 \leq d \lesssim \sqrt{\log n}$, fix $k \in \mathbb{N}_+$ with $\frac{6 \log n}{(d-2) \log(t_0^{-1})} \vee d/2 \leq k \lesssim \frac{\log n}{\log \log n}$ in Eq. (9). Then, for any $\delta \in (0,1)$, with probability at least $1 - \delta$, the excess risk of an empirical risk minimizer Eq. (7) over the neural network class $\mathcal{NN}$ satisfies that*

$$\int_{t_0}^{T} \mathbb{E}_{\boldsymbol{X}_t}\big[\|\widehat{\phi}(\boldsymbol{X}_t, t) - \nabla \log p_t(\boldsymbol{X}_t)\|_2^2\big] \mathrm{d}t \lesssim t_0^{-d/2} n^{-1} \log^{\frac{d}{2}+4} n + t_0^{-1} n^{-1} \log n \cdot \log \frac{2}{\delta}.$$

*Proof.* For notation simplicity, throughout the following, we write

$$\mathcal{NN} \equiv \big\{\mathcal{NN}(\text{width} \leq \mathcal{O}(n^{\frac{3}{2k}} \log_2 n); \text{depth} \leq \mathcal{O}(\log^2 n)) \mid \|\phi(\cdot, t)\|_\infty \lesssim \sigma_t^{-1} \sqrt{\log n}\big\}.$$

and

$$\phi^*(\boldsymbol{y}, t) := \nabla \log p_t(\boldsymbol{y}), \quad \text{for all } \boldsymbol{y} \in \mathbb{R}^d, t \in [t_0, T],$$

Recall that

$$\ell(\phi, \boldsymbol{X}_0) := \int_{t_0}^{T} \mathbb{E}_{\boldsymbol{X}_t | \boldsymbol{X}_0}[\|\phi(\boldsymbol{X}_t, t) - \nabla \log p_t(\boldsymbol{X}_t | \boldsymbol{X}_0)\|_2^2] \mathrm{d}t$$

$$= \int_{t_0}^{T} \mathbb{E}_{\boldsymbol{X}_t | \boldsymbol{X}_0}\Big[\Big\|\phi(\boldsymbol{X}_t, t) + \frac{\boldsymbol{X}_t - m_t \boldsymbol{X}_0}{\sigma_t^2}\Big\|_2^2\Big] \mathrm{d}t.$$

For all $\phi : \mathbb{R}^d \times [t_0, T] \to \mathbb{R}^d$, we have

$$\int_{t_0}^{T} \mathbb{E}_{\boldsymbol{X}_t}[\|\phi(\boldsymbol{X}_t, t) - \phi^*(\boldsymbol{X}_t, t)\|_2^2] \mathrm{d}t = \mathbb{E}_{\boldsymbol{X}_0}[\ell(\phi, \boldsymbol{X}_0)] - \mathbb{E}_{\boldsymbol{X}_0}[\ell(\phi^*, \boldsymbol{X}_0)]. \tag{230}$$

Given a set of i.i.d. samples $S := \{\boldsymbol{x}^{(i)}\}_{i=1}^n$, where $\boldsymbol{x}^{(i)} \sim P_0$, the denoising score matching estimate is defined as an empirical risk minimizer Eq. (7), i.e.,:

$$\widehat{\phi} \in \arg\min_{\phi \in \mathcal{NN}} \frac{1}{n} \sum_{i=1}^{n} \ell(\phi, \boldsymbol{x}^{(i)}).$$

Given another set of i.i.d. samples $\bar{S} := \{\bar{\boldsymbol{x}}^{(i)}\}_{i=1}^n$, where $\bar{\boldsymbol{x}}^{(i)} \sim P_0$, we write the regularized empirical score functions associated with $\bar{S}$ and $\rho_{n,t} = (2\pi \sigma_t^2)^{-d/2} e^{-1} n^{-1}$ as

$$\hat{\bar{\phi}}(\boldsymbol{y}, t) := \frac{\nabla \hat{p}_t(\boldsymbol{y})}{\hat{p}_t(\boldsymbol{y}) \vee \rho_{n,t}}. \tag{231}$$

For any $\phi \in \mathcal{NN}$, denote by

$$\widehat{\mathbb{E}}_{\boldsymbol{X}_0}\big[\ell(\phi, \boldsymbol{X}_0) - \ell(\phi^*, \boldsymbol{X}_0)\big] = \frac{1}{n} \sum_{i=1}^{n} \big(\ell(\phi, \boldsymbol{x}^{(i)}) - \ell(\phi^*, \boldsymbol{x}^{(i)})\big),$$

$$\widehat{\mathbb{E}}_{\boldsymbol{X}_0}\big[\mathbb{E}_{\bar{S}}[\ell(\phi, \boldsymbol{X}_0) - \ell(\hat{\bar{\phi}}, \boldsymbol{X}_0)]\big] = \frac{1}{n} \sum_{i=1}^{n} \mathbb{E}_{\bar{S}}\big[\ell(\phi, \boldsymbol{x}^{(i)}) - \ell(\hat{\bar{\phi}}, \boldsymbol{x}^{(i)})\big],$$

where the samples $\boldsymbol{x}^{(1)}, \ldots, \boldsymbol{x}^{(n)}$ are drawn independently from the same distribution as $\boldsymbol{X}_0$ and independent of $\bar{S}$. We aim to provide a high-probability upper bound on

$$\mathbb{E}_{\boldsymbol{X}_0}\big[\ell(\widehat{\phi}, \boldsymbol{X}_0) - \ell(\phi^*, \boldsymbol{X}_0)\big] \tag{232}$$

for the empirical risk minimizer $\widehat{\phi}$.

Recall from Lemma 5 that

$$\|\hat{\bar{\phi}}(\cdot,\cdot)\|^2_{L^\infty(\mathbb{R}^d\times[t_0,T])} \leq \|\hat{\bar{\phi}}(\cdot,\cdot)\|^2_{L^2(\mathbb{R}^d\times[t_0,T])} \leq \frac{1}{\sigma_t^2}\log\Big(\frac{(2\pi\sigma_t^2)^{-d/2}}{\rho_{n,t}}\Big) \lesssim \sigma_t^{-2}\log n, \quad (233)$$

which suggests that the random variable $\mathbb{E}_{\bar{S}}[\ell(\phi,\boldsymbol{X}_0)-\ell(\hat{\bar{\phi}},\boldsymbol{X}_0)]$ can be bounded. Therefore, instead of directly upper bounding Eq. (232), we first use Bernstein's inequality to upper bound the following excess risk:

$$\mathbb{E}_{\boldsymbol{X}_0}\big[\mathbb{E}_{\bar{S}}[\ell(\phi,\boldsymbol{X}_0)-\ell(\hat{\bar{\phi}},\boldsymbol{X}_0)]\big] - \widehat{\mathbb{E}}_{\boldsymbol{X}_0}\big[\mathbb{E}_{\bar{S}}[\ell(\phi,\boldsymbol{X}_0)-\ell(\hat{\bar{\phi}},\boldsymbol{X}_0)]\big]$$

for a fixed $\phi \in \mathcal{NN}$. Then, we provide a high probability bound for $\ell(\widehat{\phi},\cdot)-\ell(\phi^*,\cdot)$. We conduct the following steps for our purpose:

**Step 1: Bernstein's large deviation bound for $\mathbb{E}_{\bar{S}}[\ell(\phi,\cdot)-\ell(\hat{\bar{\phi}},\cdot)]$.**

We first verify Bernstein's condition for the excess loss class

$$\mathcal{L} := \{\mathbb{E}_{\bar{S}}[\ell(\phi,\cdot)-\ell(\hat{\bar{\phi}},\cdot)] : \phi \in \mathcal{NN}\}. \quad (234)$$

By Lemma 25, for any $\phi \in \mathcal{NN}$, it holds that

$$\mathbb{E}_{\boldsymbol{X}_0}\Big[\big(\mathbb{E}_{\bar{S}}[\ell(\phi,\boldsymbol{X}_0)-\ell(\hat{\bar{\phi}},\boldsymbol{X}_0)]\big)^2\Big]$$

$$\leq \mathbb{E}_{\boldsymbol{X}_0}\Big[\mathbb{E}_{\bar{S}}\big[\big(\ell(\phi,\boldsymbol{X}_0)-\ell(\hat{\bar{\phi}},\boldsymbol{X}_0)\big)^2\big]\Big] \qquad\qquad \text{(by Jensen's inequality)}$$

$$\lesssim t_0^{-1}\log n \cdot \mathbb{E}_{\boldsymbol{X}_0}\Big[\mathbb{E}_{\bar{S}}\Big[\int_{t_0}^T \mathbb{E}_{\boldsymbol{X}_t|\boldsymbol{X}_0}\big[\|\phi(\boldsymbol{X}_t,t)-\hat{\bar{\phi}}(\boldsymbol{X}_t,t)\|_2^2\big]\mathrm{d}t\Big]\Big] \qquad \text{(by Lemma 25)}$$

$$= t_0^{-1}\log n \cdot \mathbb{E}_{\bar{S}}\Big[\int_{t_0}^T \mathbb{E}_{\boldsymbol{X}_t}\big[\|\phi(\boldsymbol{X}_t,t)-\hat{\bar{\phi}}(\boldsymbol{X}_t,t)\|_2^2\big]\mathrm{d}t\Big]$$

$$\lesssim t_0^{-1}\log n \cdot \Big(\underbrace{\int_{t_0}^T \mathbb{E}_{\boldsymbol{X}_t}\big[\|\phi(\boldsymbol{X}_t,t)-\phi^*(\boldsymbol{X}_t,t)\|_2^2\big]\mathrm{d}t}_{=\text{ Eq. (230)}} + \underbrace{\mathbb{E}_{\bar{S}}\Big[\int_{t_0}^T \mathbb{E}_{\boldsymbol{X}_t}\big[\|\hat{\bar{\phi}}(\boldsymbol{X}_t,t)-\phi^*(\boldsymbol{X}_t,t)\|_2^2\big]\mathrm{d}t\Big]}_{\lesssim \text{ Theorem 4}}\Big)$$

$$\lesssim t_0^{-1}\log n \cdot \Big(\big(\mathbb{E}_{\boldsymbol{X}_0}[\ell(\phi,\boldsymbol{X}_0)]-\mathbb{E}_{\boldsymbol{X}_0}[\ell(\phi^*,\boldsymbol{X}_0)]\big) + \alpha^d t_0^{-d/2}n^{-1}\log^{\frac{d}{2}+4}n\Big)$$

$$= t_0^{-1}\log n \cdot \mathbb{E}_{\boldsymbol{X}_0}\big[\ell(\phi,\boldsymbol{X}_0)-\ell(\phi^*,\boldsymbol{X}_0)\big] + \alpha^d t_0^{-d/2-1}n^{-1}\log^{\frac{d}{2}+5}n. \quad (235)$$

First, we show that for all $\phi \in \mathcal{NN}$, the random variable $\mathbb{E}_{\bar{S}}[\ell(\phi,\boldsymbol{X}_0)-\ell(\hat{\bar{\phi}},\boldsymbol{X}_0)]$ is bounded. By Lemma 25, for any $\phi \in \mathcal{NN}$ and any $\boldsymbol{x} \in \mathbb{R}^d$, we have

$$\big(\mathbb{E}_{\bar{S}}\big[\ell(\phi,\boldsymbol{x})-\ell(\hat{\bar{\phi}},\boldsymbol{x})\big]\big)^2 \leq \mathbb{E}_{\bar{S}}\big[\big(\ell(\phi,\boldsymbol{x})-\ell(\hat{\bar{\phi}},\boldsymbol{x})\big)^2\big] \qquad \text{(by Jensen's inequality)}$$

$$\lesssim t_0^{-1}\log n \cdot \mathbb{E}_{\bar{S}}\Big[\int_{t_0}^T \mathbb{E}_{\boldsymbol{X}_t|\boldsymbol{X}_0=\boldsymbol{x}}\big[\|\phi(\boldsymbol{X}_t,t)-\hat{\bar{\phi}}(\boldsymbol{X}_t,t)\|_2^2\big]\mathrm{d}t\Big]$$

$$\lesssim t_0^{-1}\log n \cdot \mathbb{E}_{\bar{S}}\Big[\int_{t_0}^T \mathbb{E}_{\boldsymbol{X}_t|\boldsymbol{X}_0=\boldsymbol{x}}\big[\|\phi(\boldsymbol{X}_t,t)\|_2^2 + \|\hat{\bar{\phi}}(\boldsymbol{X}_t,t)\|_2^2\big]\mathrm{d}t\Big]$$

$$\lesssim t_0^{-1}\log^2 n \int_{t_0}^T \sigma_t^{-2}\mathrm{d}t \quad \text{(by } \|\phi(\boldsymbol{X}_t,t)\|_2^2, \|\hat{\bar{\phi}}(\boldsymbol{X}_t,t)\|_2^2 \lesssim \sigma_t^{-2}\log n\text{)}$$

$$\leq t_0^{-1}\log^2 n \frac{1}{e^{t_0}-1} \leq t_0^{-2}\log^2 n.$$

Applying the Bernstein's inequality (i.e., Theorem 5) with Eq. (235), we obtain that, for any fixed $\phi \in \mathcal{NN}$ and any $\delta \in (0,1)$, with probability at least $1 - \delta$ it holds that

$$
\left| \mathbb{E}_{\boldsymbol{X}_0}\big[\mathbb{E}_{\bar{S}}[\ell(\phi, \boldsymbol{X}_0) - \ell(\hat{\bar{\phi}}, \boldsymbol{X}_0)]\big] - \widehat{\mathbb{E}}_{\boldsymbol{X}_0}\big[\mathbb{E}_{\bar{S}}[\ell(\phi, \boldsymbol{X}_0) - \ell(\hat{\bar{\phi}}, \boldsymbol{X}_0)]\big] \right|
$$

$$
\lesssim \sqrt{\frac{\mathbb{E}_{\boldsymbol{X}_0}\big[\big(\mathbb{E}_{\bar{S}}[\ell(\phi, \boldsymbol{X}_0) - \ell(\hat{\bar{\phi}}, \boldsymbol{X}_0)]\big)^2\big] \log(2/\delta)}{n}} + \frac{t_0^{-1} \log n \cdot \log(2/\delta)}{n} \quad \text{(by Theorem 5)}
$$

$$
\lesssim \sqrt{\frac{t_0^{-1} \log n \cdot \big(\mathbb{E}_{\boldsymbol{X}_0}\big[\ell(\phi, \boldsymbol{X}_0) - \ell(\phi^*, \boldsymbol{X}_0)\big] + \alpha^d t_0^{-d/2} n^{-1} \log^{\frac{d}{2}+4} n\big) \log(2/\delta)}{n}}
$$

$$
+ \frac{t_0^{-1} \log n \cdot \log(2/\delta)}{n}. \tag{236}
$$

**Step 2: $\varepsilon$-net argument.**

Here, we use the standard $\varepsilon$-net argument to derive a uniform large deviation bound based on Eq. (236).

Fix a parameter $\tau > 0$ to be specified later, and define

$$
\mathcal{NN}_{\tau/\sqrt{T}} := \{\phi_j : 1 \le j \le \mathcal{N}_{\tau/\sqrt{T}}\}
$$

be the minimal $\frac{\tau}{\sqrt{T}}$-net of $\mathcal{NN}$ w.r.t. the $L^\infty$-norm on $\mathbb{R}^d \times [t_0, T]$, where $\mathcal{N}_{\tau/\sqrt{T}} := \mathcal{N}(\frac{\tau}{\sqrt{T}}, \mathcal{NN}, \|\cdot\|_\infty)$ is the cover number of $\mathcal{NN}$.

By the union bound and Eq. (236), we obtain that for any $\delta \in (0,1)$, with probability at least $1 - \delta$, it holds that

$$
\left| \mathbb{E}_{\boldsymbol{X}_0}\big[\mathbb{E}_{\bar{S}}[\ell(\phi^\circ, \boldsymbol{X}_0) - \ell(\hat{\bar{\phi}}, \boldsymbol{X}_0)]\big] - \widehat{\mathbb{E}}_{\boldsymbol{X}_0}\big[\mathbb{E}_{\bar{S}}[\ell(\phi^\circ, \boldsymbol{X}_0) - \ell(\hat{\bar{\phi}}, \boldsymbol{X}_0)]\big] \right|
$$

$$
\lesssim \sqrt{\frac{t_0^{-1} \log n \cdot \big(\mathbb{E}_{\boldsymbol{X}_0}\big[\ell(\phi^\circ, \boldsymbol{X}_0) - \ell(\phi^*, \boldsymbol{X}_0)\big] + \alpha^d t_0^{-d/2} n^{-1} \log^{\frac{d}{2}+4} n\big) \log(2\mathcal{N}_{\tau/\sqrt{T}}/\delta)}{n}}
$$

$$
+ \frac{t_0^{-1} \log n \cdot \log(2\mathcal{N}_{\tau/\sqrt{T}}/\delta)}{n}, \tag{237}
$$

simultaneously for all $\phi^\circ \in \mathcal{NN}_{\tau, t}$.

Moreover, by Lemma 25, we have for any $\phi \in \mathcal{NN}$, there exists $\phi^\circ \in \mathcal{NN}_{\tau/\sqrt{T}}$ such that

$$
\big(\ell(\phi, \boldsymbol{x}) - \ell(\phi^\circ, \boldsymbol{x})\big)^2 \lesssim t_0^{-1} \log n \int_{t_0}^T \mathbb{E}_{\boldsymbol{X}_t | \boldsymbol{X}_0 = \boldsymbol{x}}\big[\|\phi(\boldsymbol{X}_t, t) - \phi^\circ(\boldsymbol{X}_t, t)\|_2^2\big] \mathrm{d}t \quad \text{(by Lemma 25)}
$$

$$
\le \frac{\log n}{t_0} \int_{t_0}^T \frac{\tau^2}{T} \mathrm{d}t \qquad \text{(by } \phi^\circ \in \mathcal{NN}_{\tau/\sqrt{T}})
$$

$$
= \frac{\tau^2 \log n}{t_0}, \tag{238}
$$

which implies that

$$
|\ell(\phi, \boldsymbol{x}) - \ell(\phi^\circ, \boldsymbol{x})| \lesssim \tau \sqrt{\frac{\log n}{t_0}}. \tag{239}
$$

For any $\phi \in \mathcal{NN}$, let $\phi^\circ$ be its closest element in $\mathcal{NN}_{\tau/\sqrt{T}}$. Then, for any $\delta \in (0,1)$, with probability at least $1 - \delta$, we have

$$\left| \mathbb{E}_{\boldsymbol{X}_0} \big[ \mathbb{E}_{\bar{S}}[\ell(\phi, \boldsymbol{X}_0) - \ell(\hat{\hat{\phi}}, \boldsymbol{X}_0)] \big] - \widehat{\mathbb{E}}_{\boldsymbol{X}_0} \big[ \mathbb{E}_{\bar{S}}[\ell(\phi, \boldsymbol{X}_0) - \ell(\hat{\hat{\phi}}, \boldsymbol{X}_0)] \big] \right|$$

$$\leq \left| \mathbb{E}_{\boldsymbol{X}_0} \big[ \ell(\phi, \boldsymbol{X}_0) - \ell(\phi^\circ, \boldsymbol{X}_0) \big] - \widehat{\mathbb{E}}_{\boldsymbol{X}_0} \big[ \ell(\phi, \boldsymbol{X}_0) - \ell(\phi^\circ, \boldsymbol{X}_0) \big] \right|$$

$$+ \left| \mathbb{E}_{\boldsymbol{X}_0} \big[ \mathbb{E}_{\bar{S}}[\ell(\phi^\circ, \boldsymbol{X}_0) - \ell(\hat{\hat{\phi}}, \boldsymbol{X}_0)] \big] - \widehat{\mathbb{E}}_{\boldsymbol{X}_0} \big[ \mathbb{E}_{\bar{S}}[\ell(\phi^\circ, \boldsymbol{X}_0) - \ell(\hat{\hat{\phi}}, \boldsymbol{X}_0)] \big] \right|$$

$$\lesssim \tau \sqrt{\frac{\log n}{t_0}} + \left| \mathbb{E}_{\boldsymbol{X}_0} \big[ \mathbb{E}_{\bar{S}}[\ell(\phi^\circ, \boldsymbol{X}_0) - \ell(\hat{\hat{\phi}}, \boldsymbol{X}_0)] \big] - \widehat{\mathbb{E}}_{\boldsymbol{X}_0} \big[ \mathbb{E}_{\bar{S}}[\ell(\phi^\circ, \boldsymbol{X}_0) - \ell(\hat{\hat{\phi}}, \boldsymbol{X}_0)] \big] \right|$$

(by Eq. (239))

$$\lesssim \tau \sqrt{\frac{\log n}{t_0}} + \sqrt{\frac{\log n \cdot \big( \mathbb{E}_{\boldsymbol{X}_0} \big[ \ell(\phi^\circ, \boldsymbol{X}_0) - \ell(\phi^*, \boldsymbol{X}_0) \big] + \alpha^d t_0^{-d/2} n^{-1} \log^{\frac{d}{2}+4} n \big) \log(2\mathcal{N}_{\tau/\sqrt{T}}/\delta)}{n t_0}}$$

$$+ \frac{\log n \cdot \log(2\mathcal{N}_{\tau/\sqrt{T}}/\delta)}{n t_0}$$

(by Eq. (237))

$$\lesssim \sqrt{\frac{\log n \cdot \big( \tau \sqrt{t_0^{-1} \log n} + \mathbb{E}_{\boldsymbol{X}_0} \big[ \ell(\phi, \boldsymbol{X}_0) - \ell(\phi^*, \boldsymbol{X}_0) \big] + \alpha^d t_0^{-d/2} n^{-1} \log^{\frac{d}{2}+4} n \big) \log(2\mathcal{N}_{\tau/\sqrt{T}}/\delta)}{n t_0}}$$

$$+ \tau \sqrt{\frac{\log n}{t_0}} + \frac{\log n \cdot \log(2\mathcal{N}_{\tau/\sqrt{T}}/\delta)}{n t_0}. \tag{240}$$

For the first term of Eq. (240), we have

$$\sqrt{\frac{\log n \cdot \big( \tau \sqrt{t_0^{-1} \log n} + \mathbb{E}_{\boldsymbol{X}_0} \big[ \ell(\phi, \boldsymbol{X}_0) - \ell(\phi^*, \boldsymbol{X}_0) \big] + \alpha^d t_0^{-d/2} n^{-1} \log^{\frac{d}{2}+4} n \big) \log(2\mathcal{N}_{\tau/\sqrt{T}}/\delta)}{n t_0}}$$

$$\leq \sqrt{\frac{\log n \cdot \log(2\mathcal{N}_{\tau/\sqrt{T}}/\delta)}{n t_0}} \left( \sqrt{\tau \sqrt{\frac{\log n}{t_0}}} + \sqrt{\mathbb{E}_{\boldsymbol{X}_0} \big[ \ell(\phi, \boldsymbol{X}_0) - \ell(\phi^*, \boldsymbol{X}_0) \big]} + \sqrt{\alpha^d t_0^{-d/2} n^{-1} \log^{\frac{d}{2}+4} n} \right)$$

$$= \sqrt{\frac{\log n \cdot \log(2\mathcal{N}_{\tau/\sqrt{T}}/\delta)}{n t_0}} \cdot \sqrt{\tau \sqrt{\frac{\log n}{t_0}}} + \sqrt{\frac{\log n \cdot \log(2\mathcal{N}_{\tau/\sqrt{T}}/\delta)}{n t_0}} \cdot \sqrt{\alpha^d t_0^{-d/2} n^{-1} \log^{\frac{d}{2}+4} n}$$

$$+ \sqrt{\frac{\log n \cdot \mathbb{E}_{\boldsymbol{X}_0} \big[ \ell(\phi, \boldsymbol{X}_0) - \ell(\phi^*, \boldsymbol{X}_0) \big] \log(2\mathcal{N}_{\tau/\sqrt{T}}/\delta)}{n t_0}}$$

$$\leq \frac{\log n \cdot \log(2\mathcal{N}_{\tau/\sqrt{T}}/\delta)}{n t_0} + \frac{\tau}{2} \sqrt{\frac{\log n}{t_0}} + \frac{1}{2} \alpha^d t_0^{-d/2} n^{-1} \log^{\frac{d}{2}+4} n$$

$$+ \sqrt{\frac{\log n \cdot \mathbb{E}_{\boldsymbol{X}_0} \big[ \ell(\phi, \boldsymbol{X}_0) - \ell(\phi^*, \boldsymbol{X}_0) \big] \log(2\mathcal{N}_{\tau/\sqrt{T}}/\delta)}{n t_0}}. \tag{241}$$

Substituting Eq. (241) into Eq. (240) we obtain with probability at least $1 - \delta$, it holds that

$$\left| \mathbb{E}_{\boldsymbol{X}_0} \big[ \mathbb{E}_{\bar{S}}[\ell(\phi, \boldsymbol{X}_0) - \ell(\hat{\hat{\phi}}, \boldsymbol{X}_0)] \big] - \widehat{\mathbb{E}}_{\boldsymbol{X}_0} \big[ \mathbb{E}_{\bar{S}}[\ell(\phi, \boldsymbol{X}_0) - \ell(\hat{\hat{\phi}}, \boldsymbol{X}_0)] \big] \right|$$

$$\lesssim \tau \sqrt{\frac{\log n}{t_0}} + \sqrt{\frac{\log n \cdot \mathbb{E}_{\boldsymbol{X}_0} \big[ \ell(\phi, \boldsymbol{X}_0) - \ell(\phi^*, \boldsymbol{X}_0) \big] \log(2\mathcal{N}_{\tau/\sqrt{T}}/\delta)}{n t_0}} + \frac{\log n \cdot \log(2\mathcal{N}_{\tau/\sqrt{T}}/\delta)}{n t_0}$$

$$+ \alpha^d t_0^{-d/2} n^{-1} \log^{\frac{d}{2}+4} n. \tag{242}$$

**Step 3: High probability bound for $\ell(\hat{\phi}, \cdot) - \ell(\phi^*, \cdot)$**

Recall that $\hat{\phi}$ minimizes the empirical risk over the class $\mathcal{NN}$, i.e.,

$$\hat{\phi} \in \underset{\phi \in \mathcal{NN}}{\arg\min} \, \widehat{\mathbb{E}}_{\boldsymbol{X}_0}[\ell(\phi, \boldsymbol{X}_0)] = \underset{\phi \in \mathcal{NN}}{\arg\min} \, \frac{1}{n} \sum_{i=1}^{n} \ell(\phi, \boldsymbol{x}^{(i)}).$$

Let $\tilde{\phi} \in \mathcal{NN}$ be the neural network from Theorem 1 and constructed from i.i.d. samples $\bar{S} := \{\bar{x}^{(i)}\}_{i=1}^{n}$ such that

$$\mathbb{E}_{\bar{S}}\Big[\mathbb{E}_{\boldsymbol{X}_0}\big[\ell(\tilde{\phi}, \boldsymbol{X}_0) - \ell(\phi^*, \boldsymbol{X}_0)\big]\Big] = \mathbb{E}_{\bar{S}}\Big[\int_{t_0}^{T} \mathbb{E}_{\boldsymbol{X}_t}\big[\|\tilde{\phi}(\boldsymbol{X}_t, t) - \phi^*(\boldsymbol{X}_t, t)\|_2^2\big]\mathrm{d}t\Big]$$
$$\lesssim \alpha^d t_0^{-d/2} n^{-1} \log^{\frac{d}{2}+4} n.$$

Then, with probability at least $1 - \delta$, it holds that

$$\widehat{\mathbb{E}}_{\boldsymbol{X}_0}\big[\mathbb{E}_{\bar{S}}[\ell(\widehat{\phi}, \boldsymbol{X}_0) - \ell(\hat{\tilde{\phi}}, \boldsymbol{X}_0)]\big]$$

$$\leq \widehat{\mathbb{E}}_{\boldsymbol{X}_0}\big[\mathbb{E}_{\bar{S}}[\ell(\tilde{\phi}, \boldsymbol{X}_0) - \ell(\hat{\tilde{\phi}}, \boldsymbol{X}_0)]\big] \qquad\qquad\qquad \text{(by the definition of } \widehat{\phi}\text{)}$$

$$\lesssim \mathbb{E}_{\boldsymbol{X}_0}\big[\mathbb{E}_{\bar{S}}[\ell(\tilde{\phi}, \boldsymbol{X}_0) - \ell(\hat{\tilde{\phi}}, \boldsymbol{X}_0)]\big] + \sqrt{\frac{\log n \cdot \mathbb{E}_{\boldsymbol{X}_0}\big[\ell(\tilde{\phi}, \boldsymbol{X}_0) - \ell(\phi^*, \boldsymbol{X}_0)\big] \log(2\mathcal{N}_{\tau/\sqrt{T}}/\delta)}{nt_0}}$$

$$+ \tau\sqrt{\frac{\log n}{t_0}} + \frac{\log n \cdot \log(2\mathcal{N}_{\tau/\sqrt{T}}/\delta)}{nt_0} + \alpha^d t_0^{-d/2} n^{-1} \log^{\frac{d}{2}+4} n \qquad \text{(by Eq. (242))}$$

$$= \mathbb{E}_{\bar{S}}\Big[\mathbb{E}_{\boldsymbol{X}_0}\big[\ell(\tilde{\phi}, \boldsymbol{X}_0) - \ell(\phi^*, \boldsymbol{X}_0)\big]\Big] - \mathbb{E}_{\boldsymbol{X}_0}\big[\mathbb{E}_{\bar{S}}[\ell(\hat{\tilde{\phi}}, \boldsymbol{X}_0) - \ell(\phi^*, \boldsymbol{X}_0)]\big]$$

$$+ \sqrt{\frac{\log n \cdot \mathbb{E}_{\boldsymbol{X}_0}\big[\ell(\tilde{\phi}, \boldsymbol{X}_0) - \ell(\phi^*, \boldsymbol{X}_0)\big] \log(2\mathcal{N}_{\tau/\sqrt{T}}/\delta)}{nt_0}}$$

$$+ \tau\sqrt{\frac{\log n}{t_0}} + \frac{\log n \cdot \log(2\mathcal{N}_{\tau/\sqrt{T}}/\delta)}{nt_0} + \alpha^d t_0^{-d/2} n^{-1} \log^{\frac{d}{2}+4} n$$

$$= \mathbb{E}_{\bar{S}}\Big[\mathbb{E}_{\boldsymbol{X}_0}\big[\ell(\tilde{\phi}, \boldsymbol{X}_0) - \ell(\phi^*, \boldsymbol{X}_0)\big]\Big] - \underbrace{\mathbb{E}_{\bar{S}}\Big[\int_{t_0}^{T} \mathbb{E}_{\boldsymbol{X}_t}\big[\|\hat{\tilde{\phi}}(\boldsymbol{X}_t, t) - \phi^*(\boldsymbol{X}_t, t)\|_2^2\big]\mathrm{d}t\Big]}_{\geq 0}$$

$$+ \sqrt{\frac{\log n \cdot \mathbb{E}_{\boldsymbol{X}_0}\big[\ell(\tilde{\phi}, \boldsymbol{X}_0) - \ell(\phi^*, \boldsymbol{X}_0)\big] \log(2\mathcal{N}_{\tau/\sqrt{T}}/\delta)}{nt_0}}$$

$$+ \tau\sqrt{\frac{\log n}{t_0}} + \frac{\log n \cdot \log(2\mathcal{N}_{\tau/\sqrt{T}}/\delta)}{nt_0} + \alpha^d t_0^{-d/2} n^{-1} \log^{\frac{d}{2}+4} n \qquad \text{(by Eq. (230))}$$

$$\leq \mathbb{E}_{\bar{S}}\Big[\mathbb{E}_{\boldsymbol{X}_0}\big[\ell(\tilde{\phi}, \boldsymbol{X}_0) - \ell(\phi^*, \boldsymbol{X}_0)\big]\Big] + \sqrt{\frac{\log n \cdot \mathbb{E}_{\boldsymbol{X}_0}\big[\ell(\tilde{\phi}, \boldsymbol{X}_0) - \ell(\phi^*, \boldsymbol{X}_0)\big] \log(2\mathcal{N}_{\tau/\sqrt{T}}/\delta)}{nt_0}}$$

$$+ \tau\sqrt{\frac{\log n}{t_0}} + \frac{\log n \cdot \log(2\mathcal{N}_{\tau/\sqrt{T}}/\delta)}{nt_0} + \alpha^d t_0^{-d/2} n^{-1} \log^{\frac{d}{2}+4} n.$$

For the first term and the second term of the above inequality, recall from Theorem 1 that

$$\mathbb{E}_{\bar{S}}\Big[\mathbb{E}_{\boldsymbol{X}_0}\big[\ell(\tilde{\phi}, \boldsymbol{X}_0) - \ell(\phi^*, \boldsymbol{X}_0)\big]\Big] = \mathbb{E}_{\bar{S}}\Big[\int_{t_0}^{T} \mathbb{E}_{\boldsymbol{X}_t}\big[\|\tilde{\phi}(\boldsymbol{X}_t, t) - \phi^*(\boldsymbol{X}_t, t)\|_2^2\big]\mathrm{d}t\Big] \quad \text{(by Eq. (230))}$$
$$\lesssim \alpha^d t_0^{-d/2} n^{-1} \log^{\frac{d}{2}+4} n. \qquad\qquad (243)$$

Therefore, with probability at least $1 - \delta$, it holds that

$$\widehat{\mathbb{E}}_{\boldsymbol{X}_0}\big[\mathbb{E}_{\bar{S}}[\ell(\widehat{\phi}, \boldsymbol{X}_0) - \ell(\hat{\tilde{\phi}}, \boldsymbol{X}_0)]\big] \lesssim \sqrt{\frac{\log n \cdot \log(2\mathcal{N}_{\tau/\sqrt{T}}/\delta) \cdot \alpha^d t_0^{-d/2} n^{-1} \log^{\frac{d}{2}+4} n}{nt_0}} + \tau\sqrt{\frac{\log n}{t_0}}$$

$$+ \frac{\log n \cdot \log(2\mathcal{N}_{\tau/\sqrt{T}}/\delta)}{nt_0} + \alpha^d t_0^{-d/2} n^{-1} \log^{\frac{d}{2}+4} n$$

$$\lesssim \frac{\log n \cdot \log(2\mathcal{N}_{\tau/\sqrt{T}}/\delta)}{nt_0} + \alpha^d t_0^{-d/2} n^{-1} \log^{\frac{d}{2}+4} n + \tau\sqrt{\frac{\log n}{t_0}}.$$
$$(244)$$

Noticing that

$$\mathbb{E}_{\boldsymbol{X}_0}\big[\ell(\widehat{\phi}, \boldsymbol{X}_0) - \ell(\phi^*, \boldsymbol{X}_0)\big]$$

$$= \mathbb{E}_{\boldsymbol{X}_0}\big[\ell(\widehat{\phi}, \boldsymbol{X}_0) - \mathbb{E}_{\bar{S}}[\ell(\widehat{\widetilde{\phi}}, \boldsymbol{X}_0)] + \mathbb{E}_{\bar{S}}[\ell(\widehat{\widetilde{\phi}}, \boldsymbol{X}_0)] - \ell(\phi^*, \boldsymbol{X}_0)\big]$$

$$= \mathbb{E}_{\boldsymbol{X}_0}\big[\mathbb{E}_{\bar{S}}[\ell(\widehat{\phi}, \boldsymbol{X}_0) - \ell(\widehat{\widetilde{\phi}}, \boldsymbol{X}_0)]\big] + \mathbb{E}_{\boldsymbol{X}_0}\big[\mathbb{E}_{\bar{S}}[\ell(\widehat{\widetilde{\phi}}, \boldsymbol{X}_0)] - \ell(\phi^*, \boldsymbol{X}_0)\big] \qquad (245)$$

For the first term of the above inequality, according to Eq. (242), with probability at least $1 - \delta$, it holds that

$$\mathbb{E}_{\boldsymbol{X}_0}\big[\mathbb{E}_{\bar{S}}[\ell(\widehat{\phi}, \boldsymbol{X}_0) - \ell(\widehat{\widetilde{\phi}}, \boldsymbol{X}_0)]\big]$$

$$\lesssim \underbrace{\widehat{\mathbb{E}}_{\boldsymbol{X}_0}\big[\mathbb{E}_{\bar{S}}[\ell(\widehat{\phi}, \boldsymbol{X}_0) - \ell(\widehat{\widetilde{\phi}}, \boldsymbol{X}_0)]\big]}_{\lesssim \text{ Eq. (244)}} + \sqrt{\frac{\log n \cdot \mathbb{E}_{\boldsymbol{X}_0}\big[\ell(\widehat{\phi}, \boldsymbol{X}_0) - \ell(\phi^*, \boldsymbol{X}_0)\big] \log(2\mathcal{N}_{\tau/\sqrt{T}}/\delta)}{nt_0}}$$

$$+ \tau\sqrt{\frac{\log n}{t_0}} + \frac{\log n \cdot \log(2\mathcal{N}_{\tau/\sqrt{T}}/\delta)}{nt_0} + \alpha^d t_0^{-d/2} n^{-1} \log^{\frac{d}{2}+4} n$$

$$\lesssim \sqrt{\frac{\log n \cdot \mathbb{E}_{\boldsymbol{X}_0}\big[\ell(\widehat{\phi}, \boldsymbol{X}_0) - \ell(\phi^*, \boldsymbol{X}_0)\big] \log(2\mathcal{N}_{\tau/\sqrt{T}}/\delta)}{nt_0}} + \tau\sqrt{\frac{\log n}{t_0}}$$

$$+ \frac{\log n \cdot \log(2\mathcal{N}_{\tau/\sqrt{T}}/\delta)}{nt_0} + \alpha^d t_0^{-d/2} n^{-1} \log^{\frac{d}{2}+4} n. \qquad (246)$$

For the second term of Eq. (245), recall from Theorem 4 that

$$\mathbb{E}_{\boldsymbol{X}_0}\big[\mathbb{E}_{\bar{S}}[\ell(\widehat{\widetilde{\phi}}, \boldsymbol{X}_0)] - \ell(\phi^*, \boldsymbol{X}_0)\big] = \mathbb{E}_{\bar{S}}\Big[\int_{t_0}^{T} \mathbb{E}_{\boldsymbol{X}_t}\big[\|\widehat{\widetilde{\phi}}(\boldsymbol{X}_t, t) - \phi^*(\boldsymbol{X}_t, t)\|_2^2\big]\mathrm{d}t\Big] \quad \text{(by Eq. (230))}$$

$$\lesssim \alpha^d t_0^{-d/2} n^{-1} \log^{\frac{d}{2}+4} n. \qquad (247)$$

Combining Eq. (245) to (247), we obtain that with probability at least $1 - \delta$, it holds that

$$\mathbb{E}_{\boldsymbol{X}_0}\big[\ell(\widehat{\phi}, \boldsymbol{X}_0) - \ell(\phi^*, \boldsymbol{X}_0)\big] \lesssim \sqrt{\frac{\log n \cdot \mathbb{E}_{\boldsymbol{X}_0}\big[\ell(\widehat{\phi}, \boldsymbol{X}_0) - \ell(\phi^*, \boldsymbol{X}_0)\big] \log(2\mathcal{N}_{\tau/\sqrt{T}}/\delta)}{nt_0}} + \tau\sqrt{\frac{\log n}{t_0}}$$

$$+ \frac{\log n \cdot \log(2\mathcal{N}_{\tau/\sqrt{T}}/\delta)}{nt_0} + \alpha^d t_0^{-d/2} n^{-1} \log^{\frac{d}{2}+4} n.$$

Recall the fact that the inequality $x \leq 2a\sqrt{x} + b$ implies that $x \leq 4a^2 + 2b$ for non-negative $a, b$ and $x$ [28]. Therefore, with probability at least $1 - \delta$, it holds that

$$\mathbb{E}_{\boldsymbol{X}_0}\big[\ell(\widehat{\phi}, \boldsymbol{X}_0) - \ell(\phi^*, \boldsymbol{X}_0)\big] \lesssim \tau\sqrt{\frac{\log n}{t_0}} + \frac{\log n \cdot \log(2\mathcal{N}_{\tau/\sqrt{T}}/\delta)}{nt_0} + \alpha^d t_0^{-d/2} n^{-1} \log^{\frac{d}{2}+4} n. \quad (248)$$

**Step 4: Covering number evaluation for $\mathcal{N}_{\tau/\sqrt{T}}$.**

It is shown in Theorem 6 and 8 of [52] that the Pseudo-dimension of ReLU networks has two types of upper bounds: $\mathcal{O}(WL\log W)$ and $\mathcal{O}(WU)$, where $W$, $L$, and $U$ are the numbers of parameters, layers, and neurons, respectively. If we let $N$ denote the maximum width of the network, then $W = \mathcal{O}(N^2L)$ and $U = \mathcal{O}(NL)$, implying that

$$WL\log W = \mathcal{O}\big(N^2L \cdot L\log(N^2L)\big) = \mathcal{O}\big(N^2L^2\log(NL)\big)$$

$$WU = \mathcal{O}\big(N^2L \cdot NL\big) = \mathcal{O}(N^3L^2).$$

Recall that

$$\mathcal{NN} \equiv \big\{\phi \in \mathcal{NN}(\text{width} \leq \mathcal{O}(n^{\frac{3}{2k}}\log_2 n); \text{depth} \leq \mathcal{O}(\log^2 n)) \mid \|\phi(\cdot, t)\|_\infty \lesssim \sigma_t^{-1}\sqrt{\log n}\big\}.$$

With $N \leq \mathcal{O}(n^{\frac{3}{2k}} \log_2 n), L \leq \mathcal{O}(\log^2 n)$, the pseudo-dimension of $\mathcal{NN}$ satisfies that

$$
\begin{aligned}
\mathrm{Pdim}(\mathcal{NN}) &\leq \mathcal{O}(N^2 L^2 \log(NL)) \wedge \mathcal{O}(N^3 L^2) \\
&\leq \mathcal{O}\big(n^{\frac{3}{k}} \log^4 n \cdot \log_2^2 n \cdot \log(n^{\frac{3}{2k}} \log^3 n)\big) \\
&\leq \mathcal{O}\big(n^{\frac{3}{k}} \log^7 n\big).
\end{aligned}
\tag{249}
$$

Futhermore, by Theorem 6 ([53, Theorem 12.2]), the covering number $\mathcal{N}_{\tau/\sqrt{T}}$ satisfies that

$$
\begin{aligned}
\log(\mathcal{N}_{\tau/\sqrt{T}}) &\lesssim \log\Big( \sum_{l=1}^{\mathrm{Pdim}(\mathcal{NN})} \binom{d}{l} \Big(\frac{\sigma_{t_0}^{-1}\sqrt{T \log n}}{\tau}\Big)^l \Big) \\
&\lesssim \mathrm{Pdim}(\mathcal{NN}) \log\Big(\frac{\sqrt{T \log n}}{\sigma_{t_0}\tau}\Big) \\
&\lesssim n^{\frac{3}{k}} \log^7 n \cdot \log\big(\frac{n}{\sigma_{t_0}\tau}\big) && (\text{by } T = n^{\mathcal{O}(1)}) \\
&\lesssim n^{\frac{3}{k}} \log^7 n \cdot \Big(\log n + \log \frac{1}{\sqrt{t_0}} + \log \frac{1}{\tau}\Big) && (\text{by } \tfrac{1}{2}\alpha^2 n^{-2/d} \log n \leq t_0 \leq 1/2) \\
&\lesssim n^{\frac{3}{k}} \log^7 n \cdot \Big(\log n + \log \frac{1}{\tau}\Big)
\end{aligned}
\tag{250}
$$

Substituting Eq. (250) into Eq. (248), we obtain

$$
\begin{aligned}
&\mathbb{E}_{\boldsymbol{X}_0}\big[\ell(\widehat{\phi}, \boldsymbol{X}_0) - \ell(\phi^*, \boldsymbol{X}_0)\big] \\
&\lesssim \frac{\log n}{nt_0}\Big(\log(\mathcal{N}_{\tau/\sqrt{T}}) + \log \frac{2}{\delta}\Big) + \tau\sqrt{\frac{\log n}{t_0}} + \alpha^d t_0^{-d/2} n^{-1} \log^{\frac{d}{2}+4} n \\
&\lesssim \frac{\log n}{nt_0}\Big(n^{\frac{3}{k}} \log^7 n \cdot \Big(\log n + \log \frac{1}{\tau}\Big) + \log \frac{2}{\delta}\Big) + \tau\sqrt{\frac{\log n}{t_0}} + \alpha^d t_0^{-d/2} n^{-1} \log^{\frac{d}{2}+4} n.
\end{aligned}
$$

**Step 5: Determining $\tau$ and $k$**

Choosing $\tau = n^{-1}$, we obtain

$$
\mathbb{E}_{\boldsymbol{X}_0}\big[\ell(\widehat{\phi}, \boldsymbol{X}_0) - \ell(\phi^*, \boldsymbol{X}_0)\big] \lesssim t_0^{-1} n^{\frac{3}{k}-1} \log^9 n + t_0^{-1} n^{-1} \log n \cdot \log \frac{2}{\delta} + \alpha^d t_0^{-d/2} n^{-1} \log^{\frac{d}{2}+4} n.
$$

Noticing that when $d \geq 3$, if $k \geq \frac{6 \log n}{(d-2)\log(t_0^{-1})}$, we have $t_0^{-1} n^{\frac{3}{k}} \leq t_0^{-d/2}$, which ensures that the last term will dominate the first term in the above inequality. Moreover, recall that to ensure Theorem 1, we require $d/2 \leq k \lesssim \frac{\log n}{\log \log n}$ for $d \lesssim \sqrt{\log n}$.

Therefore, we obtain that for $3 \leq d \lesssim \sqrt{\log n}$, fix $k \in \mathbb{N}_+$ with $d/2 \vee \frac{6 \log n}{(d-2)\log(t_0^{-1})} \leq k \lesssim \frac{\log n}{\log \log n}$, with probability at least $1 - \delta$, it holds that

$$
\mathbb{E}_{\boldsymbol{X}_0}\big[\ell(\widehat{\phi}, \boldsymbol{X}_0) - \ell(\phi^*, \boldsymbol{X}_0)\big] \lesssim \alpha^d t_0^{-d/2} n^{-1} \log^{\frac{d}{2}+4} n + t_0^{-1} n^{-1} \log n \cdot \log \frac{2}{\delta}.
$$

$\square$

## E.2 Distribution estimation by deep neural networks

**Theorem 3** (Distribution Estimation Error of $P_{t_0}$). *Assume that the conditions of Theorem 2 hold. Then, for any $\delta \in (0, 1)$, with probability at least $1 - \delta$, it holds that*

$$\mathbb{E}_{\{\boldsymbol{x}^{(i)}\}_{i=1}^n}\big[\mathsf{TV}(P_{t_0}, \widehat{P}_{t_0}^{\gamma_d})\big] \lesssim \alpha^{d/2} t_0^{-d/4} n^{-1/2} \log^{\frac{d}{4}+2} n + t_0^{-1/2} n^{-1/2} \log^{1/2} n \cdot \sqrt{\log(2/\delta)}.$$

*Proof.* The distribution estimation error in the TV distance at time $t_0$ into the following two terms: the score estimation error, and the Gaussian induced error.

$$\mathbb{E}_{\{\boldsymbol{x}^{(i)}\}}\big[\mathsf{TV}(P_{t_0}, \widehat{P}_{t_0}^{\gamma_d})]\big] \leq \mathbb{E}_{\{\boldsymbol{x}^{(i)}\}}\big[\mathsf{TV}(P_{t_0}, \widehat{P}_{t_0})\big] + \mathbb{E}_{\{\boldsymbol{x}^{(i)}\}}\big[\mathsf{TV}(\widehat{P}_{t_0}, \widehat{P}_{t_0}^{\gamma_d})\big].$$

1) Bounding the score estimation error. By Pinsker's inequality (c.f. Lemma 31), TV distance is upper bounded by the KL divergence, i.e., $\mathsf{TV}(P_{t_0}, \widehat{P}_{t_0}) \leq \sqrt{\frac{1}{2}\mathrm{KL}(P_{t_0} \| \widehat{P}_{t_0})}$. Furthermore, by Girsanov's theorem (c.f. Theorem 7 and Corollary 6 [16, 54]), we have

$$
\begin{aligned}
&\mathbb{E}_{\{\boldsymbol{x}^{(i)}\}}\big[\mathsf{TV}(P_{t_0}, \widehat{P}_{t_0})\big] \\
&\leq \sqrt{\frac{1}{2}\mathbb{E}_{\{\boldsymbol{x}^{(i)}\}}\big[\mathrm{KL}(P_{t_0} \| \widehat{P}_{t_0})\big]} \qquad\qquad \text{(by Pinsker's inequality and Jensen's inequality)} \\
&\leq \sqrt{\mathbb{E}_{\{\boldsymbol{x}^{(i)}\}}\Big[\int_{t=t_0}^{T} \mathbb{E}_{\boldsymbol{x}_t \sim P_t}\big[\|\nabla \log p_t(\boldsymbol{x}_t) - \phi_{\text{score}}(\boldsymbol{x}_t, t)\|_2^2\big]\mathrm{d}t\Big]} \qquad \text{(by Corollary 6)} \\
&\lesssim \sqrt{\alpha^d t_0^{-d/2} n^{-1} \log^{\frac{d}{2}+4} n + t_0^{-1} n^{-1} \log n \cdot \log\frac{2}{\delta}} \qquad\qquad \text{(by Theorem 2)} \\
&\leq \alpha^{d/2} t_0^{-d/4} n^{-1/2} \log^{\frac{d}{4}+2} n + t_0^{-1/2} n^{-1/2} \log^{1/2} n \cdot \sqrt{\log(2/\delta)}.
\end{aligned}
$$

2) Bounding the Gaussian induced error. The error from the last term is induced by starting from the standard Gaussian $\gamma_d$ instead of the marginal distribution $P_T$. The convergence of the OU process [55, 22] gives that

$$\mathsf{TV}(\widehat{P}_{t_0}, \widehat{P}_{t_0}^{\gamma_d}) \leq \sqrt{\frac{1}{2}\mathrm{KL}(\widehat{P}_{t_0} \| \widehat{P}_{t_0}^{\gamma_d})} \lesssim e^{-T}. \tag{251}$$

Given that $T = n^{\mathcal{O}(1)}$, this term decays exponentially to zero. Therefore, the overall bound is dominated by the first term, which completes the proof.

$\square$

### E.3  Useful lemmas for estimation

**Lemma 24** ([56, Proposition 2.2]). *Suppose that $X$ is a sub-exponential random variable with parameters $\nu, b$, that is*

$$\mathbb{E}\big[\exp(\lambda(X - \mathbb{E}[X]))\big] \leq \exp(\nu^2\lambda^2/2), \quad \text{for all } \lambda \text{ such that } |\lambda| \leq 1/b.$$

*Then, for any $t \geq 0$, it holds that*

$$\Pr[X \geq \mathbb{E}[X] + t] \leq \exp\Big(-\frac{1}{2}\Big(\frac{t^2}{\nu^2} \wedge \frac{t}{b}\Big)\Big).$$

**Remark 5** ([56, Eamples 2.5]). *The chi-squared random variable with $d$ degrees of freedom is sub-exponential with parameters $(2\sqrt{d}, 4)$. This yields that, if $X \sim \chi^2(d)$, then*

$$\Pr[X \geq d + t] \leq \exp\Big(-\frac{1}{8}\Big(\frac{t^2}{d} \wedge t\Big)\Big) \quad \text{for all } t \geq 0.$$

**Lemma 25.** *Let $\phi : \mathbb{R}^d \times [t_0, T] \to \mathbb{R}^d$ and $\phi' : \mathbb{R}^d \times [t_0, T] \to \mathbb{R}^d$ be any Borel functions such that*

$$\|\phi(\cdot, t)\|_{L^\infty(\mathbb{R}^d)} \lesssim \sigma_t^{-1}\sqrt{\log n} \quad \text{and} \quad \|\phi'(\cdot, t)\|_{L^\infty(\mathbb{R}^d)} \lesssim \sigma_t^{-1}\sqrt{\log n}.$$

*Then, for all $\boldsymbol{x} \in \mathbb{R}^d$, it holds that*

$$\big(\ell(\phi, \boldsymbol{x}) - \ell(\phi', \boldsymbol{x})\big)^2 \lesssim t_0^{-1}d\log n \int_{t_0}^T \mathbb{E}_{\boldsymbol{X}_t|\boldsymbol{X}_0=\boldsymbol{x}}\Big[\big\|\phi(\boldsymbol{X}_t, t) - \phi'(\boldsymbol{X}_t, t)\big\|_2^2\Big]\mathrm{d}t.$$

*Proof.*

$$\ell(\phi, \boldsymbol{x}) - \ell(\phi', \boldsymbol{x})$$

$$= \int_{t_0}^T \mathbb{E}_{\boldsymbol{X}_t|\boldsymbol{X}_0=\boldsymbol{x}}\Big[\Big\|\phi(\boldsymbol{X}_t, t) + \frac{\boldsymbol{X}_t - m_t\boldsymbol{x}}{\sigma_t^2}\Big\|_2^2 - \Big\|\phi'(\boldsymbol{X}_t, t) + \frac{\boldsymbol{X}_t - m_t\boldsymbol{x}}{\sigma_t^2}\Big\|_2^2\Big]\mathrm{d}t$$

$$= \int_{t_0}^T \mathbb{E}_{\boldsymbol{X}_t|\boldsymbol{X}_0=\boldsymbol{x}}\Big[\|\phi(\boldsymbol{X}_t, t)\|_2^2 + 2\Big\langle\phi(\boldsymbol{X}_t, t), \frac{\boldsymbol{X}_t - m_t\boldsymbol{x}}{\sigma_t^2}\Big\rangle - \|\phi'(\boldsymbol{X}_t, t)\|_2^2 - 2\Big\langle\phi'(\boldsymbol{X}_t, t), \frac{\boldsymbol{X}_t - m_t\boldsymbol{x}}{\sigma_t^2}\Big\rangle\Big]\mathrm{d}t$$

$$= \int_{t_0}^T \mathbb{E}_{\boldsymbol{X}_t|\boldsymbol{X}_0=\boldsymbol{x}}\Big[\big(\phi(\boldsymbol{X}_t, t) - \phi'(\boldsymbol{X}_t, t)\big)^\top\Big(\phi(\boldsymbol{X}_t, t) + \phi'(\boldsymbol{X}_t, t) + \frac{2(\boldsymbol{X}_t - m_t\boldsymbol{x})}{\sigma_t^2}\Big)\Big]\mathrm{d}t$$

Applying the Cauchy-Schwarz inequality, we obtain that

$$|\ell(\phi, \boldsymbol{x}) - \ell(\phi', \boldsymbol{x})|$$

$$\leq \int_{t_0}^T \mathbb{E}_{\boldsymbol{X}_t|\boldsymbol{X}_0=\boldsymbol{x}}\Big[\big\|\phi(\boldsymbol{X}_t, t) - \phi'(\boldsymbol{X}_t, t)\big\|\Big\|\phi(\boldsymbol{X}_t, t) + \phi'(\boldsymbol{X}_t, t) + \frac{2(\boldsymbol{X}_t - m_t\boldsymbol{x})}{\sigma_t^2}\Big\|\Big]\mathrm{d}t$$

$$\leq \sqrt{\int_{t_0}^T \mathbb{E}_{\boldsymbol{X}_t|\boldsymbol{X}_0=\boldsymbol{x}}\Big[\big\|\phi(\boldsymbol{X}_t, t) - \phi'(\boldsymbol{X}_t, t)\big\|_2^2\Big]\mathrm{d}t} \cdot \sqrt{\int_{t_0}^T \mathbb{E}_{\boldsymbol{X}_t|\boldsymbol{X}_0=\boldsymbol{x}}\Big[\Big\|\phi(\boldsymbol{X}_t, t) + \phi'(\boldsymbol{X}_t, t) + \frac{2(\boldsymbol{X}_t - m_t\boldsymbol{x})}{\sigma_t^2}\Big\|_2^2\Big]\mathrm{d}t}$$

Noticing that for the OU process, we have $\boldsymbol{X}_t = m_t\boldsymbol{X}_0 + \sigma_t\boldsymbol{Z}$, where $\boldsymbol{Z} \sim \mathcal{N}(\boldsymbol{0}, \boldsymbol{I}_d)$. Then, we have

$$\mathbb{E}_{\boldsymbol{X}_t|\boldsymbol{X}_0=\boldsymbol{x}}\Big[\Big\|\frac{2(\boldsymbol{X}_t - m_t\boldsymbol{x})}{\sigma_t^2}\Big\|_2^2\Big] = 4\mathbb{E}_{\boldsymbol{X}_t|\boldsymbol{X}_0=\boldsymbol{x}}\Big[\Big\|\frac{\sigma_t\boldsymbol{Z}}{\sigma_t^2}\Big\|_2^2\Big] = \frac{4}{\sigma_t^2}\mathbb{E}[\|\boldsymbol{Z}\|_2^2].$$

Since $\boldsymbol{Z} = (Z_1, \ldots, Z_d)$ and $Z_i \sim \mathcal{N}(0, 1), \forall i = 1, \ldots, n$, we have $Z_i^2 \sim \chi^2(1)$ and $\mathbb{E}[Z_i^2] = 1$. Thus,

$$\mathbb{E}[\|\boldsymbol{Z}\|_2^2] = \mathbb{E}\Big[\sum_{i=1}^d Z_i^2\Big] = \sum_{i=1}^d \mathbb{E}[Z_i^2] = d,$$

which gives that

$$\mathbb{E}_{\boldsymbol{X}_t|\boldsymbol{X}_0=\boldsymbol{x}}\Big[\Big\|\frac{2(\boldsymbol{X}_t - m_t\boldsymbol{X}_0)}{\sigma_t^2}\Big\|_2^2\Big] \leq \frac{4d}{\sigma_t^2}.$$

Then it holds that

$$\sqrt{\int_{t_0}^{T} \mathbb{E}_{\boldsymbol{X}_t|\boldsymbol{X}_0=\boldsymbol{x}}\Big[\Big\|\phi(\boldsymbol{X}_t,t)+\phi'(\boldsymbol{X}_t,t)+\frac{2(\boldsymbol{X}_t-m_t\boldsymbol{x})}{\sigma_t^2}\Big\|_2^2\Big]\mathrm{d}t}$$

$$\leq \sqrt{3\int_{t_0}^{T}\Big(\mathbb{E}_{\boldsymbol{X}_t|\boldsymbol{X}_0=\boldsymbol{x}}\big[\|\phi(\boldsymbol{X}_t,t)\|_2^2+\|\phi'(\boldsymbol{X}_t,t)\|_2^2\big]+\mathbb{E}_{\boldsymbol{X}_t|\boldsymbol{X}_0=\boldsymbol{x}}\Big[\Big\|\frac{2(\boldsymbol{X}_t-m_t\boldsymbol{X}_0)}{\sigma_t^2}\Big\|_2^2\Big]\Big)\mathrm{d}t}$$

$$\lesssim \sqrt{\int_{t_0}^{T}\big(6\sigma_t^{-2}\log n+12d\sigma_t^{-2}\big)\mathrm{d}t} \qquad\qquad \text{(by } \|\phi(\boldsymbol{X}_t,t)\|_{L^\infty(\mathbb{R}^d\times[t_0,T])}\lesssim\sigma_t^{-1}\sqrt{\log n}\text{)}$$

$$= \sqrt{6(d\log n+2d)}\sqrt{\int_{t_0}^{T}\frac{1}{1-\exp(-2t)}\mathrm{d}t}$$

$$= \sqrt{3(d\log n+2d)\log\Big(\frac{(e^T-1)(e^{t_0}+1)}{(e^T+1)(e^{t_0}-1)}\Big)}$$

$$\leq \sqrt{3(d\log n+2d)\log\Big(\frac{e^{t_0}+1}{e^{t_0}-1}\Big)} \leq \sqrt{6(d\log n+2d)\frac{1}{e^{t_0}-1}} \lesssim \sqrt{t_0^{-1}d\log n}.$$

Thus, we have

$$\big(\ell(\phi,\boldsymbol{x})-\ell(\phi',\boldsymbol{x})\big)^2$$

$$\leq \int_{t_0}^{T}\mathbb{E}_{\boldsymbol{X}_t|\boldsymbol{X}_0=\boldsymbol{x}}\Big[\big\|\phi(\boldsymbol{X}_t,t)-\phi'(\boldsymbol{X}_t,t)\big\|_2^2\Big]\mathrm{d}t \cdot \int_{t_0}^{T}\mathbb{E}_{\boldsymbol{X}_t|\boldsymbol{X}_0=\boldsymbol{x}}\Big[\Big\|\phi(\boldsymbol{X}_t,t)+\phi'(\boldsymbol{X}_t,t)+\frac{2(\boldsymbol{X}_t-m_t\boldsymbol{x})}{\sigma_t^2}\Big\|_2^2\Big]\mathrm{d}t$$

$$\lesssim t_0^{-1}d\log n\int_{t_0}^{T}\mathbb{E}_{\boldsymbol{X}_t|\boldsymbol{X}_0=\boldsymbol{x}}\Big[\big\|\phi(\boldsymbol{X}_t,t)-\phi'(\boldsymbol{X}_t,t)\big\|_2^2\Big]\mathrm{d}t.$$

$\square$

### E.3.1 Bernstein's inequality

**Theorem 5** (Bernstein's inequality for bounded distributions [40]). *Let $X_1,\ldots,X_n$ be independent random variables such that $|X_i|\leq K$ for all $i\in[n]$. Then, for every $t\geq 0$, we have*

$$\Pr\Big[\Big|\sum_{i=1}^{n}(X_i-\mathbb{E}[X_i])\Big|\geq t\Big]\leq 2\exp\Big(-\frac{t^2/2}{\sum_{i=1}^{n}\mathbb{E}[X_i^2]+Kt/3}\Big).$$

*In other words, with probability at least $1-\delta$, it holds that*

$$\Big|\frac{1}{n}\sum_{i=1}^{n}(X_i-\mathbb{E}[X])\Big|\leq \frac{t}{n}\leq \frac{2K\log(2/\delta)}{3n}+\sqrt{\frac{\frac{2}{n}\sum_{i=1}^{n}\mathbb{E}[X_i^2]\log(2/\delta)}{n}}.$$

### E.3.2 Pseudo-dimension and covering number

**Definition 5** (Pseudo-Dimension [52]). *Let $\mathcal{F}$ be a class of functions from $\mathcal{X}$ to $\mathbb{R}$. The pseudo-dimension of $\mathcal{F}$, written $\mathrm{Pdim}(\mathcal{F})$, is the largest integer $m$ for which there exists $(x_1,\ldots,x_m,y_1,\ldots,y_m)\in\mathcal{X}^m\times\mathbb{R}^m$ such that for any $(b_1,\ldots,b_m)\in\{0,1\}^m$ there exists $f\in\mathcal{F}$ such that*

$$\forall i: f(x_i)>y_i \iff b_i=1.$$

**Theorem 6** (Covering Number Evaluation by Pseudo-Dimension [53, Theorem 12.2]). *Let $\mathcal{F}$ be a set of real functions from a domain $\mathcal{X}\subseteq\mathbb{R}^d$ to the bounded interval $[0,B]$. Let $\epsilon>0$ and suppose that the pseudo-dimension of $\mathcal{F}$ is $\mathrm{Pdim}$. Then,*

$$\mathcal{N}(\epsilon,\mathcal{F},\|\cdot\|_\infty)\leq \sum_{k=1}^{\mathrm{Pdim}}\binom{d}{k}\Big(\frac{B}{\epsilon}\Big)^k,$$

*which is less than $(edB/\epsilon\mathrm{Pdim})^{\mathrm{Pdim}}$ for $d\geq\mathrm{Pdim}$.*

# F    Convergence of SGMs

## F.1    Girsanov's theorem

**Theorem 7** (Girsanov's Theorem [54]). *Given a filtered probability space $(\Omega, \mathcal{F}, (\mathcal{F}_t)_{t\geq 0}, Q)$ and a Q-Brownian motion $(B_t)_{t\in[0,T]}$ under the probability measure Q. Suppose that $(b(\cdot, t))_{t\in[0,T]}$ is an adapted process w.r.t. filtration $(\mathcal{F}_t)_{t\in[0,T]}$ generated by B such that the following Novikov's condition holds:*

$$\mathbb{E}_Q\left[\exp\left(\frac{1}{2}\int_0^T \|b(B_t, t)\|_2^2 \mathrm{d}t\right)\right] < \infty, \tag{252}$$

*Consider the process:*

$$\mathcal{L}_t := \int_0^t b(B_s, s)\mathrm{d}B_s.$$

*Then, $\mathcal{L}$ is a square-integrable Q-martingale. Moreover, if we define the Doléans-Dade exponential:*

$$\mathcal{E}(\mathcal{L})_t := \exp\left(\mathcal{L}_t - \frac{1}{2}\langle \mathcal{L}, \mathcal{L}\rangle_t\right) = \exp\left(\int_0^t b(B_s, s)\mathrm{d}B_s - \frac{1}{2}\int_0^t \|b(B_s, s)\|_2^2 \mathrm{d}s\right) \quad (0 \leq t \leq T),$$

*and suppose that $\mathbb{E}_Q[\mathcal{E}(\mathcal{L})_T] = 1$, then $\mathcal{E}(\mathcal{L})$ is a Q-martingale and the process*

$$B_t' := B_t - \int_0^t b(B_s, s)\mathrm{d}s \quad (0 \leq t \leq T)$$

*is a P-Brownian motion under the new measure $P = \mathcal{E}(\mathcal{L})_T Q$.*

The following theorem, Corollary 5, provides an upper bound on the score estimation error in terms of the score matching loss, which restates the results from [16]. In our analysis, we apply Girsanov's Theorem (Theorem 7) to continuous SDE processes, whereas [16] utilizes discretized SDE processes.

**Corollary 5** (Girsanov's Theorem for SDE Processes [16]). *Let $P_0$ be any probability distribution, and let $X = (X_t)_{t\in[0,T]}, X' = (X_t')_{t\in[0,T]}$ be two different processes satisfying*

$$\mathrm{d}X_t = f(X_t, t)\mathrm{d}t + g(t)\mathrm{d}B_t \quad (0 \leq t \leq T), \quad X_0 \sim P_0, \tag{253}$$

$$\mathrm{d}X_t' = f'(X_t', t)\mathrm{d}t + g(t)\mathrm{d}B_t' \quad (0 \leq t \leq T), \quad X_0' \sim P_0. \tag{254}$$

*Denote the distributions of $X_t$ and $X_t'$ by $P_t, P_t'$ and the path measures of $X, X'$ by $\mathbb{P}, \mathbb{P}'$, respectively.*

*Suppose that the following Novikov's condition holds:*

$$\mathbb{E}_{\mathbb{P}}\left[\exp\left(\frac{1}{2}\int_0^T \int_x \frac{1}{g^2(t)}\|f(x, t) - f'(x, t)\|_2^2 \mathrm{d}x\mathrm{d}t\right)\right] < \infty. \tag{255}$$

*Then, the Radon-Nikodym's derivative of $\mathbb{P}$ w.r.t. $\mathbb{P}'$ is*

$$\frac{\mathrm{d}\mathbb{P}}{\mathrm{d}\mathbb{P}'}(X) = \exp\left(\int_0^T \frac{1}{g(t)}(f(X_t, t) - f'(X_t, t))\mathrm{d}B_t - \frac{1}{2}\int_0^T \frac{1}{g^2(t)}\|f(X_t, t) - f'(X_t, t)\|_2^2 \mathrm{d}t\right),$$

*and therefore we have that*

$$\mathrm{KL}(P_T \| P_T') \leq \mathrm{KL}(\mathbb{P}\|\mathbb{P}') = \mathbb{E}_{\mathbb{P}}\left[\log\left(\frac{\mathrm{d}\mathbb{P}}{\mathrm{d}\mathbb{P}'}\right)\right]$$

$$= \int_0^T \frac{1}{2}\int_x \frac{1}{g^2(t)}\|f(x, t) - f'(x, t)\|_2^2 p_t(x)\mathrm{d}x\mathrm{d}t.$$

*Proof.* Let $X$ be the process of Eq. (253), we define

$$b(B_t, t) := \frac{1}{g(t)}(f'(X_t, t) - f(X_t, t)), \quad \forall 0 \leq t \leq T.$$

Then, we have

$$\mathbb{E}_{\mathbb{P}}\left[\int_0^T \|b(B_t,t)\|_2^2 \mathrm{d}t\right] = \int_0^T \frac{1}{g^2(t)} \mathbb{E}_{X_t \sim P_t}\left[\|f'(X_t,t) - f(X_t,t)\|_2^2\right]\mathrm{d}t < \infty. \quad \text{(by Eq. (255))}$$

Define $\mathcal{L}_t := \int_0^t b(B_s,s)\mathrm{d}B_s$ and $\mathcal{E}(\mathcal{L})_t$ as in Theorem 7, then we have $\mathcal{L}$ is a $\mathbb{P}$-martingale

By Theorem 7, we have that under $\mathbb{P}' = \mathcal{E}(\mathcal{L})_T \mathbb{P}$, there exists a Brownian motion $(B_t')_{t \in [0,T]}$:

$$B_t' = B_t - \int_0^t b(B_s,s)\mathrm{d}s = B_t - \int_0^t \frac{1}{g(s)}\big(f'(X_s,s) - f(X_s,s)\big)\mathrm{d}s,$$

which is a $\mathbb{P}'$-martingale and we have

$$\mathrm{d}B_t' = \mathrm{d}B_t - b(B_t,t)\mathrm{d}t = \mathrm{d}B_t - \frac{1}{g(t)}(f'(X_t,t) - f(X_t,t))\mathrm{d}t. \tag{256}$$

Recall that under $\mathbb{P}$, we have

$$\mathrm{d}X_t = f(X_t,t) + g(t)\mathrm{d}B_t \quad (0 \le t \le T), \quad X_0 \sim P_0$$

Then, by Eq. (256), we have

$$\begin{aligned}
\mathrm{d}X_t &= f(X_t,t)\mathrm{d}t + g(t)\mathrm{d}B_t \\
&= f(X_t,t)\mathrm{d}t + g(t)\mathrm{d}B_t' + f'(X_t,t)\mathrm{d}t - f(X_t,t)\mathrm{d}t \quad \text{(by Eq. (256))} \\
&= f'(X_t,t)\mathrm{d}t + g(t)\mathrm{d}B_t'
\end{aligned}$$

under the measure $\mathbb{P}'$.

$$\begin{aligned}
\mathrm{KL}(\mathbb{P}\|\mathbb{P}') &= \mathbb{E}_{\mathbb{P}}\left[\log\left(\frac{\mathrm{d}\mathbb{P}}{\mathrm{d}\mathbb{P}'}\right)\right] \\
&= \mathbb{E}_{\mathbb{P}}\left[\log(\mathcal{E}(\mathcal{L})_T^{-1})\right] \quad (\text{by } \mathbb{P}' = \mathcal{E}(\mathcal{L})_T \mathbb{P}) \\
&= \mathbb{E}_{\mathbb{P}}\left[-\int_0^T b(B_s,s)\mathrm{d}B_s + \frac{1}{2}\int_0^T \|b(B_s,s)\|_2^2 \mathrm{d}s\right] \\
&= \mathbb{E}_{\mathbb{P}}\left[-\mathcal{L}_T + \frac{1}{2}\int_0^T \|b(B_s,s)\|_2^2 \mathrm{d}s\right] \\
&= \frac{1}{2}\mathbb{E}_{\mathbb{P}}\left[\int_0^T \|b(B_s,s)\|_2^2 \mathrm{d}s\right] \quad (\mathbb{E}_{\mathbb{P}}[\mathcal{L}_T] = \mathcal{L}_0 = 0 \text{ by } \mathcal{L} \text{ is a } \mathbb{P}\text{-martingale}) \\
&= \frac{1}{2}\int_0^T \frac{1}{g^2(t)}\mathbb{E}_{X_t \sim P_t}\left[\|f(X_t,t) - f'(X_t,t)\|_2^2\right]\mathrm{d}t.
\end{aligned}$$

$\square$

### F.2 Girsanov's theorem for SGMs

**Corollary 6** (Girsanov's Theorem for SGMs). *Let $P_{t_0} := \mathrm{law}(Y_{T-t_0}), \widehat{P}_{t_0} := \mathrm{law}(\widehat{Y}_{T-t_0})$ be the law of the random variables at time $t = T - t_0$ for the two processes Eq. (3) and (6), respectively, i.e.,:*

$$\begin{aligned}
\mathrm{d}Y_t &= \big(Y_t + 2\nabla \log p_{T-t}(Y_t)\big)\mathrm{d}t + \sqrt{2}\mathrm{d}B_t \quad (0 \le t \le T - t_0), \qquad Y_0 \sim P_T, \\
\mathrm{d}\widehat{Y}_t &= \big(\widehat{Y}_t + 2s_\theta(\widehat{Y}_t, T-t)\big)\mathrm{d}t + \sqrt{2}\mathrm{d}B_t \quad (0 \le t \le T - t_0), \qquad \widehat{Y}_0 \sim P_T.
\end{aligned}$$

*Then, we have*

$$\mathrm{KL}(P_{t_0}\|\widehat{P}_{t_0}) \le \int_{t_0}^T \int_{\mathbb{R}^d} \|\nabla \log p_t(x_t) - s_\theta(x_t,t)\|_2^2 p_t(x_t)\mathrm{d}x_t \mathrm{d}t.$$

*Proof.* By Corollary 5, we have

$$\mathrm{KL}(P_{t_0}\|\widehat{P}_{t_0}) \leq \int_0^{T-t_0} \frac{1}{2} \int \frac{1}{2}\|2\nabla \log p_{T-t}(y_t) - 2s_\theta(y_t, T-t)\|_2^2 p_{T-t}(y_t)\mathrm{d}y_t\mathrm{d}t$$

$$= \int_{t_0}^T \int \|\nabla \log p_t(x_t) - s_\theta(x_t, t)\|_2^2 p_t(x_t)\mathrm{d}x_t\mathrm{d}t. \qquad \text{(by } x_t = y_{T-t})$$

$\square$

# G   Controlling Early Stopping Induced Errors

## G.1   Sobolev class of density

The following assumption, adopted from [29], characterizes the Sobolev class via the Fourier transform of $f$. Unlike the usual definition (which restricts orders to integer values), this allows each $\nu_i$ to take values not only as integers but also as positive real numbers.

**Definition 6.** *For $s, K \in \mathbb{R}_+$, the Sobolev class of density is defined as*

$$\mathcal{W}_2^s(\mathbb{R}^d) := \left\{ f : \mathbb{R}^d \to \mathbb{R} | f \geq 0, \int f = 1, \forall \boldsymbol{\nu} \text{ with } \|\boldsymbol{\nu}\|_1 = s, \int |\omega^{\boldsymbol{\nu}}|^2 \mathsf{FT}[f](\omega)|^2 d\omega \leq (2\pi)^d K^2 \right\}.$$

**Lemma 26** ([29, E.1]). *Under Assumptions 1 and 2, if $s \in [0, 2], t_0 = n^{-\frac{2}{d+2s}}$ and $p_{t_0} = p_0 * \varphi_{\sigma_t}$, where $\varphi_{\sigma_t}$ is the density of Gaussian distribution in d-dimension, $\mathcal{N}(0, tI_d)$ and $*$ denote the convolution operator, then there exists a constant $C$ that depends on $p_0, s, L$ and dimension $d$ such that*

$$\mathsf{TV}(p_0, p_{t_0}) \leq C\mathrm{polylog}(n)n^{-\frac{s}{d+2s}}.$$

## G.2   Besov class of density

To define the Besov space, we introduce the modulus of smoothness.

**Definition 7** (Modulus of Smoothness). *For a function $f \in L^q(\Omega)$ for some $q \in (0, \infty]$, the r-th modulus of smoothness of f is defined by*

$$w_{r,q}(f, t) = \sup_{\|h\|_2 \leq t} \|\Delta_h^r(f)\|_q,$$

*where*

$$\Delta_h^r(f)(x) = \begin{cases} \sum_{j=0}^r \binom{r}{j}(-1)^{r-j} f(x + jh) & (x \in \Omega, \ x + rh \in \Omega), \\ 0 & (otherwise). \end{cases}$$

Based on the modulus of smoothness, the Besov space is defined as in the following definition.

**Definition 8** (Besov space ($B_{q,q'}^s(\Omega)$) [49, 22]). *For $0 < q, q' \leq \infty$, $s > 0$, $r := \lfloor s \rfloor + 1$, let the seminorm $|\cdot|_{B_{q,q'}^\alpha}$ be*

$$|f|_{B_{q,q'}^s} := \begin{cases} \left( \int_0^\infty (t^{-s} w_{r,q}(f, t))^{q'} \frac{dt}{t} \right)^{\frac{1}{q'}} & (q' < \infty), \\ \sup_{t>0} t^{-s} w_{r,q}(f, t) & (q' = \infty). \end{cases}$$

*The norm of the Besov space $B_{q,q'}^s(\Omega)$ can be defined by*

$$\|f\|_{B_{q,q'}^s} := \|f\|_q + |f|_{B_{q,q'}^s},$$

*and we have $B_{q,q'}^s(\Omega) = \{f \in L^q(\Omega) \mid \|f\|_{B_{q,q'}^s} < \infty\}$.*

Note that $q, q' < 1$ is also allowed. In that setting, the Besov space is no longer a Banach space but a quasi-Banach space. If $s > d/q, B_{q,q'}^s(\Omega)$ is continuously embedded in the set of the continuous functions. Otherwise, the elements in the space are no longer continuous.

Considering the Besov space, many well-known function classes, such as Hölder space and Sobolev space can be discussed unified. The relationship between Besov, Hölder, and Sobolev space are well known [57]:

- For $s \in \mathbb{N}, B_{q,1}^s(\Omega) \hookrightarrow \mathcal{W}_q^s(\Omega) \hookrightarrow B_{q,\infty}^s(\Omega)$.
- $B_{2,2}^s(\Omega) = \mathcal{W}_2^s(\Omega)$.
- For $s \in \mathbb{R}_+ \setminus \mathbb{Z}_+, \mathcal{C}^s(\Omega) = B_{\infty,\infty}^s(\Omega)$.

**Theorem 8** (Marchaud inequality [58]). *Let $f \in L^q(\mathbb{R}^d)$ with $1 \leq q \leq \infty$. For any integer $r \geq 1$ and $0 < k < r$, there exists a constant $C = C(r, k, d, q)$ depending only on $r, k, d, q$ such that for all $t > 0$,*

$$w_{k,q}(f, t) \leq Ct^k \int_t^\infty w_{r,q}(f, u) \frac{du}{u^{k+1}}.$$

**Lemma 27.** *Let $p \in L^q([0,1]^d) \cap B_{q,q'}([0,1]^d)$ be a probability density with $0 < s \leq 2$ and $1 \leq q \leq \infty, 0 < q' \leq \infty$. Let $\varphi_\sigma = (2\pi\sigma^2)^{-d/2}\exp(-\frac{\|\boldsymbol{x}\|_2^2}{2\sigma^2})$. Then, there exists a constant $C = C(s,d,q,q')$ such that for all $\sigma > 0$,*

$$\|p - p * \varphi_\sigma\|_1 \leq C\sigma^s |p|_{B_{q,q'}^s},$$

*and hence it holds that*

$$\mathsf{TV}(P, P * \mathcal{N}(\boldsymbol{0}, \sigma^2 \boldsymbol{I}_d)) = \frac{1}{2}\|p - p * \varphi_\sigma\|_1 \leq C\sigma^s |p|_{B_{q,q'}^s}.$$

*Proof.* We proceed in the following four steps:

**Step 1: Pointwise bound for the modulus from the Besov seminorm.**

Recall from Definition 8 that the definition of the Besov seminorm is given by

$$|p|_{B_{q,q'}^s} := \begin{cases} \left(\int_0^\infty (t^{-s} w_{r,q}(p,t))^{q'} \frac{\mathrm{d}t}{t}\right)^{\frac{1}{q'}} & (q' < \infty), \\ \sup_{t>0} t^{-s} w_{r,q}(p,t) & (q' = \infty). \end{cases}$$

For $0 < q' < \infty$, one has the elementary estimate (averaging over a dyadic annulus):

$$\sup_{t \leq k \leq 2t} k^{-s} w_{r,q}(p,k) \leq (\log 2)^{-1/q'} \left(\int_t^{2t} \left(k^{-s} w_{r,q}(p,k)\right)^{q'} \frac{\mathrm{d}k}{k}\right)^{1/q'} \leq C|p|_{B_{q,q'}^s},$$

and for $k \in [t, 2t]$ we get $w_{r,q}(p,k) \leq Ck^s |p|_{B_{q,q'}^s}$. In particular, we can take $k = t$ to obtain the pointwise bound

$$w_{r,q}(p,t) \leq Ct^s |p|_{B_{q,q'}^s}. \tag{257}$$

For $q' = \infty$, the same conclusion follows directly from the definition in Definition 8.

**Step 2: Relate $L^1$-difference to the modulus of smoothness $w_{1,1}(p,t)$.**

By the definition of convolution,

$$(p * \varphi_\sigma)(\boldsymbol{x}) = \int_{\mathbb{R}^d} p(\boldsymbol{x} - \boldsymbol{y})\varphi_\sigma(\boldsymbol{y})\mathrm{d}\boldsymbol{y},$$

we have

$$p(\boldsymbol{x}) - (p * \varphi_\sigma)(\boldsymbol{x}) = \int_{\mathbb{R}^d} \left(p(\boldsymbol{x}) - p(\boldsymbol{x} - \boldsymbol{y})\right)\varphi_\sigma(\boldsymbol{y})\mathrm{d}\boldsymbol{y}$$

Take $L^1$-norm and apply Fubini and Minkowski's inequality, we obtain

$$\|p - p * \varphi_\sigma\|_1 = \int_{[0,1]^d} \left|\int_{\mathbb{R}^d} \left(p(\boldsymbol{x}) - p(\boldsymbol{x} - \boldsymbol{y})\right)\varphi_\sigma(\boldsymbol{y})\mathrm{d}\boldsymbol{y}\right|\mathrm{d}\boldsymbol{x}$$

$$\leq \int_{\mathbb{R}^d} \int_{[0,1]^d} |p(\cdot) - p(\cdot - \boldsymbol{y})|\mathrm{d}\boldsymbol{x}\varphi_\sigma(\boldsymbol{y})\mathrm{d}\boldsymbol{y}.$$

Change variables $\boldsymbol{y} = \sigma\boldsymbol{z}$ and write $\varphi_1$ for the standard Gaussian density, we have

$$\varphi_\sigma(\boldsymbol{y})\mathrm{d}\boldsymbol{y} = (2\pi\sigma^2)^{-d/2} \exp\left(-\frac{\|\sigma\boldsymbol{z}\|_2^2}{2\sigma^2}\right)\sigma^d\mathrm{d}\boldsymbol{z} = \varphi_1(\boldsymbol{y})\mathrm{d}\boldsymbol{z}.$$

Then,

$$\|p - p * \varphi_\sigma\|_1 \leq \int_{\mathbb{R}^d} \left\|p(\cdot) - p(\cdot - \sigma\boldsymbol{z})\right\|_1 \varphi_1(\boldsymbol{z})\mathrm{d}\boldsymbol{z}. \tag{258}$$

Thus the problem reduces to bounding the $L^1$-difference $\|p(\cdot) - p(\cdot - \boldsymbol{h})\|_1$ for small shifts $\boldsymbol{h} = \sigma\boldsymbol{z}$ in terms of the modulus of smoothness. By the definition of modulus of smoothness (Definition 8),

we have

$$\|p - p * \varphi_\sigma\|_1 \leq \int_{\mathbb{R}^d} \big\|p(\cdot) - p(\cdot - \sigma z)\big\|_1 \varphi_1(z)\mathrm{d}z$$

$$\leq \int_{\mathbb{R}^d} \|\Delta^1_{\sigma z} p\|_1 \varphi_1(z)\mathrm{d}z$$

$$\leq \int_{\mathbb{R}^d} w_{1,1}(p, \sigma\|z\|_2)\varphi_1(z)\mathrm{d}z. \tag{259}$$

**Step 3: Bounding $w_{1,1}(p,t)$ by $w_{r,q}(p,t)$.**

Recall from Definition 7 that the $r$-th modulus of smoothness is defined as

$$w_{r,1}(p,t) := \sup_{\|h\|_2 \leq t} \|\Delta^r_h p\|_1,$$

where $r = \lfloor s \rfloor + 1$.

**Case 1:** $0 < s \leq 1$.

For $0 < s \leq 1$ we use $r = 1$ and the first-order difference is given by

$$\Delta^1_h p(\cdot) := p(\cdot + h) - p(\cdot)$$

Since $\Omega = [0,1]^d$ and $1 \leq q \leq \infty$, by Hölder's inequality, for any shift $h$, we have

$$\|\Delta^1_h p\|_1 \leq |\Omega|^{1-\frac{1}{q}} \|\Delta^1_h p\|_q, = \|\Delta^1_h p\|_q,$$

which implies that

$$w_{1,1}(p,t) \leq w_{1,q}(p,t). \tag{260}$$

Further, by Eq. (257), we obtain

$$w_{1,1}(p,t) \leq Ct^s |p|_{B^s_{q,q'}}. \tag{261}$$

**Case 2:** $1 < s \leq 2$.

For $1 < s \leq 2$ we use $r = 2$ and the second-order difference is given by

$$\Delta^2_h p(\cdot) := p(\cdot + 2h) - 2p(\cdot + h) + p(\cdot),$$

and the second-order modulus is

$$w_{2,q}(p,t) = \sup_{\|h\|_2 \leq t} \|\Delta^2_h p\|_q.$$

When $s > 1$, we have direct control only of the second-order modulus $w_{2,q}(p,t)$. We need a relation bounding the first-order modulus $w_{1,q}(p,t)$ by $w_{2,q}(p,t)$.

Apply Theorem 8, we obtain

$$w_{1,q}(p,t) \leq Ct \int_t^\infty w_{2,q}(p,u) \frac{\mathrm{d}u}{u^2}.$$

Further by Eq. (257) and $1 < s \leq 2$, we have

$$w_{1.q}(p,t) \leq Ct \int_t^\infty u^{s-2} |p|_{B^s_{q,q'}} \mathrm{d}u = \frac{C}{s-1} t^s |p|_{B^s_{q,q'}}.$$

Plugging into Eq. (260), we get

$$w_{1,1}(p,t) \leq Ct^s |p|_{B^s_{q,q'}}. \tag{262}$$

Combining Cases 1 and 2, we obtain that for any $0 < s \leq 2$,

$$w_{1,1}(p,t) \leq Ct^s |p|_{B^s_{q,q'}}. \tag{263}$$

**Step 4: Bounding the TV distance.**

Plugging Eq. (263) into Eq. (259) we obtain

$$\|p - p * \varphi_\sigma\|_1 \leq \int_{\mathbb{R}^d} w_{1,1}(p, \sigma\|\boldsymbol{z}\|_2)\varphi_1(\boldsymbol{z})\mathrm{d}\boldsymbol{z}$$

$$\leq C|p|_{B^s_{q,q'}} \int_{\mathbb{R}^d} \left(\sigma\|\boldsymbol{z}\|_2\right)^s \varphi_1(\boldsymbol{z})\mathrm{d}\boldsymbol{z}$$

$$= C|p|_{B^s_{q,q'}} \sigma^s \mathbb{E}\big[\|\boldsymbol{Z}\|_2^s\big]$$

$$\leq C|p|_{B^s_{q,q'}} \sigma^s \mathbb{E}\big[\|\boldsymbol{Z}\|_2^2\big], \qquad\qquad \text{(by } 0 < s \leq 2)$$

where $\boldsymbol{Z} \sim \mathcal{N}(\boldsymbol{0}, \boldsymbol{I}_d)$ and we have $\mathbb{E}[\|\boldsymbol{Z}\|_2^2] = d$. Therefore, for the law $P$ with density $p$, we obtain

$$\mathsf{TV}(P, P * \mathcal{N}(\boldsymbol{0}, \sigma^2\boldsymbol{I}_d)) = \frac{1}{2}\int_{\mathbb{R}^d} |p(\boldsymbol{x}) - (p * \varphi_\sigma)(\boldsymbol{x})|\mathrm{d}\boldsymbol{x} = \frac{1}{2}\|p - p * \varphi_\sigma\|_1 \leq C'd|p|_{B^s_{q,q'}}\sigma^s.$$

$\square$

**Theorem 9.** *Let $0 < s \leq 2$ and $1 \leq q \leq \infty, 0 < q' \leq \infty$. Consider a probability density $p \in L^q([0,1]^d) \cap U(B_{q,q'}([0,1]^d), C)$, where $U(\cdot; C)$ denotes the ball of radius $C$. Let $\{X_t\}_{t\in[t_0,T]}$ be the solutions of the process  Eq. (2). Setting $t_0 = n^{-\frac{2}{d+2s}}$, we have that*

$$\mathsf{TV}(X_0, X_{t_0}) \lesssim n^{-\frac{s}{d+2s}}$$

*Proof.* From Lemma 27, we have for all $\sigma_{t_0} > 0$,

$$\mathsf{TV}(X_0, X_{t_0}) \lesssim \sigma_{t_0}^s |p|_{B^s_{q,q'}}$$

Since $\sigma_{t_0} \lesssim \sqrt{t_0}$, substitute $t_0 = n^{-\frac{2}{d+2s}}$ gives

$$\mathsf{TV}(\boldsymbol{X}_0, \boldsymbol{X}_{t_0}) \lesssim t_0^{s/2} = n^{-\frac{s}{d+2s}}.$$

$\square$

# H  Other Lemmas and definitions

## H.1  Fourier analysis

**Lemma 28** (Fourier Transform and Inverse Fourier Transform). *The Fourier transform of a continuous function $f \in L^1(\mathbb{R})$ is defined as:*

$$\widetilde{f}(\omega) := \mathsf{FT}(f(x)) = \frac{1}{\sqrt{2\pi}} \int_{\mathbb{R}} f(x) e^{-i\omega x} \mathrm{d}x$$

*The inverse transform is defined as:*

$$f(x) = \frac{1}{\sqrt{2\pi}} \int_{\mathbb{R}} \widetilde{f}(\omega) e^{i\omega x} \mathrm{d}\omega, \quad \forall x \in \mathbb{R}.$$

**Lemma 29** (Fourier Transform of Derivative). *Suppose $f : \mathbb{R} \to \mathbb{R}$ is an absolutely continuous differentiable function, and both $f$ and its derivative $f'$ are integrable. Then the Fourier transform of $f'$ is given by*

$$\widetilde{f'}(\omega) := \mathsf{FT}(f'(x)) = i\omega \widetilde{f}(\omega).$$

*More generally, the Fourier transformation of the $k$-th derivative $f^{(k)}$ is given by*

$$\widetilde{f^{(k)}}(\omega) = \mathsf{FT}\left( \frac{\mathrm{d}^k}{\mathrm{d}x^k} f(x) \right) = (i\omega)^k \widetilde{f}(\omega).$$

**Lemma 30** (Plancherel's Identity). *For a square-integrable function $f(x) \in L^2(\mathbb{R})$, Plancherel's identity is given by*

$$\|f\|_{L^2(\mathbb{R})}^2 := \int_{\mathbb{R}} |f(x)|^2 \mathrm{d}x = \int_{\mathbb{R}} |\widetilde{f}(\omega)|^2 \mathrm{d}\omega,$$

*where $\hat{f}(\omega)$ is the Fourier transform of $f(x)$.*

## H.2  Distribution inequalities

**Definition 9.** *For distributions $P, Q \in \mathcal{P}(\mathbb{R}^d)$, and their probability density functions $p, q : \mathbb{R}^d \to \mathbb{R}$,*

- *The **total variation (TV) distance** is defined as*

$$\mathsf{TV}(P, Q) := \sup_{\mathcal{A} \subseteq \mathbb{R}^d} |p(\mathcal{A}) - q(\mathcal{A})| = \frac{1}{2} \int_{\mathbb{R}^d} |p(x) - q(x)| \mathrm{d}x. \tag{264}$$

- *The **Kullback-Leibler (KL)-divergence** is defined as*

$$\mathrm{KL}(P\|Q) := \int_{\mathbb{R}^d} p(x) \log\left( \frac{p(x)}{q(x)} \right) \mathrm{d}x. \tag{265}$$

- *The **Hellinger distance** is defined as*

$$\mathsf{H}^2(P, Q) := \int_{\mathbb{R}^d} \left( \sqrt{p(x)} - \sqrt{q(x)} \right)^2 \mathrm{d}x. \tag{266}$$

**Lemma 31** (Pinsker's Inequality [51]). *For any two probability distributions $P$ and $Q$ defined on the same probability space, we have*

$$\mathsf{TV}(P, Q) \leq \sqrt{\frac{1}{2} \mathrm{KL}(P\|Q)}.$$

**Lemma 32.** *For any two probability distributions $P$ and $Q$ defined on the same probability space, we have*

$$\mathsf{H}^2(P, Q) \leq \mathrm{KL}(P\|Q).$$

### H.3 Taylor's theorem

**Theorem 10** (Taylor's Theorem). *Let $k \in \mathbb{N}_+$ be an integer and let the function $f : \mathbb{R} \to \mathbb{R}$ be $k$ times differentiable at the point $a \in \mathbb{R}$. Then there exists a function $h_k : \mathbb{R} \to \mathbb{R}$ such that*

$$f(x) = \sum_{i=0}^{k} \frac{f^{(i)}(a)}{i!}(x-a)^i + \underbrace{h_k(x)(x-a)^k}_{:=R_k(x)},$$

*and $\lim_{x \to a} h_k(a) = 0$, which is called the Peano form of the remainder.*

**Lemma 33** (Lagrange Forms of the Reminder). *Let $f : \mathbb{R} \to \mathbb{R}$ be $k+1$ times differentiable on the open interval with $f^{(k)}$ continuous on the closed interval between $a$ and $x$. Then*

$$R_k(x) = \frac{f^{(k+1)}(\xi_L)}{(k+1)!}(x-a)^{k+1}$$

*for some real number $\xi_L$ between $a$ and $x$.*

### H.4 Nonparametric classes

**Definition 10** (Hölder Space). *[59] For $s \in \mathbb{R}_+ \setminus \mathbb{Z}_+$ and $\Omega \subset \mathbb{R}^d$, the Hölder space is a set of $\lfloor s \rfloor$ times differentiable functions*

$$\mathcal{C}^s(\Omega) := \left\{ f : \Omega \to \mathbb{R} : \sum_{\boldsymbol{\alpha}:\|\boldsymbol{\alpha}\|_1 < s} \|\partial^{\boldsymbol{\alpha}} f\|_\infty + \sum_{\boldsymbol{\alpha}:\|\boldsymbol{\alpha}\|_1 = \lfloor s \rfloor} \sup_{\boldsymbol{x},\boldsymbol{y} \in \Omega, \boldsymbol{x} \neq \boldsymbol{y}} \frac{|\partial^{\boldsymbol{\alpha}} f(\boldsymbol{x}) - \partial^{\boldsymbol{\alpha}} f(\boldsymbol{y})|}{|\boldsymbol{x} - \boldsymbol{y}|^{s - \lfloor s \rfloor}} \leq \infty \right\},$$

*where $\partial^{\boldsymbol{\alpha}} := \partial^{\alpha_1} \cdots \partial^{\alpha_d}$, with $\alpha = (\alpha_1, \ldots, \alpha_d) \in \mathbb{N}^d$.*

