# OpenReview forum: "Approximation and Generalization Abilities of Score-based Neural Network Generative Models for Sub-Gaussian Distributions"
_NeurIPS.cc/2025/Conference — NeurIPS 2025 poster_

### Official Review · Reviewer_XhMA · 2025-06-13

**Clarity:** 3
**Significance:** 3
**Originality:** 3
**Rating:** 5
**Confidence:** 3

**Summary:**

The paper studies the approximation, generalization, and distribution estimation guarantees of score-based generative models with neural network estimators, under the sole assumption that the data distribution has sub-Gaussian tail decay. In particular, they remove two common assumptions made in previous works: the Lipschitz continuity of the score function (across the whole time domain) and a strict lower bound of the density. The key technical insight of the paper is that the regularized kernel score estimator, proposed in [30], approximates the true score function even without Lipschitz continuity of the score function. This allows them to prove two important results:

1. Neural networks can approximate the score function with only the sub-Gaussianity assumption. This is proven by constructing a neural network which approximates the regularized kernel score estimator via local Taylor expansions of the kernel.
2. Neural networks trained by empirical score matching generalize well. To prove this, they reduce the generalization error to a difference between losses of the ERM and kernel score estimator built from independent ghost samples. This random variable can be controlled using elementary techniques from probability theory, whereas direct approaches to bound the generalization error often require the density lower bound.

Using these two results, the authors bound the distribution estimation error of score-based generalization errors as a sum of the score matching error, which is minimax optimal up to log factors, and early stopping error. For some common nonparametric function spaces, such as Besov spaces, the early stopping error can also be controlled, leading to end-to-end bounds.

[30] A. Wibisono, Y. Wu, and K. Y. Yang, “Optimal score estimation via empirical bayes smoothing,”
in The Thirty Seventh Annual Conference on Learning Theory, June 30 - July 3, 2023, Edmonton,
Canada, vol. 247 of Proceedings of Machine Learning Research, pp. 4958–4991, PMLR, 2024.

**Questions:**

- In Lemma 12, I believe the weights of the network $\phi_{score}$ actually depend on the sample points $\{x^{(i)}\}$ being averaged, because the construction of $\phi_{score}$ relies on approximating the kernel density estimator associated to the samples. This needs to be addressed in the proof of Theorem 1, where you drop the expectation from the bound. Essentially, I think you should clarify in which results the neural network constructed is random and in which results it is deterministic.
- Can you explain where the $O(n^{3/d}))$ depth arises from, and whether this dependence on $n$ can be improved? Also, can you explain why the depth of the network decreases with dimension? Intuitively, a higher-dimensional distribution should require a more complex network to approximate its score.
- In Lemma 2, I believe the set $\mathcal{G}$ should be defined in terms of $\rho_n$ rather than $\rho$?
- Is it possible to extend your results to allow for distributions whose tail decays slower than Gaussian? For instance, it seems the main use of the sub-Gaussianity is in bounding the second term in Lemma 2, but this term could still be controlled for distributions with fatter tails (e.g., sub-exponential or sub-Weibull).
- At the bottom of page 8, why do you introduce both $f^{(1)}$ and $f^{(2)}$ if they are equal to the same expression?
- Typo on page 43, Step 2, “estiamtors”
- On page 22, what is the meaning of “standard Gaussian-convolved densities” and “general Gaussian-convolved densities”?

**Ethical Concerns:**

["NO or VERY MINOR ethics concerns only"]

**Final Justification:**

I maintain my original assessment, which is that the paper comfortably meets the criteria for acceptance. The paper is a technically impressive theoretical work which unifies and improves several previous results on error analysis of diffusion models and develops independently-interesting approximation and generalization error techniques along the way. The only limitation is the lack of experimental validation, but in general it is extremely hard to validate such worst-case results numerically, and it is common for these kinds of theoretical papers to be accepted to the conference when the contribution is strong enough. I recommend acceptance.

**Limitations:**

Yes

**Quality:**

3

**Strengths And Weaknesses:**

Strengths: The paper makes important progress to remove commonly-made assumptions in the analysis of score-based generative models. It is the first work, to my knowledge, to provide generalization guarantees under only sub-Gaussianity of the data distribution. The intermediate results, such as removing the Lipschitz continuity assumption in the generalization error of kernel score estimators and approximating KDEs via neural networks, may be of independent interest. The proof strategies are well-explained considering the technicality of the work.

Weaknesses: Since the results apply to such a wide class of distributions, the convergence rates are quite pessimistic and probably fail to explain the real world successes of diffusion models. However, for a theoretically-focused paper, this is not a major concern. Additionally, the depth of the score network grows polynomially in the sample size $n$, which is quite large compared to related works (e.g., [22] constructs a score network with depth $O(polylog(n))$) and does not seem realistic.

[22] K. Oko, S. Akiyama, and T. Suzuki, “Diffusion models are minimax optimal distribution
estimators,” in International Conference on Machine Learning, pp. 26517–26582, PMLR, 2023.

---

> ### Author Rebuttal · Authors · 2025-07-30
>
> We are grateful for your extremely thorough and constructive review.
> Your detailed questions have helped us identify several areas for clarification.
> We address your comments in detail as follows.
>
> >**W1:**
> Pessimistic rates.
>
> We agree that our convergence rates are conservative, especially for high-dimensional settings. As our work aims to offer worst-case guarantees under minimal assumptions, this is somewhat inevitable. Nonetheless, we believe our results fill an important gap in understanding what is theoretically achievable under weak regularity.
> We see breaking the CoD through further structural assumptions as an important future direction.
>
>
> >**Q1:**
> Sample-dependence of the score network
>
> Thank you for this sharp and important question. You are correct that the score network constructed in our approximation results is sample-dependent. We clarify how this is handled in our analysis below.
>
> 1. **The approximating network in Theorem 1.**
> The neural network $\phi_{\text{score}}$ in Theorem 1 is indeed a random network.
> It is constructed to approximate the regularized empirical score function (a KDE-based estimator), and its parameters therefore depend on the specific sample used to construct the kernel estimator.
> The statement of Theorem 1 should be interpreted as follows: for any given sample $\bar{S} := \bar{x}^{(1)}, \dots, \bar{x}^{(n)} \sim P_0^{\otimes n}$, there exists a network with the specified architecture constructed from $S$ that achieves the stated approximation error.
> The expectation in the theorem's bound should be understood as an expectation taken over both the test point $x_t$ and the randomness of the sample $\bar{S}$, i.e., $E_{\bar{S}}[E_{x_t \sim P_t}[\cdot]]$.
> We will revise the statement of Theorem 1 to make this sample dependence explicit.
>
> 2. **The learned network in Theorem 2.**
> This sample dependence is a critical point that we carefully account for in our generalization analysis for the learned network $\widehat{\phi}$ in Theorem 2.
> Our proof strategy is as follows:
>
>    - We introduced an independent "ghost sample" $\bar{S}$ to construct the target regularized empirical score function, $\widehat{\bar{\phi}}$ (Line 1689-1690 Page 79), as noted by the reviewer.
>
>    - The "ideal" approximating network $\tilde{\phi}$ (Line 1725 Page 83) from Theorem 1 can then be considered to be the one constructed based on this ghost sample. This makes $\tilde{\phi}$ a random target that is independent of the actual training data.
>
>    - The final network $\widehat{\phi}$ is obtained by performing empirical risk minimization on the actual training data.
>
>     Our proof correctly bounds the excess risk by properly handling the randomness from both sample sets. Therefore, our generalization bound and proof of Theorem 2 remain valid.
>
> We will revise the manuscript to improve clarity on this crucial point.
>
> >**W2/Q2:**
> Where $\tilde{\mathcal{O}}(n^{3/d})$ depth arises from, and whether it can be improved.
>
> This is an excellent and constructive point.
> We agree that a network depth that grows polynomially with the sample size is a practical limitation.
> **This can be improved within our framework**.
>
> The depth of $\tilde{\mathcal{O}}(n^{3/d})$ (for $k \geq d/2$) arises from a specific choice of parameters $N = 1, L = \lceil n^{1/(2k)} \rceil, k \geq d/2, \epsilon = n^{-1/k}, s=k$ in the proof of Lemma 12 (see Line 1447-1448 Page 67), which are made to satisfy a technical condition from Lemma 11 and 12 (i.e., $N^{-2}L^{-2} \leq t_0 \wedge n^{-1/k}$).
> However, this choice is not fundamental.
> Our framework allows for separate control over network width and depth.
> Following the reviewer's suggestion, we can revise our construction to achieve a more realistic architecture.
> By swapping the roles of width and depth, we can choose $N = \lceil n^{1/(2k)} \rceil, L = 1$ and construct a network with width $\leq \mathcal{O}(n^{\frac{3}{2k}}\log_2n)$ and depth $\leq \mathcal{O}(\log^2n)$.
> This new architecture is more practical and, crucially, our main theoretical guarantees remain valid:
>
> - The condition $N^{-2}L^{-2} \leq t_0 \wedge n^{-1/k}$ required by Lemma 11 and 12 is still satisfied.
>
> - The generalization bound in Theorem 2 remains valid since the new network architecture's pseudo-dimension remains of the same order, which is bounded by $\tilde{\mathcal{O}}(n^{\frac{3}{k}})$.
>
> We are grateful for this suggestion and will update the manuscript to reflect this significantly improved architecture.
>
> >**Q2:** Dependence on dimension $d$.
>
> Thank you for the excellent question.
> This counter-intuitive relationship is a direct consequence of the interplay between the smoothness of $P_{t_0}$ and the network architecture in our theoretical analysis, which has a clear theoretical motivation:
>
> - **Higher dimensions force more smoothing.**
> Our condition on the early stopping time (i.e., $t_0 \geq \tilde{\mathcal{O}}(n^{-2/d})$ in Theorem 1 and 2 as well as $t_0 = n^{\frac{-2}{d+2s}}$ in Corollary 1 and 2) forces $t_0$ to be larger for higher dimension $d$ to ensure theoretical guarantees.
>
> - **More smoothing makes the marginal distribution $P_{t_0}$ easier to learn.**
> A larger $t_0$ means that $P_{t_0}$ is convolved with a Gaussian of larger variance.
> This makes $P_{t_0}$ inherently **smoother** and less complex, as fine-grained details are averaged out.
> As a result, its score can be approximated by a network whose size scales less severely with the sample size $n$.
>
> In summary, while higher dimension usually implies increased statistical difficulty (as seen in the convergence rate $n^{-s/(d+2s)}$ suffering from the curse of dimensionality), the necessity of stronger smoothing to achieve uniform control in high dimensions makes the intermediate learning problem architecturally less demanding in terms of network size. This is a reflection of the bias–variance trade-off: we reduce variance (by smoothing), but incur bias, which is ultimately what limits the rate.
> We will clarify this point in the revised manuscript.
>
>
> >**Q3:**
> Use of $\rho_n$ in $\mathcal{G}$ Definition.
>
> Yes, this should be written in terms of $\rho_n$.
> We will correct this typo.
>
>
> >**Q4:**
> Extension to sub-exponential or sub-Weibull distributions
>
> This is an excellent suggestion for future work.
> Our current analysis relies on the sub-Gaussian assumption in two critical places:
>
> - Lemma 7: To bound the score function's norm over low-density regions.
>
> - Lemma 3: To bound the KL-divergence of the Gaussian smoothed empirical distribution, which is fundamental to our main error bounds.
>
> Extending these results to broader tail classes like sub-exponential or sub-Weibull would require more refined control of tail integrals, but we agree it is plausible and an exciting avenue for future work.
> In particular, we believe that the extensions may benefit from the convergence rates of Gaussian smoothed empirical distributions under heavy-tail distribution assumptions.
> We will include the discussion in the revision.
>
>
> >**Q5:**
> Use of $f^{(1)}, f^{(2)}$.
>
> These notations correspond to different components in the DNN decomposition (e.g., numerator vs. denominator in the score approximation), and are treated separately for modular construction.
> An alternative approach would be to construct a single sub-network for this expression and reuse its output via connections to different parts of the network.
> Since both methods result in a final network of a similar size and complexity, our notation was intended to reflect this separation.
>
>
> >**Q6:**
> Typo "estiamtors"
>
> Thank you. We will correct the typo.
>
> >**Q7:**
> Clarification on “Standard” vs. “General” Gaussian-convolved Densities
>
> "Standard Gaussian-convolved" refers to convolution with a standard Gaussian distribution $\mathcal{N}(0, \mathbf{I}_d)$, whereas "general Gaussian-convolved" refers to convolution with a general isotropic Gaussian distribution $\mathcal{N}(0, \sigma^2\mathbf{I}_d)$ for any $\sigma > 0$.
> We will improve this terminology for clarity.

---

> > ### Comment · Reviewer_XhMA · 2025-07-31
> >
> > Thank you for the detailed reply. All of my questions/concerns have been addressed, and the bias-variance tradeoff you mention between score smoothing and estimation is interesting. I still feel that the paper easily meets the criteria for acceptance and I retain my initial score.

---

> > > ### Author Response · Authors · 2025-08-01
> > >
> > > Thank you again for your time and detailed feedback, which has been invaluable in helping us clarify key aspects of our work. We are pleased that our response addressed your concerns and appreciate your positive assessment and continued support for our paper.

---

### Official Review · Reviewer_tHVn · 2025-06-30

**Clarity:** 3
**Significance:** 3
**Originality:** 2
**Rating:** 4
**Confidence:** 3

**Summary:**

This paper provides a theoretical analysis of score-based generative models (SGMs) under the mild assumption that the data distribution is sub-Gaussian. It shows that ReLU neural networks can approximate score functions with nearly optimal error rates, even without assuming Lipschitz continuity or lower-bounded densities. The authors establish generalization bounds for neural network-based SGMs and demonstrate that these models achieve near-minimax optimal convergence in total variation distance for Sobolev and Besov class densities. This work broadens the applicability of SGMs to more general and realistic data settings.

**Questions:**

See weakness

**Ethical Concerns:**

["NO or VERY MINOR ethics concerns only"]

**Final Justification:**

Thanks for explaining the technical novelty. While I am not changing my score, I am leaning toward acceptance.

**Limitations:**

See weakness

**Quality:**

3

**Strengths And Weaknesses:**

Strengths:
1 The paper establishes approximation and estimation error bounds under weaker assumptions, achieving minimax optimal rates, improving upon a previous result that requires an additional Lipschitz assumption.
2 It offers a technically refined analysis of the regularized empirical score estimator, showing that it attains nearly minimax-optimal rates under the score matching loss, improving theoretical understanding in a general setting.

Weaknesses:
1 Aside from the empirical Bayes smoothing analysis, many of the technical tools employed—such as Bernstein’s inequality and Girsanov’s theorem—are standard and well-established in the literature.
2 While the relaxation of assumptions is valuable, the novelty of the contribution may be somewhat limited given the existing literature on the theoretical analysis of SGMs.

---

> ### Author Rebuttal · Authors · 2025-07-30
>
> We sincerely thank you for your thoughtful feedback and acknowledgement of the value of relaxing assumptions.
> We agree that many of the tools we employ, such as Bernstein’s inequality and Girsanov’s theorem, are well-established.
> Our primary contribution, however, lies in the novel synthesis and application of these tools to remove critical and restrictive assumptions from prior work, which required new, non-trivial technical developments. We address your comments in detail below.
>
> >**W1 \& W2:** Use of standard tools and limited novelty of the contribution
>
> While the individual components are standard, their integration within our framework is what enables our main results. Specifically, our key technical contributions are:
>
> - **A refined KL-divergence bound (Lemma 3):** Prior work provided a KL-divergence rate for smoothed empirical distributions that scaled exponentially with the inverse of the noise standard deviation $1/\sigma$. As detailed on page 8, we refine their proof to establish a rate that scales **polynomially** in $1/\sigma$. This improvement is crucial for deriving meaningful bounds in the low-noise regime.
>
> - **New application of Bernstein's inequality:** To remove the density lower-bound assumption from [22, 26, 27], we introduce a new proof strategy. Instead of analyzing the true score function, which may be **unbounded**, we analyze the excess risk for learning a **surrogate** regularized empirical score function. This surrogate is uniformly bounded by construction (Lemma 5), allowing us to verify Bernstein's condition and obtain a uniform high-probability bound without requiring a density lower bound.
>
> The combination of these non-trivial technical steps that allows us to provide the a theoretical framework establishing nearly minimax optimal rates for neural network-based SGMs under the general sub-Gaussian assumption, without the restrictive requirements of Lipschitz continuity or a density lower bound. We believe this significantly expands the applicability of SGM theory and represents a valuable contribution to the literature.

---

> > ### Comment · Reviewer_tHVn · 2025-08-05
> >
> > Thanks for explaining the technical novelty. While I am not changing my score, I am leaning toward acceptance.

---

> > > ### Author Response · Authors · 2025-08-06
> > >
> > > Thank you again for your time in reviewing our paper and providing valuable feedback. We appreciate your positive assessment.

---

### Official Review · Reviewer_rjcC · 2025-07-02

**Clarity:** 3
**Significance:** 2
**Originality:** 2
**Rating:** 4
**Confidence:** 3

**Summary:**

The paper studies the risk bound on estimation of score function in diffusion model via neural networks with i.i.d. samples and ERM algorithm where it is assumed that samples are drawn from a sub-Gaussian distribution. With estimated score function, the authors obtain an error bound on the total variation distance between the learned distribution and true distribution as a natural consequence. The main contribution of the paper is that the authors aim to remove some assumptions imposed by other works on data distribution while keep the error rate optimal.

**Questions:**

Same problem exists in all of the main results.
For example, in theorem 1, the conditions for stopping time is $t_0 \ge 1/2 \alpha^2 n^{-2/d} log n$. If you substitute this into the bound, we have $\alpha^d t_0^{-d/2} n^{-1} log^{d/2+4}n \lesssim log^4 n $.

In my understanding, the main problem is the assumption. Authors would like to obtain the risk bound without smoothness assumption, which seems unrealistic. In lemma 1, where they claim that they achieve the optimal rate with only sub-Gaussian assumption, the bound diverges with the sample size. Their results align with the results in [29]. If we bring the smoothness parameter $\beta$ to 0 in [29], we obtain something similar to the bound in this paper. But $\beta $ has to stay positive to guarantee the convergence.

**Ethical Concerns:**

["NO or VERY MINOR ethics concerns only"]

**Final Justification:**

As the authors mentioned in the response, they would clarify that the convergence result would only hold for some proper stopping time without extra assumptions. In that case, their main results remain valid.

**Limitations:**

Yes

**Paper Formatting Concerns:**

No formatting issues found.

**Quality:**

2

**Strengths And Weaknesses:**

Strengths: the paper is well-written with a clear structure. They present rather a complete picture of issues in learning via diffusion model. Also, it is fairly self-contained since they include most elements of the problem, such as how to utilize OU process to learn a target distribution, how to obtain estimation error and generalization error of score function via neural network, and how to compare learned distribution and target distribution once we have the former and etc.

Weakness: the error bound divergences with sample size n.

---

> ### Author Rebuttal · Authors · 2025-07-30
>
> We sincerely thank you for this valuable feedback.
> Your observation is correct and highlights a key feature of our framework: it accurately reflects the fundamental trade-offs in nonparametric estimation.
> The behavior you noted is expected and **does not undermine our main results**.
> We address your concern in detail as follows.
>
> >W1/Q1: Divergence of bounds with sample size $n$
>
> While the reviewer’s observation is correct, this behavior is expected within our framework and does not undermine our main results. Specifically,
>
> - **It is expected with only a sub-Gaussian distribution assumption.**
> Under only the sub-Gaussian assumption, we establish a convergence rate of $\tilde{\mathcal{O}}(t_0^{-d/2} n^{-1})$  for any $t_0 \geq 1/2\alpha^2n^{-2/d} \log n$.
> The rate indeed becomes vacuous (or diverges) if $t_0$ is taken too close to this lower limit, which is expected and consistent with the findings in [29].
> In fact, [29] also shows that without assuming smoothness, the kernel density estimator cannot achieve meaningful convergence rates either.
> Our framework correctly captures this.
> It reflects the fundamental difficulty of estimating an arbitrarily "rough" distribution.
>
> - **The bound converges to zero if we select an early stopping time $t_0$ that decreases with $n$ slightly slower than $1/2\alpha^2n^{-2/d} \log n$.**
> Noticing that we only require the early stopping time $t_0$ to be in a admissible range $t_0 \geq 1/2\alpha^2n^{-2/d} \log n$ and the bound converges to zero when the early stopping time $t_0$ is set slightly larger, for instance, $t_0 = \mathcal{O}(n^{-2/(d+\epsilon)})$ for any $\epsilon>0$. The convergence rate would be of $\tilde{\mathcal{O}}(n^{-\frac{\epsilon}{d+\epsilon}})$.
>
> - **$t_0$ can be tuned to achieve nearly-optimal rates with additional smoothness assumptions.**
> In particular, as detailed in **Corollaries 1 and 2**, when we add standard nonparametric smoothness assumptions, we can tune the early stopping time $t_0$ to balance the bias $TV(P_0, P_{t_0})$ and statistical error $TV(P_{t_0}, \widehat{P}_{t_0})$.
> For instance, with a Sobolev density, we choose $t_0 = n^{-2/(d+2s)}$, which satisfies the admissible range $t_0 \geq 1/2\alpha^2n^{-2/d}\log n$, and we recover the nearly optimal convergence rate of $\tilde{\mathcal{O}}(n^{-2s/(d+2s)})$.
>
> So in summary, our framework provides a general bound valid for all $t_0$ above the threshold, and when smoothness is available, the early stopping can be tuned to achieve the nearly minimax optimal rate.
>
> We will clarify in both the abstract and main claims that minimax-optimality holds for Sobolev/Besov or otherwise regularized classes, while for the pure sub-Gaussian case we provide the convergence guarantee for $P_{t_0}$ when $t_0 > \mathcal{O}(\alpha^2n^{-2/d}\log n)$.

---

> > ### Comment · Reviewer_rjcC · 2025-08-05
> >
> > Thank you for the response and clarification, based on which, I would like to increase the rating and lean towards acceptance.

---

> > > ### Author Response · Authors · 2025-08-05
> > >
> > > Thank you once again for your time in reviewing our paper and for your follow-up. We appreciate your valuable feedback and your willingness to reconsider your evaluation based on our clarifications.

---

### Official Review · Reviewer_oUBr · 2025-07-05

**Clarity:** 3
**Significance:** 3
**Originality:** 3
**Rating:** 5
**Confidence:** 3

**Summary:**

This paper investigates the approximation and generalization abilities of score-based (ReLU deep) neural network generative models (SGMs). The existing related theory work has two main limitations: some assumptions regarding the data remain quite restrictive, and kernel-based estimators are considered instead of neural network-based ones. To tackle the above issues, this paper provides a universal framework to characterize the approximation and generalization abilities of SGMs for sub-Gaussian distributions. Further, it also applies to the case where the target density function lies in Sobolev or Besov classes, with an appropriately early stopping strategy, demonstrating that neural network-based SGMs can attain nearly minimax convergence rates up to logarithmic factors.

**Questions:**

1. For the score estimation via empirical Bayes smoothing, what is the technical challenge to remove the Lipschitz score assumption compared with [30]?

**Ethical Concerns:**

["NO or VERY MINOR ethics concerns only"]

**Final Justification:**

The authors have addressed most of my concerns, including the discussions about the technical challenges for tackling the curve of dimensionality, the tightness of Theorems 1 & 2, and the technical challenge to remove the Lipschitz score assumption compared with [30]. The remaining concern is that there are no experimental results to illustrate the validity of the established bounds. Therefore, I keep my score.

**Limitations:**

Yes

**Quality:**

3

**Strengths And Weaknesses:**

# Strengths
1. As a dense theoretical work, this paper is overall well-written with clear notations. The motivation is clear that the existing theory work has quite restrictive assumptions on the data, and neural network-based estimators are less considered.
2. The results are promising and seem right. The proof sketches make sense, although I have not carefully checked the detailed proofs in the appendix since the proofs are rather long. Compared with related work, this work removes several restrictive assumptions. More specifically, compared with prior work [30], this work removes the assumption of the Lipschitz continuity of the score function under the sub-Gaussian distribution. In comparison with previous work [22, 26, 27], this paper derives the approximation and estimation error bound without the assumption of a strictly positive lower bound on the target density.
3. This paper has several technical insights or contributions. For neural network score and distribution estimations, to remove the density lower bound condition, this paper proposes a new proof strategy (Lemma 5).

# Weaknesses
1. As the authors discussed, the obtained results can have the issue of the curse of dimensionality, and a low-rank or manifold assumption on data can solve this. Besides, more discussions about additional technical challenges can be added, where the related work [1*,2*] might be helpful.
2. The tightness of these bounds (e.g., Theorem 1 and 2) are not discussed.  More discussions can be added.
3. Although this is a pure theoretical work, experimental results can be added to illustrate the validity of these bounds.

[1*] Low-dimensional adaptation of diffusion models: Convergence in total variation

[2*] Linear Convergence of Diffusion Models Under the Manifold Hypothesis

---

> ### Author Rebuttal · Authors · 2025-07-30
>
> We sincerely thank you for your insightful comments and constructive feedback, which have helped us improve the quality of our paper.
> We address your comments in detail below.
>
> >**W1:**
> More discussions about the technical challenges for tackling the curve of dimensionality; Suggestion to discuss [1*, 2*].
>
> We thank the reviewer for this constructive suggestion.
> Indeed, our current theoretical guarantees suffer from the curse of dimensionality (CoD), as we explicitly acknowledge in the conclusion.
> Incorporating structural assumptions, such as manifold assumption explored in [1*, 2*], or low-rank structures)  is a crucial next step to mitigate the CoD.
> This would require non-trivial extensions to our analysis, particularly in adapting our empirical Bayes smoothing techniques and neural network approximation theory to efficiently exploit low-dimensional geometry.
> In particular, our analysis relies on an isotropic Gaussian kernel for the empirical score estimator. To effectively leverage a manifold structure, this would need to be replaced with an estimator that respects the underlying geometry, such as one using anisotropic or manifold-intrinsic kernels (e.g., heat kernels), to prevent smoothing data off the manifold.
> A manifold-aware analysis, following, e.g., the path of [1*, 2*], would require establishing new bounds for score estimation and approximation that depend on the intrinsic dimension of the data, rather than the ambient dimension.
> While this is beyond the scope of the current work, we view it as a promising direction.
> We thank the reviewer for this direction and will add a substantive discussion in the revision, referring to the cited works for context.
>
>
> >**W2:**
> Discussing the tightness of Theorems 1 & 2.
>
> Thank you for pointing this out.
> While we do not provide a full minimax lower bound for the neural network-based score estimation in Theorems 1 and 2, we establish the near-optimality of our framework in two key aspects:
>
> - Our rates in Theorem 2 match the minimax-optimal score estimation rate for regularized kernel-based estimators (see Lemma 1), up to logarithmic factors.
> In particular, our results align with the optimal rate derived in [29] for a similar truncated estimator, indicating that this foundational part of our analysis is indeed tight.
>
> - When we supplement our sub-Gaussian assumption with standard nonparametric smoothness assumptions (i.e., Sobolev or Besov classes), our framework demonstrates that neural network-based SGMs can achieve nearly minimax optimal convergence rates for distribution estimation in total variation distance. This is detailed in Corollaries 1 and 2.
> These results show that our framework is fundamentally sound and capable of achieving optimal rates when appropriate conditions are considered.
> We will clarify this tightness explicitly in the revised manuscript.
>
> >**W3:**
> Adding experiments to illustrate the validity of the established bounds.
>
> We appreciate the suggestion.
> As our work is purely theoretical and aims to answer fundamental questions about SGMs under minimal assumptions, we prioritized analytical clarity.
> We agree that illustrative experiments would be valuable for validating bounds empirically and will consider adding such results in future versions or follow-up work.
>
>
>
> >**Q1:**
> For the score estimation via empirical Bayes smoothing, what is the technical challenge to remove the Lipschitz score assumption compared with [30]?
>
> The primary challenge lies in controlling the score matching loss without causing the error bounds to depend sub-optimally on the noise variance $\sigma$.
> In particular, Lemma 2 of [30] established a convergence rate of the smoothed empirical distribution in Hellinger distance $H^2$ under the Lipschitz score assumption.
> As discussed in [30] following their Lemma 2, one can instead bound the Hellinger distance by KL divergence by the fact $H^2 \leq KL$ without the need of the Lipschitz score assumption. However, the SOTA convergence rate of the smoothed empirical measure in KL-divergence established in existing works, see [42], has a constant scale exponential in $1/\sigma$, i.e., $\mathcal{O}(\exp(1/\sigma)\frac{\log^dn}{n})$.  We improve this KL-divergence convergence rate in Lemma 3 for the constant from exponential dependence to **polynomial** dependence on $1/\sigma$, i.e., $\mathcal{O}(\sigma^{-d}\frac{\log^{d/2}n}{n})$. This crucial improvement allows our bounds to hold even for small noise levels without needing the Lipschitz assumption to control the error. We believe this result on the convergence of smoothed empirical measures may be of independent interest.
>
>
> [42] Rate of convergence of the smoothed empirical Wasserstein distance. 2022

---

> > ### Comment · Reviewer_oUBr · 2025-08-07
> >
> > Thanks for the detailed response, which has addressed most of my concerns. I have read all the reviews and will keep the score.

---

> > > ### Author Response · Authors · 2025-08-07
> > >
> > > Thank you once again for taking the time to review our paper and provide constructive feedback. We appreciate your positive assessment and continued support for our paper.

---

### Decision · Program_Chairs · 2025-09-17

**Decision:**

Accept (poster)

**Comment:**

This paper studies theoretical properties of score-based generative models, and establishes nearly optimal rates under weaker assumptions than existing work. The paper is technically dense but well-written, as noted by most reviewers. Several comments came up during review:
- The assumptions that are relaxed (Lipschitz, LSI, density lower bound) are typically made out of convenience, and are not fundamental road blocks in existing papers. To quote one expert reviewer: _""While the relaxation of assumptions is valuable, the novelty of the contribution may be somewhat limited given the existing literature on the theoretical analysis of SGMs"_
- The techniques used to remove these assumptions are standard: Empirical Bayes, Bernstein’s inequality, Girsanov’s theorem, etc
- Although the authors comment that "the theoretical foundations of SGMs have been less thoroughly explored", their own literature review (citing 20+ theoretical papers) suggests otherwise. Indeed, this problem is extensively studied.
- One reviewer raised a substantial concern about the potential vacuosity of the bounds, however, during the discussion, this was clarified. In particular, without smoothness assumptions (eg only subgaussian), the rates must degrade out of necessity, and the authors show that under smoothness conditions this can be corrected (consistent with existing results on density estimation).

During the discussion, the latter concern was addressed. In the end, all reviewers support accepting the paper. The concerns about techniques and assumptions remain, but the successful application of these ideas to SGMs should be of broad interest to the NeurIPS community.